# High-Order Flow Matching:
# Unified Framework and Sharp Statistical Rates

**Maojiang Su**[*†]  **Jerry Yao-Chieh Hu**[*†]  **Yi-Chen Lee**[*‡]  **Ning Zhu**[§]  **Jui-Hui Chung**[♯]

**Shang Wu**[†]  **Zhao Song**[♭]  **Minshuo Chen**[†]  **Han Liu**[†]

[†]Northwestern University  [‡]National Taiwan University  [§]University of Glasgow  [♯]Princeton University
[♭]Simon Institute of Computing, UC Berkeley

{smj,jhu,shangwu2028}@u.northwestern.edu, b10202055@ntu.edu.tw
zhuning0519@gmail.com, juihui@princeton.edu, {minshuo.chen, hanliu}@northwestern.edu

## Abstract

Flow matching is an emerging generative modeling framework that learns continuous-time dynamics to map noise into data. To enhance expressiveness and sampling efficiency, recent works have explored incorporating high-order trajectory information. Despite the empirical success, a holistic theoretical foundation is still lacking. We present a unified framework for standard and high-order flow matching that incorporates trajectory derivatives up to an arbitrary order $K$. Our key innovation is establishing the marginalization technique that converts the intractable $K$-order loss into a simple conditional regression with exact gradients and identifying the consistency constraint. We establish sharp statistical rates of the $K$-order flow matching implemented with transformer networks. With $n$ samples, flow matching estimates nonparametric distributions at a rate $\widetilde{O}(n^{-\Theta(1/d)})$, matching minimax lower bounds up to logarithmic factors.

## 1 Introduction

We present a unified theoretical framework and establish sharp statistical rates for standard and variant flow-matching generative models with high-order velocity fields. A rigorous theoretical understanding of such models is crucial in the current era of rapidly advancing generative AI. Flow-based generative models, particularly those employing Flow Matching (FM) principles [Lipman et al., 2022, Liu et al., 2022], have emerged as a powerful class of methods, achieving state-of-the-art performance across diverse domains such as image, speech, and video generation [Esser et al., 2024, Le et al., 2023, Polyak et al., 2024]. Standard flow matching has focused on learning first-order trajectory dynamics by matching the instantaneous velocity field [Lipman et al., 2022, Liu et al., 2022, Lipman et al., 2024, Gat et al., 2024, Chen and Lipman, 2023].

However, there is a growing interest in leveraging richer dynamical information, such as high-order time derivatives of the trajectory, with the intuition that this could lead to more expressive models, smoother generation paths, improved physical plausibility, or more efficient sampling strategies. This trend is evident in recent empirical works. For instance, High-Order Matching for One-Step Shortcut Diffusion (HOMO) [Chen et al., 2025] and Force Matching (ForM) [Cao et al., 2025] have shown that supervising on acceleration and jerk leads to improved smoothness, stability, and precision in generative tasks, particularly in high-curvature regions where first-order methods falter.

---

[*]Equal contribution. Version: October 28, 2025. Please see arXiv for the full and latest version.

39th Conference on Neural Information Processing Systems (NeurIPS 2025).

Despite these promising empirical explorations into high-order dynamics, there lacks a comprehensive theoretical framework that incorporates derivatives up to an arbitrary order $K$. Rigorous understanding of its statistical properties is also missing. This paper addresses these gaps by introducing High-Order Flow Matching, a generalized theoretical framework for flow-based generative modeling. Specifically, High-Order Flow Matching defines a $K$-order velocity field $f_t$. This field is constructed by concatenating $K$ individual $d$-dimensional column vector fields $u^1, \ldots, u^K$. Each $u^k$ component is designed to capture aspects of the flow dynamics, with $u^1$ representing the primary velocity and $u^k$ (with $k > 1$) capturing higher-order temporal information of an underlying flow. To complete the theoretical foundation of High-Order Flow Matching, we analyze its statistical rates when implemented with transformers [Vaswani et al., 2017] to align with modern developments in practice.

**Contributions.** Our contributions are two-fold:

- **High-Order Flow Matching: A Unified Theoretical Framework.** We present a unified framework for Flow Matching models. We first introduce the flow ODEs of any order (Definition 3.1) and the mass conservation formula (Theorem 3.2). A key technical innovation is the high-order marginalization technique (Theorem 3.3). This approach, incorporating a consistency constraint, leads to a tractable loss for $K$-order flow matching (Theorem 3.4). We then prove that High-Order Flow Matching subsumes standard first-order Flow Matching (when $K = 1$, Proposition 3.1) and provides a unified theoretical foundation for understanding emerging high-order flow model approaches. For example, the objective in HOMO [Chen et al., 2025], which target velocity and acceleration, are instantiated by High-Order Flow Matching for $K = 2$.

- **Statistical Rates for High-Order Flow Matching with Transformers.** We provide the first rigorous statistical analysis of the High-Order Flow Matching framework when implemented with transformer architectures. We establish sharp approximation rates for transformers learning the $K$ velocity components $u^1, \ldots u^K$ (Theorem 4.1), derive corresponding estimation error rates (Theorem 4.2), and further provide end-to-end distribution estimation rates under the 2-Wasserstein metric (Theorem 4.3). In addition, we show that these rates are nearly minimax optimal up to logarithmic factors (Theorem 4.4). Importantly, our rates match the established near-minimax optimal rates of standard flow matching [Jiao et al., 2024, Fukumizu et al., 2024].

**Organization.** Section 2 reviews preliminary concepts about standard flow matching. Section 3 details the High-Order Flow Matching framework, its properties, and its connections to existing methods. Section 4 presents statistical results. Section 5 summarizes our work and discusses the implications of our findings. The appendix includes the supplementary theoretical backgrounds (Appendix B), the detailed proofs of the main text (Appendices C to G), the statistical rates for standard first-order flow matching transformers (Appendix I) and its proof (Appendices J to N).

**Notation.** We denote the index set $\{1, \ldots, I\}$ by $[I]$. Let $x[i]$ denote the $i$-th component of a vector $x$. Let $\mathbb{Z}$ denote integers and $\mathbb{Z}_+$ denote positive integers. Given random variables $X$ and $Y$ with marginal densities $\mu_x$ and $\mu_y$ respectively, we denote the 2-Wasserstein distance between $\mu_x$ and $\mu_y$ by $W_2(\mu_x, \mu_y)$. Given a matrix $Z \in \mathbb{R}^{d \times L}$, $\|Z\|_2$ and $\|Z\|_F$ denote the 2-norm and the Frobenius norm. Let $u^k \in \mathbb{R}^d$ be column vectors for $k \in [K]$, we denote $\mathrm{col}(u^1, \ldots, u^K) \in \mathbb{R}^{kd}$ as the vertical concatenation of $u^1, \ldots, u^K$. Let $\mathrm{Div} \cdot$ be the divergence operator.

## 2 Preliminaries

In this section, we provide a high-level overview of the Flow Model and Flow Matching (FM).

**Flow Model.** The flow model transforms $X_0 = x_0$ from a source distribution $P$ (e.g., the Gaussian distribution) into samples $X_1 = x_1$ from a target distribution $Q$. A flow $\psi : [0, 1] \times \mathbb{R}^d \to \mathbb{R}^d$ is a time-dependent mapping implementing $\psi : (t, x) \mapsto \psi_t(x)$. The flow model is a continuous-time Markov process $(X_t)_{0 \le t \le 1}$ defined by applying a flow $\psi_t$ to the random variable $X_0 \sim P$:

$$X_t = \psi_t(X_0), \quad t \in [0, 1].$$

On the other hand, a time-dependent velocity field $u : [0, 1] \times \mathbb{R}^d \to \mathbb{R}^d$ implementing $u : (t, x) \mapsto u_t$ defines a unique flow $\psi$ via the following ordinary differential equation (ODE):

$$\frac{\mathrm{d}\psi_t}{\mathrm{d}t} = u_t(\psi_t(x)) \quad \text{with initial condition} \quad \psi_0(x) = x. \tag{2.1}$$

Given a flow $\psi_t$, the marginal probability density function (PDF) of flow model $X_t = \psi_t(X_0) \sim p_t$ is a continuous-time probability path $(p_t)_{0 \leq t \leq 1}$. The probability path $p_t$ follows push-forward equation:

$$p_t(x) = [\psi_t]_* p_0(x) := p_0(\psi_t^{-1}(x)) \cdot \left| \det \left[ \frac{\partial \psi_t^{-1}}{\partial x} \right] \right|. \tag{2.2}$$

Further, by the equivalence of flows and velocity fields [Lipman et al., 2024], given invertible $C^1$ diffeomorphism $\psi_t$, there exists a unique smooth conditional velocity field $u_t$ taking form:

$$u_t(x) = \dot{\psi}_t(\psi_t^{-1}(x)), \quad \text{with} \quad \dot{\psi}_t = \frac{\mathrm{d}}{\mathrm{d}t} \psi_t. \tag{2.3}$$

For an arbitrary probability path $p_t$, we define a velocity field $u_t$ that *generates* $p_t$ if its flow $\psi_t$ satisfies (2.2). Continuous Normalizing Flow [Chen et al., 2018] models the velocity field $u_t$ with a neural network $u^\theta$. Once we obtain a well-trained $u^\theta$, we generate samples from solving ODE (2.1).

**Flow Matching.** Instead of training flow model by maximizing the log-likelihood of training data [Chen et al., 2018], flow matching [Lipman et al., 2022] is a simulation-free framework to train flow generative models without the need of solving ODEs during training. The Flow Matching objective is designed to match the probability path $(p_t)_{0 \leq t \leq 1}$, which allows us to flow from source $p_0 = P$ to target $p_1 = Q$. Suppose $u_t$ generates such probability path $p_t$, the flow matching loss is

$$\mathcal{L}_{\mathrm{FM}}(\theta) = \mathop{\mathbb{E}}_{t, X_t \sim p_t} [\|u^\theta(X_t, t) - u_t(X_t)\|_2^2], \tag{2.4}$$

where $t \sim U[0,1]$, $u^\theta$ is a neural network with parameter $\theta$. Flow Matching simplifies the problem of designing a probability path $p_t$ and its corresponding velocity field $u_t$ by adopting a conditional strategy. Formally, conditioning on any arbitrary random vector $Z \in \mathbb{R}^m$ with PDF $p_Z$, the marginal probability path $p_t$ satisfies

$$p_t(x) = \int p_t(x|z) p_Z(z) \mathrm{d}z. \tag{2.5}$$

Suppose conditional velocity field $u_t(x|z)$ generates $p_t(x|z)$, Lipman et al. [2022] show that following marginal velocity field $u_t$ generates marginal probability path $p_t$ under mild assumptions:

$$u_t(x) := \int u_t(x|z) p_{Z|t}(z|x) \mathrm{d}z \quad \text{with} \quad p_{Z|t}(z|x) = \frac{p_t(x|z) p_Z(z)}{p_t(x)}, \tag{2.6}$$

where the second equation follows from the Bayes' rule. Combining above, the tractable conditional flow matching loss $\mathcal{L}_{\mathrm{CFM}}$, which satisfies $\nabla_\theta \mathcal{L}_{\mathrm{CFM}}(\theta) = \nabla_\theta \mathcal{L}_{\mathrm{FM}}(\theta)$, is defined as:

$$\mathcal{L}_{\mathrm{CFM}}(\theta) = \mathop{\mathbb{E}}_{t, Z \sim p_Z, X_t \sim p_t(\cdot|Z)} [\|u^\theta(X_t, t) - u_t(X_t|Z)\|_2^2]. \tag{2.7}$$

**Affine Conditional Flows.** The conditional flow matching loss works with any choice of conditional probability path and conditional velocity fields. In this paper, we consider the affine conditional flow with independent data coupling following [Lipman et al., 2022, 2024]:

$$\psi_t(x|x_1) = \mu_t x_1 + \sigma_t x, \tag{2.8}$$

where $\mu_t, \sigma_t : [0,1] \to [0,1]$ are monotone smooth functions satisfying

$$\mu_0 = \sigma_1 = 0, \ \mu_1 = \sigma_0 = 1, \quad \text{and} \quad \frac{\mathrm{d}\mu_t}{\mathrm{d}t}, -\frac{\mathrm{d}\sigma_t}{\mathrm{d}t} > 0 \quad \text{for} \quad t \in (0,1). \tag{2.9}$$

Setting $Z = X_1 \sim Q$, $X_0 \sim N(0, I)$, the flow $\psi_t$ induces the probability flow $p_t(X_t|X_1) = N(\mu_t X_1, \sigma_t^2 I)$ and velocity field

$$u_t(x|x_1) = \dot{\psi}_t(\psi_t^{-1}(x|x_1)|x_1) = \frac{\dot{\sigma}_t(x - \mu_t x_1)}{\sigma_t} + \dot{\mu}_t x_1. \tag{2.10}$$

Further, using the law of unconscious statistician with $X_t = \psi_t(X_0|X_1)$, the conditional flow matching loss takes the form

$$\mathcal{L}_{\mathrm{CFM}}(\theta) = \mathop{\mathbb{E}}_{t, X_1 \sim q, X_0 \sim N(0, I)} \left[ \|u^\theta(\mu_t X_1 + \sigma_t X_0, t) - (\dot{\mu}_t X_1 + \dot{\sigma}_t X_0)\|_2^2 \right]. \tag{2.11}$$

In practice, for collected i.i.d. data points $\{x_i\}_{i=1}^n$, (2.11) is implemented with Monte-Carlo simulation. To avoid instability, we often clip the interval $[0, 1]$ with $t_0$ and $T$. Namely, for any velocity estimator $u^\theta$, we consider the empirical loss function $\widehat{\mathcal{L}}_{\mathrm{CFM}}(u^\theta)$:

$$\widehat{\mathcal{L}}_{\mathrm{CFM}}(u^\theta) := \frac{1}{n} \sum_{i=1}^n \int_{t_0}^T \frac{1}{T - t_0} \mathop{\mathbb{E}}_{X_0 \sim N(0, I)} \big[ \|u^\theta(\mu_t x_i + \sigma_t X_0, t) - (\dot{\mu}_t x_i + \dot{\sigma}_t X_0)\|_2^2 \big] \mathrm{d}t. \quad (2.12)$$

**Transformers.** Throughout the paper, we parameterize $u^\theta$ by transformers. Due to space limit, we defer formal definition of transformer networks to Appendix B.

## 3 High-Order Flow Matching

This section extends the flow matching framework in Section 2 to incorporate high-order trajectory information. Recall that these high-order dynamics are proven to be relevant to further improving the performance and stability of flow matching. Specifically, in Section 3.1, we first define a high-order velocity field $f_t$ using an ODE system and subsequently prove its equivalence to the mapping flow $\psi_t$ (Theorem 3.1). Furthermore, we derive the corresponding Liouville's equation (Theorem 3.2), which demonstrates mass conservation for this high-order system. Building on this foundation, Section 3.2 addresses the learning objective. We first propose the high-order Flow Matching loss (Definition 3.2).

However, similar to flow matching [Lipman et al., 2022], direct optimization is intractable. To address this, we establish the high-order marginalization trick under consistency constraint (Theorem 3.3). The method allows us to derive a tractable high-order conditional flow matching loss that preserves the original loss's gradients (Theorem 3.4). Section 3.3 clarify how that High-Order Flow provides a unifying theory. Specifically, we demonstrate that high-order flow matching subsumes existing flow-based generative modeling techniques, with standard Flow Matching serving as a foundational instance within our framework.

### 3.1 High-Order Flow Model

For $t \in [0, 1]$, let $\psi_t$ and $p_t$ be the time-dependent flow mapping and probability paths follows Section 2. Instead of using velocity field $u_t$ to construct flow $\psi_t$ via the ODE (2.1), we propose using $K$-order velocity field $f_t : \mathbb{R}^{Kd} \to \mathbb{R}^{Kd}$ to construct $\psi_t$:

**Definition 3.1** (High-Order Velocity). Let $t \in [0, 1]$, a flow $\psi_t$ can define a $K$-order velocity field $f_t : \mathbb{R}^{Kd} \to \mathbb{R}^{Kd}$ via the following ODE:

$$\frac{\mathrm{d}}{\mathrm{d}t} y_t = \begin{bmatrix} \frac{\mathrm{d}^1}{\mathrm{d}t^1} \psi_t(x) \\ \frac{\mathrm{d}^2}{\mathrm{d}t^2} \psi_t(x) \\ \vdots \\ \frac{\mathrm{d}^K}{\mathrm{d}t^K} \psi_t(x) \end{bmatrix} = \begin{bmatrix} u^1(x_t^{(0)}, t) \\ u^2(x_t^{(0)}, t) \\ \vdots \\ u^K(x_t^{(0)}, t) \end{bmatrix} = f_t(y_t) \quad \text{with} \quad \psi_0(x) = x, \quad (3.1)$$

where $y_t = \mathrm{col}(\psi_t(x), \frac{\mathrm{d}}{\mathrm{d}t} \psi_t(x), \dots, \frac{\mathrm{d}^{K-1}}{\mathrm{d}t^{K-1}} \psi_t(x)) := \mathrm{col}(x_t^{(0)}, x_t^{(1)}, \dots, x_t^{(K-1)}) \in \mathbb{R}^{Kd}$ and $u^k : \mathbb{R}^{kd} \times [0, 1] \to \mathbb{R}^d$ is $k$-th order velocity field for all $k \in [K]$. Moreover, notice that $X_t^{(0)} = \psi_t(X_0)$ is random variable since $X_0 \sim p$. Then, the extended state variable of order $K$ is the random vector

$$Y_t = \mathrm{col}(X_t^{(0)}, \dots, X_t^{(K-1)}) \in \mathbb{R}^{Kd} \quad \text{with} \quad X_t^{(k)} := \frac{\mathrm{d}^k}{\mathrm{d}t^k} \psi_t(x)|_{x=X_0^{(0)}}. \quad (3.2)$$

For $k = 0, \dots, K - 1$, define $p_t^k : \mathbb{R}^d \to \mathbb{R}$ as the probability density function of $X_t^{(k)}$. Denote $\rho_t : \mathbb{R}^{Kd} \to \mathbb{R}$ as the probability density function of $Y_t = [X_t^{(0)}, \dots, X_t^{(K-1)}]^\top$ at time $t$. For simplification, we define $Y_t$ satisfy $\frac{\mathrm{d}}{\mathrm{d}t} Y_t = f_t(Y_t)$ if (3.1) and (3.2) hold.

**Remark 3.1** (Total Derivative Constraints). The ODE (3.1) imposes a sequence of total derivative constraints on the velocity fields $u^1(x_t^{(0)}, t), \dots, u^K(x_t^{(0)}, t)$, for any $k \in [K]$:

$$u^k(x_t^{(0)}, t) = \frac{\mathrm{d}^k}{\mathrm{d}t^k} \psi_t(x) = \frac{\mathrm{d}}{\mathrm{d}t} u^{k-1}(x_t^{(0)}, t) = \frac{\partial}{\partial t} u^{k-1}(x_t^{(0)}, t) + \nabla u^{k-1}(x_t^{(0)}, t) \cdot u^1(x_t^{(0)}, t),$$
$$(3.3)$$

where $u^0(x_t^{(0)}, t) = x_t^{(0)}$. This recursive relation reveals that the velocity fields induced by the flow $\psi_t$ are not independent, but instead coupled through the structure of the ODE via (3.3).

Remark 3.1 guarantees the equivalence between flows $\psi_t$ and $K$-order velocity field $f_t$.

**Theorem 3.1** (Flow–Velocity Equivalence via ODE). Define the class of structured $k$-order velocity fields as those of the form:

$$f_t(y_t) = \mathrm{col}(u^1(x_t^{(0)}, t), \ldots, u^K(x_t^{(0)}, t)) \in \mathbb{R}^{Kd}, \quad y_t = \mathrm{col}(x_t^{(0)}, \ldots, x_t^{(K-1)}) \in \mathbb{R}^{Kd},$$

where $u^k : \mathbb{R}^{Kd} \times [0, 1] \to$ is locally lipschitz in $y_t$ and continues in $t$ for any $k \in [K]$. Suppose the velocity fields $u^1(x_t^{(0)}, t), \ldots, u^K(x_t^{(0)}, t)$ satisfy total derivative constraints (3.3). Then, for any initial condition $y_0 \in \mathbb{R}^{Kd}$, the ODE $\frac{\mathrm{d}}{\mathrm{d}t} y_t = f_t(y_t)$ exists a unique local solution $y_t$, which defines a $K$-times differentiable flow $\psi_t(x) := x_t^{(0)}$ and satisfy $\frac{\mathrm{d}^k}{\mathrm{d}t^k} \psi_t(x) = x_t^{(k)}$ for all $k \in [K]$.
Conversely, any $K$-times differentiable flow $\psi_t : \mathbb{R}^d \to \mathbb{R}^d$ defines a velocity field $f_t$ via (3.1).

*Proof.* Please see Appendix C.1 for a detailed proof. □

Recalling from Section 2 and the flow-velocity equivalence established in Theorem 3.1, the $K$-order velocity field $f_t$ governs the evolution of the probability density $\rho_t$ for the $K$-order state $Y_t$. The precise relationship describing this evolution is captured by the mass conservation formula:

**Theorem 3.2** (Mass Conservation of High-Order Flow). Let $y_t = (x_t^{(0)}, \ldots, x_t^{(K-1)})^\top \in \mathbb{R}^{Kd}$. Let velocity field $f_t(y_t) = (u^1(x_t^{(0)}, t), \ldots, u^K(x_t^{(0)}, t))^\top \in \mathbb{R}^{Kd}$, where $u^k(x_t^{(0)}, t)$ is locally Lipschitz and integrable for all $k \in [K]$. Let $\rho_t : \mathbb{R}^{Kd} \to \mathbb{R}$ be a time-varying probability density over the extended state $Y_t \in \mathbb{R}^{Kd}$ follows Definition 3.1. Then the following statements are equivalent:

1. The pair $(f_t, \rho_t)$ satisfies the Liouville's equation on the extended space:

$$\frac{\partial}{\partial t} \rho_t(y) + \nabla_y \cdot (\rho_t(y) f_t(y)) = 0, \quad \text{for all } t \in [0, 1).$$

2. Following Definition 3.1, the probability law of $Y_t$ evolves under the flow:

$$\frac{\mathrm{d}}{\mathrm{d}t} Y_t = f_t(Y_t), \quad \text{with} \quad Y_0 \sim \rho_0, \quad Y_t \sim \rho_t. \tag{3.4}$$

For some arbitrary probability path $\rho_t$, we define $f_t$ *generates* $\rho_t$ if (3.4) holds.

*Proof.* Please see Appendix C.2 for a detailed proof. □

## 3.2 High-Order Flow Matching

To model the $K$-order velocity field $f_t$, we introduce following high-order flow matching loss:

**Definition 3.2** (High-Order Flow Matching Loss). Let $f_t$ denote the ground truth $K$-order velocity field and $f_t^\theta$ be its estimator parameterized by a neural network. Let $\rho_t$ be the probability density function of $Y_t$. Then, the $K$-order Flow Matching objective minimizes the following regression loss:

$$\mathcal{L}_{\mathrm{FM}}^K(\theta) = \mathbb{E}_{t, Y_t \sim \rho_t} [D(f_t(Y_t), f_t^\theta(Y_t))],$$

where $D$ is a dissimilarity measure between vectors, such as the squared $\ell_2$-norm.

Similar to standard flow matching, the ground truth velocity $f_t$ is intractable. To address this, we adopt the conditional flow matching loss to train our model, leveraging the equivalence between the flow matching loss and its conditional counterpart. As a preliminary step, we introduce the marginalization trick for high-order flow matching.

**Theorem 3.3** (Marginalization). Recall that for some arbitrary probability path $\rho_t$, $f_t$ *generates* $\rho_t$ if $Y_t \sim \rho_t$ for all $t \in [0, 1)$. Let $Z$ be a random variable, if $f_t(x|z)$ is conditionally integrable and generates the conditional probability path $\rho_t(\cdot|z)$, then the marginal velocity $f_t := \int f_t(y|z) p_t(z|y) \mathrm{d}z$ generates the marginal probability path $p_t$.[2]

*Proof.* Please see Appendix C.3 for a detailed proof. □

Now we are ready to prove the higher version of the equivalence between the flow matching loss and conditional flow matching loss. We first define the tractable $K$-order conditional flow matching loss:

$$\mathcal{L}_{\text{CFM}}^K(\theta) = \mathop{\mathbb{E}}_{t, Z, Y_t \sim \rho_{t|Z}(\cdot|Z)} [D(f_t(Y_t|Z), f_t^\theta(Y_t))]. \tag{3.5}$$

Following Lipman et al. [2024], we specify the dissimilarity metric $D(\cdot, \cdot)$ as a Bregman divergence, which measures the distance between vectors $u, v \in \mathbb{R}^{Kd}$ as $D(u, v) := \Phi(u) - [\Phi(v) + (u - v)^\top \nabla \Phi(v)]$ where $\Phi : \mathbb{R}^{Kd} \to \mathbb{R}$ is a strictly convex function defined on a convex domain $\Omega \subset \mathbb{R}^{Kd}$. Bregman divergences possess a key property allowing interchanging gradients and expectations [Holderrieth et al., 2025, Lipman et al., 2024]:

$$\nabla_v D(\mathbb{E}[Y], v) = \mathbb{E}[\nabla_v D(Y, v)] \quad \text{for any random vector} \quad Y \in \mathbb{R}^{Kd}. \tag{3.6}$$

This property implies that the gradients of the flow matching loss and the conditional flow matching loss are identical, making the two objectives equivalent for training.

**Theorem 3.4** (Gradient Equivalence of Losses). Let the Flow Matching loss $\mathcal{L}_{\text{FM}}^K$ be defined as in Definition 3.2, and the Conditional Flow Matching loss $\mathcal{L}_{\text{CFM}}^K$ be defined as in (3.5). Then, when $D(\cdot, \cdot)$ is a Bregman divergence, the gradients of the two losses coincide:

$$\nabla \mathcal{L}_{\text{FM}}^K(\theta) = \nabla \mathcal{L}_{\text{CFM}}^K(\theta).$$

*Proof.* Please see Appendix C.4 for a detailed proof. □

We now consider training the model using the pre-constructed conditional flow $\psi_t(x \mid x_1)$ as described in Section 2. By the equivalence between flows and high-order velocity fields (Theorem 3.1), there exists a unique smooth conditional $K$-order velocity field $f_t$ such that the conditional trajectory $y_t$ satisfies the ODE: $\frac{\mathrm{d}}{\mathrm{d}t} y_t = f_t(y_t)$, in accordance with (3.1). Following Definition 3.1, we specify $\psi_t(x \mid x_1) = \mu_t x_1 + \sigma_t x$, which induces a family of $k$-th order velocity fields $u^k$. By Definition 3.1, for all $k \in [K]$, we have

$$u^k(x_t^{(0)}, t) = \frac{\mathrm{d}^k}{\mathrm{d}t^k} x_t^{(0)} = \frac{\mathrm{d}^k}{\mathrm{d}t^k} \psi_t(x). \qquad (\text{By Definition 3.1})$$

Because $\psi_t$ is an invertible diffeomorphism, we define $x' = \psi_t^{-1}(x)$ and obtain

$$u^k(\psi_t(x), t) = u^k(x', t) = \frac{\mathrm{d}^k}{\mathrm{d}t^k} \psi_t(\psi_t^{-1}(x')).$$

Extending this to the conditional setting, the conditional $k$-th order velocity field becomes

$$u^k(x, t|X_1^{(0)}) = \frac{\mathrm{d}^k}{\mathrm{d}t^k} \psi_t(\psi_t^{-1}(x|X_1^{(0)})|X_1^{(0)}). \tag{3.7}$$

Combining the results above, we now revisit the tractable training loss by setting $Z = X_1^{(0)} \sim q$:

$$\mathcal{L}_{\text{CFM}}^K(\theta) = \mathop{\mathbb{E}}_{t, X_1^{(0)} \sim q, Y_t \sim \rho_{t|X_1^{(0)}}(\cdot|X_1^{(0)})} [D(f_t(Y_t|X_1^{(0)}), f_t^\theta(Y_t))]. \qquad (\text{By (3.5)})$$

For further simplifications, we adopt the squared $\ell_2$ norm as the Bregman divergence. Let $u^k$ denote the $k$-th order velocity field, and $u^{k,\theta}$ be its estimator parameterized by a neural network. Denoting the distribution of the $k$-th order state as $X_t^{(k)} \sim p_t^k$, the training objective becomes

$$\mathcal{L}_{\text{CFM}}^K(\theta) = \mathop{\mathbb{E}}_{t, X_1^{(0)} \sim q, Y_t \sim \rho_{t|X_1^{(0)}}(\cdot|X_1^{(0)})} \left[ \|f_t(Y_t|X_1^{(0)}) - f_t^\theta(Y_t)\|_2^2 \right] \qquad (\text{By (3.5)})$$

$$= \mathop{\mathbb{E}}_{t, X_1^{(0)} \sim q, Y_t \sim \rho_{t|X_1^{(0)}}(\cdot|X_1^{(0)})} \left[ \sum_{k=1}^K \|u^k(Y_t, t|X_1^{(0)}) - u^{k,\theta}(Y_t, t)\|_2^2 \right] \quad (\text{By Definition 3.1})$$

---

[2] The marginal velocity $f_t$ implies a consistency constraint: $u_t^k(y) = \int u_t^k(y|z) \cdot p_t(z|y) \mathrm{d}z$ for all $k \in [K]$.

$$= \mathbb{E}_{t,X_1^{(0)} \sim q} \left[ \sum_{k=1}^{K} \mathbb{E}_{X_0^{(0)} \sim p(\cdot|X_1^{(0)})} \| \frac{\mathrm{d}^k}{\mathrm{d}t^k} \psi_t(X_0^{(0)}|X_1^{(0)}) - u^{k,\theta}(X_t^{(0)},t) \|_2^2 \right] \qquad \text{(By (3.7))}$$

$$= \sum_{k=1}^{K} \mathbb{E}_{t,X_1^{(0)} \sim q, X_0^{(0)} \sim p(\cdot|X_1^{(0)})} \left[ \| \frac{\mathrm{d}^k}{\mathrm{d}t^k} \psi_t(X_0^{(0)}|X_1^{(0)}) - u^{k,\theta}(X_t^{(0)},t) \|_2^2 \right]. \qquad (3.8)$$

The intermediate states $X_t^{(1)}, \ldots, X_t^{(k-1)}$ are determined by $X_0^{(0)}$ via the relation $X_t^{(k)} := \frac{\mathrm{d}^k}{\mathrm{d}t^k} \psi_t(x)\big|_{x=X_0^{(0)}}$. Therefore, the inside expectation only needs to be taken over $X_0^{(0)}$.

Now, we consider the affine conditional flow $\psi_t(x|x_1) = \mu_t x_1 + \sigma_t x$ follows Section 2. Applying (3.8), the high-order conditional flow matching loss takes the form

$$\mathcal{L}_{\mathrm{CFM}}^K(\theta) = \sum_{k=1}^{K} \mathbb{E}_{t,X_1^{(0)} \sim q, X_0^{(0)} \sim p(\cdot|X_1^{(0)})} \left[ \| (\mu_t^{(k)} X_1^{(0)} + \sigma_t^{(k)} X_0^{(0)}) - u^{k,\theta}(X_t^{(0)},t) \|_2^2 \right].$$

In practice, we train the general high-order velocity estimator $u^{1,\theta}, \ldots, u^{K,\theta}$ with i.i.d samples $\{x_i\}_{i=1}^n$ by optimizing the empirical high-order conditional flow matching loss:

$$\widehat{\mathcal{L}}_{\mathrm{CFM}}^K := \frac{1}{n} \sum_{i=1}^{n} \sum_{k=1}^{K} \frac{1}{T-t_0} \int_{t_0}^{T} \mathbb{E}_{X_0 \sim p(\cdot|X_1^{(0)})} \left[ \| (\mu_t^{(k)} x_i + \sigma_t^{(k)} X_0^{(0)}) - u^{k,\theta}(X_t^{(0)},t) \|_2^2 \right] \mathrm{d}t. \quad (3.9)$$

A significant theoretical consequence of learning the complete $K$-order velocity field $f_t$ is the ability to employ high-order numerical integration schemes for sampling. For instance, to solve the ODE (3.1), we use $K$-th order Taylor expansion with step size $h$ for the numerical integration:

$$x_{t+h}^{(0)} = x_t + h u^{1,\theta}(x_t^{(0)}, t) + \frac{h^2}{2!} u^{2,\theta}(x_t^{(0)}, t) + \cdots + \frac{h^K}{K!} u^{K,\theta}(x_t^{(0)}, t). \qquad (3.10)$$

## 3.3 Unified Perspective on High-Order Flow Dynamics

We show that our $K$-order flow matching framework offers a significant unification perspective and a theoretical foundation on existing flow-based generative modeling. Firstly, our framework subsumes standard first-order Flow Matching [Lipman et al., 2022] as a direct special case.

**Proposition 3.1** (Reduction to Standard First-Order Flow Matching). When $K = 1$, the entire $K$-order flow matching framework, including the governing ODE, the probability path definition via the continuity equation, and the $K$-order flow matching objective, becomes precisely equivalent to the standard first-order Flow Matching framework as detailed in [Lipman et al., 2022, 2024].

*Proof.* Please see Appendix C.5 for a detailed proof. □

Proposition 3.1 establishes our $K$-order framework as a strict generalization of standard first-order Flow Matching. Beyond encompassing established methods, our $K$-order framework provides a robust theoretical structure for understanding models that leverage high-order trajectory dynamics.

For instance, HOMO framework [Chen et al., 2025] defines its training objective ([Chen et al., 2025, Definition 4.3]) by matching network predictions against the true velocity $\dot{x}$ and acceleration $\ddot{x}$ of trajectories. Removing the regularization term (aligns with our total derivative constraints Remark 3.1), their loss is also a direct instantiation of our $K$-order framework's objective (Definition 3.2) for $K = 2$. Furthermore, while the Force Matching (ForM) model [Cao et al., 2025] introduces specific relativistic constraints, its fundamental generative mechanism involves matching a target "force" field ([Cao et al., 2025, Definition 4.1]). Given that force is proportional to acceleration, if separated from its relativistic regularization, aligns with matching the second-order information captured within our $K = 2$ framework.

In summary, the $K$-order flow matching framework serves as a unifying theoretical structure. It not only subsumes standard flow matching but also provides formal grounding for models that have intuitive or empirical benefits of incorporating richer, high-order dynamical information. The subsequent statistical analysis in Section 4 builds upon this unified perspective.

# 4 Statistical Rates of High-Order Flow Matching Transformers

This section characterizes sharp statistical rates for $K$-order flow matching transformers. Building on Section 2 and Section 3, we consider the case of affine conditional flow with independent data coupling. We focus on transformer architectures as Flow matching (FM) with transformers powers today's best generative models, including MovieGen [Polyak et al., 2024] and Voicebox [Le et al., 2023] by Meta, and Rectified Flow [Esser et al., 2024] by Stability AI. Section 4.1 and Section 4.2 establish bounds for the approximation and estimation of the $K$-order velocity. Based on the $K$-order velocity estimation rates, Section 4.3 analyzes the distribution estimation rate under the 2-Wasserstein metric. Finally, Section 4.4 presents the nearly minimax optimality of the $K$-order velocity estimators.

**Transformers.** We defer standard definition of transformer to Appendix B due to the page limit.

## 4.1 High-Order Velocity Approximation

To establish a statistical theory for $K$-order flow matching transformers, we first investigate an approximation theory for the $K$-order velocity under sub-Gaussian assumption. In particular, we characterize the regularity of the target density function $q(x_1)$ with Hölder smoothness, defined by:

**Definition 4.1** (Hölder Space). Let $\alpha \in \mathbb{Z}_+^d$, and let $\beta = k_1 + \gamma$ denote the smoothness parameter, where $k_1 = \lfloor \beta \rfloor$ and $\gamma \in [0, 1)$. Given a function $f : \mathbb{R}^d \to \mathbb{R}$, the Hölder space $\mathcal{H}^\beta(\mathbb{R}^d)$ is defined as the set of $\alpha$-differentiable functions satisfying: $\mathcal{H}^\beta(\mathbb{R}^d) := \{ f : \mathbb{R}^d \to \mathbb{R} \mid \|f\|_{\mathcal{H}^\beta(\mathbb{R}^d)} < \infty \}$, where the Hölder norm $\|f\|_{\mathcal{H}^\beta(\mathbb{R}^d)}$ satisfies:

$$\|f\|_{\mathcal{H}^\beta(\mathbb{R}^d)} := \sum_{\|\alpha\|_1 < k_1} \sup_x |\partial^\alpha f(x)| + \max_{\alpha:\|\alpha\|_1 = k_1} \sup_{x \neq x'} \frac{|\partial^\alpha f(x) - \partial^\alpha f(x')|}{\|x - x'\|_\infty^\gamma}.$$

Also, we define the Hölder ball of radius $B$ by $\mathcal{H}^\beta(\mathbb{R}^d, B) := \{ f : \mathbb{R}^d \to \mathbb{R} \mid \|f\|_{\mathcal{H}^\beta(\mathbb{R}^d)} < B \}$.

With Definition 4.1, we state our assumption on the target density function $q(x_1)$:

**Assumption 4.1** (Sub-Gaussian Property and Hölder Smoothness of Target Distribution). The target distribution $q(x_1) \in \mathcal{H}^\beta(\mathbb{R}^{d_x}, B)$. Further, there exist two positive constants $C_1$ and $C_2$ such that $q(x_1) \leq C_1 \exp(-C_2 \|x_1\|_2^2 / 2)$.

Assumption 4.1 provides a tail bound for the approximation error, and we leverage it to address the error outside the bounded domain where our transformer approximation applies. We now present the approximation theory for high-order flow matching transformers.

**Theorem 4.1** ($K$-order Velocity Approximation with Transformers). Assume Assumption 4.1. Suppose the $k$-th order velocity field $u^k(x, t)$ is $L_k$-Lipschitz for all $k \in 0, \ldots, K-1$ in $\ell_2$-distance. Let $\epsilon \in (0, 1)$ be the precision parameter satisfying $\epsilon \leq O(N^{-\beta})$ for some $N \in \mathbb{N}$ and smoothness parameter $\beta > 0$. Then, there exists transformers $u^{1,\theta}(x, t), \ldots, u^{K,\theta}(x, t) \in \mathcal{T}_R^{h,s,r}$ such that for any $x \in \mathbb{R}^{d_x}$ and $t \in [0, 1]$, it holds:

$$\sum_{k=1}^K \int_{t_0}^T \int_{\mathbb{R}^{d_x}} \|u^{k,\theta}(x, t) - u^k(x, t)\|_2^2 \cdot p_t(x) \mathrm{d}x \mathrm{d}t = O\left(N^{-2\beta} \cdot (\log N)^{\frac{d_x}{2} - 1}\right).$$

Further, for all $k \in [K]$, the parameter bounds in transformer network class satisfy

$$C_{KQ}, C_{KQ}^{2,\infty} = O(\lambda^{-1} N^{2\beta(2d+1)} (\log N)^{2d+1}); \quad C_{OV}, C_{OV}^{2,\infty} = O(N^{-\beta});$$

$$C_F, C_F^{2,\infty} = O(N^\beta \sqrt{\log N} L_{k-1}); \quad C_E = O(1); \quad C_\mathcal{T} = O(L_{k-1}),$$

where $\lambda^{-1} = O(N^\beta \log N)^{4d+3}$ is the inverse-temperature scaling in the softmax function and $O(\cdot)$ hides all polynomial factors depending on $d_x, d, L, \beta, C_1, C_2$.

*Proof.* Please see Appendix D for a detailed proof. □

## 4.2 High-Order Velocity Estimation

In this section, we apply the approximation results in Section 4.1 to derive $K$-order velocity estimation rates (Theorem 4.2). Given a set of i.i.d samples $\{x_i\}_{i=1}^n$, we train transformer networks

$u^{1,\theta}, \ldots, u^{K,\theta}$ by minimizing the high-order empirical conditional flow matching loss (3.9):

$$\widehat{\mathcal{L}}_{\text{CFM}}^K = \frac{1}{n} \sum_{i=1}^n \sum_{k=1}^K \frac{1}{T-t_0} \int_{t_0}^T \mathop{\mathbb{E}}_{X_0 \sim N(0,I)} \left[ \|(\mu_t^{(k)} x_i + \sigma_t^{(k)} X_0^{(0)}) - u^{k,\theta}(X_t^{(0)}, t)\|_2^2 \right] \, \mathrm{d}t.$$

We evaluate the performance of estimators $u^{1,\theta}, \ldots, u^{K,\theta}$ through the $K$-order flow matching risk:

**Definition 4.2** (High-Order Flow Matching Risk). Let $u^{k,\theta}$ be the estimator of the $k$-th order velocity field $u^k$. Let $\Theta$ be the collection of parameters of $u^{1,\theta}, \ldots, u^{K,\theta}$. We define the flow matching risk $\mathcal{R}_K(\Theta)$ as the sum of the expected mean-squared difference between $u^{k,\theta}$ and $u^k$:

$$\mathcal{R}_K(\Theta) := \sum_{k=1}^K \frac{1}{T-t_0} \int_{t_0}^T \mathop{\mathbb{E}}_{x \sim p_t^0} \left[ \|u^k(x,t) - u^{k,\theta}(x,t)\|_2^2 \right] \, \mathrm{d}t,$$

where the density function $p_t^0$ represents the probability density function of $X_t^{(0)}$ (Definition 3.1).

Further, we assume the path coefficients of the affine conditional flow preserve regularity.

**Assumption 4.2** (Path Regularity). Consider the affine conditional flow $\psi_t(x|X_1^{(0)}) = \mu_t X_1^{(0)} + \sigma_t x$, the $k$-th derivative of path coefficients $\sigma_t$ and $\mu_t$ are continuous on $[t_0, T]$, where $t_0, T \in [0, 1]$.

Assuming $k$-th order velocity Lipschitz continuity and affine path regularity (Assumption 4.2), the following theorem presents the upper bounds on estimation error $\mathcal{R}_K(\Theta)$ with sample size $n$.

**Theorem 4.2** (High-Order Velocity Estimation with Transformer). Assume Assumption 4.1 and Assumption 4.2. Let $\widehat{u}^{k,\theta} \in \mathcal{T}_R^{h,s,r}$ be the estimator of the $k$-th order velocity field $u^k$ trained by minimizing the high-order empirical conditional flow matching loss (3.9). Let $\widehat{\Theta}$ be the collection of parameters of $\widehat{u}^{k,\theta}$ for $k \in [K]$. Suppose the $k$-th order velocity field $u^k(x,t)$ is $L_k$ Lipschitz for all $k = 0, \ldots, K-1$. Suppose we choose the transformers as in Theorem 4.1, then

$$\mathop{\mathbb{E}}_{\{x_i\}_{i=1}^n} \left[ \mathcal{R}_K(\widehat{\Theta}) \right] = O\left( n^{-\frac{1}{10d}} \cdot (\log n)^{10d_x} \right),$$

where $d$ is the feature dimension.

*Proof.* Please see Appendix E for a detailed proof. $\square$

### 4.3 High-Order Distribution Estimation

Based on the $K$-order velocity estimation result in Theorem 4.2, we further analyze the distribution estimation rate for $K$-order flow matching transformer. The next theorem presents the upper bounds on the expectation of 2-Wasserstein distance between the target and estimated distribution induced by estimators $u^{k,\theta}$ trained by optimizing the empirical conditional loss (3.9).

**Theorem 4.3** (High-Order Distribution Estimation under 2-Wasserstein Distance). Assume Assumption 4.1 and Assumption 4.2. Let $\widehat{P}_T^K$ be the estimated distribution at time $T$. Then, it holds

$$\mathop{\mathbb{E}}_{\{x_i\}_{i=1}^n} [W_2(\widehat{P}_T^K, P_T^K)] = O\left( n^{-\frac{1}{18d}} \cdot (\log n)^{6d_x} \right),$$

where $d$ is the feature dimension.

*Proof.* Please see Appendix F for a detailed proof. $\square$

### 4.4 High-Order Minimax Optimal Estimation

We show that the $K$-order flow matching transformers achieves nearly minimax optimal rate:

**Theorem 4.4** (Minimax Optimality of High-Order Flow Matching Transformers). Assume that the target density function satisfies $q(x_1) \in \mathcal{H}^\beta([-1,1]^{d_x}, B)$ and $q(x_1) \geq C$ for some positive constant $C$. Then, under the setting of $18d(\beta+1) = d_x + 2\beta$, the distribution estimation rate of flow matching transformers presented in Theorem 4.3 matches the minimax lower bound of Hölder distribution class in 2-Wasserstein distance up to a $\log n$ and Lipschitz constants factors.

*Proof.* Please see Appendix G for a detailed proof. □

**Remark 4.1** (Comparison with Existing Works). Flow matching with ReLU networks is nearly minimax-optimal on Besov densities in $W_2$ [Fukumizu et al., 2024], and kernel methods achieve comparable rates in $W_1$ [Kunkel and Trabs, 2025]. We extend these results to all orders $K$ and to the major powerhouse in practice: transformer architectures. Our analysis proves that flow-matching transformers attain near-minimax rates on Hölder densities in $W_2$ with assuming Lipschitz velocities, subsuming the first-order case at $K = 1$. Please see Appendix I for details.

## 5 Discussion, Limitation, and Open Question

Section 3 and Section 4 establish a unified theoretical framework for High-Order Flow Matching and offer a sharp statistical analysis of High-Order Flow Matching transformers. As discussed in Section 3.3, this framework subsumes the not only original first-order [Lipman et al., 2024, 2022] but also many high-order flow matching models [Chen et al., 2025, Cao et al., 2025]. Furthermore, the established sharp statistical rates provide rigorous support for all models under this unified framework. This broad theoretical guarantee, covering both first-order and high-order approaches, helps explain the empirical success of the high-order flow models.

While our analysis provides foundational statistical guarantees, the compelling empirical evidence and our current theoretical framework present an intriguing open question: it does not elucidate a significant improvement in statistical rates with increasing order $K$. In addition, while our framework offers a unified perspective for numerous empirical studies, these often assume the validity of the consistency constraint within the marginalization process (Theorem 3.3). Our research indicates that the general validity of this constraint, or indeed the derivation of similar conclusions under broader conditions, remains an open question. We identify three primary directions for future work stemming from these considerations: (i) Sampling Efficiency: The High-Order Flow Matching framework enables the use of a $K$-th order Taylor expansion sampler. This sampler achieves a local truncation error of $O(h^{K+1})$ per step, with all $K$ velocity components $u^{k,\theta}$ evaluable in parallel. Future empirical work should investigate whether this high-order accuracy per step translates into practical benefits, such as requiring fewer function evaluations for a target sample quality or faster convergence to high-fidelity samples. (ii) Stable Approximation Error Propagation: In standard flow matching using Runge-Kutta Methods, the sequential nature means approximation errors in $u_\theta$ evaluations may propagate and amplify within a single step as they influence subsequent intermediate calculations. However, our $K$-order flow matching approach solves the ODE without this feedback loop, which might leads to more stable error propagation. (iii) Relaxing the Consistency Constraint: A significant direction for future research involves exploring methods to either remove or relax the consistency constraint highlighted in Theorem 3.3.

## 6 Concluding Remarks

In this work, we introduce High-Order Flow Matching, a generalized theoretical framework for flow-based generative modeling. Specifically, we characterize the relationship between flow $\psi_t$, $K$-order velocity field $f_t$, probability path $\rho_t$ through governing ODE and mass conservation formula (Definition 3.1 and Theorem 3.2). Then we purpose the $K$-order flow matching loss and establish a tractable equivalent conditional $K$-order flow matching loss (Theorem 3.4) via high-order marginalization trick (Theorem 3.3). Further, we prove that High-Order Flow Matching subsumes standard first-order Flow Matching for $K = 1$ (Proposition 3.1) and providing a unified theoretical foundation for understanding emerging high-order flow model approaches such as HOMO [Chen et al., 2025]. Our second primary contribution is the first rigorous statistical analysis of this High-Order Flow Matching framework when implemented with transformers. We establish sharp approximation, estimation, and distribution learning rates (Theorems 4.1 to 4.3), and demonstrate their near-minimax optimality up to logarithmic factors (Theorem 4.4).

**Related Work.** We defer an extended discussion on related work to Appendix A due to page limits.

## Acknowledgments

JH would like to thank Sitan Chen, Mimi Gallagher, Sara Sanchez, T.Y. Ball, Dino Feng and Andrew Chen for valuable conversations; David Ting-Chun Liu, Sophia Pi and Mingcheng Lu for pointing out typos; Mingcheng Lu, Weimin Wu, Yibo Wen and David Liu for collaborations on related topics; and the Red Maple Family for support. The authors would like to thank the anonymous reviewers and program chairs for constructive comments.

Lastly, JH dedicates this work to the memory of Prof. Bi-Ming Tsai, who passed away during its camera-ready preparation (Oct 2025). Her teachings on health, cooking, good taste in food, and the art of thoughtful living continue to inspire JH.

JH is partially supported by the Walter P. Murphy Fellowship. Han Liu is partially supported by NIH R01LM1372201, NSF AST-2421845, Simons Foundation MPS-AI-00010513, AbbVie, Dolby and Chan Zuckerberg Biohub Chicago Spoke Award. This research was supported in part through the computational resources and staff contributions provided for the Quest high performance computing facility at Northwestern University which is jointly supported by the Office of the Provost, the Office for Research, and Northwestern University Information Technology. The content is solely the responsibility of the authors and does not necessarily represent the official views of the funding agencies.

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

# Appendix

## Impact Statement

This theoretical work advances the fundamental understanding of flow matching generative models and presents no foreseeable negative social impacts.

## A    Related Work

In the following, we discuss the recent success of the techniques used in our work. We begin with the universal approximation theory of transformers. Then, we discuss the recent theoretical progress in flow matching framework, including approximation, estimation and minimax optimality theories.

**Universality of Transformers.**    The universality of transformers refers to their capability to approximate arbitrary sequence-to-sequence functions with any desired precision. It is a key to connect flow matching Transformer architectures with meaningful statistical estimations in this work. Yun et al. [2019] first prove this capability with deep stacks of self-attention and feed-forward layers through the idea of *contextual mapping* by assuming a minimal separation among all hidden representations. Subsequent work by [Alberti et al., 2023] extend the guarantee to variants that employ sparse attention mechanisms. Building upon these works, Hu et al. [2025b], Kajitsuka and Sato [2023] show that a transformer block with a single self-attention layer is sufficient to achieve universal approximation with refined contextual mapping techniques. Beyond contextual mapping, Jiang and Li [2024] derive explicit Jackson-type approximation rates for single-layer, one-head Transformers, with errors governed by the low-rank structure of the target attention kernel and by head/FFN budgets. Hu et al. [2025a], Liu et al. [2025] show sequence-to-sequence universality of minimal Transformers (attention-only) via interpolation and max-affine partition constructions. In particular, Liu et al. [2025] prove that a single-head self- or cross-attention module achieves universal approximation under $L_\infty$ for continuous targets and extends to $L_p$.

**Remark A.1** (Approximation with High- vs Low-Temperature Softmax Transformers).    We remark that, the current statistical rates rely on high-temperature region of softmax attention (i.e., $\mathrm{Softmax}_\lambda(\cdot)$ with small $\lambda$, as in Lemma H.2, Theorem H.1 and (H.25)) to cancel the double-exponential factor reported in [Hu et al., 2024, Remark 3.4]. Is it possible to circumvent with above mentioned different approximation results. For example, Hu et al. [2025a] establish universal approximation in the low-temperature regime (large $\lambda$). It suggests that analogous statistical guarantees may also hold without resorting to high-temperature softmax scaling. We leave this for future explorations.

**Flow Matching and High-Order Flow Matching.**    Flow Matching generative modeling [Lipman et al., 2024, Gat et al., 2024, Chen and Lipman, 2023, Lipman et al., 2022, Liu et al., 2022] has advanced the state-of-the-art in various fields and applications, including images [Esser et al., 2024] , speeches [Le et al., 2023], audios [Polyak et al., 2024] and biomedical data [Huguet et al., 2024]. These standard flow matching frameworks learn first-order trajectory dynamics (velocity field) to smoothly transport a simple source distribution to the target data distribution. However, there is a growing interest for the role of high-order dynamics in generative modeling with improved accuracy and efficiency, which has been applied in various empirical explorations. For instance, Cao et al. [2025] integrate special relativistic mechanics to enhance the stability of generative modeling by supervising on second-order dynamics (acceleration) to ensure sample velocities remain bounded within a safe limit. Similarly, Liang et al. [2025] also augment flow auto-regressive transformers with second-order supervision by capturing complex dependencies through high-order dynamics.

**Statistical Rates and Minimax Optimality of Flow Models.**    Benton et al. [2023], Albergo and Vanden-Eijnden [2022] measure the convergence of flow models by the $L_2$-risk of the velocity field but omit explicit convergence rates. Jiao et al. [2024] work in the latent space of an autoencoder and derive explicit convergence rates for flow models; however, they do not consider the smoothness of the target density class. Su et al. [2025] establish statistical rates for discrete flow matching by deriving a model-agnostic, intrinsic error bound. Fukumizu et al. [2024] demonstrate that flow matching achieves nearly minimax-optimal distribution estimation rates in Besov density function spaces under the 2-Wasserstein distance using $\mathrm{ReLU}$ network architectures. Kunkel and Trabs [2025] establish similar results under the 1-Wasserstein distance by employing the kernel density estimators. In this work, we provide the first theoretical evidence of the minimax optimality of any order flow matching using transformer architectures, and our results recover the first order case as a special instance. Notably, we show that flow matching transformers (FMTs) achieve nearly minimax optimal rates

in Hölder density function spaces under the 2-Wasserstein distance without imposing the Lipschitz continuity assumption on the velocity field. Please see Appendix I for a detailed analysis.

# B  Supplementary Background: Transformer Block

In this section, we introduce the transformer architecture and its Lipschitzness property. Appendix B.1 provides a formal definition of the transformer network class that we use throughout the paper. Further, Appendix B.2 shows that a transformer block is Lipschitz continuous over a compact domain.

## B.1  Transformers

Our notation follows [Hu et al., 2025c, 2024]. To begin with, given a matrix $Z \in \mathbb{R}^{d \times L}$, we denote the $i$-th column and the $j$-th row by $Z_{:i}$ and $Z_{j:}$ respectively.

**Transformer Block.** Let $\mathcal{F}^{(\mathrm{SA})} : \mathbb{R}^{d \times L} \to \mathbb{R}^{d \times L}$ denote the self-attention layer. We use $h$ and $s$ to denote the number of heads and hidden dimension in the self-attention layer, and then we have

$$\mathcal{F}^{(\mathrm{SA})}(Z) := Z + \sum_{i=1}^{h} W_O^i \cdot (W_V^i Z) \operatorname{Softmax}((W_K^i Z)^\top (W_Q^i Z)), \tag{B.1}$$

where $\operatorname{Softmax}(\cdot)$ is the column-wise softmax function, $W_V^i, W_K^i, W_Q^i \in \mathbb{R}^{s \times d}$, and $W_O^i \in \mathbb{R}^{d \times s}$ are the weight matrices. Let $r$ be the MLP dimension. Then, we define the feed-forward layer:

$$\mathcal{F}^{(\mathrm{FF})}(Z) := Z + W_2 \operatorname{ReLU}(W_1 Z + b_1) + b_2, \tag{B.2}$$

where $W_1 \in \mathbb{R}^{r \times d}$ and $W_2 \in \mathbb{R}^{d \times r}$ are weight matrices, and $b_1 \in \mathbb{R}^r$, and $b_2 \in \mathbb{R}^d$ are bias.

**Definition B.1** (Transformer Block). We define a transformer block of $h$-head, $s$-hidden dimension, $r$-MLP dimension, and with positional encoding $E \in \mathbb{R}^{d \times L}$ as

$$\mathcal{F}^{h,s,r}(Z) := \mathcal{F}^{(\mathrm{FF})}\left(\mathcal{F}^{(\mathrm{SA})}(Z + E)\right) : \mathbb{R}^{d \times L} \mapsto \mathbb{R}^{d \times L}.$$

Now, we define the transformer networks as compositions of transformer blocks.

**Definition B.2** (Transformer Network Function Class). Let $\mathcal{T}^{h,s,r}$ denote the transformer network function class where each function $f \in \mathcal{T}^{h,s,r}$ is a composition of transformer blocks $\mathcal{F}^{h,s,r}$, i.e.,

$$\mathcal{T}^{h,s,r} := \{f_{\mathcal{T}} : \mathbb{R}^{d \times L} \mapsto \mathbb{R}^{d \times L} \mid f_{\mathcal{T}} = \mathcal{F}^{h,s,r} \circ \cdots \circ \mathcal{F}^{h,s,r}\}.$$

**Flow Matching Transformer.** Following architecture of diffusion transformers (DiTs) in [Hu et al., 2025c, 2024, Peebles and Xie, 2023], we adopt the reshape layer $R$ that converts a vector input $x \in \mathbb{R}^{d_x}$ into the sequential matrix input format $Z \in \mathbb{R}^{d \times L}$ for transformer with $d_x = d \cdot L$.

**Definition B.3** (Reshape Layer). The reshape layer $R(\cdot) : \mathbb{R}^{d_x} \to \mathbb{R}^{d \times L}$ transforms $d_x$-dimensional input into a $d \times L$ matrix. For any $d_x = i \times i$ image input, $R(\cdot)$ converts it into a sequence representation with feature dimension $d := p^2$ $(p \geq 2)$ and sequence length $L := (i/p)^2$. Further, We define the reverse reshape layer $R^{-1}(\cdot) : \mathbb{R}^{d \times L} \to \mathbb{R}^{d_x}$ as the inverse of $R(\cdot)$.

Finally, we define the following transformer network function class with the reshape layer. To simplify, we define $W_{KQ} := (W_K)^\top W_Q$ and $W_{OV} := W_O W_V$.

**Definition B.4** (Transformer Network Function Class with Reshape Layer $\mathcal{T}_R^{h,s,r}$). The transformer network class with reshape layer $\mathcal{T}_R^{h,s,r}(C_{\mathcal{T}}, C_{KQ}^{2,\infty}, C_{KQ}, C_{OV}^{2,\infty}, C_{OV}, C_E, C_F^{2,\infty}, C_F, L_{\mathcal{T}})$ satisfies:

- $\mathcal{T}_R^{h,s,r} := \{R^{-1} \circ f_{\mathcal{T}} \circ R : \mathbb{R}^{d_x} \to \mathbb{R}^{d_x} \mid f_{\mathcal{T}} \in \mathcal{T}^{h,s,r}\}$;
- Transformer network output bound: $\sup_Z \|f_{\mathcal{T}}(Z)\|_2 \leq C_{\mathcal{T}}$;

- Parameter bound in $\mathcal{F}^{\text{(FF)}}$: $\max\{\|W_1\|_{2,\infty}, \|W_2\|_{2,\infty}\} \leq C_F^{2,\infty}$, $\max\{\|W_1\|_2, \|W_2\|_2\} \leq C_F^2$;
- Parameter bound in $\mathcal{F}^{\text{(SA)}}$: $\|W_{KQ}\|_2 \leq C_{KQ}$, $\|W_{OV}\|_2 \leq C_{OV}$, $\|W_{KQ}\|_{2,\infty} \leq C_{KQ}^{2,\infty}$, $\|W_{OV}\|_{2,\infty} \leq C_{OV}^{2,\infty}$, $\|E^\top\|_{2,\infty} \leq C_E$, where $2,\infty$-norm follows $\|\cdot\|_{2,\infty} := \max_{j \in [L]} \|Z_{:j}\|_2$;
- Lipschitz of $f_{\mathcal{T}} \in \mathcal{T}^{h,s,r}$: $\|f_{\mathcal{T}}(Z_1) - f_{\mathcal{T}}(Z_2)\|_F \leq L_{\mathcal{T}}\|Z_1 - Z_2\|_F$, for any $Z_1, Z_2 \in \mathbb{R}^{d \times L}$.

We remark that these norm bounds are critical to quantify the complexity of the network class.

## B.2 Lipschitzness of Transformer Network

In this section, we show the Lipschitzness for our transformer network class Definition B.2. We begin with a helper lemma and the Lipschitzness of softmax function under inverse-temperature scaling:

**Lemma B.1** (Lipschitzness from Bounded Jacobian, Lemma A.6 of [Edelman et al., 2022]). Let $\Delta^{d-1} := \{x \in \mathbb{R}^d \mid x \geq 0, \|x\|_1 = 1\}$ and $c_f > 0$ be some constant. Suppose that $f : \mathbb{R}^d \to \Delta^{d-1}$ is a differentiable function satisfying $\|Jf(x)\|_{1,1} \leq c_f$ for all $x$. Then, for any $x_1, x_2 \in \mathbb{R}^d$, it holds

$$\|f(x_1) - f(x_2)\|_1 \leq c_f \|x_1 - x_2\|_\infty.$$

We then give the Lipschitz property of the softmax function:

**Lemma B.2** (Lipschitzness of Softmax with Inverse Temperature, Modified from Corollary A.7 of [Edelman et al., 2022]). Let $y, z \in \mathbb{R}^d$. Denote the softmax function with inverse temperature $\lambda > 0$ by $\text{Softmax}_\lambda(x)[i] := \exp(\lambda x[i])/\sum_k \exp(\lambda x[k])$ for $x \in \mathbb{R}^d, k \in [d]$. Then, it holds

$$\|\text{Softmax}_\lambda(y) - \text{Softmax}_\lambda(z)\|_1 \leq 2\lambda\|y - z\|_\infty.$$

*Proof.* Our proof follows [Edelman et al., 2022] and incorporates inverse temperature scaling $\beta > 0$. Define $\text{Softmax}_\lambda(x)[i] := p_i$ for $i \in [d]$. Jacobian has $J_{i,i} = \lambda p_i(1 - p_i)$ and $J_{i,j} = -\lambda p_i p_j$ for $i \neq j$. Then, this yields $J = \lambda(\text{diga}(p) - pp^\top)$ and $\|J\|_{1,1} \leq 2\lambda$. Then we apply Lemma B.1.

This completes the proof. $\square$

Therefore, Lemma B.2 implies the Lipschitzness of a self-attention layer given bounded weight matrices $W_{KQ}$ and $W_{OV}$. Furthermore, with bounded $W_1, W_2$ in the feed-forward layer and $|\text{ReLU}(x) - \text{ReLU}(y)| \leq |x - y|$ for any $x, y \in \mathbb{R}$, we have Lipschitzness of a transformer block.

# C Proofs in Section 3

In this section, we formalize the high-order flow matching. Appendix C.1 establishes the flow–velocity equivalence through an ordinary differential equation argument (Theorem 3.1). Appendix C.2 ensures the mass conservation in high-order flows (Theorem 3.2). Appendix C.3 derives the marginalization property (Theorem 3.3). Appendix C.4 shows the gradient equivalence between the flow matching and conditional flow matching objectives (Theorem 3.4). Finally, Appendix C.5 unifies the framework by proving that $K$-order flow matching collapses to the standard first-order case (Proposition 3.1).

## C.1 Proof of Theorem 3.1

In this section, we present the main proof of Theorem 3.1.

> **Theorem C.1** (Theorem 3.1 Restated: Flow–Velocity Equivalence via ODE). Define the class of structured $k$-order velocity fields as those of the form:
>
> $$f_t(y_t) = \mathrm{col}(u^1(x_t^{(0)}, t), \ldots, u^K(x_t^{(0)}, t)) \in \mathbb{R}^{Kd}, \quad y_t = \mathrm{col}(x_t^{(0)}, \ldots, x_t^{(K-1)}) \in \mathbb{R}^{Kd},$$
>
> where $u^k : \mathbb{R}^{Kd} \times [0,1] \to$ is locally lipschitz in $y_t$ and continues in $t$ for any $k \in [K]$. Suppose the velocity fields $u^1(x_t^{(0)}, t), \ldots, u^K(x_t^{(0)}, t)$ satisfy total derivative constraints (3.3). Then, for any initial condition $y_0 \in \mathbb{R}^{Kd}$, the ODE $\frac{\mathrm{d}}{\mathrm{d}t} y_t = f_t(y_t)$ exists a unique local solution $y_t$, which defines a $K$-times differentiable flow $\psi_t(x) := x_t^{(0)}$ and satisfy $\frac{\mathrm{d}^k}{\mathrm{d}t^k} \psi_t(x) = x_t^{(k)}$ for all $k \in [K]$. Conversely, any $K$-times differentiable flow $\psi_t : \mathbb{R}^d \to \mathbb{R}^d$ defines a velocity field $f_t$ via (3.1).

*Proof.* We prove both directions:

**From velocity field $f_t$ to flow $\psi_t$:** Let $y_0 = (x_0^{(0)}, \ldots, x_0^{(K-1)})^\top \in \mathbb{R}^{Kd}$ be any initial condition. Then, the system (3.1)

$$\frac{\mathrm{d}}{\mathrm{d}t} y_t = f_t(y_t), \quad \text{with initial condition} \quad y_0,$$

is a standard autonomous first-order ODE on $\mathbb{R}^{Kd}$ with a Lipschitz right-hand side. By the Picard–Lindelöf theorem, there exists a unique local solution $y_t$. Let us define the flow $\psi_t(x) := x_t^{(0)}$ and since $y_t$ is differentiable, $\psi_t$ is differentiable. By repeatedly applying the total derivative constraint (3.3), we can establish that $\frac{\mathrm{d}^k}{\mathrm{d}t^k} \psi_t(x) = x_t^{(k)}$ for all $k \in [K]$. Specifically, for any $k \in [K]$, we have:

$$
\begin{aligned}
x_t^{(k)} &= u^k(x_t^{(0)}, t) & \left(\text{By definition of the ODE}\right) \\
&= \frac{\mathrm{d}}{\mathrm{d}t} u^{k-1}(x_t^{(0)}, t) & \left(\text{By (3.3)}\right) \\
&= \frac{\mathrm{d}}{\mathrm{d}t} x_t^{(k-1)} & \left(\text{By definition of the ODE}\right) \\
&= \frac{\mathrm{d}^k}{\mathrm{d}t^k} \psi_t(x). & \left(\text{By induction}\right)
\end{aligned}
$$

This confirms that the $k$-th order velocity field corresponds exactly to the $k$-th time derivative of the flow $\psi_t$.

**From flow $\psi_t$ to velocity field $f_t$:** Suppose there is a $K$-times differentiable flow $\psi_t$. Define

$$y_t = [\psi_t(x), \frac{\mathrm{d}}{\mathrm{d}t} \psi_t(x), \ldots, \frac{\mathrm{d}^{K-1}}{\mathrm{d}t^{K-1}} \psi_t(x)]^\top,$$

$$f_t(y_t) = \mathrm{col}(\frac{\mathrm{d}}{\mathrm{d}t} \psi_t(x), \ldots, \frac{\mathrm{d}^K}{\mathrm{d}t^K} \psi_t(x)).$$

Then, by direct differentiation:

$$\frac{\mathrm{d}}{\mathrm{d}t} y_t = f_t(y_t).$$

This completes the proof of the bidirectional equivalence. $\qquad\square$

## C.2 Proof of Theorem 3.2

In this section, we provide the proof of Theorem 3.2.

**Theorem C.2** (Theorem 3.2 Restated: Mass Conservation of High-Order Flow). Let $y_t = (x_t^{(0)}, \ldots, x_t^{(K-1)})^\top \in \mathbb{R}^{Kd}$. Let velocity field $f_t(y_t) = (u^1(x_t^{(0)}, t), \ldots, u^K(x_t^{(0)}, t))^\top \in \mathbb{R}^{Kd}$, where $u^k(x_t^{(0)}, t)$ is locally Lipschitz and integrable for all $k \in [K]$. Let $\rho_t : \mathbb{R}^{Kd} \to \mathbb{R}$ be a time-varying probability density over the extended state $Y_t \in \mathbb{R}^{Kd}$ follows Definition 3.1. Then the following statements are equivalent:

1. The pair $(f_t, \rho_t)$ satisfies the Liouville's equation on the extended space:

$$\frac{\partial}{\partial t} \rho_t(y) + \nabla_y \cdot (\rho_t(y) f_t(y)) = 0, \quad \text{for all } t \in [0, 1).$$

2. Following Definition 3.1, the probability law of $Y_t$ evolves under the flow:

$$\frac{\mathrm{d}}{\mathrm{d}t} Y_t = f_t(Y_t), \quad \text{with} \quad Y_0 \sim \rho_0, \quad Y_t \sim \rho_t. \tag{C.1}$$

For some arbitrary probability path $\rho_t$, we define $f_t$ *generates* $\rho_t$ if (C.1) holds.

*Proof.* We prove both directions:

**From ODE** (C.1) **to Liouville's Equation:** Let $\phi : \mathbb{R}^{Kd} \to \mathbb{R}$ be any smooth function with compact support (i.e., a test function). We first compute the time derivative of following quantity

$$\mathbb{E}[\phi(Y_t)] = \int \phi(y) \rho_t(y) \mathrm{d}y. \tag{C.2}$$

Since the $Y_t$ satisfy the ODE (C.1), the derivative of the expectation becomes:

$$
\begin{aligned}
\frac{\mathrm{d}}{\mathrm{d}t} \mathbb{E}[\phi(Y_t)] &= \mathbb{E}[\frac{\mathrm{d}}{\mathrm{d}t}\phi(Y_t)] && \left(\text{By swiching the expectation and derivation}\right) \\
&= \mathbb{E}[\nabla_y \phi(Y_t) \cdot \frac{\mathrm{d}}{\mathrm{d}t} Y_t] && \left(\text{By the chain rule}\right) \\
&= \mathbb{E}[\nabla_y \phi(Y_t) \cdot f_t(Y_t)] && \left(\text{By the ODE (C.1)}\right) \\
&= \int \nabla_y \phi(y) \cdot f_t(y) \rho_t(y) \mathrm{d}y \\
&= -\int \phi(y) \boldsymbol{\nabla} \cdot (f_t(y) \rho_t(y)) \mathrm{d}y. && \left(\text{By the integration by parts}\right)
\end{aligned}
$$

Therefore, for any test function $\phi_t$, it holds

$$\int \frac{\mathrm{d}}{\mathrm{d}t} \phi(y) \rho_t(y) + \phi(y) \boldsymbol{\nabla} \cdot (f_t(y) \rho_t(y)) \mathrm{d}y = 0,$$

which leads to Liouville's equation

$$\frac{d}{dt} \rho_t(y) + \nabla_y \cdot (\rho_t(y) f_t(y)) = 0.$$

**From Liouville's Equation to ODE** (C.1)**:** According to the equivalence between the flow $\psi_t$ and its associated velocity field $f_t$ (Theorem 3.1), the ODE (C.1) admits a unique local solution $\widetilde{y}_t$, which defines a unique flow $\widetilde{\psi}_t$. By the pushforward formula and the definition in Definition 3.1, this flow induces the distribution $\widetilde{Y}_t \sim \widetilde{\rho}_t$. Moreover, $\widetilde{\rho}_t$ satisfies the Liouville equation associated with the velocity field $f_t$.

Since the Liouville equation admits a unique solution in the space of probability densities starting from the same initial distribution $\rho_0$, and both $\rho_t$ and $\widetilde{\rho}_t$ solve the same continuity equation with initial condition $\rho_0$, we conclude that $\rho_t = \widetilde{\rho}_t$. This completes the proof. $\qquad\square$

### C.3 Proof of Theorem 3.3

This section presents the proof of Theorem 3.3.

**Theorem C.3** (Theorem 3.3 Restated: Marginalization)**.** Recall that for some arbitrary probability path $\rho_t$, $f_t$ generates $\rho_t$ if $Y_t \sim \rho_t$ for all $t \in [0,1)$. Let $Z$ be a random variable, if $f_t(x|z)$ is conditionally integrable and generates the conditional probability path $\rho_t(\cdot|z)$, then the marginal velocity $f_t := \int f_t(y|z)p_t(z|y)\mathrm{d}z$ generates the marginal probability path $p_t$.

*Proof.* Applying the mass conservation follows Theorem 3.2, we only need to verify that the $f_t$ and $\rho_t$ satisfy high-order continuity equation, i.e. Liouville's Equation:

$$
\begin{aligned}
\frac{\mathrm{d}}{\mathrm{d}t}\rho_t(y) &= \int \frac{\mathrm{d}}{\mathrm{d}t}\rho_{t|Z}(y|z)p_Z(z)\mathrm{d}z && \left(\text{By the law of total probability}\right) \\
&= \int -\boldsymbol{\nabla} \cdot [f_t(y|z)\rho_t(y|z)]p_Z(z)\mathrm{d}z && \left(\text{By Liouville's equation}\right) \\
&= -\boldsymbol{\nabla} \cdot \int f_t(y|z)\rho_t(y|z)p_Z(z)\mathrm{d}z && \left(\text{By switching differentiation and integration}\right) \\
&= -\boldsymbol{\nabla} \cdot \int [f_t(y|z)\rho_t(y|z)p_Z(z)/\rho_t(y)] \cdot \rho_t(y)\mathrm{d}z && \\
&= -\boldsymbol{\nabla} \cdot [f_t(y)\rho_t(y)]. && \left(\text{By the definition of } f_t(y) \text{ and the Bayes' rule}\right)
\end{aligned}
$$

This completes the proof. $\qquad\square$

### C.4 Proof of Theorem 3.4

In this section, we prove Theorem 3.4.

**Theorem C.4** (Theorem 3.4 Restated: Gradient Equivalence of Losses)**.** Let the Flow Matching loss $\mathcal{L}_{\text{FM}}^K$ be defined as in Definition 3.2, and the Conditional Flow Matching loss $\mathcal{L}_{\text{CFM}}^K$ be defined as in (3.5). Then, when $D(\cdot,\cdot)$ is a Bregman divergence, the gradients of the two losses coincide:

$$
\nabla\mathcal{L}_{\text{FM}}^K(\theta) = \nabla\mathcal{L}_{\text{CFM}}^K(\theta).
$$

*Proof.* Similar to the Theorem 4 of [Lipman et al., 2024], the result follows from the Marginalization Trick (Theorem 3.3) and the expectation-swapping property of Bregman divergences (3.6). A direct computation then shows that:

$$
\begin{aligned}
\nabla_\theta \mathcal{L}_{\text{FM}}^K(\theta) &= \nabla_\theta \mathop{\mathbb{E}}_{t,Y_t\sim\rho_t} D(f_t(Y_t), f_t^\theta(Y_t)) && \left(\text{By the definition of Flow Matching Loss}\right) \\
&= \mathop{\mathbb{E}}_{t,Y_t\sim\rho_t} \nabla_\theta D(f_t(Y_t), f_t^\theta(Y_t)) && \\
&&& \left(\text{By swaping the expectation and the gradient computation}\right) \\
&= \mathop{\mathbb{E}}_{t,Y_t\sim\rho_t} \nabla_v D(f_t(Y_t), f_t^\theta(Y_t))\nabla_\theta f_t^\theta(Y_t) && \left(\text{By the chain rule}\right) \\
&= \mathop{\mathbb{E}}_{t,Y_t\sim\rho_t} \nabla_v D\big(\mathop{\mathbb{E}}_{Z\sim p_{z|t}(\cdot|y)}[f_t(Y_t|Z)], f_t^\theta(Y_t)\big)\nabla_\theta f_t^\theta(Y_t) && \\
&&& \left(\text{By the marginalization trick follows Theorem 3.3}\right)
\end{aligned}
$$

$$
\begin{aligned}
&= \underset{t,Y_t \sim \rho_t}{\mathbb{E}} \underset{Z \sim p_{z|t}(\cdot|y)}{\mathbb{E}} [\nabla_v D([f_t(Y_t|Z)], f_t^\theta(Y_t))\nabla_\theta f_t^\theta(Y_t)] \\
&\qquad\qquad\qquad\qquad\qquad\qquad \left(\text{By the property of Bregman divergence follows (3.6)}\right) \\
&= \underset{t,Y_t \sim \rho_t}{\mathbb{E}} \underset{Z \sim p_{z|t}(\cdot|y)}{\mathbb{E}} [\nabla_\theta D([f_t(Y_t|Z)], f_t^\theta(Y_t))] \qquad\qquad \left(\text{By the chain rule}\right) \\
&= \nabla_\theta \underset{t,Z,Y_t \sim \rho_{t|Z}(\cdot|Z)}{\mathbb{E}} [D(f_t(Y_t|Z), f_t^\theta(Y_t))] \qquad\qquad\quad \left(\text{By the Bayes' rule}\right) \\
&= \nabla_\theta \mathcal{L}_{\text{CFM}}^K(\theta).
\end{aligned}
$$

This completes the proof. $\qquad\qquad\qquad\qquad\qquad\qquad\qquad\qquad\qquad\qquad\qquad\quad$ $\square$

### C.5 Proof of Proposition 3.1

This section gives the main proof of Proposition 3.1.

> **Proposition C.1** (Proposition 3.1 Restated: Reduction to Standard First-Order Flow Matching).
> When $K = 1$, the entire $K$-order flow matching framework, including the governing ODE, the probability path definition via the continuity equation, and the $K$-order flow matching objective, becomes precisely equivalent to the standard first-order Flow Matching framework as detailed in [Lipman et al., 2022, 2024].

*Proof.* The equivalence follows by setting $K = 1$ in the definitions of our $K$-order framework.

1. **State Variable and ODE:** From Definition 3.1, when $K = 1$, $Y_t = X_t^{(0)} = X_t$. The ODE system $\frac{\mathrm{d}}{\mathrm{d}t}Y_t = f_t(Y_t)$ simplifies to $\frac{\mathrm{d}}{\mathrm{d}t}X_t = u^1(X_t)$, which is the governing ODE for standard flow models ([Lipman et al., 2022, 2024]). The $K$-order velocity field $f_t$ becomes $u^1$.

2. **Probability Path and Continuity Equation:** The $K$-order mass conservation formula (Theorem 3.2) for $K = 1$ reduces to the standard Mass Conservation Formula (Theorem 2 in [Lipman et al., 2024]).

3. **Loss Objective:** The $K$-order flow matching loss (Definition 3.2), which targets matching $f_t^\theta$ to $f_t$ simplifies to matching only the $u^1$ component: $\mathbb{E}_{t,X_t \sim p_t}[D(u_t^1(X_t), u_t^{1,\theta}(X_t))]$. This is the standard Flow Matching objective (Eq. (5) in [Lipman et al., 2022]). The conditional formulation via Theorem 3.3 similarly simplifies to the conditional Flow Matching loss used for standard FM.

Thus, all core components of the $K$-order framework align with standard Flow Matching. $\qquad$ $\square$

# D    Proof of Theorem 4.1

In this section, we prove Theorem 4.1 following steps similar to the velocity approximation in Appendix J: (i) applying the universal approximation of transformers (ii) leveraging the sub-Gaussian property of the target distribution to bound the approximation error of the $K$ order velocity field.

**Organizations.**    Appendix D.1 introduces helper lemmas. Appendix D.2 presents the main proof.

## D.1    Auxiliary Lemmas

In this section, we introduce four auxiliary lemmas. In Lemma D.1, we give the lower-bound and upper-bounds on $p_t(x)$. In Lemma D.2, we state the classical Gaussian tail bounds. In Lemma D.3, we approximate the $k$-th order velocity field over a bounded domain. To control the error in unbounded regions, we exploit the sub-gaussian assumption of the target distribution $q(x_1)$ in Lemma D.4.

We begin with the bounds on $p_t(x)$.

**Lemma D.1** (Bounds on the Density Function, Lemma A.9 of [Fu et al., 2024]).    Recall that $p_t(x) = \int_{\mathbb{R}^{d_x}} p_t(x|x_1)q(x_1)\mathrm{d}x_1$ and $p_t(x|x_1) = \frac{1}{\sigma_t^{d_x}(2\pi)^{d_x/2}} \exp\big(-\|x - \mu_t x_1\|_2^2/2\sigma_t^2\big)$. Assume Assumption 4.1. Then, there exist a positive constant $C_4$ such that

$$\frac{C_4}{\sigma_t^{d_x}} \cdot \exp\left(-\frac{\|x\|_2^2 + 1}{\sigma_t^2}\right) \le p_t(x) \le \frac{C_1}{(\mu_t^2 + C_2\sigma_t^2)^{d_x/2}} \cdot \exp\left(-\frac{C_2\|x\|_2^2}{2(\mu_t^2 + C_2\sigma_t^2)}\right).$$

Then, we apply standard results for Gaussian tail bounds. We remark that the main purpose of stating Lemma D.2 is to streamline the main proof of Theorem 4.1 in Appendix D.2.

**Lemma D.2** (Gaussian Tail Bounds).    Consider a random vector $X := (X_1, \ldots, X_{d_x})^\top \sim N(0, \sigma_t^2 I)$. Let $\omega_{d_x} := 2\pi^{\frac{d_x}{2}}/\Gamma(\frac{d_x}{2})$. Then, the following two inequalities hold:

$$\int_{\|X\|>D} \exp\left(-\frac{\|X\|_2^2}{2\sigma_t^2}\right)\mathrm{d}X \le \omega_{d_x}\sigma_t^2 D^{d_x-2}\exp\left(-\frac{D^2}{2\sigma_t^2}\right),$$

$$\int_{\|X\|>D} \|X\|_2^2 \exp\left(-\frac{\|X\|_2^2}{2\sigma_t^2}\right)\mathrm{d}X \le \omega_{d_x} \cdot (\sigma_t^2 D^{d_x} + d_x\sigma_t^4 D^{d_x-2})\exp\left(-\frac{D^2}{2\sigma_t^2}\right).$$

*Proof.* We first express the integral in spherical coordinates for $X$

$$\int_{\|X\|>D} \exp\big(-\|X\|_2^2/2\sigma_t^2\big)\mathrm{d}X = \omega_{d_x} \int_D^\infty r^{d_x-1}\exp\left(-\frac{r^2}{2\sigma_t^2}\right)\mathrm{d}r.$$

Let $J_D := \int_D^\infty r^{d_x-1}\exp\left(-\frac{r^2}{2\sigma_t^2}\right)\mathrm{d}r$. Setting $u := r^{d_x-2}$ and $\mathrm{d}v := r\exp\left(-\frac{r^2}{2\sigma_t^2}\right)\mathrm{d}r$, we have

$$\mathrm{d}u = (d_x - 2)r^{d_x-3}\mathrm{d}r, \quad \text{and} \quad v = -\sigma_t^2\exp\left(-\frac{r^2}{2\sigma_t^2}\right).$$

Then,

$$\begin{aligned} J(D) &= \left[-r^{d_x-2}\sigma_t^2\exp\left(-\frac{r^2}{2\sigma_t^2}\right)\right]_{r=D}^\infty + (d_x-2)\sigma_t^2\int_D^\infty r^{d_x-3}\exp\left(-\frac{r^2}{2\sigma_t^2}\right)\mathrm{d}r \quad \text{(D.1)}\\ &= \sigma_t^2 D^{d_x-2}\exp\left(-\frac{D^2}{2\sigma_t^2}\right) + (d_x-2)\sigma_t^2\int_D^\infty r^{d_x-3}\exp\left(-\frac{r^2}{2\sigma_t^2}\right)\mathrm{d}r \\ &\qquad\qquad\qquad\qquad\qquad\qquad\qquad\qquad\qquad \left(\text{By integration by parts}\right)\\ &\le \sigma_t^2 D^{d_x-2}\exp\left(-\frac{D^2}{2\sigma_t^2}\right). \qquad\qquad\qquad\qquad \left(\text{By dropping the second term}\right) \end{aligned}$$

We obtain the final bound

$$\int_{\|X\|>D} \exp\left(-\frac{\|X\|_2^2}{2\sigma_t^2}\right) \mathrm{d}X \leq \omega_{d_x} \sigma_t^2 D^{d_x-2} \exp\left(-\frac{D^2}{2\sigma_t^2}\right).$$

This completes the proof of the first inequality. For the second inequality, we have

$$\int_{\|X\|>D} \|X\|_2^2 \exp\left(-\frac{\|X\|_2^2}{2\sigma_t^2}\right) \mathrm{d}X$$

$$= \omega_{d_x} \int_D^\infty r^2 r^{d_x-1} \exp\left(-\frac{r^2}{2\sigma_t^2}\right) \mathrm{d}r$$

$$= \omega_{d_x} \int_D^\infty r^{d_x+1} \exp\left(-\frac{r^2}{2\sigma_t^2}\right) \mathrm{d}r.$$

Let $K(D) := \int_D^\infty r^{d_x+1} \exp\left(-\frac{r^2}{2\sigma_t^2}\right) \mathrm{d}r$, $u := r^d$ and $\mathrm{d}v := r \exp\left(-\frac{r^2}{2\sigma_t^2}\right) \mathrm{d}r$. Then,

$$\mathrm{d}u = d_x r^{d_x-1} \mathrm{d}r, \quad \text{and} \quad v = -\sigma_t^2 \exp\left(-\frac{r^2}{2\sigma_t^2}\right).$$

Therefore, the integration by parts gives

$$K(D)$$
$$= \left[-r^{d_x}\sigma_t^2 \exp\left(-\frac{r^2}{2\sigma_t^2}\right)\right]_{r=D}^\infty + \int_D^\infty \sigma_t^2 \exp\left(-\frac{r^2}{2\sigma_t^2}\right) d_x r^{d_x-1} \mathrm{d}r$$
$$= \sigma_t^2 D^{d_x} \exp\left(-\frac{D^2}{2\sigma_t^2}\right) + d_x\sigma_t^2 \int_D^\infty r^{d_x-1} \exp\left(-\frac{r^2}{2\sigma_t^2}\right) \mathrm{d}r.$$

Recalling (D.1)

$$J_D := \int_D^\infty r^{d_x-1} \exp\left(-\frac{r^2}{2\sigma_t^2}\right) \mathrm{d}r, \quad \text{and} \quad J_D \leq \sigma_t^2 D^{d_x-2} \exp\left(-\frac{D^2}{2\sigma_t^2}\right),$$

we have

$$K(D)$$
$$= \sigma_t^2 D^{d_x} \exp\left(-\frac{D^2}{2\sigma_t^2}\right) + d_x\sigma_t^2 J_D$$
$$\leq \sigma_t^2 D^{d_x} \exp\left(-\frac{D^2}{2\sigma_t^2}\right) + d_x\sigma_t^2 \cdot \left(\sigma_t^2 D^{d_x-2} \exp\left(-\frac{D^2}{2\sigma_t^2}\right)\right) \qquad \text{(By the bound on } J_D)$$
$$= \left(\sigma_t^2 D^{d_x} + d_x\sigma_t^4 D^{d_x-2}\right) \exp\left(-\frac{D^2}{2\sigma_t^2}\right).$$

Then we obtain the final bound

$$\int_{\|X\|>D} \|X\|_2^2 \exp\left(-\frac{\|X\|_2^2}{2\sigma_t^2}\right) \mathrm{d}X \leq \omega_{d_x} \cdot \left(\sigma_t^2 D^{d_x} + d_x\sigma_t^4 D^{d_x-2}\right) \exp\left(-\frac{D^2}{2\sigma_t^2}\right).$$

This completes the proof of the second inequality. $\qquad\square$

Applying the universal approximation of transformers (Theorem H.2), we first approximate the $k$-th order velocity field $u^k$ over a bounded domain with transformers $u^{k,\theta}$.

**Lemma D.3** (Approximate $k$-th Order Flow with Transformers). Assume Assumption 4.1. Let $D$ be an absolute positive constant. Then, for any $x \in [-I, I]^{d_x}$, $t \in [0, 1]$ and $\epsilon \in (0, 1)$, there exist a transformer $u^{k,\theta}(x, t) \in \mathcal{T}_R^{h,s,r}$ such that

$$\int_0^1 \int_{[-I,I]^{d_x}} p_t(x) \cdot \|u^{k,\theta}(x,t) - u^k(x,t)\|_2^2 \mathrm{d}x \mathrm{d}t \le \epsilon^2,$$

for all $k \in [K]$. Furthermore, the parameter bounds in the transformer network class satisfy

$$C_{KQ}, C_{KQ}^{2,\infty} = O(\lambda^{-1} I^{4d+2} \epsilon^{-4d-2}); C_{OV}, C_{OV}^{2,\infty} = O(\epsilon);$$
$$C_F, C_F^{2,\infty} = O(I\epsilon^{-1} L_{k-1}); C_E = O(I); C_{\mathcal{T}} = O(L_{k-1})$$

where $\lambda^{-1} = O(I/\epsilon)^{4d+3}$ is the inverse-temperature scaling in the softmax function and and $O(\cdot)$ hides all polynomial factors depending on $d_x, d, L, \beta, C_1, C_2$.

*Proof.* By specifying the target function as $f = u^k$ and the transformer-based estimator as $g = u^{k,\theta}$ in Theorem H.2, and applying the bound $p_t(x) \le 1$, the proof follows Theorem H.2 since the reshape layer (Definition B.3) does not harm the uniform continuity. Further, by the Lipschitzness of the $k$-th order flow, we have $\|u^k(x,t)\|_2 \le L_{k-1}$. Then, the parameter bounds in transformer network follow Lemma H.5, where we set the model output bound $C_{\mathcal{T}} = O(L_{k-1})$. This completes the proof. $\square$

To control the approximation error over an unbounded domain, we introduce tail bounds for the probability flow $p_t(x)$ and the weighted squared norms of the $u^k$, given by $\|u^k(x,t)\|_2^2 \cdot p_t(x)$.

**Lemma D.4** (Truncation of $x$, Modified from Lemma A.1 of [Fu et al., 2024]). Assume Assumption 4.1. Suppose the $k$-th order velocity field $u^k(x,t)$ is Lipschitz continuous for all $k = 0, \ldots, K-1$. Let $L_k$ denote the Lipschitz constant of $u^k$, and then the velocity fields are uniformly bounded as $|u^k(x,t)| \le L_{k-1}$ for any $k \in [K]$. Then, for any $R_1, t > 0$ and $k \in [K]$, the following hold

$$\int_{\|x\|_\infty > R_1} p_t(x)\mathrm{d}x \lesssim R_1^{d_x-2} \exp\left(-\frac{C_2 R_1^2}{2(\mu_t^2 + C_2\sigma_t^2)}\right),$$
$$\int_{\|x\|_\infty > R_1} \|u^k(x,t)\|_2^2 \cdot p_t(x)\mathrm{d}x \lesssim L_{k-1}^2 R_1^{d_x-2} \exp\left(-\frac{C_2 R_1^2}{2(\mu_t^2 + C_2\sigma_t^2)}\right).$$

*Proof.* For the first inequality, it follows

$$\int_{\|x\|_\infty > R_1} p_t(x)\mathrm{d}x$$
$$\le \int_{\|x\|_\infty > R_1} \exp\left(-\frac{C_2\|x\|_2^2}{2(\mu_t^2 + C_2\sigma_t^2)}\right)\mathrm{d}x \qquad \text{(By Lemma D.1)}$$
$$\le \int_{\|x\|_2 > R_1} \exp\left(-\frac{C_2\|x\|_2^2}{2(\mu_t^2 + C_2\sigma_t^2)}\right)\mathrm{d}x \qquad \text{(By } \|x\|_2 \ge \|x\|_\infty)$$
$$\lesssim R_1^{d_x-2} \exp\left(-\frac{C_2 R_1^2}{2(\mu_t^2 + C_2\sigma_t^2)}\right). \qquad \text{(By Lemma D.2)}$$

For the second inequality, it follows

$$\int_{\|x\|_\infty \ge R_1} \|u^k(x,t)\|_2^2 \cdot p_t(x)\mathrm{d}x$$
$$\lesssim \int_{\|x\|_\infty \ge R_1} \|u^k(x,t)\|_2^2 \cdot \exp\left(-\frac{C_2\|x\|_2^2}{2(\mu_t^2 + C_2\sigma_t^2)}\right)\mathrm{d}x \qquad \text{(By Lemma D.1)}$$
$$\lesssim \int_{\|x\|_\infty \ge R_1} L_{k-1}^2 \exp\left(\frac{-C_2\|x\|_2^2}{2(\mu_t^2 + C_2\sigma_t^2)}\right)\mathrm{d}x \qquad \text{(By the Lipchitzness of the } k\text{-th order flow)}$$

$$\lesssim L_{k-1}^2 R_1^{d_x - 2} \exp\left(-\frac{C_2 R_1^2}{2(\mu_t^2 + C_2 \sigma_t^2)}\right). \hspace{2cm} \text{(By Lemma D.2)}$$

This completes the proof. □

## D.2  Main Proof of Theorem 4.1

We now present the formal proof of Theorem 4.1.

**Theorem D.1** (Theorem 4.1 Restated: $K$-order Velocity Approximation with Transformers)**.** Assume Assumption 4.1. Suppose the $k$-th order velocity field $u^k(x,t)$ is $L_k$-Lipschitz for all $k \in 0, \ldots, K-1$ in $\ell_2$-distance. Let $\epsilon \in (0,1)$ be the precision parameter satisfying $\epsilon \leq O(N^{-\beta})$ for some $N \in \mathbb{N}$ and smoothness parameter $\beta > 0$. Then, there exists transformers $u^{1,\theta}(x,t), \ldots, u^{K,\theta}(x,t) \in \mathcal{T}_R^{h,s,r}$ such that for any $x \in \mathbb{R}^{d_x}$ and $t \in [0,1]$, it holds:

$$\sum_{k=1}^K \int_{t_0}^T \int_{\mathbb{R}^{d_x}} \|u^{k,\theta}(x,t) - u^k(x,t)\|_2^2 \cdot p_t(x) \mathrm{d}x \mathrm{d}t = O\big(N^{-2\beta} \cdot (\log N)^{\frac{d_x}{2} - 1}\big).$$

Further, for all $k \in [K]$, the parameter bounds in transformer network class satisfy

$$C_{KQ}, C_{KQ}^{2,\infty} = O(\lambda^{-1} N^{2\beta(2d+1)} (\log N)^{2d+1}); \quad C_{OV}, C_{OV}^{2,\infty} = O(N^{-\beta});$$
$$C_F, C_F^{2,\infty} = O(N^\beta \sqrt{\log N} L_{k-1}); \quad C_E = O(I); \quad C_\mathcal{T} = O(L_{k-1}),$$

where $\lambda^{-1} = O(N^\beta \log N)^{4d+3}$ is the inverse-temperature scaling in the softmax function and $O(\cdot)$ hides all polynomial factors depending on $d_x, d, L, \beta, C_1, C_2$.

*Proof of Theorem 4.1.* For $u^{1,\theta}(x,t), \ldots, u^{K,\theta}(x,t) \in \mathcal{T}_R^{h,s,r}$, we set the transformer output bound $C_\mathcal{T} = O(L_{k-1})$ for the $k$-th network and let $R_3$ and $\epsilon_{\text{low}}$ be two positive numbers to be chosen.

First, we decompose the target into three components and bound each of them

$$\sum_{k=1}^K \int_{t_0}^T \int_{\mathbb{R}^{d_x}} \|u^{k,\theta}(x,t) - u^k(x,t)\|_2^2 \cdot p_t(x) \mathrm{d}x \mathrm{d}t$$

$$= \underbrace{\sum_{k=1}^K \int_{t_0}^T \int_{\|x\|_\infty > R_3} \|u^{k,\theta}(x,t) - u^k(x,t)\|_2^2 \cdot p_t(x) \mathrm{d}x \mathrm{d}t}_{(\mathrm{T}_1)}$$

$$+ \underbrace{\sum_{k=1}^K \int_{t_0}^T \int_{\|x\|_\infty \leq R_3} \|u^{k,\theta}(x,t) - u^k(x,t)\|_2^2 \cdot p_t(x) \mathrm{d}x \mathrm{d}t}_{(\mathrm{T}_2)}.$$

- **Bound on** $(\mathrm{T}_1)$**.** It holds

$(\mathrm{T}_1)$
$$= \sum_{k=1}^K \int_{t_0}^T \int_{\|x\|_\infty > R_3} \|u^{k,\theta}(x,t) - u^k(x,t)\|_2^2 \cdot p_t(x) \mathrm{d}x \mathrm{d}t$$

$$\leq 2\sum_{k=1}^K \int_{t_0}^T \int_{\|x\|_\infty > R_3} \|u^{k,\theta}(x,t)\|_2^2 \cdot p_t(x) \mathrm{d}x \mathrm{d}t + 2\sum_{k=1}^K \int_{t_0}^T \int_{\|x\|_\infty > R_3} \|u^k(x,t)\|_2^2 \cdot p_t(x) \mathrm{d}x \mathrm{d}t$$

$$\text{(By expanding } \ell_2\text{-norm)}$$

$$\lesssim \sum_{k=1}^{K} L_{k-1}^2 \int_{t_0}^{T} \int_{\|x\|_\infty > R_3} p_t(x) \mathrm{d}x \mathrm{d}t + \sum_{k=1}^{K} \int_{t_0}^{T} \int_{\|x\|_\infty > R_3} \|u^k(x,t)\|_2^2 \cdot p_t(x) \mathrm{d}x \mathrm{d}t$$

$$\left( \text{By } C_{\mathcal{T}} = O(L_{k-1}) \right)$$

$$\lesssim \sum_{k=1}^{K} L_{k-1}^2 \int_{t_0}^{T} \int_{\|x\|_\infty > R_3} p_t(x) \mathrm{d}x \mathrm{d}t \qquad \left( \text{By the Lipschitzness of the } k\text{-th order flow} \right)$$

$$\lesssim R_3^{d_x - 2} \exp\left( -\frac{C_2 R_3^2}{2(\mu_t^2 + C_2 \sigma_t^2)} \right) \sum_{k=1}^{K} L_{k-1}^2 \int_{t_0}^{T} \mathrm{d}t. \qquad \left( \text{By Lemma D.4} \right)$$

$$\leq R_3^{d_x - 2} \exp\left( -\frac{C_2 R_3^2}{2(\mu_t^2 + C_2 \sigma_t^2)} \right) \sum_{k=1}^{K} L_{k-1}^2. \qquad \left( \text{By } t_0, T \in (0,1) \right)$$

- **Bound on** $(\mathrm{T}_2)$. For any $\epsilon \in (0,1)$, it holds

$$(\mathrm{T}_2) = \sum_{k=1}^{K} \int_{t_0}^{T} \int_{\|x\|_\infty \leq R_3} \|u^{k,\theta}(x,t) - u^k(x,t)\|_2^2 \cdot p_t(x) \mathrm{d}x \mathrm{d}t \leq K \epsilon^2. \qquad \left( \text{By Lemma D.3} \right)$$

By the upper-bound on $(\mathrm{T}_1)$ and $(\mathrm{T}_2)$, we have

$$\sum_{k=1}^{K} \int_{t_0}^{T} \int_{\mathbb{R}^{d_x}} \|u^{k,\theta}(x,t) - u^k(x,t)\|_2^2 \cdot p_t(x) \mathrm{d}x \mathrm{d}t$$

$$= (\mathrm{T}_1) + (\mathrm{T}_2)$$

$$\lesssim R_3^{d_x - 2} \exp\left( -\frac{C_2 R_3^2}{2(\mu_t^2 + C_2 \sigma_t^2)} \right) \sum_{k=1}^{K} L_{k-1}^2 + K \epsilon^2$$

$$\lesssim \max\left\{ R_3^{d_x - 2} \exp\left( -\frac{C_2 R_3^2}{2(\mu_t^2 + C_2 \sigma_t^2)} \right), \epsilon^2 \right\}.$$

Finally, for some $N \in \mathbb{N}$ and $\beta > 0$, we set

$$R_3 := \sqrt{\frac{4\beta(\mu_t^2 + C_2 \sigma_t^2) \log N}{C_2}} \quad \text{and} \quad \epsilon := N^{-\beta}.$$

This gives

$$\sum_{k=1}^{K} \int_{t_0}^{T} \int_{\mathbb{R}^{d_x}} \|u^{k,\theta}(x,t) - u^k(x,t)\|_2^2 \cdot p_t(x) \mathrm{d}x \mathrm{d}t = O\left( N^{-2\beta} \cdot (\log N)^{\frac{d_x}{2} - 1} \right)$$

The transformer parameter bounds follow Lemma D.3 with $I = O(\sqrt{\log N})$ and $\epsilon = N^{-\beta} > 0$:

$$C_{KQ}, C_{KQ}^{2,\infty} = O(\lambda^{-1} N^{2\beta(2d+1)} (\log N)^{2d+1}); C_{OV}, C_{OV}^{2,\infty} = O(N^{-\beta});$$

$$C_F, C_F^{2,\infty} = O(N^\beta \sqrt{\log N} L_{k-1}); C_E = O(I); C_{\mathcal{T}} = O(L_{k-1}), \qquad (\text{D.2})$$

where $\lambda^{-1} = O(N^\beta \log N)^{4d+3}$ is the inverse-temperature scaling in the softmax function.

This completes the proof. $\qquad \square$

# E  Proof of Theorem 4.2

In this section, we derive the estimation rate of the $K$ order flow matching using transformers. We decompose the proof of Theorem 4.2 into the following three parts due to its complexity.

- **Step 0: Preliminaries.** We introduce several essential definitions, including the $K$ order conditional flow matching loss, $K$ order empirical risk and their domain truncation. These definitions are the extensions from the velocity estimation analysis (see Appendix I.3 and Appendix L).

- **Step 1: Controlling Error from Loss Function outside of the Truncated Domain.** By leveraging the sub-Gaussian tail bound and the Lipschitz continuity of the $k$-th order velocity field, we derive an upper bound on the loss function outside of the truncated domain in Lemma E.1.

- **Step 2: Upper Bound on the Covering Number.** We present a unified upper bound on the covering number that holds across $K$ transformer networks $u^{1,\theta}, \ldots, u^{K,\theta}$ in Lemma E.2.

- **Step 3: Generalization Error.** We apply the covering number technique to bound the deviation between the $K$ order empirical risk and the $K$ order true risk in Lemma E.3.

**Organizations.**  Appendix E.1 includes preliminaries on the framework of estimators' quality evaluation. Appendix E.2 introduces auxiliary lemmas. Appendix E.3 presents the main proof.

## E.1  Preliminaries

In this section, we consider affine conditional $\psi_t(x|X_1^{(0)}) = \mu_t X_1^{(0)} + \sigma_t x$ following Section 2. Given $k$-th order velocity estimator $u^{k,\theta}$, we aim to bound the flow matching risk $\mathcal{R}_K(\Theta)$:

$$
\mathcal{R}_K(\Theta) := \sum_{k=1}^{K} \frac{1}{T - t_0} \int_{t_0}^{T} \mathbb{E}_{x \sim p_t^0} [\|u^{k,\theta}(x,t) - u^k(x,t)\|_2^2] \mathrm{d}t,
$$

where the density function $p_t$ and the $k$-th order flow are induced by the flow $\psi_t$ (Definition 3.1).

In practice, we use the $K$ order conditional flow matching loss to train $u^{1,\theta}, \ldots, u^{K,\theta} \in \mathcal{T}_R^{h,s,r}$.

---

**Definition E.1** (High-Order Conditional Flow Matching Loss).   Let $q$ be the ground truth distribution and the normal distribution $N(0, I)$ be the source distribution $p$. Considering affine conditional flows $\psi_t(x|X_1) = \mu_t X_1 + \sigma_t x$, we define the $K$ order conditional flow matching loss:

$$
\mathcal{L}_{\mathrm{CFM}}^K(\Theta) := \sum_{k=1}^{K} \frac{1}{T - t_0} \int_{t_0}^{T} \mathbb{E}_{X_1^{(0)} \sim q, X_0^{(0)} \sim p} [\|(\mu_t^{(k)} X_1^{(0)} + \sigma_t^{(k)} X_0^{(0)}) - u^{k,\theta}(X_t^{(0)}, t)\|_2^2] \mathrm{d}t.
$$

Further, we define the $K$ order loss function

$$
\ell_K(x; u^{1,\theta}, \ldots, u^{K,\theta}) := \sum_{k=1}^{K} \frac{1}{T - t_0} \int_{t_0}^{T} \mathbb{E}_{X_0^{(0)} \sim p} [\|(\mu_t^{(k)} x + \sigma_t^{(k)} X_0^{(0)}) - u^{k,\theta}(X_t^{(0)}, t)\|_2^2] \mathrm{d}t.
$$

---

Given a set of i.i.d sample $\{x_i\}_{i=1}^n$, we obtain transformers $u^{1,\theta}, \ldots, u^{K,\theta}$ by optimizing the empirical conditional flow matching loss:

$$
\widehat{\mathcal{L}}_{\mathrm{CFM}}^K := \frac{1}{n} \sum_{i=1}^{n} \sum_{k=1}^{K} \frac{1}{T - t_0} \int_{t_0}^{T} \mathbb{E}_{X_0 \sim N(0,I)} [\|(\mu_t^{(k)} x_i + \sigma_t^{(k)} X_0^{(0)}) - u^{k,\theta}(X_t^{(0)}, t)\|_2^2] \mathrm{d}t.
$$

Then, we define the $K$-order empirical risk:

**Definition E.2** (High-Order Empirical Risk). Let $u^{k,\theta}$ be the estimator of the $k$-th order velocity field $u^k$. Further, consider i.i.d training samples $\{x_i\}_{i=1}^n$ and empirical conditional flow matching loss $\widehat{\mathcal{L}}_{\mathrm{CFM}}^K = \frac{1}{n}\sum_{i=1}^n \ell_K(x_i; \cdot)$. Then, we define the $K$ order empirical risk:

$$\widehat{\mathcal{R}}_K(\Theta) := \frac{1}{n}\sum_{i=1}^n \ell_K(x_i; u^{1,\theta}, \dots, u^{K,\theta}) - \frac{1}{n}\sum_{i=1}^n \ell_K(x_i; u^1, \dots, u^K).$$

**Remark E.1.** Let $\mathcal{R}_K(f_t)$ be the ground truth inputs of the high-order risk; that is, $u^{k,\theta} = u^k$ for any $k \in [K]$. Then, by the definition of high-order velocity field in Definition 3.1, $\mathcal{R}_K(f_t) = 0$ since $f_t(y_t) = (u^1, \dots, u^K)$ is the collection of $K$ order ground truth velocity fields. Further, the gradient equivalence Theorem 3.4 implies that $\mathcal{R}_K(\Theta) = \mathcal{R}_K(\Theta) - \mathcal{R}_K(f_t) = \mathcal{L}_{\mathrm{CFM}}^K(\Theta) - \mathcal{L}_{\mathrm{CFM}}(f_t)$.

**Remark E.2.** We use $\widehat{\mathcal{L}}_{\mathrm{CFM}}^{K'}$ and $\widehat{\mathcal{R}}_K'$ to denote the conditional flow matching loss and empirical risk with training samples $\{x_i'\}_{i=1}^n$. Then, by the i.i.d assumption on the training sample, we have $\mathbb{E}_{\{x_i'\}_{i=1}^n}[\widehat{\mathcal{L}}_{\mathrm{CFM}}^{K'}(\Theta)] = \mathcal{L}_{\mathrm{CFM}}(\Theta)$, and therefore $\mathbb{E}_{\{x_i'\}_{i=1}^n}[\widehat{\mathcal{R}}_K'(\Theta)] = \mathcal{R}_K(\Theta)$.

To obtain finite covering number, we introduce the $K$ truncated loss and truncated risk.

**Definition E.3** (Domain Truncation of High-Order Loss and Risk). Let $D > 0$ be constant. Given the $K$ order conditional flow matching loss $\ell_K(x; u^{1,\theta}, \dots, u^{K,\theta})$ defined in Definition E.1, we define its truncated counterparts on a bounded domain $\mathcal{D} := [-D, D]^{d_x}$ by

$$\ell_K^{\mathrm{trunc}}(x; u^{1,\theta}, \dots, u^{K,\theta}) := \ell_K(x; u^{1,\theta}, \dots, u^{K,\theta})\mathbb{1}\{\|x\|_\infty \le D\}.$$

Given the $K$ order conditional flow matching risk and the $K$ order empirical risk, we define

$$\mathcal{R}_K^{\mathrm{trunc}}(\Theta) := \mathcal{R}_K(\Theta)\mathbb{1}\{\|x\|_\infty \le D\}, \quad \widehat{\mathcal{R}}_K^{\mathrm{trunc}}(\Theta) := \widehat{\mathcal{R}}_K(\Theta)\mathbb{1}\{\|x\|_\infty \le D\}.$$

## E.2 Auxiliary Lemmas

We follow the proof of velocity estimation in Appendix L.2 and Appendix L.3 to bound the $K$ order flow matching estimation error. Since direct computation of risk is infeasible, we first decompose the $K$ order flow matching risk $\mathcal{R}_K$ into four terms. Then, we leverage the sub-Gaussian property (Assumption I.1) and the Lipschitzness of transformer network class (Definition B.2) to bound each term. Specifically, we introduce three lemmas to bound

1. the error from the domain truncation of loss function class (Lemma E.1),

2. the log covering number of loss function class (Lemma E.2), and

3. the generalization error bound (Lemma E.3).

**Risk Decomposition.** For simplicity, we shorthand $\mathcal{R}_K(u^{1,\theta}, \dots, u^{K,\theta})$ with $\mathcal{R}_K$. Let $\{x_i'\}_{i=1}^n$ be a different set of i.i.d samples independent of the training sample $\{x_i\}_{i=1}^n$. Then we decompose:

$$\begin{aligned}
\mathbb{E}_{\{x_i\}_{i=1}^n}[\mathcal{R}_K] = {}& \underbrace{\mathbb{E}_{\{x_i\}_{i=1}^n}\big[\mathbb{E}_{\{x_i'\}_{i=1}^n}[\widehat{\mathcal{R}}_K' - \widehat{\mathcal{R}}_K'^{\mathrm{trunc}}]\big]}_{(\mathrm{I})} \\
& + \underbrace{\mathbb{E}_{\{x_i\}_{i=1}^n}\big[\mathbb{E}_{\{x_i'\}_{i=1}^n}[\widehat{\mathcal{R}}_K'^{\mathrm{trunc}} - \widehat{\mathcal{R}}_K^{\mathrm{trunc}}]\big]}_{(\mathrm{II})} \\
& + \underbrace{\mathbb{E}_{\{x_i\}_{i=1}^n}[\widehat{\mathcal{R}}_K^{\mathrm{trunc}} - \widehat{\mathcal{R}}_K]}_{(\mathrm{III})} + \underbrace{\mathbb{E}_{\{x_i\}_{i=1}^n}[\widehat{\mathcal{R}}_K]}_{(\mathrm{IV})},
\end{aligned}$$

where we use the fact that $\mathbb{E}_{\{x_i\}_{i=1}^n}[\widehat{\mathcal{R}}_K(\Theta)] = \mathcal{R}_K(\Theta)$ (Remark E.1). This decomposition follows standard statistical learning theory technique, formulated in [Hu et al., 2025c, Fu et al., 2024].

**High-Order Truncation Loss.** We begin with the bounds on term (I) and term (III).

---

**Lemma E.1** (Upper Bound on the High-Order Truncation Error). Let $u^{1,\theta}, \ldots, u^{K,\theta} \in \mathcal{T}_R^{h,s,r}$ be transformers in Theorem 4.1. Then, for any $t \in [t_0, T]$ it holds

$$\mathbb{E}_x\big[|\ell_K(x; u^{1,\theta}, \ldots, u^{K,\theta}) - \ell_K^{\text{trunc}}(x; u^{1,\theta}, \ldots, u^{K,\theta})|\big] \lesssim KD^{d_x} \exp\left(-\frac{1}{2}C_2D^2\right)\max_k\{L_k^2\}.$$

---

*Proof.* By Theorem 4.1, we have transformers output bounds $C_{\mathcal{T}} = O(L_{k-1})$ for all $k$.

For all $k \in [K]$, we define

$$\ell_k(x; u^{k,\theta}) := \frac{1}{T-t_0}\int_{t_0}^T \mathbb{E}_{X_0 \sim N(0,I)}[\|u^{k,\theta}(X_t^{(0)}, t) - (\mu_t^{(k)}x + \sigma_t^{(k)}X_0^{(0)})\|_2^2]\mathrm{d}t$$

$$\ell_k^{\text{trunc}}(x; u^{k,\theta}) := \frac{1}{T-t_0}\int_{t_0}^T \mathbb{E}_{X_0 \sim N(0,I)}[\|u^{k,\theta}(X_t^{(0)}, t) - (\mu_t^{(k)}x + \sigma_t^{(k)}X_0^{(0)})\|_2^2]\mathrm{d}t\mathbb{1}\{\|x\|_\infty \le D\}.$$

Then, it holds

$$\mathbb{E}_x\big[|\ell_k(x; u^{k,\theta}) - \ell_k^{\text{trunc}}(x; u^{k,\theta})|\big] \tag{E.1}$$

$$= \mathbb{E}_x\big[|\ell_k(x; u^{k,\theta})\mathbb{1}[\|x\| \ge D]|\big] \qquad \text{(By Definition E.3)}$$

$$= \frac{1}{T-t_0}\int_{t_0}^T \int_{\|x\|>D} \mathbb{E}_{X_0 \sim N(0,I)}[\|u^{k,\theta}(X_t^{(0)}, t) - (\mu_t^{(k)}x + \sigma_t^{(k)}X_0^{(0)})\|_2^2]q(x)\mathrm{d}x\mathrm{d}t$$
$$\text{(By Definition E.1)}$$

$$\lesssim \frac{1}{T-t_0}\int_{t_0}^T \int_{\|x\|\ge D} \mathbb{E}_{X_0 \sim N(0,I)}[\|u^{k,\theta}(X_t^{(0)}, t)\|_2^2 + \|\mu_t^{(k)}x + \sigma_t^{(k)}X_0^{(0)}\|_2^2]q(x)\mathrm{d}x\mathrm{d}t$$
$$\text{(By expanding the } \ell_2\text{-norm)}$$

$$\lesssim \frac{1}{T-t_0}\int_{t_0}^T \int_{\|x\|\ge D} \mathbb{E}_{X_0 \sim N(0,I)}[\|u^{k,\theta}(X_t^{(0)}, t)\|_2^2 + \|\mu_t^{(k)}x + \sigma_t^{(k)}X_0^{(0)}\|_2^2]\exp\left(-\frac{1}{2}C_2\|x\|_2^2\right)\mathrm{d}x\mathrm{d}t$$
$$\text{(By Assumption I.1)}$$

$$\lesssim \frac{1}{T-t_0}\int_{t_0}^T \int_{\|x\|\ge D} \mathbb{E}_{X_0 \sim N(0,I)}[\max_k\{L_k^2\} + \|\mu_t^{(k)}x + \sigma_t^{(k)}X_0^{(0)}\|_2^2]\exp\left(-\frac{1}{2}C_2\|x\|_2^2\right)\mathrm{d}x\mathrm{d}t$$
$$\text{(By } C_{\mathcal{T}} = O(\max_k\{L_k\}))$$

$$\lesssim \frac{1}{T-t_0}\int_{t_0}^T \int_{\|x\|\ge D} (\max_k\{L_k^2\} + (\sigma_t^{(k)})^2 d_x + (\mu_t^{(k)})^2\|x\|_2^2)\exp\left(-\frac{1}{2}C_2\|x\|_2^2\right)\mathrm{d}x\mathrm{d}t$$
$$(x_0 \sim N(0,I))$$

$$\lesssim \frac{D^{d_x-2}\exp\left(-\frac{1}{2}C_2D^2\right)}{T-t_0}\int_{t_0}^T \left(\max_k\{L_k^2\} + (\sigma_t^{(k)})^2 d_x\right)\mathrm{d}t + \frac{D^{d_x}\exp\left(-\frac{1}{2}C_2D^2\right)}{T-t_0}\int_{t_0}^T (\mu_t^{(k)})^2\mathrm{d}t$$
$$\text{(By Lemma D.2)}$$

$$\lesssim D^{d_x}\exp\left(-\frac{1}{2}C_2D^2\right)\max_k\{L_k^2\}. \qquad \text{(By Assumption I.2)}$$

Therefore,

$$\mathbb{E}_x\big[|\ell_K(x; u^{1,\theta}, \ldots, u^{K,\theta}) - \ell_K^{\text{trunc}}(x; u^{1,\theta}, \ldots, u^{K,\theta})|\big]$$

$$\leq \sum_{k=1}^{K} \mathbb{E}_x[|\ell_k(x; u^{k,\theta}) - \ell_k^{\mathrm{trunc}}(x; u^{k,\theta})|] \qquad \text{(By triangle inequality)}$$

$$\lesssim K D^{d_x} \exp\left(-\frac{1}{2} C_2 D^2\right) \max_k \{L_k^2\}. \qquad \text{(By (E.1))}$$

This completes the proof. $\qquad\qquad\qquad\qquad\qquad\qquad\qquad\qquad\qquad\qquad\qquad\qquad\qquad\square$

**Covering Number of High-Order Loss Function Class with Transformers.** The next lemma extends Lemma L.5 to its higher-order counterpart. Please see Definition L.4 for a precise definition.

---

**Lemma E.2** (Covering Number Bounds for $\mathcal{S}(D)$, Lemma K.2 of [Hu et al., 2025c], Theorem A.17 of [Edelman et al., 2022]). Let $\epsilon_c > 0$. We define the loss function class by $\mathcal{S}(D) := \{\ell_K(x; u^{1,\theta}, \dots, u^{K,\theta}) : \mathcal{D} \to \mathbb{R} \mid u^{1,\theta}, \dots, u^{K,\theta} \in \mathcal{T}_R^{h,s,r}\}$. Further, we define the norm of loss functions by $\|\ell_K\|_{\infty\mathcal{D}} := \max_{x \in [-D,D]^{d_x}} |\ell_K|$. Then, under transformer parameter configuration in Theorem 4.1 the $\epsilon_c$-covering number of $\mathcal{S}(D)$ with respect to $\|\cdot\|_{\infty\mathcal{D}}$ satisfies:

$$\log \mathcal{N}(\epsilon_c, \mathcal{S}(D), \|\cdot\|_{\infty\mathcal{D}}) \leq O\left(\frac{\log(nL/\epsilon_c)}{\epsilon_c^2} D^2 N^{\beta(16d+12)} (\log N)^{8d+8}\right).$$

---

*Proof.* We first derive the log covering number of transformers $u^{1,\theta}, \dots, u^{K,\theta}$ in Theorem 4.1. Then, we extend the results to $K$ order loss function class.

- **Log-Covering Number of Transformers Network Class.** From (D.2), for all $k \in [K]$, we have

$$C_{KQ}, C_{KQ}^{2,\infty} = O(\lambda^{-1} I^{4d+2} \epsilon^{-4d-2}); C_{OV}, C_{OV}^{2,\infty} = O(\epsilon);$$
$$C_F, C_F^{2,\infty} = O(I\epsilon^{-1} L_{k-1}); C_E = O(I); C_{\mathcal{T}} = O(L_{k-1}),$$

where $\lambda^{-1} = O(N^\beta \log N)^{4d+3}$ is the inverse-temperature scaling in the softmax function, $I = O(\sqrt{\log N})$ and $\epsilon = N^{-\beta} > 0$ some $N \in \mathbb{N}$ and $\beta > 0$.

By Lemma L.5, the bounds on log-covering number follow

$$\log \mathcal{N}(\epsilon_c, \mathcal{T}_R^{h,s,r}, \|\cdot\|_2)$$
$$\leq \frac{\alpha^2 \log(nL/\epsilon_c)}{\epsilon_c^2} \left((C_F^{2\infty})^{\frac{4}{3}} + (\lambda(C_F)^2 C_{OV} C_{KQ}^{2,\infty})^{\frac{2}{3}} + ((C_F)^2 C_{OV}^{2,\infty})^{\frac{2}{3}}\right)^3$$
$$\lesssim \frac{\alpha^2 \log(nL/\epsilon_c)}{\epsilon_c^2} \Big(\underbrace{I^{4/3}\epsilon^{-4/3}}_{(C_F^{2,\infty})^{\frac{4}{3}}} + \underbrace{I^{4/3}\epsilon^{-4/3}}_{(C_F)^{4/3}} \underbrace{\epsilon^{2/3}}_{(C_{OV})^{2/3}} \underbrace{I^{(8d+4)/3}\epsilon^{-8d/3-4/3}}_{(\lambda C_{KQ}^{2,\infty})^{2/3}} + \underbrace{I^{4/3}\epsilon^{-4/3}}_{(C_F)^{\frac{4}{3}}} \underbrace{\epsilon^{2/3}}_{(C_{OV}^{2,\infty})^{\frac{2}{3}}} \Big)^3$$
$$\lesssim \frac{\alpha^2 \log(nL/\epsilon_c)}{\epsilon_c^2} \cdot (I^{(8d+8)/3}\epsilon^{-8d/3-2})^3$$
$$= \frac{\alpha^2 \log(nL/\epsilon_c)}{\epsilon_c^2} \cdot I^{8d+8}\epsilon^{-8d-6}.$$

By Lemma L.5, we have

$$\alpha \lesssim (C_F)^2 C_{OV} C_{KQ}(D + C_E)$$
$$\lesssim \underbrace{I^2\epsilon^{-2}}_{(C_F)^2} \cdot \underbrace{\epsilon}_{(C_{OV})} \cdot \underbrace{I^{4d+2}\epsilon^{-4d-2}}_{(C_{KQ})} \cdot (D + C_E) \qquad \text{(By the definition of } \alpha)$$
$$= DI^{4d+4}\epsilon^{-4d-3}.$$

Altogether, for all $u^{k,\theta} \in \mathcal{T}_R^{h,s,r}$, we have

$$\log \mathcal{N}(\epsilon_c, \mathcal{T}_R^{h,s,r}, \|\cdot\|_2) \lesssim \frac{\log(nL/\epsilon_c)}{\epsilon_c^2} D^2 I^{16d+16} \epsilon^{-16d-12}.$$

Further, by $\|\cdot\|_\infty \le \|\cdot\|_2$, we have

$$\log \mathcal{N}(\epsilon_c, \mathcal{T}_R^{h,s,r}, \|\cdot\|_\infty) \lesssim \frac{\log(nL/\epsilon_c)}{\epsilon_c^2} D^2 I^{16d+16} \epsilon^{-16d-12}. \tag{E.2}$$

for all $u^{k,\theta} \in \mathcal{T}_R^{h,s,r}$.

- **Log-Covering Number of Loss Function Class.** Let $\delta > 0$. Let $u := \{u^{1,\theta}, \dots, u^{K,\theta}\}$ and $\bar{u} := \{\bar{u}^{1,\theta}, \dots, \bar{u}^{K,\theta}\}$ be two sets of transformers network satisfying $\|u^{k,\theta} - u^{s,\theta}\|_\infty \le \delta$ on domain $x \in [-D, D]^{d_x}$ for all $u^{k,\theta} \in u$ and $u^{s,\theta} \in \bar{u}$. Further, let $\psi_{t,k}^\star$ denote the ground truth $k$-th order conditional velocity field (Definition E.1):

$$\psi_{t,k}^\star := \mu_t^{(k)} x + \sigma_t^{(k)} X_0^{(0)}.$$

Then, the distance between two $K$ order conditional loss functions $\ell_{K,1}(x; u^{1,\theta}, \dots, u^{K,\theta})$ and $\ell_{K,2}(x; \bar{u}^{1,\theta}, \dots, \bar{u}^{K,\theta})$ follows:

$$\left| \ell_{K,1}(x; u^{1,\theta}, \dots, u^{K,\theta}) - \ell_{K,2}(x; \bar{u}^{1,\theta}, \dots, \bar{u}^{K,\theta}) \right| \tag{E.3}$$

$$= \frac{1}{T - t_0} \left| \sum_{k=1}^K \int_{t_0}^T \mathop{\mathbb{E}}_{X_0 \sim N(0,I)} [\|u^{k,\theta} - \psi_{t,k}^\star\|_2^2] \mathrm{d}t - \sum_{s=1}^K \int_{t_0}^T \mathop{\mathbb{E}}_{X_0 \sim N(0,I)} [\|\bar{u}^{s,\theta} - \psi_{t,k}^\star\|_2^2] \mathrm{d}t \right|$$
$$\left( \text{By Definition E.1} \right)$$

$$\le \sum_{k=1}^K \frac{1}{T - t_0} \left| \int_{t_0}^T \mathop{\mathbb{E}}_{X_0 \sim N(0,I)} [(u^{k,\theta} + \bar{u}^{k,\theta} - 2\psi_{t,k}^\star)^\top (u^{k,\theta} - \bar{u}^{k,\theta})] \mathrm{d}t \right| \quad \left( \text{By triangle inequality} \right)$$

$$\le \sum_{k=1}^K \frac{\delta}{T - t_0} \int_{t_0}^T \mathop{\mathbb{E}}_{X_0 \sim N(0,I)} [\|u^{k,\theta} + \bar{u}^{k,\theta} - 2\psi_{t,k}^\star\|] \mathrm{d}t \quad \left( \text{By } \|u^{k,\theta} - \bar{u}^{k,\theta}\|_\infty \le \delta \right)$$

$$\le \sum_{k=1}^K \frac{\delta}{T - t_0} \int_{t_0}^T \sqrt{2 \mathop{\mathbb{E}}_{X_0 \sim N(0,I)} [\|u^{k,\theta} + \bar{u}^{k,\theta}\|_2^2 + 2\|\psi_{t,k}^\star\|_2^2]} \mathrm{d}t \quad \left( \text{By Jensen's inequality} \right)$$

$$\lesssim \sum_{k=1}^K \frac{\delta}{T - t_0} \int_{t_0}^T \sqrt{\max_k \{L_k^2\} + 2\|\psi_{t,k}^\star\|_2^2} \mathrm{d}t \quad \left( \text{By } C_\mathcal{T} = O(\max_k\{L_k\}) \right)$$

$$\lesssim \sum_{k=1}^K \frac{\delta \max_k\{L_k\}}{T - t_0} \int_{t_0}^T \mathrm{d}t \quad \left( \text{By the Lipschitzness of } k\text{-th order flow} \right)$$

$$\lesssim \delta \max_k\{L_k\}.$$

Finally, we extend the log covering number to the loss function class $\mathcal{S}(D)$ by setting

$$\epsilon_c' := \Omega\big(\epsilon_c \max_k\{L_k\}\big).$$

This gives

$$\log \mathcal{N}(\epsilon_c', \mathcal{S}(D), \|\cdot\|_{\infty\mathcal{D}}) \le \log \mathcal{N}(\epsilon_c, \mathcal{T}_R^{h,s,r}, \|\cdot\|_\infty). \tag{By E.3}$$

Therefore,

$$
\begin{aligned}
&\log \mathcal{N}(\epsilon_c', \mathcal{S}(D), \|\cdot\|_{\infty \mathcal{D}}) \\
&\leq \log \mathcal{N}(\epsilon_c, \mathcal{T}_R^{h,s,r}, \|\cdot\|_\infty) \\
&\lesssim \frac{\log(nL/\epsilon_c)}{\epsilon_c^2} D^2 I^{16d+16} \epsilon^{-16d-12} && (\text{By (E.2)}) \\
&= O\Big(\frac{\log(nL/\epsilon_c)}{(\epsilon_c')^2} D^2 I^{16d+16} \epsilon^{-16d-12} \max_k \{L_k^2\}\Big). && (\text{By the definition of } \epsilon_c')
\end{aligned}
$$

Finally, we substitute $I = O(\sqrt{\log N})$ and $\epsilon = N^{-\beta} > 0$. This completes the proof. $\qquad\square$

**Generalization Bound.** Based on covering number bounds results in Lemma E.2, we now analyze the upper bound of generalization error $\left|\mathbb{E}_{\{x_i\}_{i=1}^n}[\mathcal{R}_K^{\mathrm{trunc}}(\widehat{\Theta}) - \widehat{\mathcal{R}}_K^{\mathrm{trunc}}(\widehat{\Theta})]\right|$.

**Lemma E.3** (Generalization Bound on $K$ Order Flow Matching Risk). For $\epsilon_c > 0$, let $\mathcal{N} := \mathcal{N}(\epsilon_c, \mathcal{S}(D), \|\cdot\|_{\infty \mathcal{D}})$ be the covering number of function class of loss $\mathcal{S}(D)$ following Lemma E.2. Let $\widehat{\Theta}$ be the collection of parameters of transformers trained by optimizing $\mathcal{L}_{\mathrm{CFM}}(\Theta)$ following Definition E.1 with i.i.d training samples $\{x_i\}_{i=1}^n$. Then we bound the generalization error:

$$
\mathbb{E}_{\{x_i\}_{i=1}^n}\Big[\mathcal{R}_K^{\mathrm{trunc}}(\widehat{\Theta}) - \widehat{\mathcal{R}}_K^{\mathrm{trunc}}(\widehat{\Theta})\Big] \leq \widehat{\mathcal{R}}_k^{\mathrm{trunc}}(\widehat{\Theta}) + O(\tfrac{1}{n}\log\mathcal{N} + \epsilon_c).
$$

*Proof.* Let $\widehat{u}^{k,\theta} \in \mathcal{T}_R^{h,s,r}$ be the approximator of the $k$-th velocity field $u^k$ obtained from minimizing the high-order empirical conditional flow matching loss:

$$
\widehat{\mathcal{L}}_{\mathrm{CFM}}^K := \frac{1}{n}\sum_{i=1}^n \sum_{k=1}^K \frac{1}{T-t_0}\int_{t_0}^T \mathbb{E}_{X_0 \sim N(0,I)}[\|(\mu_t^{(k)} x_i + \sigma_t^{(k)} X_0^{(0)}) - u^{k,\theta}(X_t^{(0)}, t)\|_2^2]\mathrm{d}t.
$$

Further, we define

$$
\mathcal{R}_k^{\mathrm{trunc}}(\widehat{u}^{k,\theta}) := \frac{1}{T-t_0}\int_{t_0}^T \mathbb{E}_{x \sim p_t}[\|u^k(x,t) - u^{k,\theta}(x,t)\|_2^2]\mathbb{1}\{\|x\|_\infty \leq D\}\mathrm{d}t,
$$

and

$$
\begin{aligned}
&\widehat{\mathcal{R}}_k^{\mathrm{trunc}}(\widehat{u}^{k,\theta}) \\
&:= \frac{1}{n}\sum_{i=1}^n \frac{1}{T-t_0}\int_{t_0}^T \mathbb{E}_{X_0^{(0)} \sim p}[\|(\mu_t^{(k)} x_i + \sigma_t^{(k)} X_0^{(0)}) - u^{k,\theta}(X_t^{(0)}, t)\|_2^2]\mathrm{d}t \cdot \mathbb{1}\{\|x_i\|_\infty \leq D\} \\
&\quad - \frac{1}{n}\sum_{i=1}^n \frac{1}{T-t_0}\int_{t_0}^T \mathbb{E}_{X_0^{(0)} \sim p}[\|(\mu_t^{(k)} x_i + \sigma_t^{(k)} X_0^{(0)}) - u^k(X_t^{(0)}, t)\|_2^2]\mathrm{d}t \cdot \mathbb{1}\{\|x_i\|_\infty \leq D\}
\end{aligned}
$$

Since every network configurations and log covering number are identical across all $K$ order velocity fields from Theorem 4.1 and Lemma E.2, for any $k \in [K]$, Lemma L.8 extends to

$$
\left|\mathbb{E}_{\{x_i\}_{i=1}^n}[\mathcal{R}_k^{\mathrm{trunc}}(\widehat{u}^{k,\theta}) - \widehat{\mathcal{R}}_k^{\mathrm{trunc}}(\widehat{u}^{k,\theta})]\right| \leq \frac{1}{2}\mathbb{E}_{\{x_i\}_{i=1}^n}[\mathcal{R}_k^{\mathrm{trunc}}(\widehat{u}^{k,\theta})] + O(\tfrac{1}{n}\log\mathcal{N} + \epsilon_c).
$$

Therefore,

$$
\left|\mathbb{E}_{\{x_i\}_{i=1}^n}[\mathcal{R}_k^{\mathrm{trunc}}(\widehat{\Theta}) - \widehat{\mathcal{R}}_k^{\mathrm{trunc}}(\widehat{\Theta})]\right|
$$

$$\leq \sum_{k=1}^{K} \left| \mathop{\mathbb{E}}_{\{x_i\}_{i=1}^{n}} [\mathcal{R}_k^{\mathrm{trunc}}(\widehat{u}^{k,\theta}) - \widehat{\mathcal{R}}_k^{\mathrm{trunc}}(\widehat{u}^{k,\theta})] \right| \qquad \text{(By the triangle inequality)}$$

$$\lesssim \sum_{k=1}^{K} \frac{1}{2} \mathop{\mathbb{E}}_{\{x_i\}_{i=1}^{n}} [\mathcal{R}_k^{\mathrm{trunc}}(\widehat{u}^{k,\theta})] + O(\frac{1}{n}\log\mathcal{N} + \epsilon_c) \qquad \text{( By Lemma L.8)}$$

$$= \frac{1}{2} \cdot \mathop{\mathbb{E}}_{\{x_i\}_{i=1}^{n}} [\mathcal{R}_K^{\mathrm{trunc}}(\widehat{\Theta})] + O(\frac{1}{n}\log\mathcal{N} + \epsilon_c).$$

This implies

$$\mathop{\mathbb{E}}_{\{x_i\}_{i=1}^{n}} [\mathcal{R}_k^{\mathrm{trunc}}(\widehat{\Theta})] \leq 2 \cdot \widehat{\mathcal{R}}_k^{\mathrm{trunc}}(\widehat{\Theta}) + O(\frac{1}{n}\log\mathcal{N} + \epsilon_c).$$

Finally, we conclude that

$$\mathop{\mathbb{E}}_{\{x_i\}_{i=1}^{n}} [\mathcal{R}_k^{\mathrm{trunc}}(\widehat{\Theta}) - \widehat{\mathcal{R}}_k^{\mathrm{trunc}}(\widehat{\Theta})] \leq \widehat{\mathcal{R}}_k^{\mathrm{trunc}}(\widehat{\Theta}) + O(\frac{1}{n}\log\mathcal{N} + \epsilon_c).$$

This completes the proof. □

## E.3 Main Proof of Theorem 4.2

We now present the main proof of Theorem 4.2.

**Theorem E.1** (Theorem 4.2 Restated: High-Order Velocity Estimation with Transformer). Assume Assumption 4.1 and Assumption 4.2. Let $\widehat{u}^{k,\theta} \in \mathcal{T}_R^{h,s,r}$ be the estimator of the $k$-th order velocity field $u^k$ trained by minimizing the high-order empirical conditional flow matching loss (3.9). Let $\widehat{\Theta}$ be the collection of parameters of $\widehat{u}^{k,\theta}$ for $k \in [K]$. Suppose the $k$-th order velocity field $u^k(x,t)$ is $L_k$ Lipschitz for all $k = 0, \ldots, K-1$. Suppose we choose the transformers as in Theorem 4.1, then

$$\mathop{\mathbb{E}}_{\{x_i\}_{i=1}^{n}} [\mathcal{R}_K(\widehat{\Theta})] = O\left( n^{-\frac{1}{10d}} \cdot (\log n)^{10d_x} \right),$$

where $d$ is the feature dimension.

*Proof of Theorem 4.2.* Let $\{x_i'\}_{i=1}^{n}$ be a different set of i.i.d samples independent of the training sample $\{x_i\}_{i=1}^{n}$. Further, we use $\widehat{\mathcal{R}}'$ to denote the empirical risk with samples $\{x_i'\}_{i=1}^{n}$.

Then, we decompose $\mathbb{E}_{\{x_i\}_{i=1}^{n}}[\mathcal{R}_K(\widehat{\Theta})]$ as:

$$\mathop{\mathbb{E}}_{\{x_i\}_{i=1}^{n}} \left[ \mathcal{R}_K(\widehat{\Theta}) \right] = \underbrace{\mathop{\mathbb{E}}_{\{x_i\}_{i=1}^{n}} \left[ \mathop{\mathbb{E}}_{\{x_i'\}_{i=1}^{n}} \left[ \widehat{\mathcal{R}}_K'(\widehat{\Theta}) - \widehat{\mathcal{R}}_K'^{\mathrm{trunc}}(\widehat{\Theta}) \right] \right]}_{\text{(I)}}$$

$$+ \underbrace{\mathop{\mathbb{E}}_{\{x_i\}_{i=1}^{n}} \left[ \mathop{\mathbb{E}}_{\{x_i'\}_{i=1}^{n}} \left[ \widehat{\mathcal{R}}_K'^{\mathrm{trunc}}(\Theta) \right] - \widehat{\mathcal{R}}_K^{\mathrm{trunc}}(\widehat{\Theta}) \right]}_{\text{(II)}}$$

$$+ \underbrace{\mathop{\mathbb{E}}_{\{x_i\}_{i=1}^{n}} \left[ \widehat{\mathcal{R}}_K^{\mathrm{trunc}}(\widehat{\Theta}) - \widehat{\mathcal{R}}_K(\widehat{\Theta}) \right]}_{\text{(III)}} + \underbrace{\mathop{\mathbb{E}}_{\{x_i\}_{i=1}^{n}} \left[ \widehat{\mathcal{R}}_K(\widehat{\Theta}) \right]}_{\text{(IV)}},$$

Then, we bound each term and incorporate them to obtain the bound on the estimation error.

- **Bound on** (I) **and** III**.** By Lemma E.1, (I) and (III) are upper bounded by

$$(\mathrm{I}), (\mathrm{III}) \lesssim K D^{d_x} \exp\left(-\frac{1}{2} C_2 D^2\right) \max_k \{L_k^2\}.$$

- **Bound on** (II)**.** By the generalization error bound (Lemma E.3), we have

$$
\begin{aligned}
&(\mathrm{II})\\
&= \mathop{\mathbb{E}}_{\{x_i\}_{i=1}^n} \big[ \mathop{\mathbb{E}}_{\{x_i'\}_{i=1}^n} [\widehat{\mathcal{R}}_K'^{\mathrm{trunc}}(\Theta)] - \widehat{\mathcal{R}}_K^{\mathrm{trunc}}(\Theta) \big]\\
&= \mathop{\mathbb{E}}_{\{x_i\}_{i=1}^n} [\mathcal{R}_K^{\mathrm{trunc}}(\widehat{\Theta}) - \widehat{\mathcal{R}}_K^{\mathrm{trunc}}(\widehat{\Theta})] && (\text{By Remark E.2})\\
&\leq \mathop{\mathbb{E}}_{\{x_i\}_{i=1}^n} [\widehat{\mathcal{R}}_K^{\mathrm{trunc}}(\widehat{\Theta})] + O(\frac{1}{n}\log\mathcal{N} + \epsilon_c) && (\text{By Lemma E.3})\\
&\lesssim (\mathrm{IV}) + D^{d_x} \exp\left(-\frac{1}{2}C_2 D^2\right) \max_k\{L_k^2\} + O(\frac{1}{n}\log\mathcal{N} + \epsilon_c) && (\text{By Lemma E.1})
\end{aligned}
$$

- **Bound on** (IV)**.** Recall Remark E.1, Remark E.2. We have $\widehat{\mathcal{R}}_K(\Theta) := \widehat{\mathcal{L}}_{\mathrm{CFM}}(\Theta) - \widehat{\mathcal{L}}_{\mathrm{CFM}}(f_t)$, where the collection of parameters of $K$ transformers $\widehat{\Theta}$ is trained by optimizing $\widehat{\mathcal{L}}_{\mathrm{CFM}}(\Theta)$.

  Therefore, it holds

$$\widehat{\mathcal{R}}_K(\widehat{\Theta}) \leq \widehat{\mathcal{L}}_{\mathrm{CFM}}(\Theta) - \widehat{\mathcal{L}}_{\mathrm{CFM}}(f_t) = \widehat{\mathcal{R}}_K(\Theta).$$

  Then, for any velocity estimator $\Theta$, it holds

$$\mathop{\mathbb{E}}_{\{x_i\}_{i=1}^n} [\widehat{\mathcal{R}}_K(\widehat{\Theta})] \leq \mathop{\mathbb{E}}_{\{x_i\}_{i=1}^n} [\widehat{\mathcal{R}}_K(\Theta)] = \mathcal{R}_K(\Theta). \tag{E.4}$$

  This implies

$$(\mathrm{IV}) \leq \mathcal{R}_K(\Theta) \lesssim N^{-2\beta} \cdot (\log N)^{\frac{d_x}{2}-1}. \qquad (\text{By Theorem 4.1})$$

Altogether, the estimation error is upper bounded by

$$
\begin{aligned}
&\mathop{\mathbb{E}}_{\{x_i\}_{i=1}^n} [\mathcal{R}_K(\widehat{\Theta})] && (\text{E.5})\\
&= (\mathrm{I}) + (\mathrm{II}) + (\mathrm{III}) + (\mathrm{IV})\\
&\lesssim \underbrace{D^{d_x} \exp\left(-\frac{1}{2}C_2 D^2\right)}_{(\mathrm{T}_1)} + \underbrace{O(\frac{1}{n}\log\mathcal{N} + \epsilon_c)}_{(\mathrm{T}_2)} + \underbrace{N^{-2\beta} \cdot (\log N)^{\frac{d_x}{2}-1}}_{(\mathrm{T}_3)},
\end{aligned}
$$

where

$$\log\mathcal{N} = O\Big(\frac{\log(nL/\epsilon_c)}{\epsilon_c^2} D^2 N^{\beta(16d+12)}(\log N)^{8d+8}\Big). \qquad (\text{By Lemma E.2})$$

Let $\gamma := 16d + 12$. Then, we set $N := n^{\eta_1/(\gamma\beta)}$, $\epsilon_c := n^{-\eta_2}$ and $D := \sqrt{(2\eta_3 \log n)/C_2}$, where $\eta_1, \eta_2, \eta_3 \geq 0$ are constants satisfying $0 \leq \eta_1 + 2\eta_2 < 1$.[3] This gives

$$(\mathrm{T}_1) = D^{d_x} \exp\left(-\frac{1}{2}C_2 D^2\right) \lesssim n^{-\eta_3}(\log n)^{\frac{d_x}{2}}.$$

---

[3]The constraint $0 \leq \eta_1 + 2\eta_2 < 1$ is imposed in order to ensure $(\mathrm{T}_2)$ converges as $n \to \infty$.

Further, we have

$$\log \mathcal{N} = O(n^{\eta_1 + 2\eta_2} (\log n)^{8d_x + 11}),$$

implying

$$(\mathrm{T}_2) = O(\frac{1}{n} \log \mathcal{N} + \epsilon_c) = O(n^{\eta_1 + 2\eta_2 - 1} (\log n)^{8d_x + 11} + n^{-\eta_2}).$$

Further,

$$(\mathrm{T}_3) = n^{-\frac{2\eta_1}{\gamma}} (\log n)^{\frac{d_x}{2} - 1}.$$

Then, (E.5) becomes

$$\underset{\{x_i\}_{i=1}^n}{\mathbb{E}} [\mathcal{R}_K(\Theta)]$$
$$\lesssim (\mathrm{T}_1) + (\mathrm{T}_2) + (\mathrm{T}_3)$$
$$= O\Big(n^{-\min\left\{1 - (\eta_1 + 2\eta_2), \ \eta_2, \ \frac{2\eta_1}{\gamma}\right\}} \cdot (\log n)^{8d_x + 11}\Big).$$

For any $\eta_1$ and $\eta_2$ satisfying

$$0 < \eta_1 + 2\eta_2 < 1,$$

we consider solving

$$\min \left\{1 - (\eta_1 + 2\eta_2), \ \eta_2, \ \frac{2\eta_1}{\gamma}\right\}.$$

The linear programming problem has simple solution

$$1 - (\eta_1 + 2\eta_2) = \eta_2 = \frac{2\eta_1}{\gamma}.$$

This gives

$$\eta_1 = \frac{\gamma}{\gamma + 6}, \quad \text{and} \quad \eta_2 = \frac{2}{\gamma + 6},$$

and $\eta_1 + 2\eta_2 \in (0, 1)$ is satisfied for any $\eta_1, \eta_2 > 0$.

Finally, by $\gamma = 16d + 12$, these free parameters achieves balance and gives

$$\underset{\{x_i\}_{i=1}^n}{\mathbb{E}} [\mathcal{R}_K(\widehat{\Theta})] \lesssim O\Big(n^{-\frac{2}{\gamma + 6}} \cdot (\log n)^{8d_x + 11}\Big) = O\Big(n^{-\frac{1}{8d + 9}} \cdot (\log n)^{8d_x + 11}\Big)$$

This completes the proof. □

# F  Proof of Theorem 4.3

We now present the main proof of Theorem 4.3.

---

**Theorem F.1** (Theorem 4.3 Restated: High-Order Distribution Estimation under 2-Wasserstein Distance). Assume Assumption 4.1 and Assumption 4.2. Let $\widehat{P}_T^K$ be the estimated distribution at time $T$. Then, it holds

$$\mathop{\mathbb{E}}_{\{x_i\}_{i=1}^n} [W_2(\widehat{P}_T^K, P_T^K)] = O\Big(n^{-\frac{1}{18d}} \cdot (\log n)^{6d_x}\Big),$$

where $d$ is the feature dimension.

---

*Proof of Theorem 4.3.* We first consider two general ODE functions that describe the ground truth velocity field and estimated velocity field respectively:

$$\frac{\mathrm{d}}{\mathrm{d}t} y_t = \begin{bmatrix} u^1(x_t^{(0)}, t) \\ u^2(x_t^{(0)}, t) \\ \vdots \\ u^K(x_t^{(0)}, t) \end{bmatrix} := f(y, t), \quad \frac{\mathrm{d}}{\mathrm{d}t} y_t = \begin{bmatrix} u^{1,\theta}(x_t^{(0)}, t) \\ u^{2,\theta}(x_t^{(0)}, t) \\ \vdots \\ u^{K,\theta}(x_t^{(0)}, t) \end{bmatrix} := f^\theta(y, t),$$

where the first $d$ rows of $y_t \in \mathbb{R}^{Kd}$ construct $x_t^{(0)} \in \mathbb{R}^d$.

According to the existence uniqueness theorem of ODEs, these two functions can induce another following two corresponding flows $\phi(\cdot) \in \mathbb{R}^{Kd}$ and $\phi^\theta(\cdot) \in \mathbb{R}^{Kd}$ defined for $t \geq s$ that satisfy:

$$\frac{\mathrm{d}}{\mathrm{d}t} \phi(y, s, t) = f(\phi(y, s, t), t), \quad \phi(y, s, s) = y,$$

$$\frac{\mathrm{d}}{\mathrm{d}t} \phi^\theta(y, s, t) = f^\theta(\phi^\theta(y, s, t), t), \quad \phi^\theta(y, s, s) = y.$$

We define the first $d$ rows of $\phi(\cdot)$ construct the flow function $\psi_t(x)$ and the first $d$ rows of $\phi^\theta(\cdot)$ construct the flow function $\psi_t^\theta(x)$. By applying Lemma M.2, it holds that

$$\phi^\theta(y, t_0, T) - \phi(y, t_0, T) = \int_{t_0}^T D\phi^\theta(\phi(y, t_0, s), s, T)(f^\theta(\phi(y, t_0, s), s) - f(\phi(y, t_0, s), s))\mathrm{d}s.$$

We extract the first $d$ rows of left-hand-side and it holds:

$$\psi^\theta(x, t_0, T) - \psi(x, t_0, T) = \int_{t_0}^T D\phi^\theta(\phi(y, t_0, s), s, t))[: d](f^\theta(\phi(y, t_0, s), s) - f(\phi(y, t_0, s), s))\mathrm{d}s,$$

where $D\phi^\theta(\phi(y, t_0, s), s, t))[: d]$ denotes the first $d$ rows of the Jacobian matrix.

We then bound $\psi^\theta(x, t_0, T) - \psi(x, t_0, T)$ by using similar techniques in proof of Theorem I.4. It shows that

$$\frac{\partial}{\partial t} \|D\phi^\theta(\phi(y, t_0, s), s, t))[: d]\|_2$$

$$\leq \|\frac{\partial}{\partial t} D\phi^\theta(\phi(y, t_0, s), s, t))[: d]\|_2 \qquad \text{(By triangle inequality)}$$

$$= \|Du^{1,\theta}(\psi^\theta(\psi(x, t_0, s), s, t) D\phi^\theta(\phi(y, t_0, s), s, t))[: d])\|_2 \qquad \text{(By chain rule)}$$

$$\leq L_\mathcal{T} \|D\phi^\theta(\phi(y, t_0, s), s, t))[: d])\|_2. \qquad \text{(By Lipschitz constant of transformer)}$$

Therefore,

$$\|D\phi^\theta(\phi(y,t_0,s),s,t))[:d])\|_2 \lesssim \exp\{\int_s^t L_\mathcal{T}\mathrm{d}u\} \leq \exp\{\int_0^1 L_\mathcal{T}\mathrm{d}u\} =: M. \quad \text{(By Lemma M.1)}$$

Now we have

$$\|\psi^\theta(x,t_0,T) - \psi(x,t_0,T)\|_2^2$$

$$\leq M^2 \int_{t_0}^T (f^\theta(\phi(y,t_0,s),s) - f(\phi(y,t_0,s),s))^2 \mathrm{d}s \qquad \text{(By Lemma M.1)}$$

$$= M^2(\int_{t_0}^T (\sum_{k=1}^K \|u^{k,\theta}(\psi(x,t_0,s),s) - u^k(\psi(x,t_0,s),s)\|_2)\mathrm{d}s)^2 \qquad \text{(By definition of } f^\theta)$$

$$\leq M^2 \int_{t_0}^T (\sum_{k=1}^K \|u^{k,\theta}(\psi(x,t_0,s),s) - u^k(\psi(x,t_0,s),s)\|_2^2)\mathrm{d}s. \quad \text{(By Cauchy Schwarz inequality)}$$

Then, we take expectation with $x \sim p_{t_0}^0$ on both sides

$$\mathop{\mathbb{E}}_{x\sim p_{t_0}^0}[\|\psi^\theta(x,t_0,T) - \psi(x,t_0,T)\|_2^2]$$

$$\leq M^2 \sum_{k=1}^K \mathop{\mathbb{E}}_{x\sim p_{t_0}^0}[\int_{t_0}^T \|u^{k,\theta}(\psi(x,t_0,s),s) - u^k(\psi(x,t_0,s),s)\|_2^2\mathrm{d}s]$$

$$= M^2(T-t_0)\mathcal{R}_K(\Theta). \qquad \text{(By definition of higher order risk in Definition 4.2)}$$

Finally, we bound the 2-Wasserstein distance between the estimated and true distributions following Appendix M. By using the definition of the 2-Wasserstein metric, it follows that

$$W_2(\widehat{P}_T^K, P_T^K) \leq \sqrt{\mathop{\mathbb{E}}_{x\sim p_{t_0}}[\|\psi^\theta(x,t_0,T) - \psi(x,t_0,T)\|_2^2]} \lesssim \sqrt{\mathcal{R}_K(\Theta)}$$

Then,

$$\mathop{\mathbb{E}}_{\{x_i\}_{i=1}^n}[W_2(\widehat{P}_T^K, P_T^K)] \lesssim \sqrt{\mathcal{R}_K(\widehat{\Theta})}$$

We apply the high-order velocity estimation results in Theorem 4.2

$$\mathop{\mathbb{E}}_{\{x_i\}_{i=1}^n}[\mathcal{R}_K(\widehat{\Theta})] = O\Big(n^{-\frac{1}{8d+9}} \cdot (\log n)^{8d_x+11}\Big).$$

This implies

$$\mathop{\mathbb{E}}_{\{x_i\}_{i=1}^n}[W_2(\widehat{P}_T^K, P_T^K)] \lesssim \mathop{\mathbb{E}}_{\{x_i\}_{i=1}^n}[\sqrt{\mathcal{R}_K(\Theta)}] = O\Big(n^{-\frac{1}{16d+18}} \cdot (\log n)^{4d_x+6}\Big).$$

This completes the proof. □

# G  Proof of Theorem 4.4

Recall the Hölder density function class and its minimax optimal rate under 2-Wasserstein distance:

**Lemma G.1** (Lemma N.2 Restated: Modified from Theorem 3 of [Niles-Weed and Berthet, 2022]).
Consider the task of estimating a probability distribution $P(x_1)$ with density function belonging to the space

$$\mathcal{P} := \big\{ q(x_1) \mid q(x_1) \in \mathcal{H}^\beta([-1,1]^{d_x}, B), q(x_1) \geq C \big\},$$

Then, for any $r \geq 1$, $\beta > 0$ and $d_x > 2$, we have

$$\inf_{\widehat{P}} \sup_{q(x_1) \in \mathcal{P}} \mathbb{E}_{\{x_i\}_{i=1}^n} [W_r(\widehat{P}, P)] \gtrsim n^{-\frac{\beta+1}{d_x+2\beta}},$$

where $\{x_i\}_{i=1}^n$ is a set of i.i.d samples drawn from distribution $P$, and $\widehat{P}$ runs over all possible estimators constructed from the data.

We now give the formal proof of Theorem 4.4.

**Theorem G.1** (Theorem 4.4 Restated: Minimax Optimality of High-Order Flow Matching Transformers).  Assume that the target density function satisfies $q(x_1) \in \mathcal{H}^\beta([-1,1]^{d_x}, B)$ and $q(x_1) \geq C$ for some constant $C$. Then, under the setting of $18d(\beta+1) = d_x + 2\beta$, the distribution estimation rate of flow matching transformers presented in Theorem 4.3 matches the minimax lower bound of Hölder distribution class in 2-Wasserstein distance up to a $\log n$ and Lipchitz constants factors.

*Proof of Theorem 4.4.* Since the bounded support $[-1,1]^{d_x}$ guarantees the sub-Gaussian property in Assumption I.1, the distribution estimation Theorem 4.3 holds under $q(x_1) \in \mathcal{H}^\beta([-1,1]^{d_x}, B)$:

$$\mathbb{E}_{\{x_i\}_{i=1}^n} [W_2(\widehat{P}_T, P_T)] \lesssim O\left( n^{-\frac{1}{18d}} \cdot (\log n)^{6d_x} \right).$$

Then, by Lemma N.2, the distribution rates matches the minimax lower bound up to a $\log n$ and Lipschitz constant factors under the setting

$$18d(\beta+1) = d_x + 2\beta.$$

This completes the proof. $\qquad\qquad\qquad\qquad\qquad\qquad\qquad\qquad\qquad\qquad\qquad\square$

**Remark G.1.**  Since $d_x = d \cdot L$, the condition $18d(\beta+1) = d_x + 2\beta$ implies

$$d(18\beta + 18 - L) = 2\beta \quad \text{and} \quad \beta(18d - 2) = d(L - 18),$$

the transformesr achieve minimax optimal rate with reshape layer such that $18 \leq L \leq 18\beta + 18$.

# H   Preliminaries: Universal Approximation of Transformers

Prior works [Hu et al., 2025b, 2024, Kajitsuka and Sato, 2023, Yun et al., 2019] develop the universal approximation of transformers for continuous functions. Here we revisit these methods to establish a foundation for our analysis of $k$-th order flow matching transformers. Specifically, we revisit (i) the ability of the transformer function class (defined in Appendix B) to approximate any continuous function on a compact domain with arbitrary error, (ii) the parameter norm bounds required to achieve the universal approximation. Notably, controlling the magnitude of these norm bounds is essential for subsequent analysis on the velocity estimation error and distribution error.

**Background: Contextual Mapping.**   Recall the reshape layer Definition B.3. Let $Z \in \mathbb{R}^{d \times L}$ represent input embeddings. where $Z_{:,k} \in \mathbb{R}^d$ denotes the $k$-th token (column) of each $Z$ sequence. Further, given $M$ embeddings $Z^{(1)}, \ldots, Z^{(M)} \in \mathbb{R}^{d \times L}$, we say $Z^{(i)}$ is the $i$-th sequence for $i \in [M]$. Then, we define the *vocabulary* corresponding to the $i$-th sequence at the $k$-th index in Definition H.1.

**Definition H.1** (Vocabulary).   We define the $i$-th vocabulary set for $i \in [M]$ by $\mathcal{V}^{(i)} = \bigcup_{k \in [L]} Z_{:,k}^{(i)} \subset \mathbb{R}^d$, and the whole vocabulary set $\mathcal{V}$ is defined by $\mathcal{V} = \bigcup_{i \in [N]} \mathcal{V}^{(i)} \subset \mathbb{R}^d$.

In line with prior works [Hu et al., 2025b, 2024, Kajitsuka and Sato, 2023, Kim et al., 2022, Yun et al., 2019], we assume the embeddings separateness to be $(\gamma_{\min}, \gamma_{\max}, \delta)$-separated,

**Definition H.2** (Tokenwise Separateness).   Let $Z^{(1)}, \ldots, Z^{(M)} \in \mathbb{R}^{d \times L}$ be embeddings. Then, we say $Z^{(1)}, \ldots, Z^{(M)}$ are tokenwise $(\gamma_{\min}, \gamma_{\max}, \delta)$-separated if the following three conditions hold.

1. For any $i \in [M]$ and $k \in [n]$, $\|Z_{:,k}^{(i)}\| > \gamma_{\min}$ holds.

2. For any $i \in [M]$ and $k \in [n]$, $\|Z_{:,k}^{(i)}\| < \gamma_{\max}$ holds.

3. For any $i, j \in [M]$ and $k, l \in [n]$ if $Z_{:,k}^{(i)} \neq Z_{:,l}^{(j)}$, then $\|Z_{:,k}^{(i)} - Z_{:,l}^{(j)}\| > \delta$ holds.

Further, we say $Z^{(1)}, \ldots, Z^{(N)}$ is $(\gamma, \delta)$-separated when only conditions (ii) and (iii) hold. Also, if only condition (iii) holds, we denote it as $(\delta)$-separateness.

Building on the token separateness, we introduce the *contextual mapping*, that characterizes the ability of transformers' self-attention to capture the relationships among tokens across different sequences. This allows transformers to utilize self-attention for full context representation.

**Definition H.3** (Contextual Mapping).   Let $Z^{(1)}, \ldots, Z^{(M)} \in \mathbb{R}^{d \times L}$ be embeddings. Then, we say a map $T : \mathbb{R}^{d \times L} \to \mathbb{R}^{d \times L}$ is a $(\gamma, \delta)$-contextual mapping if the following two conditions hold:

1. For any $i \in [M]$ and $k \in [L]$, it holds

$$\|T(Z^{(i)})_{:,k}\| < \gamma.$$

2. For any $i, j \in [M]$ and $k, l \in [L]$ such that $\mathcal{V}^{(i)} \neq \mathcal{V}^{(j)}$ or $Z_{:,k}^{(i)} \neq Z_{:,l}^{(j)}$, it holds

$$\|T(Z^{(i)})_{:,k} - T(Z^{(j)})_{:,l}\| > \delta.$$

We introduce results from [Hu et al., 2025b] in Theorem H.1, which shows that a one-layer single-head attention mechanism is a contextual mapping.

**Helper Lemmas.**   Before presenting Theorem H.1, we restate several helper lemmas from [Hu et al., 2025b, Kajitsuka and Sato, 2023] to simplify the proof.

**Lemma H.1** (Boltz Preserves Distance, Lemma 1 of [Kajitsuka and Sato, 2023]).   Given $(\gamma, \delta)$-tokenwise separated vectors $z^{(1)}, \ldots, z^{(N)} \in \mathbb{R}^n$ with no duplicate entries in each vector:

$$z_s^{(i)} \neq z_t^{(i)},$$

where $i \in [N]$ and $s, t \in [L], s \neq t$. Also, let

$$\delta \geq 4 \ln n.$$

Then, the outputs of the Boltzmann operator has the following properties:

$$\left| \text{Boltz}\left(z^{(i)}\right) \right| \leq \gamma,$$

$$\left| \text{Boltz}\left(z^{(i)}\right) - \text{Boltz}\left(z^{(j)}\right) \right| > \delta' = \ln^2(n) \cdot e^{-2\gamma}$$

for all $i, j \in [N], i \neq j$.

We remark that Lemma H.1 gives the key step for constructing a contextual mapping via the self-attention layer Lemma H.4. We extend the results to the case with inverse-temperature scaling.

**Lemma H.2** (Boltz Preserves Distance with Finite Inverse-Temperature Scaling in Softmax). Let $z^{(1)}, \ldots, z^{(M)} \in \mathbb{R}^n$ be $(\gamma, \delta)$-tokenwise separated vectors with no duplicate entries in each vector:

$$z_s^{(i)} \neq z_t^{(i)},$$

where $n \geq 2, \delta \geq 4 \ln n / \lambda$ for some $\lambda > 0, i \in [M]$ and $s, t \in [L], s \neq t$. Further, we define

$$\text{Boltz}_\lambda(z^{(i)}) := z^{(i)\top} \text{Softmax}_\lambda(z^{(i)}) \quad \text{for all} \quad i \in [M],$$

where $\text{Softmax}_\lambda(z^{(i)})[k] = \exp\left(\lambda z^{(i)}[k]\right) / \sum_j \exp\left(\lambda z^{(i)}[j]\right)$ is the softmax function with inverse-temperature scaling $\lambda > 0$. Then, the outputs of the Boltzmann operator has the following properties:

$$\left| \text{Boltz}_\lambda\left(z^{(i)}\right) \right| \leq \gamma, \tag{H.1}$$

$$\left| \text{Boltz}_\lambda\left(z^{(i)}\right) - \text{Boltz}_\lambda\left(z^{(j)}\right) \right| > \delta' = \frac{\ln^2(n) \cdot e^{-2\lambda\gamma}}{\lambda} \tag{H.2}$$

for all $i, j \in [M], i \neq j$.

*Proof.* We first scale the input vectors by $y^{(i)}[j] := z^{(i)}[j] \cdot \lambda$ for all $i \in [M]$ and all $j \in [n]$. This gives a $(R, \Delta)$-tokenwise separated vectors $y^{(1)}, \ldots, y^{(M)} \in \mathbb{R}^n$, where $R = \lambda\gamma$ and $\Delta = \lambda\delta$.

Next, notice that $\text{softmax}_\lambda(\cdot) = \text{softmax}(\lambda\cdot)$ and $\text{Boltz}_\lambda(\cdot) = \text{Boltz}(\lambda\cdot)/\lambda$. Therefore, since the condition $\delta \geq 4 \ln n / \lambda$ ensures $\Delta \geq 4 \ln n$, Lemma H.1 implies both (H.1) and (H.2).

This completes the proof. $\qquad \square$

**Lemma H.3** (Lemma 13 of [Park et al., 2021]). For any finite subset $\mathcal{X} \subset \mathbb{R}^d$, there exists at least one unit vector $u \in \mathbb{R}^d$ such that

$$\frac{1}{|\mathcal{X}|^2} \sqrt{\frac{8}{\pi d}} \|x - x'\| \leq \left| u^\top (x - x') \right| \leq \|x - x'\|,$$

for any $x, x' \in \mathcal{X}$.

Lemma H.3 shows the existence of a unit vector $u \in \mathbb{R}^d$ that bounds the inner product of the difference between points in a finite subset $\mathcal{X} \subset \mathbb{R}^d$. Next, we restate the construction of rank-$\rho$ weight matrices in a self-attention layer following [Hu et al., 2025b] in Lemma H.4.

**Lemma H.4** (Construction of Weight Matrices, Lemma D.2 of [Hu et al., 2025b]). Given $(\gamma_{\min}, \gamma_{\max}, \epsilon)$-separated input embeddings $Z^{(1)}, \ldots, Z^{(M)} \in \mathbb{R}^{d \times L}$ with finite vocabulary set

$\mathcal{V} \subset \mathbb{R}^d$. There exists rank-$\rho$ weight matrices $W_K, W_Q \in \mathbb{R}^{s \times d}$ such that

$$\left| (W_K v_a)^\top (W_Q v_c) - (W_K v_b)^\top (W_Q v_c) \right| > \delta,$$

for any $\delta > 0$, any $\min(d, s) \geq \rho \geq 1$ and any $v_a, v_b, v_c \in \mathcal{V}$ with $v_a \neq v_b$. In addition, the matrices are constructed as

$$W_K = \sum_{i=1}^{\rho} p_i q_i^\top \in \mathbb{R}^{s \times d}, \quad W_Q = \sum_{j=1}^{\rho} p_j' q_j'^\top \in \mathbb{R}^{s \times d},$$

where $q_i, q_i' \in \mathbb{R}^d$ are unit vectors that satisfy Lemma H.3 for at least one $i$, and $p_i, p_i' \in \mathbb{R}^s$ satisfies

$$\left| p_i^\top p_i' \right| = 5 \left( |\mathcal{V}| + 1 \right)^4 d \frac{\delta}{\epsilon \gamma_{\min}}.$$

**Any-Rank Attention is Contextual Mapping.** The next lemma shows that the any rank self-attention mechanisms of transformers serve as contextual mappings (Definition H.3).

**Theorem H.1** (Any-Rank Attention is $(\gamma, \delta)$-Contextual Mapping, Lemma 2.2 of [Hu et al., 2025b])**.** Consider $(\gamma_{\min}, \gamma_{\max}, \epsilon)$-tokenwise separated embeddings $Z^{(1)}, \ldots, Z^{(M)} \in \mathbb{R}^{d \times L}$ and vocabulary set $\mathcal{V} = \bigcup_{i \in [N]} \mathcal{V}^{(i)} \subset \mathbb{R}^d$. Let $Z^{(1)}, \ldots, Z^{(N)} \in \mathbb{R}^{d \times L}$ be embedding sequences with no duplicate word token in each sequence; that is, $Z_{:,k}^{(i)} \neq Z_{:,l}^{(i)}$, for any $i \in [M]$ and $k, l \in [L]$. Then, there exists a 1-layer single head attention with weight matrices $W_O \in \mathbb{R}^{d \times s}$ and $W_V, W_K, W_Q \in \mathbb{R}^{s \times d}$, that is a $(\gamma, \delta)$-contextual mapping for embeddings $Z^{(1)}, \ldots, Z^{(M)}$ with

$$\gamma = \gamma_{\max} + \epsilon/4, \quad \delta = \exp\left(-5\lambda\epsilon^{-1}|\mathcal{V}|^4 d\kappa\gamma_{\max} \log L\right)/\lambda,$$

where $\kappa = \gamma_{\max}/\gamma_{\min}$ and $\lambda$ is the inverse-temperature scaling in the column-wise $\mathrm{Softmax}$ function.

We restate the proof of Theorem H.1 since it is crucial for subsequent analysis.

*Proof.* For completeness, we restate the proof from Lemma 2.2 of [Hu et al., 2025c].

The proof consists of two steps:

- **Construct the Softmax Attention.** We ensure that different input tokens are mapped to unique contextual embeddings by configuring the weight matrices in Lemma H.4.

- **Handle Identical Tokens in Different Contexts.** We show that the construction from Lemma H.4 are able to handle identical tokens in different contexts by applying Lemma H.2.

We proceed the proof with these two steps.

**Step 1: Attention Construction.** We show the construction of matrices: $W_K, W_Q, W_O$ and $W_V$.

- **Weight Matrices $W_K$ and $W_Q$.** First, we construct $W_K$ and $W_Q$ by:

$$W_K = \sum_{i=1}^{\rho} p_i q_i^\top \in \mathbb{R}^{s \times d}; \quad W_Q = \sum_{j=1}^{\rho} p_j' q_j'^\top \in \mathbb{R}^{s \times d},$$

where $p_i, p_j' \in \mathbb{R}^s$ and $q_i, q_j' \in \mathbb{R}^d$. In addition, let $\delta = 4 \ln n$ and $p_1, p_1' \in \mathbb{R}^s$ be an arbitrary vector pair that satisfies

$$\left| p_1^\top p_1' \right| = \left( |\mathcal{V}| + 1 \right)^4 d \frac{\delta}{\epsilon \gamma_{\min}}. \tag{H.3}$$

- **Weight Matrices $W_V$ and $W_O$.** Next, we construct $W_O \in \mathbb{R}^{d \times s}$ and $W_V \in \mathbb{R}^{s \times d}$ by:

$$W_V = \sum_{i=1}^{\rho} p_i'' q_i''^{\top} \in \mathbb{R}^{s \times d}, \tag{H.4}$$

where $q'' \in \mathbb{R}^d$, $q_1'' = q_1$ and $p_i'' \in \mathbb{R}^s$ is some nonzero vector that satisfies

$$\|W_O p_i''\| = \frac{\epsilon}{4\rho\gamma_{\max}}. \tag{H.5}$$

This can be accomplished, e.g., $W_O = \sum_{i=1}^{\rho} p_i''' p_i''^{\top}$ for any vector $p_i'''$ which satisfies $\|p_i'''\| = \epsilon/(4\rho^2\gamma_{\max}\|p_i''\|^2)$ for any $i \in [\rho]$.

For simplicity, we define $s_{k'}^k := \text{Softmax}\left[\left(W_K Z^{(i)}\right)^{\top}\left(W_Q Z_{:,k}^{(i)}\right)\right]_{k'}$.

Then, we combine the above weights construction and obtain

$$\|W_O\left(W_V Z^{(i)}\right)\text{Softmax}\left[\left(W_K Z^{(i)}\right)^{\top}\left(W_Q Z_{:,k}^{(i)}\right)\right]\| \tag{H.6}$$

$$= \|\sum_{k'=1}^{L} s_{k'}^k W_O\left(W_V Z^{(i)}\right)\| \qquad \left(\text{By the definition of } s_{k'}^k\right)$$

$$\leq \sum_{k'=1}^{L} \|s_{k'}^k W_O\left(W_V Z^{(i)}\right)\| \qquad \left(\text{By triangle inequality}\right)$$

$$\leq \max_{k' \in [L]} \|W_O\left(W_V Z^{(i)}\right)\| \qquad \left(\text{By } \sum_{k'=1}^{L} s_{k'}^k = 1\right)$$

$$\leq \frac{\epsilon}{4\gamma_{\max}} \cdot \max_{k' \in [L]} \left|q^{\top} Z_{:,k'}^{(i)}\right| \qquad \left(\text{By (H.4) and (H.5)}\right)$$

$$\leq \frac{\epsilon}{4\gamma_{\max}} \cdot \max_{k' \in [L]} \left|Z_{:,k'}^{(i)}\right| \qquad \left(\text{By Lemma H.3}\right)$$

$$\leq \frac{\epsilon}{4}. \qquad \left(\text{By the } (\gamma_{\min}, \gamma_{\max}, \epsilon) \text{ separateness}\right)$$

for $i \in [M]$ and $k \in [L]$.

**Step 2: The Case of Identical Tokens in Different Contexts.** For the second part, we show that with the constructed weight matrices $W_O, W_V, W_K, W_Q$, the attention layer distinguishes duplicate input tokens with different context, $Z_{:,k}^{(i)} = Z_{:,l}^{(j)}$ with different vocabulary sets $\mathcal{V}^{(i)} \neq \mathcal{V}^{(j)}$.

We define $a^{(i)}, a^{(j)}$ as

$$a^{(i)} = \left(W_K Z^{(i)}\right)^{\top}\left(W_Q Z_{:,k}^{(i)}\right) \in \mathbb{R}^n, \quad a^{(j)} = \left(W_K Z^{(j)}\right)^{\top}\left(W_Q Z_{:,l}^{(j)}\right) \in \mathbb{R}^n,$$

where $a^{(i)}$ and $a^{(j)}$ are tokenwise $(\gamma, \delta)$-separated. Specifically, the following inequality holds

$$|a_{k'}^{(i)}| \leq (|\mathcal{V}| + 1)^4 d\frac{\delta}{\epsilon\gamma_{\min}}\gamma_{\max}^2.$$

Since $\mathcal{V}^{(i)} \neq \mathcal{V}^{(j)}$ and there is no duplicate token in $Z^{(i)}$ and $Z^{(j)}$, we use Lemma H.1 and obtain

$$\left|\text{Boltz}\left(a^{(i)}\right) - \text{Boltz}\left(a^{(j)}\right)\right|$$

$$= \left|\left(a^{(i)}\right)^{\top}\text{Softmax}\left[a^{(i)}\right] - \left(a^{(j)}\right)^{\top}\text{Softmax}\left[a^{(j)}\right]\right| \tag{H.7}$$

$$> \delta'$$
$$= (\ln n)^2 e^{-2\gamma}.$$

Additionally, using Lemma H.4 and (H.3), and assuming $Z_{:,k}^{(i)} = Z_{:,l}^{(j)}$, we have

$$\left| \left( a^{(i)} \right)^\top \mathrm{Softmax}\left[ a^{(i)} \right] - \left( a^{(j)} \right)^\top \mathrm{Softmax}\left[ a^{(j)} \right] \right| \tag{H.8}$$
$$\leq \sum_{i=1}^{\rho} \gamma_{\max} \cdot (|\mathcal{V}| + 1)^4 \frac{\pi d}{8} \frac{\delta}{\epsilon \gamma_{\min}} \cdot \left| \left( q_i^\top Z^{(i)} \right) \mathrm{Softmax}\left[ a^{(i)} \right] - \left( q_i^\top Z^{(j)} \right) \mathrm{Softmax}\left[ a^{(j)} \right] \right|.$$

By combining (H.7) and (H.8), we have

$$\sum_{i=1}^{\rho} \left| \left( q_i^\top Z^{(i)} \right) \mathrm{Softmax}\left[ a^{(i)} \right] - \left( q_i^\top Z^{(j)} \right) \mathrm{Softmax}\left[ a^{(j)} \right] \right| > \frac{\delta'}{(|\mathcal{V}| + 1)^4} \frac{\epsilon \gamma_{\min}}{d \delta \gamma_{\max}}. \tag{H.9}$$

Using (H.5) and (H.9). we derive the lower bound of the difference between the self-attention outputs of $Z^{(i)}, Z^{(j)}$ as follows:

$$\| \mathcal{F}_S^{(\mathrm{SA})}\left( Z^{(i)} \right)_{:,k} - \mathcal{F}_S^{(\mathrm{SA})}\left( Z^{(j)} \right)_{:,l} \| > \frac{\epsilon}{4\gamma_{\max}} \frac{\delta'}{(|\mathcal{V}| + 1)^4} \frac{\epsilon \gamma_{\min}}{d \delta \gamma_{\max}},$$

where $\delta = 4 \ln L$ and $\delta' = \ln^2(L) e^{-2\gamma}$ with $\gamma = (|\mathcal{V}| + 1)^4 d \delta \gamma_{\max}^2 / (\epsilon \gamma_{\min})$. Finally, we extend the bound to softmax function with inverse-temperature scaling $\lambda > 0$ using Lemma H.2.

This completes the proof. □

Notably, Theorem H.1 shows that, for identical embeddings $Z_{:,k}^{(i)} = Z_{:,l}^{(j)}$ with distinct vocabularies $\mathcal{V}^{(i)} \neq \mathcal{V}^{(j)}$, any-rank self-attention is able to distinguish two identical tokens in distinct contexts.

**Universal Approximation of Transformer.** We introduce the universal approximation result for transformers with a single self-attention layer from [Hu et al., 2025b, Kajitsuka and Sato, 2023].

**Theorem H.2** (Transformer Universal Approximation, Theorem B.1 of [Hu et al., 2025b] and Proposition 1 of [Kajitsuka and Sato, 2023]). Let $\epsilon \in (0, 1)$ and $p \in [1, \infty)$. Let $\mathcal{F}_1^{(\mathrm{FF})}, \mathcal{F}_2^{(\mathrm{FF})}$ be two feed-forward layers and $\mathcal{F}^{(\mathrm{SA})}$ be a single-head self-attention layer with softmax function defined in (B.1) and (B.2). Then, for any permutation equivariant continuous function $f$ on a compact domain and any $\epsilon$, there exists a $g(Z) = \mathcal{F}_2^{(\mathrm{FF})} \circ \mathcal{F}^{(\mathrm{SA})} \circ \mathcal{F}_1^{(\mathrm{FF})}(Z) \in \mathcal{T}_R^{h,s,r}$ such that $d_p(f(Z), g(Z)) < \epsilon$, where $d_p := (\int \| f(Z) - g(Z) \|_p^p \mathrm{d}Z)^{1/p}$ and $\| \cdot \|_p$ is the element-wise $\ell_p$-norm.

*Proof.* Since the universal approximation of transformer over any bounded domain differs only by scaling and shifting the transformer's parameters in $\mathcal{F}_1^{(\mathrm{FF})}$ and $\mathcal{F}_2^{(\mathrm{FF})}$, Hu et al. [2025b], Kajitsuka and Sato [2023] prove Theorem H.2 assuming that the target function $f$ is normalized on domain $[0, 1]^{d \times L}$ for simplicity. To support subsequent derivations of transformer parameter bounds required for $\epsilon$-precision (Lemma H.5), we provide the proof on a more general bounded domains.

The proof consists of three steps: (i) Quantization by the First Feed-Forward Layer (ii) Contextual Mapping by the Self-Attention Layer (iii) Memorization by the Second Feed-Forward Layer.

Let $\Omega := [-I, I]^{d \times L}$ be the domain of $f$. Without loss of generality, we consider $I \in \mathbb{N}$.

**First Step Quantization.** First, we define a grid $\mathbb{G}_D$:

$$\mathbb{G}_D := \left\{ C \in \Omega \mid C_{t,k} = -I + \frac{s_{t,k}}{D}, \ s_{t,k} = 1, \ldots, 2ID \ \text{for all} \ t \in [d], k \in [L] \right\}, \tag{H.10}$$

where $D \in \mathbb{N}$ is the granularity of $\mathbb{G}_D$.

Then, for $Z \in \Omega$, we construct a piece-wise constant function approximator:

$$g_1(Z) \coloneqq \sum_{C \in \mathbb{G}_D} f(C) \mathbb{1}\{Z \in C + [-1/D, 0)^{d \times L}\} \qquad \text{(H.11)}$$

By the uniform continuity of $f$, for any $\epsilon > 0$, there exist a $D$ such that

$$d_p(f(Z), g_1(Z)) < \epsilon/3. \qquad \text{(H.12)}$$

We use a transformer to approximate $g_1(Z)$ using two feed forward layers and a self-attention layer. First, we introduce the *quantization function* that discretizes the input into $\mathbb{G}_D$:

**Quantization Function.** We define the quantization function $\text{quant}_D : \mathbb{R} \to \mathbb{R}$:

$$\text{quant}_D(z) \coloneqq \begin{cases} -I & z < -I \\ -I + 1/D & -I \le z < -I + 1/D \\ \vdots & \vdots \\ I & I - 1/D \le z. \end{cases}$$

Within $\Omega$, $\text{quant}_D^{d \times L}(z)$ outputs regions identical to $C + [-1/D, 0)^{d \times L}$ defined by the indicator function in (H.11). We extend the quantization to $\text{quant}_D^{d \times L}(Z) : \mathbb{R}^{d \times L} \to \mathbb{R}^{d \times L}$, where $\text{quant}_D(z)$ is applied coordinate-wise. Then, we approximate $\text{quant}_D(z)$ with the following network[4]:

$$f_1(z) \coloneqq -I + \sum_{t=-ID}^{ID-1} \frac{\text{ReLU}\left[z/\delta - t/\delta D\right] - \text{ReLU}\left[z/\delta - 1 - t/\delta D\right]}{D} \approx \text{quant}_D(z), \quad \text{(H.13)}$$

where $\delta > 0$ determines the steepness of the change from one quantized level to the next. For $z \in \mathbb{R} \setminus [-I, I]$, we add and subtract the first and last step functions scaled by $I$ to obtain zero output:

$$\begin{aligned} f_1^{\text{FF}}(z) \coloneqq f_1(z) - I \cdot \Big( \text{ReLU}\left[z/\delta - I/\delta\right] - \text{ReLU}\left[z/\delta - 1 - I/\delta\right] \Big) \\ + I \cdot \Big( \text{ReLU}\left[-z/\delta - I/\delta\right] - \text{ReLU}\left[-z/\delta - 1 - I/\delta\right] \Big), \qquad \text{(H.14)} \end{aligned}$$

where $f_1^{\text{FF}}(z)$ approximately quanitzes $[-I, I]$ into $\{-I + 1/D, \dots, I\}$ and project $\mathbb{R} \setminus [-I, I]$ to $0$ by taking sufficiently small $\delta$. Since every element shares identical weights, we realize (H.14) using $W_2' \text{ReLU}\left[W_1' Z + b_1'\right] + b_2'$ by constructing $W_1' \in \mathbb{R}^{L \cdot (4ID+4) \times d}$ and $b_1' \in \mathbb{R}^{L \cdot (4ID+4)}$ as:

$$W_1' \coloneqq \begin{pmatrix} W_1^{(1)} \\ W_1^{(2)} \\ \vdots \\ W_1^{(L)} \end{pmatrix}, \quad b_1' \coloneqq \begin{pmatrix} b_1^{(1)} \\ b_1^{(2)} \\ \vdots \\ b_1^{(L)} \end{pmatrix}, \quad \text{where} \quad b_1^{(i)} = \begin{pmatrix} \frac{I}{\delta} \\ \frac{I}{\delta} - 1 \\ \vdots \\ -\frac{I}{\delta} + \frac{1}{D} \\ -\frac{I}{\delta} - 1 + \frac{1}{D} \\ -\frac{I}{\delta} \\ -\frac{I}{\delta} - 1 \\ -\frac{I}{\delta} \\ -\frac{I}{\delta} - 1 \end{pmatrix} \quad \text{for all } i \in [L], \quad \text{(H.15)}$$

---

[4]This is by the shifting and stacking step functions from [Yun et al., 2019].

and $W_1^{(1)}, \ldots, W_1^{(L)} \in \mathbb{R}^{(4ID+4) \times d}$ take the form:

$$
W_1^{(1)} = \begin{pmatrix} 1/\delta & 0 & 0 & \cdots & 0 \\ 1/\delta & 0 & 0 & \cdots & 0 \\ \vdots & \vdots & \vdots & \ddots & \vdots \\ 1/\delta & 0 & 0 & \cdots & 0 \\ 1/\delta & 0 & 0 & \cdots & 0 \\ 1/\delta & 0 & 0 & \cdots & 0 \\ 1/\delta & 0 & 0 & \cdots & 0 \\ -1/\delta & 0 & 0 & \cdots & 0 \\ -1/\delta & 0 & 0 & \cdots & 0 \end{pmatrix}, W_1^{(2)} = \begin{pmatrix} 0 & 1/\delta & 0 & \cdots & 0 \\ 0 & 1/\delta & 0 & \cdots & 0 \\ \vdots & \vdots & \vdots & \ddots & \vdots \\ 0 & 1/\delta & 0 & \cdots & 0 \\ 0 & 1/\delta & 0 & \cdots & 0 \\ 0 & 1/\delta & 0 & \cdots & 0 \\ 0 & 1/\delta & 0 & \cdots & 0 \\ 0 & -1/\delta & 0 & \cdots & 0 \\ 0 & -1/\delta & 0 & \cdots & 0 \end{pmatrix}, \cdots, W_1^{(L)} = \begin{pmatrix} 0 & 0 & 0 & \cdots & 1/\delta \\ 0 & 0 & 0 & \cdots & 1/\delta \\ \vdots & \vdots & \vdots & \ddots & \vdots \\ 0 & 0 & 0 & \cdots & 1/\delta \\ 0 & 0 & 0 & \cdots & 1/\delta \\ 0 & 0 & 0 & \cdots & 1/\delta \\ 0 & 0 & 0 & \cdots & 1/\delta \\ 0 & 0 & 0 & \cdots & -1/\delta \\ 0 & 0 & 0 & \cdots & -1/\delta \end{pmatrix}.
$$

The first $4ID$ rows of $W_1'$ and $b_1'$ implement the ReLU terms in (H.13), and the last four rows implement the ReLU terms in (H.14). Then, we construct $W_2' \in \mathbb{R}^{d \times L \cdot (4ID+4)}$ and $b_2 \in \mathbb{R}^d$ by

$$
W_2' = \begin{pmatrix} 1/D & -1/D & \cdots & 1/D & -1/D & -I & I & I & -I \\ 1/D & -1/D & \cdots & 1/D & -1/D & -I & I & I & -I \\ \vdots & \vdots & \vdots & \vdots & \vdots & \vdots & \vdots & \ddots & \vdots \\ 1/D & -1/D & \cdots & 1/D & -1/D & -I & I & I & -I \end{pmatrix}, \quad b_2' = \begin{pmatrix} -I \\ -I \\ \vdots \\ -I \end{pmatrix}. \quad \text{(H.16)}
$$

Besides the quantization, we include an additional penalty function to signify the case where the inputs are not on the target domain $[-I, I]^{d \times L}$.

**Penalty Function.** We define the penalty function $\text{penalty} : \mathbb{R} \to \mathbb{R}$

$$
\text{penalty}(z) = \begin{cases} -2I & z < -I \\ 0 & z \in [-I, I] \\ -2I & z > I. \end{cases} \quad \text{(H.17)}
$$

Again, we extend the penalty function to $\text{penalty}^{d \times L}(z) : \mathbb{R}^{d \times L} \to \mathbb{R}^{d \times L}$ where $\text{penalty}(z)$ is applied coordinate-wise. By taking sufficiently small $\delta$, we approximate $\text{penalty}(z)$ by

$$
f_2^{\text{FF}}(z) \approx \text{penalty}(z),
$$

where

$$
f_2^{\text{FF}} := \underbrace{-2I\Big(\text{ReLU}\left[(z-I)/\delta\right] + \text{ReLU}\left[(z-I)/\delta - 1\right]\Big)}_{\text{Approximate step from 0 to } -2I \text{ at } z = I} \quad \text{(H.18)}
$$
$$
\underbrace{-2I\Big(\text{ReLU}\left[(-z-I)/\delta\right] + \text{ReLU}\left[(-z-I)/\delta - 1\right]\Big)}_{\text{Approximate step from 0 to } -2I \text{ at } z = -I}.
$$

Let $\mathcal{I} \in \mathbb{R}^{d \times L}$ be a matrix with all entries equal to $I$. Altogether, we define $g_2(Z) : \mathbb{R}^{d \times L} \to \mathbb{R}^{d \times L}$

$$
g_2(Z) = \underbrace{\text{quant}_D^{d \times L}(Z) + \mathcal{I}}_{(A)} + \underbrace{dL \cdot \text{penalty}^{d \times L}(Z)}_{(B)}. \quad \text{(H.19)}
$$

Term (A) first quantizes input $[-I, I]^{d \times L}$ into $\mathbb{G}_D$ and scales the grid to $[0, 2I]^{d \times L}$, denoted by $\mathbb{G}_D^\circ$. Term (B) ensures non-positive outputs for any $Z \in \mathbb{R}^{d \times L} \setminus [-I, I]^{d \times L}$.

Together, we approximate (H.19) using the first feed-forward block $W_2 \text{ReLU}\left[W_1 Z + b_1\right] + b_2$ with $W_1 \in \mathbb{R}^{L \cdot (4ID+8) \times d}$, $b_1 \in \mathbb{R}^{L \cdot (4ID+8)}$, $W_2 \in \mathbb{R}^{d \times L \cdot (4ID+8)}$ and $b_2 \in \mathbb{R}^d$, where we stack $4L$ rows to $W_1', b_1'$ to implement the penalty function, scale weights in $W_2'$, and set $b_2$ as a zero vector.

**Second Step Contextual Mapping.** Let $\widetilde{\mathbb{G}}_D \subseteq \mathbb{G}_D^\circ$ denote the sub-grid on $[0, 2I]^{d \times L}$:

$$\widetilde{\mathbb{G}}_D := \left\{ G \in \mathbb{G}_D^\circ \mid G_{:,k} \neq G_{:,l} \ \text{ for all } \ k, l \in [L] \right\}. \tag{H.20}$$

By the construction of $\mathbb{G}_D^\circ$, the sub-grid $\widetilde{\mathbb{G}}_D$ is a collection of grids with pairwise distinct tokens, and every $G \in \widetilde{\mathbb{G}}_D$ is a token-wise $(1/D, 2I\sqrt{d}, 1/D)$-separated sequences.

From the construction of $\mathcal{F}^{(\mathrm{SA})}$ in (H.6), it holds:

$$\|\mathcal{F}^{(\mathrm{SA})}(Z)_{:,k} - Z_{:,k}\| < \frac{1}{8I\sqrt{d}D} \max_{k' \in [L]} \|Z_{:,k'}\|,$$

Recall that the magnitude of every entry in $\mathcal{F}_1^{\mathrm{FF}}(Z)$ is at most $2IdL$ by the construction of the penalty function. Therefore, for all $k' \in [L]$, it holds:

$$\max_{Z \in \mathbb{R}^{d \times L}} \|\mathcal{F}_1^{\mathrm{FF}}(Z)_{:,k'}\| \leq 2IdL \times \sqrt{d}.$$

This implies

$$\|\mathcal{F}^{(\mathrm{SA})}(Z)_{:,k} \circ \mathcal{F}_1^{\mathrm{FF}}(Z)_{:,k} - \mathcal{F}_1^{\mathrm{FF}}(Z)_{:,k}\| < \frac{dL}{4D}$$

Taking sufficiently large $D$, every element of output of $Z \in \mathbb{R}^{d \times L} \setminus [-I, I]^{d \times L}$ is upper-bounded by:

$$\mathcal{F}^{(\mathrm{SA})} \circ \mathcal{F}_1^{\mathrm{FF}}(Z)_{t,k} < \frac{1}{4D} \quad \text{for all} \quad t \in [d], k \in [L].$$

Also, for input on $Z \in [0, 1]^{d \times L}$, we have lower bound:

$$\mathcal{F}^{(\mathrm{SA})} \circ \mathcal{F}_1^{\mathrm{FF}}(Z)_{t,k} > \frac{3}{4D} \quad \text{for all} \quad t \in [d], k \in [L].$$

Then, it remains to map sequences on $(3/4D, \infty)^{d \times L}$ to their corresponding target value, and map sequences on on $(-\infty, 1/4D)^{d \times L}$ to zero.

**Third Step Memorization.** For a fixed $\overline{C} \in \widetilde{\mathbb{G}}_D$ and a network input $Z$, we define $u := \mathcal{F}^{(\mathrm{SA})}(\overline{C})$ and $S := \mathcal{F}^{(\mathrm{SA})} \circ \mathcal{F}_1^{\mathrm{FF}}(Z)$. The goal is to use $\mathcal{F}_2^{\mathrm{FF}}$ to implement $g_3 : \mathbb{R}^{d \times L} \to \mathbb{R}^{d \times L}$, where

$$g_3(S)_{t,k} = f(\overline{C} - \mathcal{I})_{t,k} \cdot \mathbb{1}\left[u_{t,k} = S_{t,k}\right] \ \text{ for all } \ t \in [d], k \in [L],$$

and $f$ is the target function. We use the bump function with three-piece ReLU to achieve this:

$$\begin{aligned}
&\mathrm{bump}_R(S)_{t,k} \tag{H.21}\\
&= \mathrm{ReLU}\left[R_{\mathrm{FF}}(S_{t,k} - u_{t,k}) - 1\right] - 2\mathrm{ReLU}\left[R_{\mathrm{FF}}(S_{t,k} - u_{t,k})\right] + \mathrm{ReLU}\left[R_{\mathrm{FF}}(S_{t,k} - u_{t,k}) + 1\right],
\end{aligned}$$

where inputs with a deviation from the correct grid point $u_{t,k}$ greater than $1/R_{\mathrm{FF}}$ are mapped to zero. Therefore, by taking sufficiently large $R_{\mathrm{FF}}$, (H.21) implements $\mathbb{1}\left[u_{t,k} = S_{t,k}\right]$ exactly.

Suppose that there exist a feed-forward block $\mathcal{F}_2^{\mathrm{FF}}$ such that (H.21) holds for all $t \in [d], k \in [L]$. By choosing the granularity $D$ sufficiently large such that $\left|\mathbb{G}_D \setminus \widetilde{\mathbb{G}}_D\right|$ is negligible, it holds

$$d_p\left(\mathcal{F}_2^{\mathrm{FF}} \circ \mathcal{F}^{(\mathrm{SA})} \circ \mathcal{F}_1^{\mathrm{FF}}, g_1\right) < 2\epsilon/3 \tag{H.22}$$

Combining (H.12), we have

$$d_p\left(\mathcal{F}_2^{\mathrm{FF}} \circ \mathcal{F}^{(\mathrm{SA})} \circ \mathcal{F}_1^{\mathrm{FF}}, f\right) < \epsilon.$$

It remains to show that there exists a $\mathcal{F}_2^{\mathrm{FF}}$ that implements (H.21) for all $t \in [d]$, $k \in [L]$. This is possible since Theorem H.1 ensures that $(\gamma_{\min}, \gamma_{\max}, \epsilon)$-tokenwise separated sequences are at least $\delta'$ apart from one another after the self-attention mapping, where $\delta' = \exp\left(-5\lambda\epsilon^{-1}|\mathcal{V}|^4 d\gamma_{\max}^2 \log L/\gamma_{\min}\right)$, $\lambda > 0$ is the inverse-temperature constant and $|\mathcal{V}|$ is the size of the vocabulary set.

Specifically, for a fixed $\overline{C}$ and $u = \mathcal{F}^{\mathrm{SA}}(\overline{C})$, we construct a $W_1^{(i)} \in \mathbb{R}^{3dL \times L}$ and a $b_1^{(i)} \in \mathbb{R}^{3dL}$:

$$
W_1^{(i)} = \begin{pmatrix}
R_{\mathrm{FF}} & 0 & 0 & \cdots & 0 \\
R_{\mathrm{FF}} & 0 & 0 & \cdots & 0 \\
R_{\mathrm{FF}} & 0 & 0 & \cdots & 0 \\
0 & R_{\mathrm{FF}} & 0 & \cdots & 0 \\
0 & R_{\mathrm{FF}} & 0 & \cdots & 0 \\
0 & R_{\mathrm{FF}} & 0 & \cdots & 0 \\
0 & 0 & R_{\mathrm{FF}} & \cdots & 0 \\
0 & 0 & R_{\mathrm{FF}} & \cdots & 0 \\
0 & 0 & R_{\mathrm{FF}} & \cdots & 0 \\
\vdots & \vdots & \vdots & \ddots & \vdots \\
0 & 0 & 0 & \cdots & R_{\mathrm{FF}} \\
0 & 0 & 0 & \cdots & R_{\mathrm{FF}} \\
0 & 0 & 0 & \cdots & R_{\mathrm{FF}} \\
\vdots & \vdots & \vdots & \vdots & \vdots \\
\vdots & \vdots & \vdots & \ddots & \vdots \\
R_{\mathrm{FF}} & 0 & 0 & \cdots & 0 \\
R_{\mathrm{FF}} & 0 & 0 & \cdots & 0 \\
R_{\mathrm{FF}} & 0 & 0 & \cdots & 0 \\
R_{\mathrm{FF}} & 0 & \cdots & 0 & \\
0 & R_{\mathrm{FF}} & 0 & \cdots & 0 \\
0 & R_{\mathrm{FF}} & 0 & \cdots & 0 \\
0 & 0 & R_{\mathrm{FF}} & \cdots & 0 \\
0 & 0 & R_{\mathrm{FF}} & \cdots & 0 \\
0 & 0 & R_{\mathrm{FF}} & \cdots & 0 \\
\vdots & \vdots & \vdots & \ddots & \vdots \\
0 & 0 & 0 & \cdots & R_{\mathrm{FF}} \\
0 & 0 & 0 & \cdots & R_{\mathrm{FF}} \\
0 & 0 & 0 & \cdots & R_{\mathrm{FF}}
\end{pmatrix}, \quad
b_1^{(i)} = \begin{pmatrix}
-R_{\mathrm{FF}}u_{1,1} \\
-R_{\mathrm{FF}}u_{1,1} - 1 \\
-R_{\mathrm{FF}}u_{1,1} + 1 \\
-R_{\mathrm{FF}}u_{2,1} \\
-R_{\mathrm{FF}}u_{2,1} - 1 \\
-R_{\mathrm{FF}}u_{2,1} + 1 \\
-R_{\mathrm{FF}}u_{3,1} \\
-R_{\mathrm{FF}}u_{3,1} - 1 \\
-R_{\mathrm{FF}}u_{3,1} + 1 \\
\vdots \\
-R_{\mathrm{FF}}u_{d,1} \\
-R_{\mathrm{FF}}u_{d,1} - 1 \\
-R_{\mathrm{FF}}u_{d,1} + 1 \\
\vdots \\
\vdots \\
-R_{\mathrm{FF}}u_{1,L} \\
-R_{\mathrm{FF}}u_{1,L} - 1 \\
-R_{\mathrm{FF}}u_{1,L} + 1 \\
-R_{\mathrm{FF}}u_{2,L} \\
-R_{\mathrm{FF}}u_{2,L} - 1 \\
-R_{\mathrm{FF}}u_{2,L} + 1 \\
-R_{\mathrm{FF}}u_{3,L} \\
-R_{\mathrm{FF}}u_{3,L} - 1 \\
-R_{\mathrm{FF}}u_{3,L} + 1 \\
\vdots \\
-R_{\mathrm{FF}}u_{d,L} \\
-R_{\mathrm{FF}}u_{d,L} - 1 \\
-R_{\mathrm{FF}}u_{d,L} + 1
\end{pmatrix}
\tag{H.23}
$$

where the first $3L$ rows of $W_1^{(i)}$ are repeated $d$ times and $S = \mathcal{F}^{(\mathrm{SA})} \circ \mathcal{F}_1^{\mathrm{FF}}(Z)$ is mapped to $\mathbb{R}^{3dL \times L}$:

$$W_1^{(i)} S + b_1^{(i)}$$

$$
= \begin{pmatrix}
R_{\text{FF}}(S_{1,1} - u_{1,1}) & R_{\text{FF}}(S_{1,2} - u_{1,1}) & \cdots & R_{\text{FF}}(S_{1,L} - u_{1,1}) \\
R_{\text{FF}}(S_{1,1} - u_{1,1}) - 1 & R_{\text{FF}}(S_{1,2} - u_{1,1}) - 1 & \cdots & R_{\text{FF}}(S_{1,L} - u_{1,1}) - 1 \\
R_{\text{FF}}(S_{1,1} - u_{1,1}) + 1 & R_{\text{FF}}(S_{1,2} - u_{1,1}) + 1 & \cdots & R_{\text{FF}}(S_{1,L} - u_{1,1}) + 1 \\
R_{\text{FF}}(S_{2,1} - u_{2,1}) & R_{\text{FF}}(S_{2,2} - u_{2,1}) & \cdots & R_{\text{FF}}(S_{2,L} - u_{2,1}) \\
R_{\text{FF}}(S_{2,1} - u_{2,1}) - 1 & R_{\text{FF}}(S_{2,2} - u_{2,1}) - 1 & \cdots & R_{\text{FF}}(S_{2,L} - u_{2,1}) - 1 \\
R_{\text{FF}}(S_{2,1} - u_{2,1}) + 1 & R_{\text{FF}}(S_{2,2} - u_{2,1}) + 1 & \cdots & R_{\text{FF}}(S_{2,L} - u_{2,1}) + 1 \\
\vdots & \vdots & \ddots & \vdots \\
R_{\text{FF}}(S_{d,1} - u_{d,1}) & R_{\text{FF}}(S_{d,2} - u_{d,1}) & \cdots & R_{\text{FF}}(S_{d,L} - u_{d,1}) \\
R_{\text{FF}}(S_{d,1} - u_{d,1}) - 1 & R_{\text{FF}}(S_{d,2} - u_{d,1}) - 1 & \cdots & R_{\text{FF}}(S_{d,L} - u_{d,1}) - 1 \\
R_{\text{FF}}(S_{d,1} - u_{d,1}) + 1 & R_{\text{FF}}(S_{d,2} - u_{d,1}) + 1 & \cdots & R_{\text{FF}}(S_{d,L} - u_{d,1}) + 1 \\
\vdots & \vdots & \vdots & \vdots \\
\vdots & \vdots & \ddots & \vdots \\
R_{\text{FF}}(S_{1,1} - u_{1,L}) & R_{\text{FF}}(S_{1,2} - u_{1,L}) & \cdots & R_{\text{FF}}(S_{1,L} - u_{1,L}) \\
R_{\text{FF}}(S_{1,1} - u_{1,L}) - 1 & R_{\text{FF}}(S_{1,2} - u_{1,L}) - 1 & \cdots & R_{\text{FF}}(S_{1,L} - u_{1,L}) - 1 \\
R_{\text{FF}}(S_{1,1} - u_{1,L}) + 1 & R_{\text{FF}}(S_{1,2} - u_{1,L}) + 1 & \cdots & R_{\text{FF}}(S_{1,L} - u_{1,L}) + 1 \\
R_{\text{FF}}(S_{2,1} - u_{2,L}) & R_{\text{FF}}(S_{2,2} - u_{2,L}) & \cdots & R_{\text{FF}}(S_{2,L} - u_{2,L}) \\
R_{\text{FF}}(S_{2,1} - u_{2,L}) - 1 & R_{\text{FF}}(S_{2,2} - u_{2,L}) - 1 & \cdots & R_{\text{FF}}(S_{2,L} - u_{2,L}) - 1 \\
R_{\text{FF}}(S_{2,1} - u_{2,L}) + 1 & R_{\text{FF}}(S_{2,2} - u_{2,L}) + 1 & \cdots & R_{\text{FF}}(S_{2,L} - u_{2,L}) + 1 \\
\vdots & \vdots & \ddots & \vdots \\
R_{\text{FF}}(S_{d,1} - u_{d,L}) & R_{\text{FF}}(S_{d,2} - u_{d,L}) & \cdots & R_{\text{FF}}(S_{d,L} - u_{d,L}) \\
R_{\text{FF}}(S_{d,1} - u_{d,L}) - 1 & R_{\text{FF}}(S_{d,2} - u_{d,L}) - 1 & \cdots & R_{\text{FF}}(S_{d,L} - u_{d,L}) - 1 \\
R_{\text{FF}}(S_{d,1} - u_{d,L}) + 1 & R_{\text{FF}}(S_{d,2} - u_{d,L}) + 1 & \cdots & R_{\text{FF}}(S_{d,L} - u_{d,L}) + 1
\end{pmatrix}.
$$

Then, we construct the second matrix by $W_2^{(i)} := W_2^{''(i)} W_2^{'(i)} \in \mathbb{R}^{d \times 3dL}$ with $W_2^{'(i)} \in \mathbb{R}^{dL \times 3dL}$ and $W_2^{''(i)} \in \mathbb{R}^{d \times dL}$. For $W_2^{'(i)}$, we set

$$
W_2^{'(i)} = \begin{pmatrix}
-2f(\overline{C} - \mathcal{I})_{1,1} & f(\overline{C} - \mathcal{I})_{1,1} & f(\overline{C} - \mathcal{I})_{1,1} & 0 & \cdots & 0 \\
\vdots & \vdots & \vdots & \vdots & \vdots & \vdots \\
\vdots & \vdots & \vdots & \vdots & \ddots & \vdots \\
0 & \cdots & 0 & -2f(\overline{C} - \mathcal{I})_{d,L} & f(\overline{C} - \mathcal{I})_{d,L} & f(\overline{C} - \mathcal{I})_{d,L}
\end{pmatrix},
$$

which maps the output of the first layer after ReLU operation to $\mathbb{R}^{dL \times L}$:

$$
\begin{aligned}
& W_2^{'(i)} \text{ReLU}\big[W_1^{(i)} S + b_1^{(i)}\big] \\
& = \begin{pmatrix}
f(\overline{C} - \mathcal{I})_{1,1}\text{bump}(S_{1,1} - u_{1,1}) & f(\overline{C} - \mathcal{I})_{1,1}\text{bump}(S_{1,2} - u_{1,1}) & \cdots & f(\overline{C} - \mathcal{I})_{1,1}\text{bump}(S_{1,L} - u_{1,1}) \\
f(\overline{C} - \mathcal{I})_{2,1}\text{bump}(S_{2,1} - u_{2,1}) & f(\overline{C} - \mathcal{I})_{2,1}\text{bump}(S_{2,2} - u_{2,1}) & \cdots & f(\overline{C} - \mathcal{I})_{2,1}\text{bump}(S_{2,L} - u_{2,1}) \\
\vdots & & \ddots & \vdots \\
f(\overline{C} - \mathcal{I})_{d,1}\text{bump}(S_{d,1} - u_{d,1}) & f(\overline{C} - \mathcal{I})_{d,1}\text{bump}(S_{d,2} - u_{d,1}) & \cdots & f(\overline{C} - \mathcal{I})_{d,1}\text{bump}(S_{d,L} - u_{d,1}) \\
f(\overline{C} - \mathcal{I})_{1,2}\text{bump}(S_{1,1} - u_{1,2}) & f(\overline{C} - \mathcal{I})_{1,2}\text{bump}(S_{1,2} - u_{1,2}) & \cdots & f(\overline{C} - \mathcal{I})_{1,2}\text{bump}(S_{1,L} - u_{1,2}) \\
f(\overline{C} - \mathcal{I})_{2,2}\text{bump}(S_{2,1} - u_{2,2}) & f(\overline{C} - \mathcal{I})_{2,2}\text{bump}(S_{2,2} - u_{2,2}) & \cdots & f(\overline{C} - \mathcal{I})_{2,2}\text{bump}(S_{2,L} - u_{2,2}) \\
\vdots & & \ddots & \vdots \\
f(\overline{C} - \mathcal{I})_{d,2}\text{bump}(S_{d,1} - u_{d,2}) & f(\overline{C} - \mathcal{I})_{d,2}\text{bump}(S_{d,2} - u_{d,2}) & \cdots & f(\overline{C} - \mathcal{I})_{d,2}\text{bump}(S_{d,L} - u_{d,2}) \\
\vdots & \vdots & \vdots & \vdots \\
\vdots & \vdots & \ddots & \vdots \\
f(\overline{C} - \mathcal{I})_{1,L}\text{bump}(S_{1,1} - u_{1,L}) & f(\overline{C} - \mathcal{I})_{1,L}\text{bump}(S_{1,2} - u_{1,L}) & \cdots & f(\overline{C} - \mathcal{I})_{1,L}\text{bump}(S_{1,L} - u_{1,L}) \\
f(\overline{C} - \mathcal{I})_{2,L}\text{bump}(S_{2,1} - u_{2,L}) & f(\overline{C} - \mathcal{I})_{2,L}\text{bump}(S_{2,2} - u_{2,L}) & \cdots & f(\overline{C} - \mathcal{I})_{2,L}\text{bump}(S_{2,L} - u_{2,L}) \\
\vdots & & \ddots & \vdots \\
f(\overline{C} - \mathcal{I})_{d,L}\text{bump}(S_{d,1} - u_{d,L}) & f(\overline{C} - \mathcal{I})_{d,L}\text{bump}(S_{d,2} - u_{d,L}) & \cdots & f(\overline{C} - \mathcal{I})_{d,L}\text{bump}(S_{d,L} - u_{d,L})
\end{pmatrix}.
\end{aligned}
\tag{H.24}
$$

Recall that we have tokenwise $(1/D, 2I\sqrt{d}, 1/D)$-separated sequence after the quantization of the target domain $\Omega$, and $|V| \leq (2ID)^d$. Therefore, it suffices to set the scale of the bump function as

$$R_{\mathrm{FF}} := O(1/\delta') = O\left(\exp\left(640\lambda(ID)^{4d+2}d^2\log L\right)\right).$$

(H.24) is a partition of $L$ vertical blocks, each of dimension $d \times L$. Then, if $S$ is a permutation of $u$, a single column within each block is comprised of correct target output since their bump function all evaluates to one, and all other columns containing zeros. Also, The position of this all-ones column is distinct for every block. Conversely, if $S$ is not a permutation of $u$, (H.24) becomes the zero matrix.

Lastly, we construct $W_2^{''(i)}$ by $(I_d, \quad I_d, \quad \cdots \quad, I_d)$ to sum over every column in each block matrix, where the $d \times d$ identity matrix is concatenated $L$ times.

Up to permutation equivariance, there are total $q_2 = (2ID)^{dL}/L!$ possible $u$, so we stack $W_1^{(i)}, b_1^{(i)}$ and $W_2^{(ii)}$ with weights corresponding to each distinct $u$ using the identical construction.

In sum, there exist a feed-forward block $\mathcal{F}_2^{\mathrm{FF}}$ with $W_1 \in \mathbb{R}^{3dLq_2 \times d}$, a $b_1 \in \mathbb{R}^{3dLq_2}$ and a $W_2 \in \mathbb{R}^{d \times 3dLq_2}$ that implements the bump function (H.21) for all $t \in [d]$ and $k \in [L]$.

This completes the proof. $\qquad\square$

**Remark H.1.** Note that Theorem H.2 uses 2 FFN layers and $g \in \mathcal{T}_R^{1,1,r}$, where hidden dimension $r = O(dL(ID)^{dL}/L!)$. By Definition B.2, $\mathcal{T}_R^{1,1,r}$ belongs to our transformer network class.

To eliminate the permutation equivariance required for the target function, we incorporate the positional encoding to Theorem H.2 to break the symmetry following [Yun et al., 2019].

**Corollary H.2.1** (Universal Approximation of Transformers with Positional Encoding, [Kajitsuka and Sato, 2023])**.** Let $\epsilon \in (0,1)$ and $p \in [1, \infty)$. Let $\mathcal{F}_1^{(\mathrm{FF})}, \mathcal{F}_2^{(\mathrm{FF})}$ be two feed-forward layers and $\mathcal{F}^{(\mathrm{SA})}$ be a single-head self-attention layer with softmax function defined in (B.1) and (B.2). Then, for any continuous function $f$ on a compact domain and any $\epsilon$, there exists a positional encoding $E \in \mathbb{R}^{d \times L}$ and a $h(Z) = \mathcal{F}_2^{(\mathrm{FF})} \circ \mathcal{F}^{(\mathrm{SA})} \circ \mathcal{F}_1^{(\mathrm{FF})}(Z + E) \in \mathcal{T}_R^{h,s,r}$ such that $d_p(f, h) < \epsilon$, where $d_p := (\int \|f(Z) - g(Z)\|_p^p dZ)^{1/p}$ and $\|\cdot\|_p$ is the element-wise $\ell_p$-norm.

*Proof.* It suffices to show that the universal approximation remains valid with the inclusion of a positional encoding $E \in \mathbb{R}^{d \times L}$ to the weight matrices constructed in Theorem H.2.

Recall the (H.14) and (H.20). We have token-wise $(1/D, 2I\sqrt{d}, 1/D)$-separated sequences on $[0, 2I]^{d \times L}$ after quantization. Then, we add the positional encoding to the quantized sequences:

$$E := \begin{pmatrix} 2I & 4I & \cdots & 2IL \\ \vdots & \vdots & \ddots & \vdots \\ 2I & 4I & \cdots & 2IL \end{pmatrix},$$

giving token-wise $(2I\sqrt{d}, 2(L+1)I\sqrt{d}, 1/D)$-separated sequences, where the first column is in $[0, 4I]^d$, the second is in $[0, 6I]^d$ and so on. For each row, entries are monotonically increasing.

The second step contextual mapping remains valid by scaling the penalty function (H.18) to $2I(L+1)$. By Theorem H.1, columns after the self-attention mapping are at least $\delta'$ apart (in $\ell_2$-distance) from each other, where $\delta' = \exp\left(-5\lambda\epsilon^{-1}|\mathcal{V}|^4 d\gamma_{\max}^2 \log L/\gamma_{\min}\right)$ and $|\mathcal{V}| = O((ID)^d)$.

Recall the bump function (H.21) and (H.24). The construction of $W_1^{(i)}, b_1^{(i)}$ and $W_2^{(i)}$ follows by setting the scale parameter $R_{\mathrm{FF}} = O(1/\delta)$ . Because of the lack of permutation equivariance here, it necessitates to stack these matrices and biases for all $q_2 = (2ID)^{dL}$ possible values $u$.

This completes the proof. $\qquad\square$

**Parameter Norm Bounds for Transformer Approximation.** Next lemma provides matrices norm bounds required to achieve the universal approximation of transformer with any error $\epsilon$.

**Lemma H.5** (Transformer Matrices Bounds, Modified from Lemma F.4 and Lemma F.5 of [Hu et al., 2025c]). Let $\epsilon \in (0, 1)$. Let $Z \in [-I, I]^{d \times L}$ be an input sequence, where $I$ is an absolute positive constant and $L \geq 2$. Let $f(Z) : [-I, I]^{d \times L} \to \mathbb{R}^{d \times L}$ be any Lipchitz continuous function with respect to some norm $d_Y$. Then, for $g \in \mathcal{T}_R^{r,h,s}$ that approximates $f$ within $\epsilon$ precision, i.e., $d_Y(f, g) < \epsilon$, the parameter bounds in the transformer network class follow:

$$C_{KQ}, C_{KQ}^{2,\infty} = O(\lambda^{-1} I^{4d+2} \epsilon^{-4d-2}); \quad C_{OV}, C_{OV}^{2,\infty} = O(\epsilon);$$
$$C_F, C_F^{2,\infty} = O\left(I \epsilon^{-1} \cdot \max \|f(Z)\|_F\right); \quad C_E = O(I),$$

where $\lambda^{-1} = O(I/\epsilon)^{4d+3}$ is the inverse-temperature scaling in the softmax function and $O(\cdot)$ hides polynomial and logarithmic factors depending on $d$ and $L$. Further, for all feed-forward layers,

$$\max\{\|b_1\|_\infty, \|b_2\|_\infty\} = O(I \epsilon^{-1}).$$

*Proof.* Hu et al. [2025c] provide parameter bounds for the universal approximation of transformers on domain $[0, 1]^{d \times L}$. We specify these bounds for approximation on domain $[-I, I]^{d \times L}$

Recall the construction of weight matrices in the proof of Theorem H.2. We achieve the universal approximation by choosing "sufficiently large" granularity $D$ and "sufficiently small" $\delta$ (H.13).

To prove Lemma H.5, we first identify the order of $\delta$, $D$ and $R_{\text{FF}}$ in terms $\epsilon$. Then, we derive norm bounds on matrices in two feed-forward layers $\mathcal{F}_1^{\text{FF}}, \mathcal{F}_2^{\text{FF}}$, and the self attention layer $\mathcal{F}^{\text{SA}}$.

**Bound on $\delta$.** Recall the approximation of quantization function in (H.13). In each step function, we have extra partition $(1/D, 1/D + \delta)$. Therefore, it suffices to take $\delta = o(1/D)$.

**Bound on the Granularity $D$.** Recall the contextual mapping step in the proof of Theorem H.2. The total omitted duplicated points in the grid $\mathbb{G}_D^\circ$ are $\left|\mathbb{G}_D^\circ \setminus \widetilde{\mathbb{G}}_D\right| = \left|D^{-d} \cdot (2ID)^{dL}\right|$, where $\widetilde{\mathbb{G}}_D \subseteq \mathbb{G}_D^\circ$ is the sub-grid consisting of sequences with non-duplicated tokens. Further, by the extreme value theorem, $\|f\|_p^p \leq B_{\mathcal{T}}$ for a constant $B_{\mathcal{T}} > 0$. Then, the difference between the target function $f$ and the piece-wise constant approximator $g_1$ with granularity $D$ is bounded by

$$d_p(f, g_1)$$
$$= \left(\int \|f(Z) - g_1(Z)\|_p^p \mathrm{d}Z\right)^{\frac{1}{p}}$$
$$= O\left(\left(D^{-d}(2ID)^{dL} \cdot B_{\mathcal{T}}(1/D)^{dL}\right)^{\frac{1}{p}}\right)$$
$$= O(D^{-d/p} \cdot I^{dL}).$$

For $p \in [1, \infty)$, we have that $\epsilon = O(D^{-d/p} \cdot I^{dL})$. This implies $D = O(\epsilon^{-p/d} \cdot I^{-L/p})$. Without loss of generality, we drop $I^{-L/p} \in (0, 1)$ and drop the constant $p$. Then, we have that $D = O(\epsilon^{-1/d})$.

Next, recall the piece-wise constant approximation (H.10), (H.11) and (H.12).

For Lipchitz continuous target function $f$, there exist a grid $\mathbb{G}_D$ on domain $[-I, I]^{d \times L}$ such that

$$d_p(f(Z), g_1(Z)) < L_f \|Z - Z'\|_2 \leq L_f \|Z - Z'\|_F \leq \sqrt{dL} L_f / D,$$

where $Z' \in \mathbb{G}_D$ and $L_f$ is the Lipchitz constant with respect to the matrix 2-norm. Therefore, it suffices to take $\epsilon = \sqrt{dL} L_f / D$. Altogether, we take $D = O(\epsilon^{-1})$ such that Theorem H.2 holds.

Next, we derive the norm bounds on transformer weight matrices.

- **Bounds on $W_Q$ and $W_K$ in $\mathcal{F}^{\text{SA}}$.** For the self-attention layer, we denote the separatedness of the input tokens by $(\gamma_{\min}, \gamma_{\max}, \epsilon_s)$ and the separatedness of the output tokens by $(\gamma, \delta_s)$.

Recall Theorem H.1. We construct rank $\rho$ matrix $W_Q$ and $W_K$ in the self-attention layer by

$$W_K = \sum_{i=1}^{\rho} p_i q_i^{\top} \in \mathbb{R}^{s \times d}, \quad W_Q = \sum_{i=1}^{\rho} p_i' q_i'^{\top} \in \mathbb{R}^{s \times d},$$

with the identity $p_i^{\top} p_i' = (|\mathcal{V}| + 1)^4 d\delta_s / (\epsilon_s \gamma_{\min})$. Then, the bounds on $W_{KQ}$ follows

$$\|W_{KQ}\|_2 \leq \|W_{KQ}\|_F = \|(W_K)^{\top} W_Q\|_F = O\Big(\frac{\delta_s |\mathcal{V}|^4}{\epsilon_s \gamma_{\min}}\Big),$$

$$\|W_{KQ}\|_{2,\infty} = \|(W_K)^{\top} W_Q\|_{2,\infty} = O\Big(\frac{\delta_s |\mathcal{V}|^4}{\epsilon_s \gamma_{\min}}\Big).$$

We identify the order of each terms. Recall the first step quantization (H.13). We have total $(DI)^{dL}$ input that are token-wise $(1/D, 2I\sqrt{d}, 1/D)$-separated.

Further, since there are at most $DI$ possible values that each entry can take, we have vocabulary $|\mathcal{V}| = O((DI)^d)$ and $\gamma_{\min}, \epsilon_s = (2DI)^{-1}$. Further, from the proof of the second step contextual mapping in Theorem H.1, we construct the self-attention such that $\delta_s \cdot \lambda = 4 \log L$, where we specify the choice of $\lambda$ in (H.25). Finally, by $D = \mathcal{O}(\epsilon^{-1})$ the bounds on $W_{KQ}$ follows

$$\|W_{KQ}\|_2 \leq C_{KQ} = O(\lambda^{-1} \epsilon^{-4d-2} \cdot I^{4d+2}); \quad \|W_{KQ}\|_{2,\infty} \leq C_{KQ}^{2,\infty} = O(\lambda^{-1} \epsilon^{-4d-2} \cdot I^{4d+2}).$$

- **Bounds on $W_O$ and $W_V$ in $\mathcal{F}^{\mathrm{SA}}$.** From the proof of contextual mapping Theorem H.1, we have,

$$W_V = \sum_{i=1}^{\rho} p_i'' q_i''^{\top} \in \mathbb{R}^{s \times d}; \quad W_O = \sum_{i=1}^{\rho} p_i''' p_i''^{\top} \in \mathbb{R}^{d \times s},$$

with the identity $\|p_i'''\| \lesssim \epsilon_s / (4\rho \gamma_{\max} \|p_i''\|)$ from (H.5), and $p_i'' \in \mathbb{R}^s$ is any nonzero vector.

With the $(\gamma_{\min} = 1/D, \gamma_{\max} = \sqrt{d}, \epsilon_s = 1/D)$ separateness and $D = \mathcal{O}(\epsilon^{-1})$, we have

$$\|W_V\|_2 = \sup_{\|x\|_2=1} \|W_V x\|_2 \leq C_V = \mathcal{O}(\sqrt{\rho}) = \mathcal{O}\left(\sqrt{d}\right),$$

$$\|W_V\|_{2,\infty} = \max_{1 \leq i \leq L} \|(W_V)_{(:,i)}\|_2 \leq C_V^{2,\infty} = \mathcal{O}(\rho) = \mathcal{O}(d),$$

$$\|W_O\|_2 = \sup_{\|x\|_2=1} \|W_O x\|_2 \leq C_O = \mathcal{O}(\sqrt{\rho} \cdot \rho^{-1} \cdot \gamma_{\max}^{-1} \cdot \epsilon_s) = \mathcal{O}\left(d^{-1}\epsilon\right)$$

$$\|W_O\|_{2,\infty} = \max_{1 \leq i \leq L} \|(W_O)_{(:,i)}\|_2 \leq C_O^{2,\infty} = \mathcal{O}(\rho \cdot \rho^{-1} \cdot \gamma_{\max}^{-1} \cdot \epsilon_s) = \mathcal{O}\left(d^{-1/2}\epsilon\right).$$

Therefore,

$$\|W_{OV}\|_2 = \|W_O W_V\|_2 \leq C_{OV} = O(\epsilon); \quad \|W_{OV}\|_{2,\infty} = \|W_O W_V\|_{2,\infty} \leq C_{OV}^{2,\infty} = O(\epsilon).$$

- **Bounds on $W_1$ and $W_2$ in $\mathcal{F}_1^{\mathrm{FF}}$** Recall(H.15). By $\delta = o(1/D)$ and $D = O(\epsilon^{-1})$, we have

$$\max\{\|W_1\|_2, \|W_2\|_2\} \leq C_F^2 = O(\epsilon^{-1} \cdot IDL);$$

$$\max\{\|W_1\|_{2,\infty}, \|W_2\|_{2,\infty}\} \leq C_F^{2,\infty} = O(\epsilon^{-1} \cdot IDL).$$

- **Bounds on $W_1$ and $W_2$ in $\mathcal{F}_2^{\mathrm{FF}}$.** Recall the construction of bump function (H.24). We take

$$R_{\mathrm{FF}} = O\big(\exp\big(640\lambda(ID)^{4d+2} d^2 \log L\big)\big),$$

where $(\gamma_{\min}, \gamma_{\max}, \epsilon) = (2I\sqrt{d}, 2(L+1)I\sqrt{d}, 1/D)$. Recall that $D = O(\epsilon^{-1})$. Then, we take

$$\lambda = \frac{(ID)^{-4d-2} \cdot \log \epsilon^{-1}}{640d^2 \cdot \log L}. \tag{H.25}$$

This gives

$$\max\{\|W_1\|_2, \|W_2\|_2\} \le C_F^2 = O\big(I\epsilon^{-1} \cdot \max \|f(Z)\|_F\big),$$
$$\max\{\|W_1\|_{2,\infty}, \|W_2\|_{2,\infty}\} \le C_F^{2,\infty} = O\big(I\epsilon^{-1} \cdot \max \|f(Z)\|_F\big),$$

where $\Omega = [-I, I]^{d \times L}$ is the domain of the target function $f$.

- **Bounds on Positional Encoding Matrix $E$.** By Corollary H.2.1, we have:

$$E := \begin{pmatrix} 2I & 4I & \cdots & 2IL \\ \vdots & \vdots & \ddots & \vdots \\ 2I & 4I & \cdots & 2IL \end{pmatrix},$$

Therefore, we have $C_E = O(I)$.

Finally, recall that (i) $u = \mathcal{F}^{(\mathrm{SA})}(\overline{C})$ from (H.23) (ii) $C_{OV} = O(\epsilon)$ and (iii) $R_{\mathrm{FF}} = \epsilon^{-1}$ by (H.25). Then, the bound on the bias holds by the construction in (H.15), (H.16) and (H.23).

This completes the proof. $\qquad\square$

# I  Statistical Rates of Flow Matching Transformers (FMTs)

In this section, we present statistical rates for the first order flow, i.e., the velocity field, $u_t(x)$.

Specifically, we consider the target density function $q_1(x)$ in the Hölder space (Definition I.1) with sub-Gaussian property. Then, we bound the approximation and estimation error for $u_t(x)$. Further, we extend these results to derive distribution estimation rates under the 2-Wasserstein distance. Compared to high-order flow matching statistical rates Section 4, we remove the requirement of Lipschitz continuousness of the velocity field $u_t(x)$.

**Organizations.**  Appendix I.1 presents velocity approximation under a generic Hölder smoothness assumption. Appendix I.2 adopts a stronger Hölder smoothness assumption; this yields tighter approximation error bounds toward minimax optimality in velocity estimation. Appendix I.3 utilizes these approximation results to develop velocity estimation bounds and distribution estimation rates. Finally, Appendix I.4 establishes the nearly minimax optimality of flow matching transformers.

## I.1  Velocity Approximation: Generic Hölder Smooth Data Distributions

Establishing our statistical theory begins with approximating the velocity using transformers. We present the corresponding velocity approximation theory under the Hölder smoothness assumption on the initial data [Fu et al., 2024]. This theory ensures our approximation rate adaptive to the initial data's smoothness. First, we restate the definition of Hölder space and Hölder ball.

**Definition I.1** (Definition 4.1 Restated: Hölder Space).  Let $\alpha \in \mathbb{Z}_+^d$, and let $\beta = k_1 + \gamma$ denote the smoothness parameter, where $k_1 = \lfloor \beta \rfloor$ and $\gamma \in [0, 1)$. For a function $f : \mathbb{R}^d \to \mathbb{R}$, the Hölder space $\mathcal{H}^\beta(\mathbb{R}^d)$ is defined as the set of $\alpha$-differentiable functions satisfying: $\mathcal{H}^\beta(\mathbb{R}^d) := \{f : \mathbb{R}^d \to \mathbb{R} \mid \|f\|_{\mathcal{H}^\beta(\mathbb{R}^d)} < \infty\}$, where the Hölder norm $\|f\|_{\mathcal{H}^\beta(\mathbb{R}^d)}$ satisfies:

$$\|f\|_{\mathcal{H}^\beta(\mathbb{R}^d)} := \max_{\alpha : \|\alpha\|_1 < k_1} \sup_x |\partial^\alpha f(x)| + \max_{\alpha : \|\alpha\|_1 = k_1} \sup_{x \neq x'} \frac{|\partial^\alpha f(x) - \partial^\alpha f(x')|}{\|x - x'\|_\infty^\gamma}.$$

Also, we define the Hölder ball of radius $B$ by $\mathcal{H}^\beta(\mathbb{R}^d, B) := \{f : \mathbb{R}^d \to \mathbb{R} \mid \|f\|_{\mathcal{H}^\beta(\mathbb{R}^d)} < B\}$.

Before presenting the main result of velocity approximation, we state our two assumptions: (i) the Generic Hölder Smooth assumption on the target distribution $q(x_1)$. (ii) the regularity assumption on the first derivative of path coefficients. In particular, (i) and (ii) are the counterparts of Assumption 4.1 and Assumption 4.2 in the $K$ order flow matching framework (Section 4) respectively. Notably, we remove the Lipschitzness assumption via a more fine-grained analysis on the velocity field $u_t(x)$.

**Assumption I.1** (Generic Hölder Smooth Data).  The density function $q(x_1)$ belongs to Hölder ball of radius $B > 0$ with Hölder index $\beta > 0$ (Definition 4.1), denoted by $q(x_1) \in \mathcal{H}^\beta(\mathbb{R}^{d_x}, B)$. Also, there exist constant $C_1, C_2 > 0$ such that $q(x_1) \leq C_1 \exp\left(-C_2 \|x_1\|_2^2 / 2\right)$.

**Assumption I.2** (Path Regularity).  Consider the affine conditional flow $\psi_t(x|x_1) = \mu_t x_1 + \sigma_t x$. The first-derivative of path coefficients $\dot{\sigma}_t$ and $\dot{\alpha}_t$ are continuous on $[t_0, T]$, where $t_0, T \in (0, 1)$.

**Remark I.1.**  We remark that such path assumption is general and applies to a number of common scenarios. For instance, Lipman et al. [2024] present: (i) the conditional optimal transport schedule: $\psi_t(x|x_1) = tx_1 + (1-t)x$, (ii) the polynomial schedule: $\psi_t(x|x_1) = t^n x_1 + (1-t^n)x$, (iii) the linear variance preserving schedule: $\psi_t(x|x_1) = tx_1 + \sqrt{1-t^2}x$. These cases satisfy Assumption I.2.

We now present the velocity approximation for flow matching transformers.

**Theorem I.1** (Velocity Approximation with Transformers under Generic Hölder Smoothness).  Assume Assumption I.1 and Assumption I.2. For any precision parameter $0 < \epsilon < 1$ and smoothness parameter $\beta > 0$, let $\epsilon \leq O(N^{-\beta})$ for some $N \in \mathbb{N}$. Then, for all $t \in [t_0, T]$ with $t_0, T \in (0, 1)$,

there exists a transformer $u_\theta(x, t) \in \mathcal{T}_R^{h,s,r}$ such that

$$\int_{t_0}^{T} \int_{\mathbb{R}^{d_x}} \|u_t(x) - u_\theta(x,t)\|_2^2 \cdot p_t(x)\, \mathrm{d}x\mathrm{d}t = O\left(B^2 N^{-\beta} \cdot (\log N)^{d_x + \frac{\beta}{2} + 1}\right).$$

Let $d$ be the feature dimension and $L$ be the sequence length defined by the flow matching reshape layer in Definition B.3. Then, the parameter bounds in transformer network $\mathcal{T}_R^{h,s,r}$ satisfy

$$C_{KQ}, C_{KQ}^{2,\infty} = O\left(\lambda^{-1} N^{4\beta d + 2\beta}(\log N)^{4d_x + 2}\right); C_{OV}, C_{OV}^{2,\infty} = O\left(N^{-\beta}\right);$$

$$C_F, C_F^{2,\infty} = O\left(N^\beta (\log N)^{\frac{d_x + \beta}{2} + 1}\right); \ C_E = O(1); \ C_{\mathcal{T}} = O(\sqrt{\log N}).$$

where $\lambda^{-1} = O(N^\beta \cdot \log N)^{4d+3}$ is the inverse-temperature scaling in the softmax function and $O(\cdot)$ hides all polynomial factors depending on $d_x, d, L, \beta, C_1, C_2$.

---

*Proof Sketch.* We adopt the following strategy:

- **Step 1: Approximation on a Compact Domain via Transformer Universality.** To reflect the Hölder smoothness of the target density $q(x_1)$, we begin by applying a multivariate Taylor expansion to construct a compactly supported approximation of velocity field $u_t(x)$. We then approximate this function on a compact domain using the universal approximation of transformers.

- **Step 2: Extension to the Full Domain via Sub-Gaussian Tails.** We exploit the sub-Gaussian tail behavior of the target distribution to control the approximation error outside the compact region. Combining the errors from both regions yields the final approximation rate for the velocity field.

Please see Appendix J for a detailed proof. □

## I.2 Velocity Approximation: Stronger Hölder Smooth Data Distributions

We obtain tighter velocity approximation rates than Appendix I.1 by imposing stronger Hölder smoothness assumption on the target distribution $q(x_1)$.

**Assumption I.3** (Stronger Hölder Smooth Data). Let $C, C_1$ and $C_2$ be positive constants. The density function satisfies $q(x_1) = \exp\left(-C_2\|x_1\|_2^2/2\right) \cdot f(x_1)$, where $f$ belongs to Hölder space $f(x_1) \in \mathcal{H}^\beta(\mathbb{R}^{d_x}, B)$ (Definition 4.1) and satisfies $C_1 \geq f(x_1) \geq C$ for all $x_1$.

The density lower bound prevents $f(x)$ from taking small values, ensuring well-conditioned approximation. Without this bound, small values of $f(x)$ require a chosen threshold to maintain uniform approximation. A positive lower bound eliminates the need for such adjustments, keeping the approximation error controlled across the domain and enabling efficient convergence.

Assuming Assumption I.3, we derive the velocity approximation for flow matching transformers.

**Theorem I.2** (Velocity Approximation with Transformers under Stronger Hölder Smoothness). Assume Assumption I.3 and Assumption I.2. For any precision parameter $0 < \epsilon < 1$ and smoothness parameter $\beta > 0$, let $\epsilon \leq O(N^{-\beta})$ for some $N \in \mathbb{N}$. Then, for all $t \in [t_0, T]$ with $t_0, T \in (0, 1)$, there exists a transformer $u_\theta(x, t) \in \mathcal{T}_R^{h,s,r}$ such that

$$\int_{t_0}^{T} \int_{\mathbb{R}^{d_x}} \|u_t(x) - u_\theta(x,t)\|_2^2 \cdot p_t(x)\mathrm{d}x\mathrm{d}t = O\left(B^2 N^{-2\beta}(\log N)^{d_x + \beta}\right),$$

Further, the parameter bounds in the transformer network class follows Theorem I.1.

---

*Proof Sketch.* The proof strategy closely follows Theorem I.1:

- **Step 0: Velocity Decomposition.** We invoke Assumption I.3 to decompose the velocity field into a target function that is lower bounded. This step mitigates the influence of low-density regions and enables a more refined approximation analysis, in contrast to the setting under Assumption I.1.

- **Step 1: Approximation with Transformer Universality on Compact Domain.** To capture the Hölder regularity of the target density $q(x_1)$, we construct a compactly supported function as an intermediary to approximate the velocity field $u_t(x)$ using multivariate Taylor expansion. We then apply the universal approximation of transformers to approximate the constructed function.

- **Step 2: Full Domain Approximation.** We extend the approximation to the full space by leveraging the sub-Gaussian tail behavior, ensuring that the error outside the compact region remains controlled. Then, we incorporate all errors terms to achieve the final approximation rates for $u_t(x)$.

Please see Appendix K for a detailed proof. $\qquad\square$

## I.3   Velocity Estimation and Distribution Estimation

In this section, we study the statistical estimation problems and develop sample complexity results based on the established approximation results in Appendix I.1 and Appendix I.2. Specifically, we present the estimation error bound of flow matching transformers in Theorem I.3. Applying the velocity estimation rates, we further study the distribution estimation in Theorem I.4.

**Velocity Estimation**   Building on the transformer-based velocity approximation, we evaluate the performance of the velocity estimator $u_\theta$ trained with i.i.d. data points $\{x_i\}_{i=1}^n$ by optimizing the empirical loss (2.12). To quantify this, we define flow matching risk:

**Definition I.2** (Flow Matching Risk). Let $q$ be the target distribution and $X_1 \sim q$. Given a velocity estimator $u_\theta$, we define the flow matching risk $\mathcal{R}(u_\theta)$ as the expectation of the mean-squared difference between the $u_\theta$ and the ground truth $u_t$:

$$\mathcal{R}(u_\theta) := \frac{1}{T - t_0} \int_{t_0}^{T} \underset{x_t \sim p_t}{\mathbb{E}} [\|u_\theta(x_t, t) - u_t(x_t)\|_2^2] \, \mathrm{d}t,$$

where marginal probability path $p_t$ and marginal velocity field $u_t$ are induced by affine conditional flow $\psi_t(x|x_1) = \mu_t x_1 + \sigma_t x$ follows (2.2), (2.3), (2.5) and (2.6).

Let $\widehat{u}_\theta$ be the trained velocity estimator with i.i.d samples $\{x_i\}_{i=1}^n$. Then the following theorem presents upper bounds on the expectation of $\mathcal{R}(\widehat{u}_\theta)$ w.r.t training samples $\{x_i\}_{i=1}^n$, where $x_i \sim q$.

**Theorem I.3** (Velocity Estimation with Transformer). Let $d$ be the feature dimension. Suppose we choose the transformers as in Theorem I.1 and Theorem I.2 correspondingly, then we have

- Assume Assumption I.1 and Assumption I.2. Then,

$$\underset{\{x_i\}_{i=1}^n}{\mathbb{E}} [\mathcal{R}(\widehat{u}_\theta)] = O(n^{-\frac{1}{16d+15}} (\log n)^{20d_x + 4\beta + 20}).$$

- Assume Assumption I.2 and Assumption I.3. Then,

$$\underset{\{x_i\}_{i=1}^n}{\mathbb{E}} [\mathcal{R}(\widehat{u}_\theta)] = O(n^{-\frac{1}{8d+9}} (\log n)^{20d_x + 4\beta + 20}).$$

*Proof Sketch.* Recall (2.12) from Section 2. We obtain the velocity estimator $\widehat{u}_\theta(x, t) \in \mathcal{T}_R^{h,s,r}$ by minimizing the empirical conditional flow matching loss:

$$\widehat{\mathcal{L}}_{\mathrm{CFM}}(u_\theta) := \frac{1}{n} \sum_{i=1}^n \int_{t_0}^{T} \frac{1}{T - t_0} \underset{X_0 \sim N(0,I)}{\mathbb{E}} [\|u_\theta(\mu_t x_i + \sigma_t X_0, t) - (\dot{\mu}_t x_i + \dot{\sigma}_t X_0)\|_2^2] \mathrm{d}t.$$

To derive the estimation error, we adopt a standard strategy in empirical process theory. This involves bounding the generalization gap between empirical and true risk using covering number techniques:

- **Step 1: Domain Truncation for Risk Control.** We truncate the domain of the flow matching risk and the flow matching loss to ensure the transformer network has a finite covering number. We then control the error outside of the truncated domain by using the sub-Gaussian tail bound.

- **Step 2: Analysis on the Complexity of the Transformer Network Class via Covering Number.** Using the norm bounds on transformer parameters from Appendix I.2, we derive an upper bound on the covering number of the transformer networkfunction class. This captures the model complexity required to achieve a desired approximation rate on the compact domain.

- **Step 3: Final True Risk Upper Bound.** We apply the covering number bound to control the deviation between the empirical risk and the true risk. Lastly, we incorporate all sources of error from previous steps to derive the final estimation rate for the learned velocity field $\widehat{u}_\theta(x, t) \in \mathcal{T}_R^{h,s,r}$ via the minimization of the empirical conditional flow matching loss $\widehat{\mathcal{L}}_{\mathrm{CFM}}(u_\theta)$ in (2.12).

Please see Appendix L for a detailed proof. $\qquad\square$

**Distribution Estimation.** Next, we analyze the distribution estimation rate for the velocity estimator $\widehat{u}_\theta$ through the 2-Wasserstein distance between estimated and true distributions. Based on the velocity estimation results in Appendix I.3, the next theorem presents upper bounds on the 2-Wasserstein distance between the target distribution and the estimated distribution induced by the velocity estimator $\widehat{u}_\theta$ trained from optimizing the empirical conditional loss (2.12).

**Theorem I.4** (Distribution Estimation under 2-Wasserstein Distance). Let $\widehat{P}_T$ denote the estimated distribution at time $T$. Let $d$ be the feature dimension.

- Assume Assumption I.1 and Assumption I.2. It holds

$$\underset{\{x_i\}_{i=1}^n}{\mathbb{E}}[W_2(\widehat{P}_T, P_T)] = O(n^{-\frac{1}{32d+30}}(\log n)^{10d_x+2\beta+10}).$$

- Assume Assumption I.2 and Assumption I.3. It holds

$$\underset{\{x_i\}_{i=1}^n}{\mathbb{E}}[W_2(\widehat{P}_T, P_T)] = O(n^{-\frac{1}{16d+18}}(\log n)^{10d_x+2\beta+10}).$$

*Proof Sketch.* We derive the distribution estimation rate under the 2-Wasserstein distance by relating it to the velocity estimation error through the flow dynamics. Our proof follows three steps:

- **Step 1: Flow Deviation via Alekseev–Gröbner Lemma.** We apply the Alekseev–Gröbner lemma (Lemma M.2) to bound the deviation between the learned flow $\psi_\theta$ and the true flow $\psi$ in terms of the difference between the estimated velocity $\widehat{u}_\theta(x, t)$ and true velocity fields $u_t(x)$.

- **Step 2: Bounding the Jacobian via Grönwall's Inequality.** The flow deviation bound given by the Alekseev–Gröbner lemma involves the Jacobian matrix $D\psi_\theta$. To ensure the deviation remains controlled over time, we use Grönwall's inequality (Lemma M.1) along with the Lipschitz continuity of the network to upper bound the Jacobian norm by an exponential function.

- **Step 3: From Velocity Error to Wasserstein Distance.** We integrate the velocity error over time and apply the definition of the 2-Wasserstein metric to relate the flow deviation to $W_2(\widehat{P}_T, P_T)$. Substituting the velocity estimation error from Theorem I.3 then gives the final convergence rate.

Please see Appendix M for a detailed proof. $\qquad\square$

## I.4 Minimax Optimal Estimation

In Theorem I.4, we present a fine-grained analysis of distribution estimation. In this section, we further show that the derived estimation rates match the minimax lower bounds in Hölder space under the 2-Wasserstein metric in specific setting. We begin by recalling the minimax optimal rate for distribution estimation over Hölder smooth function classes.

**Lemma I.1** (Modified from Theorem 3 of [Niles-Weed and Berthet, 2022]). Consider the task of estimating a probability distribution $P(x_1)$ with density belonging to the space

$$\mathcal{P} := \big\{ q(x_1) | q(x_1) \in \mathcal{H}^\beta([-1,1]^{d_x}, B), q(x_1) \geq C \big\},$$

Then, for any $r \geq 1$, $\beta > 0$ and $d_x > 2$, we have

$$\inf_{\widehat{P}} \sup_{q(x_1) \in \mathcal{P}} \mathbb{E}_{\{x_i\}_{i=1}^n} [W_r(\widehat{P}, P)] \gtrsim n^{-\frac{\beta+1}{d_x+2\beta}},$$

where $\{x_i\}_{i=1}^n$ is a set of i.i.d samples drawn from distribution $P$, and $\widehat{P}$ runs over all possible estimators constructed from the data.

*Proof.* Please see Appendix N for a detailed proof. □

We show flow matching transformers match the minimax optimal rate under specific conditions.

**Theorem I.5** (Minimax Optimality of Flow Matching Transformers). Under the setting of $(16d + 18)(\beta + 1) = d_x + 2\beta$, the distribution estimation rate of flow matching transformers (Theorem I.4) matches the minimax lower bound of Hölder distribution class in 2-Wasserstein distance up to a $\log n$ and Lipschitz constants factors.

*Proof.* Please see Appendix N for a detailed proof. □

# J Proof of Theorem I.1

In this section, we use transformers to approximate velocity and give an upper bound of the velocity approximation error. We prove Theorem I.1 following the three steps shown in the proof sketch.

**Organizations.** Appendix J.1 introduces auxiliary lemmas. Appendix J.2 establishes a bound on the velocity approximation error over a bounded domain by applying the universal approximation of transformers. Appendix J.3 presents the main proof by incorporating the bounded-domain approximation error and controlling the unbounded region using the sub-Gaussian assumption.

## J.1 Auxiliary Lemmas

In this section, we introduce auxiliary lemmas for velocity approximation. Specifically, we decompose the velocity field $u_t(x)$ into three components in Lemma J.1 based on the setting of affine conditional flows (Section 2). To approximate each component, we clip the integral domain of $x_1$ in the integrals defining $\Phi_1(x, t)$, $\Phi_2(x, t)$, and $\Phi_3(x, t)$ to a closed and bounded region in Lemma J.2. This step allows us to perform the approximation on a bounded domain while controlling the error introduced by restricting the integral. Furthermore, we revisit the bounds on the density function $p_t(x)$ in $\ell_\infty$-distance, and extend these bounds to the velocity field $u_t(x)$ in Lemma J.3 and Lemma J.4.

**Decomposition of Velocity Field.** We present the next lemma to decompose the velocity field $u_t(x)$. Constructing an approximator for $u_t(x)$ is difficult due to its complex structure. This decomposition splits the velocity into three functions, each satisfying properties that make approximation feasible. These components allow the use of sub-Gaussian assumptions on the target distribution (Assumption I.1) and provide better control over the approximation error (Lemma J.9).

**Lemma J.1** (Decomposition of Velocity Field). Under the flow matching setting (Section 2), the velocity field follows a decomposition:

$$u_t(x) = \Phi_1(x, t)^{-1} \cdot \left( \frac{\dot{\mu}_t}{\mu_t} \cdot \Phi_2(x, t) + (\dot{\sigma}_t - \frac{\dot{\mu}_t \sigma_t}{\mu_t}) \Phi_3(x, t) \right),$$

where

$$\Phi_1(x, t) := \int_{\mathbb{R}^{d_x}} \frac{1}{\sigma_t^{d_x} (2\pi)^{d_x/2}} \exp\left( -\frac{\|x - \mu_t \cdot x_1\|^2}{2\sigma_t^2} \right) \cdot q(x_1) \, dx_1,$$

$$\Phi_2(x, t) := x \int_{\mathbb{R}^{d_x}} \frac{1}{\sigma_t^{d_x} (2\pi)^{d_x/2}} \exp\left( -\frac{\|x - \mu_t \cdot x_1\|^2}{2\sigma_t^2} \right) \cdot q(x_1) \, dx_1,$$

$$\Phi_3(x, t) := \int_{\mathbb{R}^{d_x}} \left( \frac{x - \mu_t \cdot x_1}{\sigma_t} \right) \cdot \frac{1}{\sigma_t^{d_x} (2\pi)^{d_x/2}} \exp\left( -\frac{\|x - \mu_t \cdot x_1\|^2}{2\sigma_t^2} \right) \cdot q(x_1) \, dx_1.$$

*Proof.* By (2.5), the density function $p_t(x)$ has the form

$$p_t(x) = \int p_t(x|x_1) \cdot q(x_1) \, dx_1$$

$$= \frac{1}{\sigma_t^{d_x} (2\pi)^{d_x/2}} \int \exp\left( -\frac{\|\mu_t x_1 - x\|^2}{2\sigma_t^2} \right) \cdot q(x_1) \, dx_1.$$

Therefore, we have $p_t(x) = \Phi_1(x, t)$.

Then, we rewrite the velocity field at time $t$ by

$$u_t(x)$$
$$= \frac{1}{p_t(x)} \cdot \int_{\mathbb{R}^{d_x}} u_t(x|x_1) p_t(x|x_1) q(x_1) \, dx_1$$

$$= \frac{1}{p_t(x)} \cdot \int_{\mathbb{R}^{d_x}} \left( \frac{\dot{\sigma}_t(x - \mu_t \cdot x_1)}{\sigma_t} + \dot{\mu}_t \cdot x_1 \right) \cdot p_t(x|x_1) q(x_1) \mathrm{d}x_1 \qquad \text{(By (2.6) and (2.8))}$$

$$= \frac{1}{p_t(x)} \cdot \int_{\mathbb{R}^{d_x}} \left( \frac{\dot{\sigma}_t(x - \mu_t \cdot x_1)}{\sigma_t} - \frac{\dot{\mu}_t}{\mu_t}(x - \mu_t \cdot x_1) + \frac{\dot{\mu}_t}{\mu_t} \cdot x \right) \cdot p_t(x|x_1) q(x_1) \mathrm{d}x_1$$

$$= \Phi_1(x,t)^{-1} \cdot \left( \dot{\sigma}_t \cdot \Phi_3(x,t) - \frac{\dot{\mu}_t \sigma_t}{\mu_t} \cdot \Phi_3(x,t) + \frac{\dot{\mu}_t}{\mu_t} \cdot \Phi_2(x,t) \right)$$
$$\left( \text{By the definition of } \Phi_1, \Phi_2 \text{ and } \Phi_3 \right)$$

$$= \Phi_1(x,t)^{-1} \cdot \left( \frac{\dot{\mu}_t}{\mu_t} \cdot \Phi_2(x,t) + (\dot{\sigma}_t - \frac{\dot{\mu}_t \sigma_t}{\mu_t}) \Phi_3(x,t) \right).$$

This completes the proof. $\qquad \square$

Based on decomposition, we construct separate approximators for $\Phi_1(x,t)$, $\Phi_2(x,t)$, and $\Phi_3(x,t)$. Then, we approximate $u_t(x)$ by combining these approximations in Appendix J.2.

**Clipping Integral Domain.** Next lemma handles unbounded integral domain of $\Phi_1(x,t)$, $\Phi_2(x,t)$, and $\Phi_3(x,t)$. Lemma J.2 ensures that for any small error $\epsilon > 0$ and any fixed $x \in \mathbb{R}^{d_x}$, a bounded domain $B_x$ dependent on $\epsilon$ and $x$ exists, where the integral outside $B_x$ remains bounded by $\epsilon$.

**Lemma J.2** (Clipping the Multi-Index Gaussian Integral, Lemma A.8 of [Fu et al., 2024] and Lemma F.9 of [Oko et al., 2023]). Assume Assumption I.1. Let $d_x$ be the dimension of the target data $x_1$ and $n \in \mathbb{N}$. Then, for any $\kappa \in \mathbb{Z}_+^{d_x}$ with $\|\kappa\|_1 \le n$, $x_1 \in \mathbb{R}^{d_x}$ and $0 < \epsilon \le 1/e$, there exists a constant $C(n, d_x) \ge 1$ such that

$$\int_{\mathbb{R}^{d_x} \setminus B_x} \left| \left( \frac{\mu_t \cdot x_1 - x}{\sigma_t} \right)^{\kappa} \right| \cdot \frac{q(x_1)}{\sigma_t^{d_x}(2\pi)^{d_x/2}} \cdot \exp\left( -\frac{\|\mu_t x_1 - x\|^2}{2\sigma_t^2} \right) \mathrm{d}x_1 \le \epsilon,$$

where $\left( \frac{\mu_t \cdot x_1 - x}{\sigma_t} \right)^{\kappa} := \left( \left( \frac{\mu_t \cdot x_1[1] - x[1]}{\sigma_t} \right)^{\kappa[1]}, \dots, \left( \frac{\mu_t \cdot x_1[d_x] - x[d_x]}{\sigma_t} \right)^{\kappa[d_x]} \right)$ is a *multi-index vector* and

$$B_x := \left[ \frac{x - \sigma_t C(n, d_x)\sqrt{\log(1/\epsilon)}}{\mu_t}, \frac{x + \sigma_t C(n, d_x)\sqrt{\log(1/\epsilon)}}{\mu_t} \right]$$
$$\bigcap \left[ C(n, d_x)\sqrt{\log(1/\epsilon)}, C(n, d_x)\sqrt{\log(1/\epsilon)} \right]^{d_x}. \qquad \text{(J.1)}$$

**Remark J.1.** The rationale behind this error choice follows from the need to control the clipping error, when we construct a polynomial-like approximator for the components of the decomposed velocity $\Phi_1$, $\Phi_2$, and $\Phi_3$ on the bounded domain $B_{x,N}$. Specifically, these approximations capture the smoothness of the density function in Hölder space and leads to an error of order $N^{-\beta}$ up to a logarithmic factor. Therefore, the clipping error is set to match this order.

**Bounds on Density Function and Velocity.** We introduce two lemmas that provide bounds on the density function $p_t(x)$ and the velocity field $u_t(x)$. These bounds are crucial because the maximum output of the transformer network class plays a key role in analyzing the capacity of the loss function class in estimation error analysis (Appendix I.3). We start with the bounds on $p_t(x)$ and $\nabla \log p_t(x)$.

**Lemma J.3** (Bounds on the Density Function, Lemma A.9 and Lemma A.10 of [Fu et al., 2024]). Recall that $p_t(x) = \int_{\mathbb{R}^{d_x}} p_t(x|x_1) q(x_1) \mathrm{d}x_1$ and $p_t(x|x_1) = \frac{1}{\sigma_t^{d_x}(2\pi)^{d_x/2}} \exp\left( -\|x - \mu_t x_1\|_2^2 / 2\sigma_t^2 \right)$. Assume Assumption I.1. There exist a $C_7 > 0$ such that

$$\frac{C_7}{\sigma_t^{d_x}} \cdot \exp\left( -\frac{\|x\|_2^2 + 1}{\sigma_t^2} \right) \le p_t(x) \le \frac{C_1}{(\mu_t^2 + C_2 \sigma_t^2)^{d_x/2}} \cdot \exp\left( -\frac{C_2 \|x\|_2^2}{2(\mu_t^2 + C_2 \sigma_t^2)} \right).$$

Moreover, there exist a positive constant $C_7'$ such that

$$\|\nabla \log p_t(x)\|_\infty \le \frac{C_7'}{\sigma_t^2} \cdot (\|x\|_2 + 1).$$

By Lemma J.1, the velocity field $u_t(x)$ follows the decomposition

$$u_t(x) = \Phi_1(x,t)^{-1} \cdot \left( \frac{\dot{\mu}_t}{\mu_t} \cdot \Phi_2(x,t) + (\dot{\sigma}_t - \frac{\dot{\mu}_t \sigma_t}{\mu_t}) \Phi_3(x,t) \right).$$

With this expression, we apply Lemma J.3 to obtain bound on the velocity $u_t(x)$ in $\ell_\infty$-distance.

**Lemma J.4** ($\ell_\infty$-Bounds on the Velocity Field)**.** Assume Assumption I.1. Then, there exists a positive constant $C_5$ such that

$$\|u_t(x)\|_\infty \leq \frac{|\dot{\mu}_t|}{\mu_t} \cdot \|x\|_\infty + C_5 \left| \frac{\dot{\mu}_t}{\mu_t} - \frac{\dot{\sigma}_t}{\sigma_t} \right| \cdot (\|x\|_2 + 1).$$

*Proof.* Recalling from Lemma J.1, we have the velocity decomposition

$$u_t(x) = \Phi_1(x,t)^{-1} \cdot \left( \frac{\dot{\mu}_t}{\mu_t} \cdot \Phi_2(x,t) + (\dot{\sigma}_t - \frac{\dot{\mu}_t \sigma_t}{\mu_t}) \Phi_3(x,t) \right),$$

where

$$\Phi_1(x,t) = \int_{\mathbb{R}^{d_x}} \frac{1}{\sigma_t^{d_x}(2\pi)^{d_x/2}} \exp\left( -\frac{\|x - \mu_t \cdot x_1\|^2}{2\sigma_t^2} \right) \cdot q(x_1) \, dx_1,$$

$$\Phi_2(x,t) = x \int_{\mathbb{R}^{d_x}} \frac{1}{\sigma_t^{d_x}(2\pi)^{d_x/2}} \exp\left( -\frac{\|x - \mu_t \cdot x_1\|^2}{2\sigma_t^2} \right) \cdot q(x_1) \, dx_1,$$

$$\Phi_3(x,t) = \int_{\mathbb{R}^{d_x}} \left( \frac{x - \mu_t \cdot x_1}{\sigma_t} \right) \cdot \frac{1}{\sigma_t^{d_x}(2\pi)^{d_x/2}} \exp\left( -\frac{\|x - \mu_t \cdot x_1\|^2}{2\sigma_t^2} \right) \cdot q(x_1) \, dx_1.$$

First, we rewrite the expression of $\Phi_2(x,t)$ and $\Phi_3(x,t)$. Then, we derive the bound on $u_t(x)$.

- **Step 1. Rewrite $\Phi_2(x,t)$ and $\Phi_3(x,t)$.** By the definition of $\Phi_2(x,t)$ and $\Phi_3(x,t)$, it holds

$$\Phi_2(x,t) = x \int_{\mathbb{R}^{d_x}} \frac{1}{\sigma_t^{d_x}(2\pi)^{d_x/2}} \exp\left( -\frac{\|x - \mu_t \cdot x_1\|^2}{2\sigma_t^2} \right) \cdot q(x_1) \, dx_1 = x \cdot \Phi_1(x,t).$$

Therefore, for all $i \in [d_x]$, it holds

$$\left| \frac{\dot{\mu}_t}{\mu_t} \cdot \Phi_2(x,t)[i] \right| = \left| \frac{\dot{\mu}_t x[i]}{\mu_t} \cdot \Phi_1(x,t) \right|. \tag{J.2}$$

Next, since the gradient of $p_t(x)$ has the expression

$$\nabla p_t(x) = -\int \left( \frac{x - \mu_t \cdot x_1}{\sigma_t^2} \right) \cdot \frac{1}{\sigma_t^{d_x}(2\pi)^{d_x/2}} \exp\left( -\frac{\|x - \mu_t \cdot x_1\|^2}{2\sigma_t^2} \right) q(x_1) \, dx_1,$$

we have $\Phi_3(x,t) = -\nabla p_t(x) \cdot \sigma_t$.

Therefore, for all $i \in [d_x]$, it holds

$$\left| \left( \dot{\sigma}_t - \frac{\dot{\mu}_t \sigma_t}{\mu_t} \right) \cdot \Phi_3(x,t)[i] \right| = \left| \left( \dot{\sigma}_t - \frac{\dot{\mu}_t \sigma_t}{\mu_t} \right) \sigma_t \cdot \nabla p_t(x)[i] \right|. \tag{J.3}$$

- **Step 2. Bound Velocity Field.** Based on Step 1, the following holds for all $i \in [d_x]$

$$|u_t[i]|$$

$$= \left| \Phi_1(x,t)^{-1} \cdot \left( \frac{\dot{\mu}_t}{\mu_t} \cdot \Phi_2(x,t)[i] + (\dot{\sigma}_t - \frac{\dot{\mu}_t \sigma_t}{\mu_t}) \Phi_3(x,t)[i] \right) \right|$$

$$\le \left| \Phi_1(x,t)^{-1} \cdot \left( \frac{\dot{\mu}_t}{\mu_t} \cdot \Phi_2(x,t)[i] \right) \right| + \left| \Phi_1(x,t)^{-1} \left( (\dot{\sigma}_t - \frac{\dot{\mu}_t \sigma_t}{\mu_t}) \cdot \Phi_3(x,t)[i] \right) \right|$$

$$\text{(By triangle inequality)}$$

$$= \left| \Phi_1(x,t)^{-1} \cdot \left( \frac{\dot{\mu}_t x[i]}{\mu_t} \cdot \Phi_1(x,t) \right) \right| + \left| \Phi_1(x,t)^{-1} \left( (\frac{\dot{\mu}_t \sigma_t^2}{\mu_t} - \dot{\sigma}_t \sigma_t) \cdot \nabla p_t(x)[i] \right) \right|$$

$$\text{(By (J.2) and (J.3))}$$

$$= \left| \frac{\dot{\mu}_t}{\mu_t} \cdot x[i] \right| + \left| \frac{\dot{\mu}_t \sigma_t^2}{\mu_t} - \dot{\sigma}_t \sigma_t \right| \cdot |\nabla \log p_t(x)[i]| \qquad \text{(By } \nabla \log p_t = \nabla p_t / p_t)$$

$$\le \left| \frac{\dot{\mu}_t}{\mu_t} \cdot x[i] \right| + C_5 \left| \frac{\dot{\mu}_t \sigma_t^2}{\mu_t} - \dot{\sigma}_t \sigma_t \right| \cdot \left| \frac{1}{\sigma_t^2} \cdot (\|x\|_2 + 1) \right|. \qquad \text{(By Lemma J.3)}$$

Therefore, by symmetry,

$$\|u_t(x)\|_\infty \le \frac{|\dot{\mu}_t|}{\mu_t} \cdot \|x\|_\infty + C_5 \left| \frac{\dot{\mu}_t}{\mu_t} - \frac{\dot{\sigma}_t}{\sigma_t} \right| \cdot (\|x\|_2 + 1).$$

This completes the proof. $\qquad\square$

## J.2 Velocity Approximation on Bounded Domain

In this section, we approximate the velocity field $u_t(x)$ on a bounded domain through a two-step approach. Specifically, the first step constructs three compactly supported continuous functions $\Psi_1(x,t)$, $\Psi_2(x,t)$ and $\Psi_3(x,t)$ as approximators for $\Phi_1(x,t)$, $\Phi_2(x,t)$, and $\Phi_3(x,t)$ in Lemma J.5, Lemma J.6, and Lemma J.7 respectively. Then, the second step applies the universal approximation to approximate $\Psi_1(x,t)$, $\Psi_2(x,t)$ and $\Psi_3(x,t)$ with transformers in Lemma J.8. Bu incorporating these steps, we derive the velocity approximation on a bounded domain in Lemma J.9.

Before proceeding, we reiterate on the velocity expression. By Lemma J.1, $u_t(x)$ has the form

$$u_t(x) = \Phi_1(x,t)^{-1} \cdot \left( \frac{\dot{\mu}_t}{\mu_t} \Phi_2(x,t) + (\dot{\sigma}_t - \frac{\dot{\mu}_t \sigma_t}{\mu_t}) \Phi_3(x,t) \right),$$

where

$$\Phi_1(x,t) = \int_{\mathbb{R}^{d_x}} \frac{1}{\sigma_t^{d_x} (2\pi)^{d_x/2}} \exp\left( -\frac{\|x - \mu_t \cdot x_1\|^2}{2\sigma_t^2} \right) \cdot q(x_1) \, dx_1,$$

$$\Phi_2(x,t) = x \int_{\mathbb{R}^{d_x}} \frac{1}{\sigma_t^{d_x} (2\pi)^{d_x/2}} \exp\left( -\frac{\|x - \mu_t \cdot x_1\|^2}{2\sigma_t^2} \right) \cdot q(x_1) \, dx_1,$$

$$\Phi_3(x,t) = \int_{\mathbb{R}^{d_x}} \left( \frac{x - \mu_t \cdot x_1}{\sigma_t} \right) \cdot \frac{1}{\sigma_t^{d_x} (2\pi)^{d_x/2}} \exp\left( -\frac{\|x - \mu_t \cdot x_1\|^2}{2\sigma_t^2} \right) \cdot q(x_1) \, dx_1.$$

**Approximation of $\Phi_1(x,t)$.** This step builds on [Hu et al., 2025c, Fu et al., 2024].

By the expression of $\Phi_1(x,t)$:

$$\Phi_1(x,t) = \int \frac{1}{\sigma_t^{d_x} (2\pi)^{d_x/2}} \exp\left( -\frac{\|\mu_t x_1 - x\|^2}{2\sigma_t^2} \right) \cdot q(x_1) \, dx_1,$$

we approximate $q(x_1)$ and $\exp\left(-\frac{\|\mu_t x_1 - x\|^2}{2\sigma_t^2}\right)$ with $k_1$-order Taylor polynomial and $k_2$-order Taylor polynomial on a bounded domain $B_{x,N}$, introduced in the integral clipping (Lemma J.2). Altogether, we approximate $\Phi_1$ with the local polynomial $\Psi_1(x,t)$ on $B_{x,N}$ with the expression:

$$\Psi_1(x,t) := \sum_{v \in [N]^{d_x}} \sum_{\|n_x\|_1 \leq k_1} \frac{R_B^{\|n_x\|_1}}{n_x!} \left.\frac{\partial^{n_x}\Phi_1}{\partial x^{n_x}}\right|_{x=R_B\left(\frac{v}{N}-\frac{1}{2}\right)} g_1(x, n_x, v, t), \tag{J.4}$$

where $n_x \in \mathbb{Z}^{d_x}$ is a multi-index, $R_B > 0$ is a constant depending on the Hölder ball radius $B$,

- $g_1(x, n_x, v, t) := \prod_{i=1}^{d_x} \sum_{k_2 < p} g_2(x[i], n_x[i], v[i], k_2)$, and

- $g_2(x[i], n_x[i], v[i], k_2) := \frac{1}{\sigma_t\sqrt{2\pi}} \int \left(\frac{x_1}{R_B} + \frac{1}{2} - \frac{v[i]}{N}\right)^{n_x[i]} \frac{1}{k_2!} \left(-\frac{|x[i]-\mu_t x_1[i]|^2}{2\sigma_t^2}\right)^{k_2} \mathrm{d}x_1$.

Hu et al. [2025c], Fu et al. [2024] consider the setting of conditional diffusion transformer with classifier-free guidance. In contrast, we apply (J.4) by removing the condition $y \in \mathbb{R}^{d_y}$.

Since $\Psi_1(x,t)$ is an approximator of $\Phi_1(x,t)$, we need to ensure that it is lower bounded away from zero so that the denominator of velocity $u_t(x)$ in Lemma J.1 does not blow up.

Therefore, we introduce an additional definition.

**Definition J.1** (Truncated Density Approximator). Let $\epsilon_{\text{low}}$ be a positive real number, and let $\Psi_1(x,t)$ be a scalar-valued function defined in (J.4). Then, we define

$$\Psi_1^c(x,t) := \max\{\Psi_1(x,t), \epsilon_{\text{low}}\}.$$

We specify the choice of $\epsilon_{\text{low}}$ in Lemma J.9. For now, we approximate $\Phi_1(x,t)$ with $\Psi_1(x,t)$.

**Lemma J.5** (Local Polynomial Approximation of $\Phi_1$, Lemma A.4 of [Fu et al., 2024]). Assume Assumption I.1. Let $\Psi_1(x,t)$ be the approximator of $\Phi_1(x,t)$. Then, for any $t \in [0,1]$ and $x \in \mathbb{R}^{d_x}$, it holds

$$|\Psi_1(x,t) - \Phi_1(x,t)| \lesssim BN^{-\beta}(\log N)^{\frac{d_x+k_1}{2}}.$$

Next, we approximate $\Phi_2(x,t)$.

**Approximation of $\Phi_2(x,t)$.** By Lemma J.1, the following identity holds

$$\Phi_2(x,t) = x \int_{\mathbb{R}^{d_x}} \frac{1}{\sigma_t^{d_x}(2\pi)^{d_x/2}} \exp\left(-\frac{\|x - \mu_t \cdot x_1\|^2}{2\sigma_t^2}\right) \cdot q(x_1)\,\mathrm{d}x_1 = x \cdot \Phi_1(x,t). \tag{J.5}$$

Building upon the local polynomial $\Psi_1(x,t)$, we use $x \cdot \Psi_1(x,t)$ as the approximator of $\Phi_2(x,t)$.

Next lemma gives the approximation error rate of $\Phi_2(x,t)$ using $\Psi_2(x,t) := x \cdot \Psi_1(x,t)$

**Lemma J.6** (Local Polynomial Approximation of $\Phi_2$). Assume Assumption I.1. Let $\Psi_1(x,t)$ be the local polynomial and $\Psi_2(x,t) := x\Psi_1(x,t)$. Let $C_x(d_x, \beta, C_1, C_2)$ be a positive constant. Then, for any $t \in [0,1]$ and $x \in [-C_x\sqrt{\log N}, C_x\sqrt{\log N}]^{d_x}$, it holds for all $i \in [d_x]$

$$|\Psi_2(x,t)[i] - \Phi_2(x,t)[i]|_\infty \lesssim BN^{-\beta}(\log N)^{\frac{d_x+k_1+1}{2}}.$$

*Proof.* Since $\Psi_2(x,t) = x\Psi_1(x,t)$ and $\Phi_2(x,t) = x\Phi_1(x,t)$, for all $i \in [d_x]$, it holds

$$\begin{aligned}
|\Psi_2[i] - \Phi_2[i]| &= |x\Psi_1[i] - x\Phi_1[i]| \\
&\leq |x[i]| \cdot |\Psi_1 - \Phi_1| &&\text{(By (J.5))} \\
&\lesssim x[i] \cdot BN^{-\beta}(\log N)^{\frac{d_x+k_1}{2}} &&\text{(By Lemma J.5)}
\end{aligned}$$

$$\lesssim BN^{-\beta} (\log N)^{\frac{d_x+k_1+1}{2}}. \qquad \left(\text{By } x \in [-C_x\sqrt{\log N}, C_x\sqrt{\log N}]^{d_x}\right)$$

This completes the proof. $\qquad\qquad\qquad\qquad\qquad\qquad\qquad\qquad\qquad\qquad\qquad\qquad\qquad\qquad\square$

**Approximation of $\Phi_3(x,t)$.** Similarly, we have approximation results for $\Phi_3(x,t)$.

---

**Lemma J.7** (Local Polynomial Approximation of $\Phi_3$, Lemma A.6 of [Fu et al., 2024]). Assume Assumption I.1. Let $C_x(d_x, \beta, C_1, C_2)$ be a positive constant. There exists local polynomial $\Psi_3(x,t)$ such that for all $t > 0$, $i \in [d_x]$ and $x \in [-C_x\sqrt{\log N}, C_x\sqrt{\log N}]^{d_x}$, it holds

$$|\Psi_3(x,t)[i] - |\sigma_t \nabla p_t(x)|[i]| \lesssim BN^{-\beta}(\log N)^{\frac{d_x+k_1+1}{2}}.$$

---

**Remark J.2.** We clarify that Lemma J.7 gives the approximation of $\Phi_3(x,t)$ using $\Psi_3(x,t)$. First, the density at time $t$ has the form:

$$p_t(x) = \int_{\mathbb{R}^{d_x}} \frac{1}{\sigma_t^{d_x}(2\pi)^{d_x/2}} \exp\left(-\frac{\|x - \mu_t \cdot x_1\|^2}{2\sigma_t^2}\right) \cdot q(x_1)\,\mathrm{d}x_1.$$

Then, the gradient of $p_t(x)$ with respect to $x$ has the form:

$$\nabla p_t(x) = \int_{\mathbb{R}^{d_x}} -\left(\frac{x - \mu_t \cdot x_1}{\sigma_t^2}\right) \frac{1}{\sigma_t^{d_x}(2\pi)^{d_x/2}} \exp\left(-\frac{\|x - \mu_t \cdot x_1\|^2}{2\sigma_t^2}\right) \cdot q(x_1)\,\mathrm{d}x_1.$$

By Lemma J.1, we have $\Phi_3(x,t) = |\sigma_t \nabla p_t(x)|$.
Therefore,

$$|\Psi_3(x,t)[i] - \Phi_3(x,t)[i]| \lesssim BN^{-\beta}(\log N)^{\frac{d_x+k_1+1}{2}}. \qquad (\text{By Lemma J.7})$$

**Velocity Approximation with Transformers on Bounded Domain.** We first approximate the velocity approximator constructed with $\Psi_1(x,t)$, $\Psi_2(x,t)$ and $\Psi_3(x,t)$. We reiterate that transformers take input $d \times L$ matrices, where $d \times L = d_x$. Then, the next lemma specifies the network configuration for the approximating the velocity approximator with arbitrarily small error.

---

**Lemma J.8** (Approximate Velocity Approximator with Transformers). Assume Assumption I.1. Let $C_x(d_x, \beta, C_1, C_2)$ be a positive constant. Further, let $\Psi(x,t) : [-C_x\sqrt{\log N}, C_x\sqrt{\log N}]^{d_x} \times [0,1] \to \mathbb{R}^{d_x}$ be the target function:

$$\Psi(x,t) := \frac{\dot{\mu}_t \Psi_2(x,t)/\mu_t + (\dot{\sigma}_t - \dot{\mu}_t \sigma_t/\mu_t)\Psi_3(x,t)}{\Psi_1^c(x,t)}.$$

Then, for any $t \in [0,1]$ and any $\epsilon \in (0,1)$, there exist a transformer $g(x,t) \in \mathcal{T}_R^{h,s,r}$ such that

$$\int_0^1 \int_{\|x\|_\infty \le C_x\sqrt{\log N}} \|g(x,t) - \Psi(x,t)\|_2^2 \mathrm{d}x\mathrm{d}t \le \epsilon^2.$$

Furthermore, the parameter bounds in the transformer network class $\mathcal{T}_R^{h,s,r}$ satisfy

$$C_{KQ}, C_{KQ}^{2,\infty} = O(\lambda^{-1}(\log N)^{4d+2}\epsilon^{-4d-2}); C_{OV}, C_{OV}^{2,\infty} = O(\epsilon);$$
$$C_F, C_F^{2,\infty} = O(\sqrt{\log N} \cdot \epsilon^{-1} \cdot \max\|\Psi\|_2); C_E = O(1),$$

where $\lambda^{-1} = O(\log N/\epsilon)^{4d+3}$ is the inverse-temperature scaling in the softmax function and $O(\cdot)$ hides all polynomial factors depending on $d_x, d, L, \beta, C_1, C_2$.

---

*Proof.* Since the path coefficients are smooth and the first-step approximators $\Psi_1(x,t)$, $\Psi_2(x,t)$, and $\Psi_3(x,t)$ integrate polynomials, the target function is Lipschitz continuous on a compact domain.

Further, the reshape layer Definition B.4 does not harm the continuity of the element-wise $\ell_2$-norm. This continuity ensures that the function satisfies the conditions for applying the universal approximation of transformers. Also, we concatenate $t$ as a additional sequence. Then, we apply Theorem H.2 with $p = 2$ and $Z \in [-C_x\sqrt{\log N}, C_x\sqrt{\log N}]^{d\times(L+1)}$.[5] For any $\epsilon \in (0, 1)$, it holds

$$d_2(g, f) = \Big( \int \int \|g(x, t) - \Psi(x, t)\|_2^2 \mathrm{d}x\mathrm{d}t \Big)^{1/2} \leq \epsilon. \qquad \text{(By Theorem H.2)}$$

The parameter bounds in the transformer network class follow Lemma H.5.

This completes the proof. $\qquad\square$

**Remark J.3.** Lemma J.8 modifies Lemma I.6 of [Hu et al., 2025c] by adapting the transformer approximation to decomposed velocity components (Lemma J.1), whereas their work focuses on approximating $\nabla \log p_t(x)$. Our flow matching framework eliminates the label $y$ and reduces the number of hidden dimensions to one.

Then, by analyzing the error accumulation from both the transformer approximation (Lemma J.8) and the local polynomial approximations (Lemma J.5, Lemma J.6, and Lemma J.7), we establish a bound on the velocity approximation error over a bounded domain.

---

**Lemma J.9** (Velocity Approximation with Transformers on Bounded Domain). Assume Assumption I.1 and Assumption I.3. Let $t_0, T \in (0, 1)$. Let $C_x(\beta, C_2)$ and $C_3$ be two positive constants. Let $\epsilon_{\text{low}} := C_3 N^{-\beta} (\log N)^{(d_x+k_1)/2}$. Then, there exist a transformer $u_\theta(x, t) \in \mathcal{T}_R^{h,s,r}$ such that for all $x \in [-C_x\sqrt{\log N}, C_x\sqrt{\log N}]^{d_x}$, $t \in [t_0, T]$ and $p_t(x) \geq \epsilon_{\text{low}}$, it holds

$$\int_{t_0}^T \int \|u_t(x) - u_\theta(x, t)\|_2^2 (p_t(x))^2 \mathrm{d}x\mathrm{d}t \lesssim \Big( \frac{|\dot\mu_t|}{\mu_t} + \Big| \frac{\dot\mu_t}{\mu_t} - \frac{\dot\sigma_t}{\sigma_t} \Big| \Big)^2 B^2 N^{-2\beta} (\log N)^{\frac{3d_x}{2}+k_1+1},$$

Furthermore, the transformer parameter bounds satisfy

$$C_{KQ}, C_{KQ}^{2,\infty} = O\big(\lambda^{-1} N^{4\beta d + 2\beta} (\log N)^{4d_x+2}\big); \quad C_{OV}, C_{OV}^{2,\infty} = O\big(N^{-\beta}\big);$$
$$C_F, C_F^{2,\infty} = O\big(N^\beta (\log N)^{\frac{d_x+\beta}{2}+1}\big); \quad C_E = O(I); \quad C_{\mathcal{T}} = O(\sqrt{\log N}).$$

where $\lambda^{-1} = O(N^\beta \cdot \log N)^{4d+3}$ is the inverse-temperature scaling in the softmax function and $O(\cdot)$ hides all polynomial factors depending on $d_x, d, L, \beta, C_1, C_2$.

---

*Proof.* We use the notation "$\lesssim$" in our derivation when an inequality holds up to a constant factor.

We prove Lemma J.9 with following two steps.

- **Step A: Approximate velocity with constructed function.** We approximate the components $\Phi_1(x, t)$, $\Phi_2(x, t)$, and $\Phi_3(x, t)$ using local polynomials $\Psi_1(x, t)$, $\Psi_2(x, t)$, and $\Psi_3(x, t)$, respectively. Based on the velocity decomposition given in Lemma J.1, we construct an approximation $\Psi(x, t)$ by combining these polynomial components to approximate the full velocity field $u_t(x)$.

- **Step B: Approximate with Transformers.** We leverage the universal approximation of transformers (Appendix H) to approximate the constructed function $\Psi$. Based on this approximation, we derive the final velocity approximation rates with the required bounds on model parameters.

By Lemma J.1, the velocity field $u_t(x)$ takes the form

$$u_t(x) = \Phi_1(x, t)^{-1} \cdot \Big( \frac{\dot\mu_t}{\mu_t} \cdot \Phi_2(x, t) + (\dot\sigma_t - \frac{\dot\mu_t \sigma_t}{\mu_t})\Phi_3(x, t) \Big),$$

---

[5]Please see Appendix H for a detailed proof.

where

$$\Phi_1(x,t) = \int_{\mathbb{R}^{d_x}} \frac{1}{\sigma_t^{d_x}(2\pi)^{d_x/2}} \exp\left(-\frac{\|x - \mu_t \cdot x_1\|^2}{2\sigma_t^2}\right) \cdot q(x_1)\,\mathrm{d}x_1,$$

$$\Phi_2(x,t) = x \int_{\mathbb{R}^{d_x}} \frac{1}{\sigma_t^{d_x}(2\pi)^{d_x/2}} \exp\left(-\frac{\|x - \mu_t \cdot x_1\|^2}{2\sigma_t^2}\right) \cdot q(x_1)\,\mathrm{d}x_1,$$

$$\Phi_3(x,t) = \int_{\mathbb{R}^{d_x}} \left(\frac{x - \mu_t \cdot x_1}{\sigma_t}\right) \cdot \frac{1}{\sigma_t^{d_x}(2\pi)^{d_x/2}} \exp\left(-\frac{\|x - \mu_t \cdot x_1\|^2}{2\sigma_t^2}\right) \cdot q(x_1)\,\mathrm{d}x_1.$$

Moreover, by Lemma J.4, the bound on the velocity field in $\ell_\infty$-distance follows

$$
\begin{aligned}
&\|u_t(x)\|_\infty &&\text{(J.6)}\\
&\leq \frac{|\dot\mu_t|}{\mu_t} \cdot \|x\|_\infty + \left|\frac{\dot\mu_t}{\mu_t} - \frac{\dot\sigma_t}{\sigma_t}\right| \cdot (\|x\|_2 + 1) &&\left(\text{By Lemma J.4}\right)\\
&\lesssim \frac{|\dot\mu_t|}{\mu_t} \cdot \sqrt{\log N} + \left|\frac{\dot\mu_t}{\mu_t} - \frac{\dot\sigma_t}{\sigma_t}\right| \cdot (\sqrt{\log N} + 1). &&\left(\text{By } x \in [-C_x\sqrt{\log N}, C_x\sqrt{\log N}]^{d_x}\right)
\end{aligned}
$$

Set the transformer network output bound $C_{\mathcal{T}}$ equal to the right-hand side of the expression. Then we are now ready to present the proof of Lemma J.9.

- **Step A: Approximation via Local Polynomial.**

  We construct the approximator for $u_t(x)$ based on Lemma J.5, Lemma J.6, and Lemma J.7. Specifically, we define $\Psi(x,t) \in \mathbb{R}^{d_x}$ with each element given by

  $$|\Psi(x,t)[i]| := \min\left\{\frac{\dot\mu_t\Psi_2[i]/\mu_t + (\dot\sigma_t - \dot\mu_t\sigma_t/\mu_t)\Psi_3[i]}{\Psi_1^c}, U\right\}, \tag{J.7}$$

  where $U$ is the upper-bound of the ground truth velocity $u_t(x)$ under the sub-Gaussian assumption (Assumption I.1) and

  $$U := \frac{|\dot\mu_t|}{\mu_t} \cdot \sqrt{\log N} + C_5\left|\frac{\dot\mu_t}{\mu_t} - \frac{\dot\sigma_t}{\sigma_t}\right| \cdot (\sqrt{\log N} + 1).$$
  $$\left(\text{By Lemma J.4 and } x \in [-C_x\sqrt{\log N}, C_x\sqrt{\log N}]^{d_x}\right)$$

  Notice that, for all $i \in [d_x]$, the difference between $\Psi(x,t)[i]$ and $u_t(x)[i]$ follows

  $$
  \begin{aligned}
  &|u_t(x)[i] - \Psi(x,t)[i]|\\
  &= \left|\frac{\dot\mu_t\Phi_2[i]/\mu_t + (\dot\sigma_t - \dot\mu_t\sigma_t/\mu_t)\Phi_3[i]}{\Phi_1} - \frac{\dot\mu_t\Psi_2[i]/\mu_t + (\dot\sigma_t - \dot\mu_t\sigma_t/\mu_t)\Psi_3[i]}{\Psi_1^c}\right|\\
  &\hspace{6cm}\left(\text{By the definition of } u_t \text{ and } \Psi(x,t)\right)\\
  &\leq \underbrace{\left|\frac{\dot\mu_t\Phi_2[i]/\mu_t + (\dot\sigma_t - \dot\mu_t\sigma_t/\mu_t)\Phi_3[i]}{\Psi_1^c} - \frac{\dot\mu_t\Psi_2[i]/\mu_t + (\dot\sigma_t - \dot\mu_t\sigma_t/\mu_t)\Psi_3[i]}{\Psi_1^c}\right|}_{(\mathrm{T}_1)}\\
  &\quad + \underbrace{\left|\frac{\dot\mu_t\Phi_2[i]/\mu_t + (\dot\sigma_t - \dot\mu_t\sigma_t/\mu_t)\Phi_3[i]}{\Phi_1} - \frac{\dot\mu_t\Phi_2[i]/\mu_t + (\dot\sigma_t - \dot\mu_t\sigma_t/\mu_t)\Phi_3[i]}{\Psi_1^c}\right|}_{(\mathrm{T}_2)}.\\
  &\hspace{9cm}\left(\text{By triangle inequality}\right)
  \end{aligned}
  $$

  Next, we bound $(\mathrm{T}_1)$ and $(\mathrm{T}_2)$.

– **Step A.1: Bound term** $(\mathrm{T}_1)$**.** Recall Definition J.1. By the definition of $\epsilon_{\mathrm{low}}$, we set

$$\Psi_1^c(x,t) \coloneqq \max\left\{\Psi_1(x,t), C_3 \cdot N^{-\beta}(\log N)^{\frac{d_x+k_1}{2}}\right\}.$$

By Lemma J.5, we have

$$|\Psi_1(x,t) - p_t(x)| \lesssim BN^{-\beta}(\log N)^{\frac{d_x+k_1}{2}}, \tag{J.8}$$

and (J.8) implies

$$p_t(x) - KBN^{-\beta}(\log N)^{\frac{d_x+k_1}{2}} \leq \Psi_1(x,t),$$

for some positive constant $K$. Next, recall that we consider

$$C_3 N^{-\beta}(\log N)^{\frac{d_x+k_1}{2}} \leq p_t(x).$$

By setting $C_3 = 2KB$, it holds

$$KBN^{-\beta}(\log N)^{\frac{d_x+k_1}{2}} = \frac{C_3}{2}N^{-\beta}(\log N)^{\frac{d_x+k_1}{2}} \leq p_t(x)/2,$$

leading to

$$p_t(x) - p_t(x)/2 \leq p_t(x) - KBN^{-\beta}(\log N)^{\frac{d_x+k_1}{2}} \leq \Psi_1(x,t).$$

As a result, $p_t(x)/2 \leq \Psi_1 \leq \Psi_1^c$ holds.

This allows us to replace the approximator $\Psi_1^c$ with $p_t(x)$ by dropping constant $1/2$. Then,

$$\begin{aligned}
&(\mathrm{T}_1) \tag{J.9}\\
&= \left|\frac{\dot{\mu}_t\Phi_2[i]/\mu_t + (\dot{\sigma}_t - \dot{\mu}_t\sigma_t/\mu_t)\Phi_3[i]}{\Psi_1^c} - \frac{\dot{\mu}_t\Psi_2[i]/\mu_t + (\dot{\sigma}_t - \dot{\mu}_t\sigma_t/\mu_t)\Psi_3[i]}{\Psi_1^c}\right|\\
&\leq \frac{2}{p_t}\left|\frac{\dot{\mu}_t}{\mu_t}\cdot\left(\Phi_2[i] - \Psi_2[i]\right) + \left(\dot{\sigma}_t - \frac{\dot{\mu}_t\sigma_t}{\mu_t}\right)\cdot\left(\Phi_3[i] - \Psi_3[i]\right)\right| && \left(\text{By } \Psi_1^c > p_t(x)/2\right)\\
&\leq \frac{2}{p_t}\frac{|\dot{\mu}_t|}{\mu_t}\cdot|\Phi_2[i] - \Psi_2[i]| + \frac{2}{p_t}\left|\dot{\sigma}_t - \frac{\dot{\mu}_t\sigma_t}{\mu_t}\right|\cdot|\Phi_3[i] - \Psi_3[i]| && \left(\text{By triangle inequality}\right)\\
&\lesssim \frac{1}{p_t}\cdot\left(\frac{|\dot{\mu}_t|}{\mu_t} + \left|\dot{\sigma}_t - \frac{\dot{\mu}_t\sigma_t}{\mu_t}\right|\right)\cdot BN^{-\beta}(\log N)^{\frac{d_x+k_1+1}{2}} && \left(\text{By Lemma J.6 and Lemma J.7}\right)\\
&\leq \frac{1}{p_t}\cdot\left(\frac{|\dot{\mu}_t|}{\mu_t} + \left|\frac{\dot{\sigma}_t}{\sigma_t} - \frac{\dot{\mu}_t}{\mu_t}\right|\right)\cdot BN^{-\beta}(\log N)^{\frac{d_x+k_1+1}{2}} && \left(\text{By } \sigma_t \in [0,1]\right)\\
&\lesssim \frac{1}{p_t}\cdot BN^{-\beta}(\log N)^{\frac{d_x+k_1+1}{2}}. && \left(\text{By Assumption I.2}\right)
\end{aligned}$$

Next, we bound $(\mathrm{T}_2)$.

– **Step A.2: Bound term** $(\mathrm{T}_2)$**.** By Lemma J.4 and $\|x\|_2 \lesssim \sqrt{\log N}$, it holds

$$\begin{aligned}
&|u_t(x)[i]|\\
&\leq \frac{|\dot{\mu}_t|}{\mu_t}\cdot\sqrt{\log N} + \left|\frac{\dot{\sigma}_t}{\sigma_t} - \frac{\dot{\mu}_t}{\mu_t}\right|\cdot(\sqrt{\log N} + 1)\\
&= \left(\frac{|\dot{\mu}_t|}{\mu_t} + \left|\frac{\dot{\sigma}_t}{\sigma_t} - \frac{\dot{\mu}_t}{\mu_t}\right|\right)\cdot\sqrt{\log N} + \left|\frac{\dot{\sigma}_t}{\sigma_t} - \frac{\dot{\mu}_t}{\mu_t}\right|,
\end{aligned}$$

for all $i \in [d_x]$. Next, by the decomposition of velocity in Lemma J.1, it holds

$$\frac{\dot{\mu}_t}{\mu_t}\Phi_2[i] + \left(\dot{\sigma}_t - \frac{\dot{\mu}_t\sigma_t}{\mu_t}\right)\Phi_3[i] \lesssim \Phi_1\left(\frac{|\dot{\mu}_t|}{\mu_t} + \left|\frac{\dot{\sigma}_t}{\sigma_t} - \frac{\dot{\mu}_t}{\mu_t}\right|\right) \cdot \sqrt{\log N} + \Phi_1\left|\frac{\dot{\sigma}_t}{\sigma_t} - \frac{\dot{\mu}_t}{\mu_t}\right|.$$
(J.10)

Therefore,

$$(\text{T}_2) \tag{J.11}$$
$$\leq \left|\frac{\dot{\mu}_t}{\mu_t}\Phi_2[i] + \left(\dot{\sigma}_t - \frac{\dot{\mu}_t\sigma_t}{\mu_t}\right)\Phi_3[i]\right| \cdot \left|\frac{1}{\Phi_1} - \frac{1}{\Psi_1}\right|$$
$$\lesssim \Phi_1\left(\left(\frac{|\dot{\mu}_t|}{\mu_t} + \left|\frac{\dot{\sigma}_t}{\sigma_t} - \frac{\dot{\mu}_t}{\mu_t}\right|\right)\sqrt{\log N} + \left|\frac{\dot{\sigma}_t}{\sigma_t} - \frac{\dot{\mu}_t}{\mu_t}\right|\right) \cdot \left|\frac{1}{\Phi_1} - \frac{1}{\Psi_1}\right| \qquad (\text{By (J.10)})$$
$$= \frac{1}{\Psi_1} \cdot \left(\left(\frac{|\dot{\mu}_t|}{\mu_t} + \left|\frac{\dot{\sigma}_t}{\sigma_t} - \frac{\dot{\mu}_t}{\mu_t}\right|\right)\sqrt{\log N} + \left|\frac{\dot{\sigma}_t}{\sigma_t} - \frac{\dot{\mu}_t}{\mu_t}\right|\right) \cdot |\Phi_1 - \Psi_1|$$
$$\qquad\qquad\qquad\qquad\qquad\qquad\qquad\qquad\qquad (\text{By factoring out } 1/\Phi_1 \text{ and } 1/\Psi_1)$$
$$\leq \frac{1}{p_t} \cdot \left(\left(\frac{|\dot{\mu}_t|}{\mu_t} + \left|\frac{\dot{\sigma}_t}{\sigma_t} - \frac{\dot{\mu}_t}{\mu_t}\right|\right)\sqrt{\log N} + \left|\frac{\dot{\sigma}_t}{\sigma_t} - \frac{\dot{\mu}_t}{\mu_t}\right|\right) \cdot |\Phi_1 - \Psi_1| \qquad (\text{By } \Psi_1^c > p_t(x)/2)$$
$$\lesssim \frac{1}{p_t} \cdot \left(\left(\frac{|\dot{\mu}_t|}{\mu_t} + \left|\frac{\dot{\sigma}_t}{\sigma_t} - \frac{\dot{\mu}_t}{\mu_t}\right|\right)\sqrt{\log N} + \left|\frac{\dot{\sigma}_t}{\sigma_t} - \frac{\dot{\mu}_t}{\mu_t}\right|\right) \cdot BN^{-\beta}(\log N)^{\frac{d_x+k_1}{2}}$$
$$\qquad\qquad\qquad\qquad\qquad\qquad\qquad\qquad\qquad\qquad\qquad (\text{By Lemma J.5})$$

Combining (J.9) and (J.11), we have

$$p_t \cdot |u_t[i] - \Psi[i]| \tag{J.12}$$
$$\leq (\text{T}_1) \cdot p_t + (\text{T}_2) \cdot p_t$$
$$\lesssim \left(\frac{|\dot{\mu}_t|}{\mu_t} + \left|\frac{\dot{\mu}_t}{\mu_t} - \frac{\dot{\sigma}_t}{\sigma_t}\right|\right)BN^{-\beta}(\log N)^{\frac{d_x+k_1+1}{2}}, \qquad (\text{By (J.9) and (J.11)})$$

for all $i \in [d_x]$.
Therefore,

$$p_t^2 \cdot \|u_t(x) - \Psi(x,t)\|_2^2 \tag{J.13}$$
$$\leq p_t^2 \cdot d_x \|u_t(x) - \Psi(x,t)\|_\infty^2 \qquad (\text{By } \|\cdot\|_2 \leq d_x\|\cdot\|_\infty)$$
$$\lesssim \left(\frac{|\dot{\mu}_t|}{\mu_t} + \left|\frac{\dot{\mu}_t}{\mu_t} - \frac{\dot{\sigma}_t}{\sigma_t}\right|\right)^2 B^2 N^{-2\beta}(\log N)^{d_x+k_1+1}. \qquad (\text{By (J.12)})$$

- **Step B: Approximation with Transformer.**

By Lemma J.8, there exists a transformer $u_\theta(x,t) \in \mathcal{T}_R^{h,r,s}$ such that

$$\int\int \|u_\theta(x,t) - \Psi(x,t)\|_2^2 \mathrm{d}x\mathrm{d}t \leq \epsilon^2. \tag{J.14}$$

By setting $\epsilon := N^{-\beta}$, it holds

$$\int\int p_t^2 \cdot \|u_t(x) - u_\theta(x,t)\|_2^2 \mathrm{d}x\mathrm{d}t$$
$$\leq \int\int p_t^2 \cdot \|u_t(x) - \Psi(x,t)\|_2^2 \mathrm{d}x\mathrm{d}t + \int\int p_t^2 \cdot \|\Psi(x,t) - u_\theta(x,t)\|_2^2 \mathrm{d}x\mathrm{d}t$$
$$\qquad\qquad\qquad\qquad\qquad\qquad\qquad\qquad\qquad (\text{By triangle inequality})$$

$$\leq \int \int p_t^2 \cdot \|u_t(x) - \Psi(x,t)\|_2^2 \mathrm{d}x\mathrm{d}t + \int \int \|\Psi(x,t) - u_\theta(x,t)\|_2^2 \mathrm{d}x\mathrm{d}t \quad (\text{By } 0 \leq p_t(x) \leq 1)$$

$$\lesssim \left( \frac{|\dot{\mu}_t|}{\mu_t} + \left| \frac{\dot{\mu}_t}{\mu_t} - \frac{\dot{\sigma}_t}{\sigma_t} \right| \right)^2 B^2 N^{-2\beta} (\log N)^{d_x + k_1 + 1} \int \int \mathrm{d}x\mathrm{d}t + \int \int \|\Psi(x,t) - u_\theta(x,t)\|_2^2 \mathrm{d}x\mathrm{d}t$$
$$\left( \text{By (J.13)} \right)$$

$$\leq \left( \frac{|\dot{\mu}_t|}{\mu_t} + \left| \frac{\dot{\mu}_t}{\mu_t} - \frac{\dot{\sigma}_t}{\sigma_t} \right| \right)^2 B^2 N^{-2\beta} (\log N)^{\frac{3d_x}{2} + k_1 + 1} + \int \int \|\Psi(x,t) - u_\theta(x,t)\|_2^2 \mathrm{d}x\mathrm{d}t$$
$$\left( \text{By } \|x\|_\infty \leq C_x \sqrt{\log N} \text{ and } t \in [0,1] \right)$$

$$\lesssim \left( \frac{|\dot{\mu}_t|}{\mu_t} + \left| \frac{\dot{\mu}_t}{\mu_t} - \frac{\dot{\sigma}_t}{\sigma_t} \right| \right)^2 B^2 N^{-2\beta} (\log N)^{\frac{3d_x}{2} + k_1 + 1}. \qquad \left( \text{By (J.14) and } \epsilon = N^{-\beta} \right)$$

By (J.6) and $x \in [-C_x\sqrt{\log N}, C_x\sqrt{\log N}]^{d_x}$, we have

$$U = O(\sqrt{\log N}),$$

and by (J.12) we have

$$|u_t[i] - \Psi[i]| = O(N^{-\beta}(\log N)^{\frac{d_x + k_1 + 1}{2}}).$$

This implies

$$\|\Psi(x,t)\|_2 = O(\sqrt{\log N} + N^{-\beta}(\log N)^{\frac{d_x + k_1 + 1}{2}}).$$

We take a looser bound on $\Psi(x,t)$ such that it holds for all $d_x$:

$$\|\Psi(x,t)\|_2 \leq d_x\|\Psi(x,t)\|_\infty = O((\log N)^{\frac{d_x + k_1 + 1}{2}}).$$

Then, the parameter bounds follow Lemma J.8 with $\epsilon = N^{-\beta}$.

This completes the proof. $\qquad \square$

## J.3 Main Proof of Theorem I.1

We establish the velocity approximation with transformers in Lemma J.9. However, it is valid under two settings: (i) the bounded domain $x \in [-C_x\sqrt{\log N}, C_x\sqrt{\log N}]^{d_x}$ with some constant $C_x(\beta, C_2)$ (ii) the mild and high density region $p_t(x) \geq \epsilon_{\text{low}}$. To obtain general approximation results, we introduce two additional lemmas to tackle the uncontrolled region.

**Lemma J.10** (Truncation of $x$, Modified from Lemma A.1 of [Fu et al., 2024]). Assume Assumption I.1. Then, for any $R_4 > 1$, $t > 0$, the following hold

$$\int_{\|x\|_\infty > R_4} p_t(x)\mathrm{d}x \lesssim R_4^{d_x - 2} \exp\left( -\frac{C_2 R_4^2}{2(\mu_t^2 + C_2\sigma_t^2)} \right),$$

$$\int_{\|x\|_\infty > R_4} \|u_t(x)\|_2^2 \cdot p_t(x)\mathrm{d}x \lesssim R_4^{d_x} \exp\left( -\frac{C_2 R_4^2}{2(\mu_t^2 + C_2\sigma_t^2)} \right).$$

*Proof.* For the first inequality, it follows

$$\int_{\|x\|_\infty > R_4} p_t(x)\mathrm{d}x$$

$$\lesssim \int_{\|x\|_\infty > R_4} \exp\left( -\frac{C_2\|x\|_2^2}{2(\mu_t^2 + C_2\sigma_t^2)} \right) \mathrm{d}x \qquad (\text{By Lemma J.3})$$

$$\leq \int_{\|x\|_2 > R_4} \exp\left(-\frac{C_2\|x\|_2^2}{2(\mu_t^2 + C_2\sigma_t^2)}\right) \mathrm{d}x \qquad \left(\text{By } \|x\|_2 \geq \|x\|_\infty\right)$$

$$\lesssim R_4^{d_x-2} \exp\left(-\frac{C_2 R_4^2}{2(\mu_t^2 + C_2\sigma_t^2)}\right). \qquad \left(\text{By Lemma D.2 and dropping constant terms}\right)$$

For the second inequality, it follows

$$\int_{\|x\|_\infty \geq R_4} \|u_t(x)\|_2^2 \cdot p_t(x)\mathrm{d}x$$

$$\lesssim \int_{\|x\|_\infty \geq R_4} \|u_t(x)\|_2^2 \cdot \exp\left(-\frac{C_2\|x\|_2^2}{2(\mu_t^2 + C_2\sigma_t^2)}\right)\mathrm{d}x \qquad \left(\text{By Lemma J.3}\right)$$

$$\lesssim \int_{\|x\|_\infty \geq R_4} \left(\frac{|\dot{\mu}_t|}{\mu_t}\cdot\|x\|_\infty + \left|\frac{\dot{\mu}_t}{\mu_t} - \frac{\dot{\sigma}_t}{\sigma_t}\right|\cdot(\|x\|_2+1)\right)^2 \exp\left(-\frac{C_2\|x\|_2^2}{2(\mu_t^2 + C_2\sigma_t^2)}\right)\mathrm{d}x$$
$$\left(\text{By Lemma J.4}\right)$$

$$\lesssim \int_{\|x\|_\infty \geq R_4} \|x\|_2^2 \exp\left(-\frac{C_2\|x\|_2^2}{2(\mu_t^2 + C_2\sigma_t^2)}\right)\mathrm{d}x \qquad \left(\text{By Assumption I.2}\right)$$

$$\leq \int_{\|x\|_2 \geq R_4} \|x\|_2^2 \exp\left(-\frac{C_2\|x\|_2^2}{2(\mu_t^2 + C_2\sigma_t^2)}\right)\mathrm{d}x \qquad \left(\text{By } \|x\|_2 \geq \|x\|_\infty\right)$$

$$\lesssim R_4^{d_x} \exp\left(-\frac{C_2 R_4^2}{2(\mu_t^2 + C_2\sigma_t^2)}\right). \qquad \left(\text{By Lemma D.2}\right)$$

This completes the proof. $\qquad\square$

**Lemma J.11** (Bound on Low-Density Region, Modified from Lemma A.2 of [Fu et al., 2024]). Assume Assumption I.1. Then, for any $R_5, \epsilon_{\text{low}} > 0$, the following two inequalities hold

$$\int_{\|x\|_\infty \leq R_5} \mathbb{1}\{|p_t(x)| < \epsilon_{\text{low}}\} \cdot p_t(x)\mathrm{d}x \leq R_5^{d_x} \cdot \epsilon_{\text{low}},$$

$$\int_{\|x\|_\infty \leq R_5} \mathbb{1}\{|p_t(x)| < \epsilon_{\text{low}}\} \cdot \|u_t(x)\|_2^2 \cdot p_t(x)\mathrm{d}x \lesssim R_5^{d_x+2} \cdot \epsilon_{\text{low}}.$$

*Proof.* The proof for the first inequality is identical to [Fu et al., 2024].

For the second inequality, it follows,

$$\int_{\|x\|_\infty \leq R_5} \mathbb{1}\{|p_t(x)| < \epsilon_{\text{low}}\} \cdot \|u_t(x)\|_2^2 \cdot p_t(x)\mathrm{d}x$$

$$\lesssim \int_{\|x\|_\infty \leq R_5} \mathbb{1}\{|p_t(x)| < \epsilon_{\text{low}}\} \cdot \left(\frac{|\dot{\mu}_t|}{\mu_t}\cdot\|x\|_\infty + \left|\frac{\dot{\mu}_t}{\mu_t} - \frac{\dot{\sigma}_t}{\sigma_t}\right|\cdot(\|x\|_2+1)\right)^2 \cdot p_t(x)\mathrm{d}x$$
$$\left(\text{By Lemma J.4}\right)$$

$$\leq \epsilon_{\text{low}} \int_{\|x\|_\infty \leq R_5} \left(\frac{|\dot{\mu}_t|}{\mu_t}\cdot\|x\|_\infty + \left|\frac{\dot{\mu}_t}{\mu_t} - \frac{\dot{\sigma}_t}{\sigma_t}\right|\cdot(\|x\|_2+1)\right)^2 \mathrm{d}x$$

$$\lesssim R_5^{d_x+2} \cdot \epsilon_{\text{low}}. \qquad \left(\text{By Assumption I.2}\right)$$

This completes the proof. $\qquad\square$

Next, we present the formal proof of Theorem I.1.

**Theorem J.1** (Theorem I.1 Restated: Velocity Approximation with Transformers under Generic Hölder Smoothness). Assume Assumption I.1 and Assumption I.2. For any precision parameter $0 < \epsilon < 1$ and smoothness parameter $\beta > 0$, let $\epsilon \leq O(N^{-\beta})$ for some $N \in \mathbb{N}$. Then, for all

$t \in [t_0, T]$ with $t_0, T \in (0,1)$, there exists a transformer $u_\theta(x,t) \in \mathcal{T}_R^{h,s,r}$ such that

$$\int_{t_0}^{T} \int_{\mathbb{R}^{d_x}} \|u_t(x) - u_\theta(x,t)\|_2^2 \cdot p_t(x)\mathrm{d}x\mathrm{d}t = O\left(B^2 N^{-\beta} \cdot (\log N)^{d_x + \frac{\beta}{2} + 1}\right).$$

Furthermore, the parameter bounds in transformer network $\mathcal{T}_R^{h,s,r}$ satisfy

$$C_{KQ}, C_{KQ}^{2,\infty} = O\left(\lambda^{-1} N^{4\beta d + 2\beta} (\log N)^{4d_x + 2}\right); \quad C_{OV}, C_{OV}^{2,\infty} = O\left(N^{-\beta}\right);$$
$$C_F, C_F^{2,\infty} = O\left(N^\beta (\log N)^{\frac{d_x + \beta}{2} + 1}\right); \quad C_E = O(I); \quad C_\mathcal{T} = O(\sqrt{\log N}).$$

where $\lambda^{-1} = O(N^\beta \cdot \log N)^{4d+3}$ is the inverse-temperature scaling in the softmax function and $O(\cdot)$ hides all polynomial factors depending on $d_x, d, L, \beta, C_1, C_2$.

*Proof of Theorem I.1.* Let $R_6 := C_x \sqrt{\log N}$ and $C_x := \sqrt{4\beta(\mu_t^2 + C_2\sigma_t^2)/C_2}$. Further, we have

$$C_\mathcal{T} = O(\sqrt{\log N}); \quad \epsilon_{\text{low}} = C_3 N^{-\beta} (\log N)^{(d_x + k_1)/2}. \qquad \text{(By Lemma J.9)}$$

First, we decompose the target into three components and bound each of them

$$\int_{t_0}^{T} \int \|u_\theta - u_t\|_2^2 \cdot p_t(x)\mathrm{d}x\mathrm{d}t$$

$$= \underbrace{\int_{t_0}^{T} \int_{\|x\|_\infty > R_6} \|u_\theta - u_t\|_2^2 p_t(x)\mathrm{d}x \, \mathrm{d}t}_{(\mathrm{T}_1)} + \underbrace{\int_{t_0}^{T} \int_{\|x\|_\infty \leq R_6} \mathbb{1}\{p_t(x) < \epsilon_{\text{low}}\}\|u_\theta - u_t\|_2^2 p_t(x)\mathrm{d}x \, \mathrm{d}t}_{(\mathrm{T}_2)}$$

$$+ \underbrace{\int_{t_0}^{T} \int_{\|x\|_\infty \leq R_6} \mathbb{1}\{p_t(x) \geq \epsilon_{\text{low}}\}\|u_\theta - u_t\|_2^2 p_t(x)\mathrm{d}x\mathrm{d}t}_{(\mathrm{T}_3)}.$$

- **Bound on** $(\mathrm{T}_1)$. It holds

$$\int_{\|x\|_\infty > R_6} \|u_\theta - u_t\|_2^2 \cdot p_t(x)\mathrm{d}x$$

$$\leq 2\int_{\|x\|_\infty > R_6} \|u_\theta\|_2^2 \cdot p_t(x) \, \mathrm{d}x + 2\int_{\|x\|_\infty > R_6} \|u_t\|_2^2 \cdot p_t(x)\mathrm{d}x \qquad \text{(By expanding } \ell_2\text{-norm)}$$

$$\leq 2d_x \int_{\|x\|_\infty > R_6} \|u_\theta\|_\infty^2 \cdot p_t(x)\mathrm{d}x + 2\int_{\|x\|_\infty > R_6} \|u_t\|_2^2 \cdot p_t(x)\mathrm{d}x \qquad \left(\text{By } \|\cdot\|_2^2 \leq d_x\|\cdot\|_\infty^2\right)$$

$$\lesssim \underbrace{\int_{\|x\|_\infty > R_6} \log N \cdot p_t(x)\mathrm{d}x}_{(\mathrm{T}_{1.1})} + \underbrace{\int_{\|x\|_\infty > R_6} \|u_t\|_2^2 \cdot p_t(x)\mathrm{d}x}_{(\mathrm{T}_{1.2})}.$$

$$\left(\text{By } C_\mathcal{T} = O(\sqrt{\log N}) \text{ from Lemma J.9}\right)$$

We bound $(\mathrm{T}_{1.1})$ by

$$(\mathrm{T}_{1.1})$$
$$= \log N \cdot \int_{\|x\|_\infty > R_6} p_t(x)\mathrm{d}x$$
$$\lesssim \log N \cdot R_6^{d_x - 2} \exp\left(-\frac{C_2 R_6^2}{2(\mu_t^2 + C_2\sigma_t^2)}\right) \qquad \text{(By Lemma J.10)}$$

$$\lesssim \log N \cdot (\log N)^{\frac{d_x - 2}{2}} N^{-\beta}.$$
$$\left(\text{By the choice of } R_6 = C_x\sqrt{\log N} \text{ and } C_x = \sqrt{2\beta(\mu_t^2 + C_2\sigma_t^2)/C_2}\right)$$

We bound $(\mathrm{T}_{1.2})$ by

$$(\mathrm{T}_{1.2})$$
$$= \int_{\|x\|_\infty > R_6} \|u_t\|_2^2 \cdot p_t(x)\,\mathrm{d}x$$
$$\lesssim \left(\frac{|\dot{\mu}_t|}{\mu_t} + \left|\frac{\dot{\mu}_t}{\mu_t} - \frac{\dot{\sigma}_t}{\sigma_t}\right|\right)^2 \cdot R_6^{d_x} \exp\left(-\frac{C_2 R_6^2}{2(\mu_t^2 + C_2\sigma_t^2)}\right) \qquad \text{(By Lemma J.10)}$$
$$\lesssim \left(\frac{|\dot{\mu}_t|}{\mu_t} + \left|\frac{\dot{\mu}_t}{\mu_t} - \frac{\dot{\sigma}_t}{\sigma_t}\right|\right)^2 \cdot (\log N)^{\frac{d_x}{2}} N^{-\beta}.$$
$$\left(\text{By the choice of } R_6 = C_x\sqrt{\log N} \text{ and } C_x = \sqrt{2\beta(\mu_t^2 + C_2\sigma_t^2)/C_2}\right)$$

Therefore,

$$(\mathrm{T}_1) \lesssim (\mathrm{T}_{1.1}) + (\mathrm{T}_{1.2}) \lesssim \left(\frac{|\dot{\mu}_t|}{\mu_t} + \left|\frac{\dot{\mu}_t}{\mu_t} - \frac{\dot{\sigma}_t}{\sigma_t}\right|\right)^2 \cdot (\log N)^{\frac{d_x}{2}} \cdot N^{-\beta}.$$

- **Bound on** $(\mathrm{T}_2)$. It holds

$$\int_{\|x\|_\infty \le R_6} \mathbb{1}\{p_t(x) < \epsilon_{\mathrm{low}}\} \cdot \|u_\theta - u_t\|_2^2 \cdot p_t(x)\mathrm{d}x$$
$$\le 2\int_{\|x\|_\infty \le R_6} \mathbb{1}\{p_t(x) < \epsilon_{\mathrm{low}}\} \cdot \left(\|u_\theta\|_2^2 + \|u_t\|_2^2\right) \cdot p_t(x)\mathrm{d}x \qquad \text{(By expanding } \ell_2\text{-norm)}$$
$$\le 2\int_{\|x\|_\infty \le R_6} \mathbb{1}\{p_t(x) < \epsilon_{\mathrm{low}}\} \cdot \left(d_x \cdot \|u_\theta\|_\infty^2 + \|u_t\|_2^2\right) \cdot p_t(x)\mathrm{d}x \qquad \left(\text{By } \|\cdot\|_2^2 \le d_x\|\cdot\|_\infty^2\right)$$
$$\lesssim \underbrace{\int_{\|x\|_\infty \le R_6} \mathbb{1}\{p_t < \epsilon_{\mathrm{low}}\} \cdot \|u_\theta\|_\infty^2 \cdot p_t(x)\mathrm{d}x}_{(\mathrm{T}_{2.1})} + \underbrace{\int_{\|x\|_\infty \le R_6} \mathbb{1}\{p_t < \epsilon_{\mathrm{low}}\} \cdot \|u_t\|_2^2 \cdot p_t(x)\,\mathrm{d}x}_{(\mathrm{T}_{2.2})}.$$

We bound $(\mathrm{T}_{2.1})$ by

$$(\mathrm{T}_{2.1})$$
$$= \int_{\|x\|_\infty \le R_6} \mathbb{1}\{p_t(x) < \epsilon_{\mathrm{low}}\} \cdot \|u_\theta(x)\|_\infty^2 \cdot p_t(x)\mathrm{d}x$$
$$\lesssim \log N \cdot \int_{\|x\|_\infty \le R_6} \mathbb{1}\{p_t(x) < \epsilon_{\mathrm{low}}\} \cdot p_t(x)\mathrm{d}x \qquad \left(\text{By } C_\mathcal{T} = O(\sqrt{\log N}) \text{ from Lemma J.9}\right)$$
$$\lesssim \log N \cdot \epsilon_{\mathrm{low}} R_6^{d_x} \qquad\qquad\qquad \text{(By Lemma J.11)}$$
$$\lesssim \log N \cdot (\log N)^{\frac{d_x}{2}} \cdot \epsilon_{\mathrm{low}} \quad \left(\text{By the choice of } R_6 = C_x\sqrt{\log N} \text{ and } C_x = \sqrt{2\beta(\mu_t^2 + C_2\sigma_t^2)/C_2}\right)$$
$$\lesssim \log N(\log N)^{\frac{d_x}{2}} \cdot N^{-\beta} (\log N)^{\frac{d_x+k_1}{2}} \qquad \left(\text{By the choice of } \epsilon_{\mathrm{low}} = C_3 N^{-\beta} (\log N)^{\frac{d_x+k_1}{2}}\right)$$

We bound $(\mathrm{T}_{2.2})$ by

$$(\mathrm{T}_{2.2})$$
$$= \int_{\|x\|_\infty \le R_6} \mathbb{1}\{p_t(x) < \epsilon_{\mathrm{low}}\} \cdot \|u_t\|_2^2 \cdot p_t(x)\mathrm{d}x$$

$$\lesssim \epsilon_{\text{low}} R_6^{d_x+2} \qquad\qquad\qquad\qquad\qquad\qquad\qquad\qquad \left(\text{By Lemma J.11}\right)$$

$$\lesssim \epsilon_{\text{low}} (\log N)^{\frac{d_x+2}{2}} \qquad \left(\text{By the choice of } R_6 = C_x\sqrt{\log N} \text{ and } C_x = \sqrt{2\beta(\mu_t^2 + C_2\sigma_t^2)/C_2}\right)$$

$$\leq N^{-\beta} (\log N)^{\frac{d_x+k_1}{2}} \cdot (\log N)^{\frac{d_x+2}{2}}. \qquad \left(\text{By the choice of } \epsilon_{\text{low}} = C_3 N^{-\beta} (\log N)^{\frac{d_x+k_1}{2}}\right)$$

Therefore,

$$(T_2) \lesssim (T_{2.1}) + (T_{2.2}) \lesssim N^{-\beta}(\log N)^{d_x+\frac{k_1}{2}+1}.$$

- **Bound on** $(T_3)$**.** We bound term $(T_3)$ by

$$(T_3)$$

$$= \int_{t_0}^{T} \int_{\|x\|_\infty \leq R_6} \mathbb{1}\{p_t(x) \geq \epsilon_{\text{low}}\} \cdot \|u_\theta - u_t\|_2^2 \cdot p_t(x)\mathrm{d}x\mathrm{d}t$$

$$= \int_{t_0}^{T} \int_{\|x\|_\infty \leq R_6} \frac{1}{p_t}\mathbb{1}\{p_t(x) \geq \epsilon_{\text{low}}\} \cdot d_x\|u_\theta - u_t\|_2^2 \cdot (p_t(x))^2\mathrm{d}x\mathrm{d}t \quad \left(\text{By multiplying } p_t/p_t\right)$$

$$\leq \int_{t_0}^{T} \int_{\|x\|_\infty \leq R_6} \frac{1}{\epsilon_{\text{low}}}\mathbb{1}\{p_t(x) \geq \epsilon_{\text{low}}\} \cdot d_x\|u_\theta - u_t\|_2^2 \cdot (p_t(x))^2\mathrm{d}x\mathrm{d}t \quad \left(\text{By } 1/p_t \leq 1/\epsilon_{\text{low}}\right)$$

$$\leq \frac{d_x}{\epsilon_{\text{low}}} \cdot \left(\frac{|\dot\mu_t|}{\mu_t} + \left|\frac{\dot\mu_t}{\mu_t} - \frac{\dot\sigma_t}{\sigma_t}\right|\right)^2 \cdot B^2 N^{-2\beta}(\log N)^{\frac{3d_x}{2}+k_1+1} \qquad\qquad \left(\text{By Lemma J.9}\right)$$

$$\lesssim N^{\beta}(\log N)^{\frac{-(d_x+k_1)}{2}} \cdot \left(\frac{|\dot\mu_t|}{\mu_t} + \left|\frac{\dot\mu_t}{\mu_t} - \frac{\dot\sigma_t}{\sigma_t}\right|\right)^2 \cdot B^2 N^{-2\beta}(\log N)^{\frac{3d_x}{2}+k_1+1}$$

$$\left(\text{By the choice of } \epsilon_{\text{low}}\right)$$

$$= \left(\frac{|\dot\mu_t|}{\mu_t} + \left|\frac{\dot\mu_t}{\mu_t} - \frac{\dot\sigma_t}{\sigma_t}\right|\right)^2 \cdot B^2 N^{-\beta} \cdot (\log N)^{d_x+\frac{k_1}{2}+1}.$$

By the upper-bound on $(T_1)$, $(T_2)$ and $(T_3)$, we have

$$\int_{t_0}^{T} \int \|u_t(x) - u_\theta(x,t)\|_2^2 \cdot p_t \, \mathrm{d}x\mathrm{d}t$$

$$\lesssim (T_1) + (T_2) + (T_3) \qquad\qquad\qquad\qquad\qquad\qquad \left(\text{By } t_0, T \in (0,1)\right)$$

$$= \left(\frac{|\dot\mu_t|}{\mu_t} + \left|\frac{\dot\mu_t}{\mu_t} - \frac{\dot\sigma_t}{\sigma_t}\right|\right)^2 \cdot O\left(B^2 N^{-\beta} \cdot (\log N)^{d_x+\frac{k_1}{2}+1}\right)$$

$$\leq \left(\frac{|\dot\mu_t|}{\mu_t} + \left|\frac{\dot\mu_t}{\mu_t} - \frac{\dot\sigma_t}{\sigma_t}\right|\right)^2 \cdot O\left(B^2 N^{-\beta} \cdot (\log N)^{d_x+\frac{\beta}{2}+1}\right) \qquad\qquad \left(\text{By } k_1 \leq \beta\right)$$

$$\leq O\left(B^2 N^{-\beta} \cdot (\log N)^{d_x+\frac{\beta}{2}+1}\right). \qquad\qquad\qquad\qquad \left(\text{By Assumption I.2}\right)$$

Furthermore, the transformer parameter bounds follow Lemma J.9.

This completes the proof. $\qquad\qquad\qquad\qquad\qquad\qquad\qquad\qquad\qquad\qquad\qquad\qquad \square$

# K    Proof of Theorem I.2

In this section, we derives a tighter error bound for velocity approximation using transformers.

**Organizations.**  Appendix K.1 introduces auxiliary lemmas. Appendix K.2 establishes a bound on the velocity approximation error over a bounded domain by applying the universal approximation of transformers. Appendix K.3 presents the main proof by incorporating the bounded-domain approximation error and controlling the unbounded region using the sub-Gaussian assumption.

## K.1    Auxiliary Lemmas

In this section, we introduce auxiliary lemmas for velocity field approximation. Specifically, Lemma K.1 applies a stronger Hölder assumption to decompose the density function $p_t(x)$. Lemma K.2 further decomposes the velocity into two components, differing from the decomposition under a generic Hölder assumption. Then, Lemma K.3 and Lemma K.4 establish upper and lower bounds for the decomposed components and the velocity in $\ell_\infty$-distance, respectively.

We begin with the density function decomposition.

**Lemma K.1** (Density Function Decomposition, Lemma B.1 of [Fu et al., 2024]).  Assume Assumption I.3. Then, the density function $p_t(x)$ and $\nabla \log p_t(x)$ follow the decomposition:

$$p_t(x) = \frac{1}{(\mu_t^2 + C_2 \cdot \sigma_t^2)^{d_x/2}} \exp\left( \frac{-C_2\|x\|_2^2}{2(\mu_t^2 + C_2\sigma_t^2)} \right) h(x,t),$$

$$\nabla \log p_t(x) = \frac{-C_2 x}{\mu_t^2 + C_2\sigma_t^2} + \frac{\nabla h(x,t)}{h(x,t)},$$

where $h(x,t) := \int \frac{f(x_1)}{(2\pi)^{d_x/2}\widehat{\sigma}_t^{d_x}} \exp\left( -\frac{\|x_1 - \widehat{\mu}_t x\|_2^2}{2\widehat{\sigma}_t} \right) \mathrm{d}x_1, \widehat{\sigma}_t := \frac{\sigma_t}{(\mu_t^2 + C_2\sigma_t^2)^{1/2}}$ and $\widehat{\mu}_t := \frac{\mu_t}{(\mu_t^2 + C_2\sigma_t^2)}$.

Then, we give the velocity field decomposition.

**Lemma K.2** (Velocity Decomposition under Stronger Hölder Smoothness Assumption).  Assume Assumption I.3. Then, the velocity field $u_t(x)$ follows the decomposition:

$$u_t(x) = \frac{\dot{\mu}_t}{\mu_t}x - \left(\dot{\sigma}_t\sigma_t - \frac{\dot{\mu}_t\sigma_t^2}{\mu_t}\right)\left( \frac{-C_2 x}{\mu_t^2 + C_2\sigma_t^2} + \frac{\nabla h(x,t)}{h(x,t)} \right).$$

**Remark K.1.**  The key aspect of Lemma K.2 is the velocity field $u_t(x)$ having a denominator bounded away from zero. Specifically, we apply $f(x_1) \geq C$ to derive the lower bound on $h(x,t)$ (Lemma K.3). This removes the need to impose an additional lower threshold on the density function approximator. In contrast, under Assumption I.1, the approximator is constrained to stay above the threshold $\epsilon_{\text{low}}$ to prevent explosion, and therefore leads to slower approximation rate.

*Proof.*  Our proof builds on [Fu et al., 2024].

By Lemma J.1, the velocity field $u_t(x)$ has the form

$$u_t(x) = \Phi_1(x,t)^{-1}\left( \frac{\dot{\mu}_t}{\mu_t} \cdot \Phi_2(x,t) + (\dot{\sigma}_t - \frac{\dot{\mu}_t\sigma_t}{\mu_t})\Phi_3(x,t) \right),$$

where

$$\Phi_1(x,t) = \int_{\mathbb{R}^{d_x}} \frac{1}{\sigma_t^{d_x}(2\pi)^{d_x/2}} \exp\left( -\frac{\|x - \mu_t \cdot x_1\|^2}{2\sigma_t^2} \right) \cdot q(x_1)\,\mathrm{d}x_1,$$

$$\Phi_2(x,t) = x\int_{\mathbb{R}^{d_x}} \frac{1}{\sigma_t^{d_x}(2\pi)^{d_x/2}} \exp\left( -\frac{\|x - \mu_t \cdot x_1\|^2}{2\sigma_t^2} \right) \cdot q(x_1)\,\mathrm{d}x_1,$$

$$\Phi_3(x,t) = \int_{\mathbb{R}^{d_x}} \left( \frac{x - \mu_t \cdot x_1}{\sigma_t} \right) \cdot \frac{1}{\sigma_t^{d_x}(2\pi)^{d_x/2}} \exp\left( -\frac{\|x - \mu_t \cdot x_1\|^2}{2\sigma_t^2} \right) \cdot q(x_1)\, dx_1.$$

Furthermore, we have

$$\sigma_t \nabla p_t(x)$$

$$= -\sigma_t \int \left( \frac{x - \mu_t \cdot x_1}{\sigma_t^2} \right) \cdot \frac{1}{\sigma_t^{d_x}(2\pi)^{d_x/2}} \exp\left( -\frac{\|x - \mu_t \cdot x_1\|^2}{2\sigma_t^2} \right) q(x_1)\, dx_1$$

$$= -\int \left( \frac{x - \mu_t \cdot x_1}{\sigma_t} \right) \cdot \frac{1}{\sigma_t^{d_x}(2\pi)^{d_x/2}} \exp\left( -\frac{\|x - \mu_t \cdot x_1\|^2}{2\sigma_t^2} \right) q(x_1)\, dx_1$$

$$= -\Phi_3(x,t).$$

Therefore,

$$u_t(x)$$

$$= \Phi_1^{-1}\left( \frac{\dot{\mu}_t}{\mu_t} \cdot \Phi_2 + (\dot{\sigma}_t - \frac{\dot{\mu}_t \sigma_t}{\mu_t})\Phi_3 \right)$$

$$= \frac{\dot{\mu}_t}{\mu_t} x - (\dot{\sigma}_t - \frac{\dot{\mu}_t \sigma_t}{\mu_t})\sigma_t \nabla \log p_t \qquad \left( \text{By } \Phi_2 = x\Phi_1 \text{ and } \Phi_3 = -\sigma_t \nabla p_t \right)$$

$$= \frac{\dot{\mu}_t}{\mu_t} x - (\dot{\sigma}_t \sigma_t - \frac{\dot{\mu}_t \sigma_t^2}{\mu_t})\left( \frac{-C_2 x}{\mu_t^2 + C_2 \sigma_t^2} + \frac{\nabla h(x,t)}{h(x,t)} \right). \qquad \left( \text{By Lemma K.1} \right)$$

This completes the proof. $\qquad \square$

The next lemma bounds $h(x,t)$.

**Lemma K.3** (Lemma B.8 of [Fu et al., 2024]).  Assume Assumption I.3. Then, it holds

$$C_1 \le h(x,t) \le B, \quad \left\| \frac{\widehat{\sigma}_t}{\widehat{\mu}_t} \nabla h(x,t) \right\|_\infty \le \sqrt{\frac{2}{\pi}} B.$$

Lemma K.3 ensures that $h(x,t)$ remains bounded above and below by a constant. As a result, $u_t(x)$ stays finite for all $x$. This eliminates the need for an additional threshold $\epsilon_{\text{low}}$ (Definition J.1) in the constructed approximator to prevent divergence, leading to a faster approximation rate.

**Bound on Velocity Field.** We give the $\ell_\infty$-bound on $u_t(x)$ under stronger Hölder assumption.

**Lemma K.4** (Bounds on Velocity Field).  Assume Assumption I.3. Then, there exist a positive constant $C_6$ such that

$$\|u_t(x)\|_\infty \le \left| \frac{\dot{\mu}_t}{\mu_t} + (\dot{\sigma}_t \sigma_t - \frac{\dot{\mu}_t \sigma_t^2}{\mu_t})(\frac{C_2}{\mu_t^2 + C_2 \sigma_t^2}) \right| \|x\|_\infty + C_6 \left| \dot{\sigma}_t - \frac{\dot{\mu}_t \sigma_t}{\mu_t} \right|.$$

*Proof.* Recalling from Lemma K.2 and Lemma K.3, the velocity field has the expression

$$u_t(x) = \frac{\dot{\mu}_t}{\mu_t} x - (\dot{\sigma}_t \sigma_t - \frac{\dot{\mu}_t \sigma_t^2}{\mu_t})\left( \frac{-C_2 x}{\mu_t^2 + C_2 \sigma_t^2} + \frac{\nabla h(x,t)}{h(x,t)} \right),$$

where $\widehat{\sigma}_t = \sigma_t / \sqrt{\mu_t^2 + C_2 \sigma_t^2}$, $\widehat{\mu}_t = \mu_t / (\mu_t^2 + C_2 \sigma_t^2)$ and

$$h(x,t) = \int f(x_1) \frac{1}{(2\pi)^{d_x/2} \cdot \widehat{\sigma}_t^{d_x}} \cdot \exp\left( -\frac{\|x_1 - \widehat{\mu}_t \cdot x\|_2^2}{2\widehat{\sigma}_t} \right)\, dx_1.$$

By Lemma K.3 and Assumption I.2, it holds

$$\|\frac{\nabla h(x,t)}{h(x,t)}\|_\infty \leq \frac{\widehat{\mu}_t}{\widehat{\sigma}_t} \cdot \sqrt{\frac{2}{\pi}} BC_1 = O(\frac{1}{\sigma_t}). \qquad \left(\text{By Lemma K.3}\right)$$

Therefore,

$$\|u_t(x)\|_\infty$$
$$\leq \left|\frac{\dot{\mu}_t}{\mu_t} + (\dot{\sigma}_t \sigma_t - \frac{\dot{\mu}_t \sigma_t^2}{\mu_t})(\frac{C_2}{\mu_t^2 + C_2 \sigma_t^2})\right| \cdot \|x\|_\infty + \left|\dot{\sigma}_t \sigma_t - \frac{\dot{\mu}_t \sigma_t^2}{\mu_t}\right| \cdot \left\|\frac{\nabla h(x,t)}{h(x,t)}\right\|_\infty$$
$$\left(\text{By triangle inequality}\right)$$
$$\leq \left|\frac{\dot{\mu}_t}{\mu_t} + (\dot{\sigma}_t \sigma_t - \frac{\dot{\mu}_t \sigma_t^2}{\mu_t})(\frac{C_2}{\mu_t^2 + C_2 \sigma_t^2})\right| \cdot \|x\|_\infty + C_6\left|\dot{\sigma}_t - \frac{\dot{\mu}_t \sigma_t}{\mu_t}\right|, \qquad \left(\text{By }\left(\left(\text{By Lemma K.3}\right)\right)\right)$$

for some positive constant $C_6$.

This completes the proof. $\qquad\qquad\square$

## K.2  Velocity Approximation on Bounded Domain

In this section, we approximate the velocity field $u_t(x)$ using transformers in two steps. The first step constructs two compactly supported continuous functions, $Q_1(x,t)$ and $Q_2(x,t)$, as approximations of $h(x,t)$ and $\nabla h(x,t)$ (Lemma K.5 and Lemma K.6). The second step applies the universal approximation of transformers to approximate $Q_1(x,t)$ and $Q_2(x,t)$ (Lemma K.7). Combining these steps, we present the velocity approximation on a bounded domain in Lemma K.8.

Before proceeding, we reiterate the expression of decomposed velocity under Assumption I.3.

$$u_t(x) = \frac{\dot{\mu}_t}{\mu_t} x - (\dot{\sigma}_t \sigma_t - \frac{\dot{\mu}_t \sigma_t^2}{\mu_t}) \left(\frac{-C_2 x}{\mu_t^2 + C_2 \sigma_t^2} + \frac{\nabla h(x,t)}{h(x,t)}\right).$$

Then, we construct two local polynomials as the approximators for $h(x,t)$, and $\nabla h(x,t)$.

**Approximation of $h(x,t)$ and $\nabla h(x,t)$.** The differences between

$$h(x,t) = \int f(x_1) \cdot \frac{1}{(2\pi)^{d_x/2} \cdot \widehat{\sigma}_t^{d_x}} \cdot \exp\left(-\frac{\|x_1 - \widehat{\mu}_t \cdot x\|_2^2}{2\widehat{\sigma}_t}\right) dx_1,$$

and

$$p_t(x) = \int q(x_1) \cdot \frac{1}{(2\pi)^{d_x/2} \cdot \sigma_t^{d_x}} \cdot \exp\left(-\frac{\|x_1 - \mu_t \cdot x\|_2^2}{2\sigma_t}\right) dx_1,$$

lie in (i) the target function $f(x_1)$ and $q(x_1)$ (ii) the path coefficients $\widehat{\sigma}_t$, $\widehat{\mu}_t$ and $\sigma_t$, $\mu_t$.

We define local polynomial $\Psi_1(x,t)$ as the approximator for $p_t(x)$ in (J.4). Given the differences between $h$ and $p_t$, the construction of an approximator for $h(x,t)$ follows the formulation of $\Psi_1$.

Formally, we approximate $h(x,t)$ around $x$ with:

$$Q_1(x,t) := \sum_{v \in [N]^{d_x}} \sum_{\|n_x\|_1 \leq k_1} \frac{R_B^{\|n_x\|_1}}{n_x!} \frac{\partial^{n_x} f}{\partial x^{n_x}}\bigg|_{x=R_B(\frac{v}{N} - \frac{1}{2})} g_1(x, n_x, v, t), \qquad (K.1)$$

where $n_x \in \mathbb{Z}^{d_x}$ is a multi-index, $R_B > 0$ is a constant depending on the Hölder ball radius $B$,

- $g_1(x, n_x, v, t) := \prod_{i=1}^{d_x} \sum_{k_2 < p} g_2(x[i], n_x[i], v[i], k_2)$, and

- $g_2(x[i], n_x[i], v[i], k_2) := \frac{1}{\widehat{\sigma}_t \sqrt{2\pi}} \int \left( \frac{x_1[i]}{R_B} + \frac{1}{2} - \frac{v[i]}{N} \right)^{n_x[i]} \frac{1}{k_2!} \left( -\frac{|x[i] - \widehat{\mu}_t x_1[i]|^2|^2}{2\widehat{\sigma}_t} \right)^{k_2} dx_1.$

**Remark K.2.** Given the differences between $h(x,t)$ and $p_t(x)$, we replace (i) $\partial^{n_x} \Phi_1 / \partial x^{n_x}$ with $\partial^{n_x} f / \partial x^{n_x}$ (ii) $\sigma_t$ and $\mu_t$ with $\widehat{\sigma}_t$ and $\widehat{\mu}_t$ respectively. Then, the formulation of $Q_1(x,t)$ follows constructions identical to the density function approximator $\Psi_1(x,t)$.

**Remark K.3.** When the context is clear, we refer to $Q_1(x,t)$ as a local polynomial and distinguish it from $\Psi_1(x,t)$. The generic Hölder assumption (Assumption I.1) applies to $\Psi_1(x,t)$, while the stronger Hölder assumption (Assumption I.3) applies to $Q_1(x,t)$.

Then, we approximate $h(x,t)$ using $Q_1(x,t)$.

**Lemma K.5** (Approximate of $h(x,t)$, Lemma B.4 of [Fu et al., 2024]). Assume Assumption I.3. Let $Q_1(x,t)$ be the approximator of $h(x,t)$, and $C_x(d_x, \beta, C_1, C_2)$ be a positive constant. Then, for any $t \in [0,1]$ and $x \in [-C_x \sqrt{\log N}, C_x \sqrt{\log N}]^{d_x}$, it holds

$$|Q_1(x,t) - h(x,t)| \lesssim B N^{-\beta} (\log N)^{\frac{k_1}{2}}.$$

Based on the approximation of $h(x,t)$ using local polynomial $Q_1(x,t)$, we construct a approximator of $\nabla h(x,t)$ following similar formulation

**Definition K.1** (Approximator of $\nabla h(x,t)$). We define $Q_2(x,t)$ as the approximator of $\nabla h(x,t)$, with each component $Q_2[i]$ following the form of local polynomial presented in (K.1).

Then, we approximate $h'(x,t)$ and $\nabla h(x,t)$ with $Q_2(x,t)$.

**Lemma K.6** (Approximate $\nabla h(x,t)$, Lemma B.6 of [Fu et al., 2024]). Assume Assumption I.3. Let $C_x(d_x, \beta, C_1, C_2)$ be a positive constant. Then, for all $x \in [-C_x \sqrt{\log N}, C_x \sqrt{\log N}]^{d_x}$, $i \in [d_x]$ and $t > 0$, it holds

$$\left| Q_2(x,t)[i] - \frac{\widehat{\sigma}_t}{\widehat{\mu}_t} \cdot \nabla h(x,t)[i] \right| \lesssim B N^{-\beta} (\log N)^{\frac{k_1+1}{2}}.$$

**Approximate Velocity Approximator with Transformers** Before deriving the velocity approximation with transformers on a bounded domain, we first approximate the velocity approximator constructed with $Q_1(x,t)$ and $Q_2(x,t)$ using transformers.

**Lemma K.7** (Approximate Velocity Approximators with Transformers Network). Assume Assumption I.3. Let $C_x$ be a positive constant dependent on $d_x, \beta, C_1$ and $C_2$. Then, for any $x \in [-C_x \sqrt{\log N}, C_x \sqrt{\log N}]^{d_x}$ and $t \in [0,1]$, there exist a transformer $\mathcal{T} \in \mathcal{T}_R^{h,s,r}$ such that,

$$\int_0^1 \int \left\| \mathcal{T}(x,t) - \frac{\dot{\mu}_t}{\mu_t} x + (\dot{\sigma}_t \sigma_t - \frac{\dot{\mu}_t \sigma_t^2}{\mu_t}) \left( \frac{-C_2 x}{\mu_t^2 + C_2 \sigma_t^2} + \frac{\widehat{\mu}_t \nabla Q_2[i]}{\widehat{\sigma}_t Q_1} \right) \right\|_2^2 dx dt \leq \epsilon^2.$$

Further, the parameter bounds in the transformer network class follows Lemma J.8.

*Proof.* The proof closely follows Lemma J.8. □

**Approximate Velocity with Transformers on Bounded Domain.** We incorporate the approximations with $Q_1, Q_2$ and $\mathcal{T}(x,t)$ to derive the velocity approximation on a bounded domain.

**Lemma K.8** (Velocity Approximation with Transformers on Bounded Domain). Assume Assumption I.3. Then, for any $x \in [-C_x \sqrt{\log N}, C_x \sqrt{\log N}]^{d_x}$ and $t \in [t_0, T]$ with a positive constant $C_x(d_x, \beta, C_1, C_2)$ and $t_0, T \in (0,1)$, there exist a $u_\theta(x,t) \in \mathcal{T}_R^{h,s,r}$ such that

$$\int_{t_0}^T \int_{\|x\|_\infty \leq C_x \sqrt{\log N}} \|u_t(x) - u_\theta(x,t)\|_2^2 p_t(x) dx dt \lesssim B^2 N^{-2\beta} (\log N)^{k_1+d_x}.$$

Further, the parameter bounds in the transformer network class follows Lemma J.9.

*Proof.* Building upon [Hu et al., 2025c, Fu et al., 2024], we prove Lemma K.8 with two steps.

- **Step 1: Approximate velocity with constructed function.** We approximate the decomposed velocity field (Lemma K.2) and its components with approximator $Q_1(x,t)$ and $Q_2(x,t)$.

- **Step 2: Approximate with transformers.** We apply the universal approximation of transformers presented in Appendix H to approximate the constructed function in Step 1.

Before proceeding, we recall some previous lemmas to prepare our proof.

By Lemma K.2, the velocity follows the decomposition under Assumption I.3:

$$u_t(x) = \frac{\dot{\mu}_t}{\mu_t}x - (\dot{\sigma}_t\sigma_t - \frac{\dot{\mu}_t\sigma_t^2}{\mu_t})\left(\frac{-C_2 x}{\mu_t^2 + C_2\sigma_t^2} + \frac{\nabla h(x,t)}{h(x,t)}\right),$$

where $\hat{\sigma}_t = \sigma_t/(\mu_t^2 + C_2\sigma_t^2)^{1/2}$, $\hat{\mu}_t = \mu_t/(\mu_t^2 + C_2\sigma_t^2)$ and

$$h(x,t) = \int f(x_1)\frac{1}{(2\pi)^{d_x/2}\cdot\hat{\sigma}_t^{d_x}}\cdot\exp\left(-\frac{\|x_1 - \hat{\mu}_t\cdot x\|_2^2}{2\hat{\sigma}_t}\right)\,\mathrm{d}x_1.$$

Furthermore, by Lemma K.4, the bound on $u_t(x)$ in $\ell_\infty$-distance follows

$$\|u_t(x)\|_\infty \le \left|\frac{\dot{\mu}_t}{\mu_t} + (\dot{\sigma}_t\sigma_t - \frac{\dot{\mu}_t\sigma_t^2}{\mu_t})(\frac{C_2}{\mu_t^2 + C_2\sigma_t^2})\right|\cdot\|x\|_\infty + C_6\left|\dot{\sigma}_t - \frac{\dot{\mu}_t\sigma_t}{\mu_t}\right|.$$

First, we apply $\|x\|_2 \lesssim \sqrt{\log N}$ to Lemma K.4. Next, we apply Lemma K.4 and Lemma K.6 to construct the first-step approximator $Q(x,t)\in\mathbb{R}^{d_x}$, with each element defined by:

$$\|Q[i]\| := \min\left\{\frac{\dot{\mu}_t}{\mu_t}x - (\dot{\sigma}_t\sigma_t - \frac{\dot{\mu}_t\sigma_t^2}{\mu_t})\left(\frac{-C_2 x}{\mu_t^2 + C_2\sigma_t^2} + \frac{\hat{\mu}_t\nabla Q_2[i]}{\hat{\sigma}_t Q_1}\right), \|u_t(x)\|_\infty\right\}. \tag{K.2}$$

The first element consists of approximators for $h(x,t)$ and $\nabla h(x,t)$. The second element ensures that $\Psi(x,t)$ does not output value larger than the maximum of $u_t(x)$ in $\ell_\infty$.

- **Step A: Approximation via Local Polynomial.**

  By symmetry, for all $i\in[d_x]$, the difference between $Q(x,t)[i]$ and $u_t(x)[i]$ follows

$$|u_t[i] - Q[i]|$$
$$= \left|(\dot{\sigma}_t\sigma_t - \frac{\dot{\mu}_t\sigma_t^2}{\mu_t})\left(\frac{\nabla h[i]}{h} - \frac{\hat{\mu}_t Q_2[i]}{\hat{\sigma}_t Q_1}\right)\right|$$
$$\le \left|\dot{\sigma}_t\sigma_t - \frac{\dot{\mu}_t\sigma_t^2}{\mu_t}\right|\cdot\left|\left(\frac{\nabla h[i]}{h} - \frac{\hat{\mu}_t Q_2[i]}{\hat{\sigma}_t Q_1}\right)\right|$$
$$= \left|\dot{\sigma}_t\sigma_t - \frac{\dot{\mu}_t\sigma_t^2}{\mu_t}\right|\cdot\underbrace{\left|\frac{\nabla h[i]}{h} - \frac{\nabla h[i]}{Q_1})\right|}_{(\text{T}_1)} + \left|\dot{\sigma}_t\sigma_t - \frac{\dot{\mu}_t\sigma_t^2}{\mu_t}\right|\cdot\underbrace{\left|\frac{\nabla h[i]}{Q_1} - \frac{\hat{\mu}_t Q_2[i]}{\hat{\sigma}_t Q_1}\right|}_{(\text{T}_2)}.$$

(By triangle inequality)

  Next, we bound $(\text{T}_1)$ and $(\text{T}_2)$.

**Step A.1: Bound** $(\mathrm{T}_1)$**.** By Lemma K.3, we have $C_1 \leq h \leq B$ and

$$\left\| \frac{\widehat{\sigma}_t}{\widehat{\mu}_t} \nabla h(x,t) \right\|_\infty \leq \sqrt{\frac{2}{\pi}} B.$$

Moreover, by Lemma K.5, it holds

$$|Q_1(x,t) - h(x,t)| \lesssim BN^{-\beta} (\log N)^{\frac{k_1}{2}}.$$

It implies that

$$h(x,t) - K'BN^{-\beta} (\log N)^{\frac{k_1}{2}} \leq Q_1(x,t),$$

for some positive constant $K'$. This gives

$$|Q_1(x,t)| \lesssim BN^{-\beta}(\log N)^{\frac{k_1}{2}}. \tag{K.3}$$

Therefore,

$$
\begin{aligned}
(\mathrm{T}_1) &= \left| \frac{\nabla h[i]}{h} - \frac{\nabla h[i]}{Q_1} \right| \\
&\leq |\nabla h[i]| \cdot \left| \frac{h - Q_1}{hQ_1} \right| \\
&\leq \sqrt{\frac{2}{\pi}} \frac{\widehat{\mu}_t}{\widehat{\sigma}_t} B \left| \frac{h - Q_1}{hQ_1} \right| && \left(\text{By Lemma K.3}\right) \\
&\lesssim \frac{B}{\sigma_t} N^{-\beta} (\log N)^{\frac{k_1}{2}}.
\end{aligned}
$$

**Step 1.2: Bound** $(\mathrm{T}_2)$**.** It holds

$$
\begin{aligned}
(\mathrm{T}_2) &= \left| \frac{\nabla h[i]}{Q_1} - \frac{\widehat{\mu}_t Q_2[i]}{\widehat{\sigma}_t Q_1} \right| \\
&\leq \frac{\widehat{\mu}_t}{\widehat{\sigma}_t} \left| \frac{Q_2[i] - \frac{\widehat{\sigma}_t}{\widehat{\mu}_t} \nabla h[i]}{Q_1} \right| && \left(\text{By factoring out } \widehat{\mu}_t/\widehat{\sigma}_t\right) \\
&\lesssim \frac{B}{\sigma_t} N^{-\beta} (\log N)^{\frac{k_1+1}{2}}. && \left(\text{By (K.3) and Lemma K.6}\right)
\end{aligned}
$$

Combining bounds on $(\mathrm{T}_1)$ and $(\mathrm{T}_2)$, it holds

$$
\begin{aligned}
&|u_t[i] - Q[i]| && \text{(K.4)} \\
&\leq \left| \dot{\sigma}_t \sigma_t - \frac{\dot{\mu}_t \sigma_t^2}{\mu_t} \right| \cdot ((\mathrm{T}_1) + (\mathrm{T}_2)) \\
&\lesssim \left| \dot{\sigma}_t \sigma_t - \frac{\dot{\mu}_t \sigma_t^2}{\mu_t} \right| BN^{-\beta} (\log N)^{\frac{k_1+1}{2}},
\end{aligned}
$$

for all $i \in [d_x]$.

Therefore, by symmetry, it holds

$$
\begin{aligned}
&\|u_t(x) - Q(x,t)\|_2^2 && \text{(K.5)} \\
&\leq d_x \|u_t(x) - Q(x,t)\|_\infty^2 && \left(\text{By } \| \cdot \|_2 \leq d_x \| \cdot \|_\infty\right)
\end{aligned}
$$

$$\lesssim \left| \dot{\sigma}_t \sigma_t - \frac{\dot{\mu}_t \sigma_t^2}{\mu_t} \right|^2 B^2 N^{-2\beta} (\log N)^{k_1}. \qquad \text{(By (K.4))}$$

- **Step B: Approximation with Transformers.**

  By Lemma K.7, there exists a transformer $u_\theta(x, t) \in \mathcal{T}_R^{h,r,s}$ such that

  $$\int \int \left\| u_\theta(x, t) - \frac{\dot{\mu}_t}{\mu_t} x + (\dot{\sigma}_t \sigma_t - \frac{\dot{\mu}_t \sigma_t^2}{\mu_t})(\frac{-C_2 x}{\mu_t^2 + C_2 \sigma_t^2} + \frac{\widehat{\mu}_t \nabla Q_2[i]}{\widehat{\sigma}_t Q_1}) \right\|_2^2 \mathrm{d}x \mathrm{d}t \leq \epsilon^2.$$

  By setting $\epsilon := N^{-\beta}$, the velocity approximation using transformers follows

  $$\int \int p_t \| u_t(x) - u_\theta(x, t) \|_2^2 \mathrm{d}x \mathrm{d}t$$

  $$\leq \int \int p_t \| u_t(x) - Q(x, t) \|_2^2 \mathrm{d}x \mathrm{d}t + \int \int p_t \| Q(x, t) - u_\theta(x, t) \|_2^2 \mathrm{d}x \mathrm{d}t$$
  $$\text{(By triangle inequality)}$$

  $$\leq \int \int \| u_t(x) - Q(x, t) \|_2^2 \mathrm{d}x \mathrm{d}t + \int \int \| Q(x, t) - u_\theta(x, t) \|_2^2 \mathrm{d}x \mathrm{d}t \qquad \text{(By } 0 \leq p_t(x) \leq 1\text{)}$$

  $$\lesssim \left| \dot{\sigma}_t \sigma_t - \frac{\dot{\mu}_t \sigma_t^2}{\mu_t} \right|^2 B^2 N^{-2\beta} (\log N)^{k_1} \int \int \mathrm{d}x \mathrm{d}t + \int \| Q(x, t) - u_\theta(x, t) \|_2^2 \mathrm{d}x \mathrm{d}t \quad \text{(By (K.5))}$$

  $$\leq \left| \dot{\sigma}_t \sigma_t - \frac{\dot{\mu}_t \sigma_t^2}{\mu_t} \right|^2 B^2 N^{-2\beta} (\log N)^{k_1 + d_x} + \int \| Q(x, t) - u_\theta(x, t) \|_2^2 \mathrm{d}x \mathrm{d}t$$
  $$\left( \text{By } t \in (0, 1) \text{ and } \| x \|_\infty \leq C_x \sqrt{\log N} \right)$$

  $$\lesssim \left| \dot{\sigma}_t \sigma_t - \frac{\dot{\mu}_t \sigma_t^2}{\mu_t} \right|^2 B^2 N^{-2\beta} (\log N)^{k_1 + d_x}, \qquad \text{(By Lemma K.7)}$$

  $$\lesssim B^2 N^{-2\beta} (\log N)^{k_1 + d_x}. \qquad \text{(By Assumption I.2)}$$

  By Lemma K.4, it holds

  $$\| u_t(x) \|_\infty$$
  $$\leq \left| \frac{\dot{\mu}_t}{\mu_t} + (\dot{\sigma}_t \sigma_t - \frac{\dot{\mu}_t \sigma_t^2}{\mu_t})(\frac{C_2}{\mu_t^2 + C_2 \sigma_t^2}) \right| \cdot \| x \|_\infty + C_6 \left| \dot{\sigma}_t - \frac{\dot{\mu}_t \sigma_t}{\mu_t} \right|. \qquad \text{(By Lemma K.4)}$$
  $$\lesssim O(\sqrt{\log N}). \qquad \text{(By Assumption I.2)}$$

  Therefore, we set transformer output bound $C_\mathcal{T} := O(\| u_t(x) \|_\infty)$. Then, the parameter bounds in the transformer network follow Lemma J.9.

  This completes the proof. $\qquad \square$

## K.3 Main Proof of Theorem I.2

In Lemma J.9, we give the velocity field approximation using transformer on a bounded domain $x \in [-C_x \sqrt{\log N}, C_x \sqrt{\log N}]^{d_x}$ under stronger Hölder assumption. To obtain general approximation result, we introduce the next lemma that bounds the uncontrolled region.

**Lemma K.9** (Truncation of $x$, Modified from Lemma B.2 of [Fu et al., 2024]). Assume Assumption I.3. Then, for any $R_7 > 1$ and $t > 0$, the following hold

$$\int_{\| x \|_\infty \geq R_7} p_t(x) \mathrm{d}x \lesssim R_7^{d_x - 2} \exp\left( -\frac{C_2 R_7^2}{2(\mu_t^2 + C_2 \sigma_t^2)} \right),$$

$$\int_{\|x\|_\infty \geq R_7} \|u_t(x)\|_2^2 \cdot p_t(x) \mathrm{d}x \lesssim R_7^{d_x} \exp\left(-\frac{C_2 R_7^2}{2(\mu_t^2 + C_2\sigma_t^2)}\right).$$

*Proof.* The first part of the proof is identical to Lemma J.10

Recall Lemma J.3. The density function at time $t$ is upper bounded by

$$p_t \leq \frac{C_1}{(\mu_t^2 + C_2\sigma_t^2)^{d_x/2}} \cdot \exp\left(-\frac{C_2\|x\|_2^2}{2(\mu_t^2 + C_2\sigma_t^2)}\right) \qquad \text{(By dropping constant term)}$$

$$\lesssim \exp\left(-\frac{C_2\|x\|_2^2}{2(\mu_t^2 + C_2\sigma_t^2)}\right).$$

Furthermore, by Lemma K.4 we have

$$\|u_t(x)\|_\infty \qquad\qquad\qquad\qquad\qquad\qquad\qquad\qquad (\text{K.6})$$

$$\leq \left|\frac{\dot{\mu}_t}{\mu_t} + (\dot{\sigma}_t\sigma_t - \frac{\dot{\mu}_t\sigma_t^2}{\mu_t})(\frac{C_2}{\mu_t^2 + C_2\sigma_t^2})\right| \cdot \|x\|_\infty + C_6\left|\dot{\sigma}_t - \frac{\dot{\mu}_t\sigma_t}{\mu_t}\right|$$

$$\lesssim \|x\|_\infty \qquad\qquad\qquad\qquad\qquad\qquad (\text{By Assumption I.2})$$

$$\leq \|x\|_2. \qquad\qquad\qquad\qquad\qquad\qquad (\text{By } \|\cdot\|_\infty \leq \|\cdot\|_2)$$

Therefore, the second inequality follows

$$\int_{\|x\|_\infty \geq R_7} \|u_t(x)\|_2^2 p_t(x)\mathrm{d}x$$

$$\leq d_x \int_{\|x\|_\infty \geq R_7} \|u_t(x)\|_\infty^2 p_t(x)\mathrm{d}x \qquad\qquad \left(\text{By } \|\cdot\|_2 \leq d_x\|\cdot\|_\infty\right)$$

$$\lesssim \int_{\|x\|_\infty \geq R_7} \|x\|_2^2 \cdot p_t(x)\mathrm{d}x \qquad\qquad\qquad\qquad (\text{By (K.6)})$$

$$\lesssim \int_{\|x\|_\infty \geq R_7} \|x\|_2^2 \cdot \exp\left(-\frac{C_2\|x\|_2^2}{2(\mu_t^2 + C_2\sigma_t^2)}\right)\mathrm{d}x \qquad (\text{By Lemma J.3})$$

$$\lesssim \int_{\|x\|_2 \geq R_7} \|x\|_2^2 \cdot \exp\left(-\frac{C_2\|x\|_2^2}{2(\mu_t^2 + C_2\sigma_t^2)}\right)\mathrm{d}x \qquad (\text{By } \|x\|_2 \geq \|x\|_\infty)$$

$$\lesssim R_7^{d_x} \exp\left(-\frac{C_2 R_7^2}{2(\mu_t^2 + C_2\sigma_t^2)}\right). \qquad\qquad\qquad (\text{By Lemma D.2})$$

This completes the proof. □

Next, we present the main proof of Theorem I.2

**Theorem K.1** (Theorem I.2 Restated: Velocity Approximation with Transformers under Stronger Hölder Smoothness)**.** Assume Assumption I.3 and Assumption I.2. For any precision parameter $0 < \epsilon < 1$ and smoothness parameter $\beta > 0$, let $\epsilon \leq O(N^{-\beta})$ for some $N \in \mathbb{N}$. Then, for all $t \in [t_0, T]$ with $t_0, T \in (0, 1)$, there exists a transformer $u_\theta(x, t) \in \mathcal{T}_R^{h,s,r}$ such that

$$\int_{t_0}^{T} \int_{\mathbb{R}^{d_x}} \|u_t(x) - u_\theta(x,t)\|_2^2 \cdot p_t(x)\mathrm{d}x\mathrm{d}t = O\left(B^2 N^{-2\beta}(\log N)^{d_x+\beta}\right),$$

Further, the parameter bounds in the transformer network class follows Theorem I.1.

*Proof of Theorem I.2.* Recall Lemma K.8, Lemma K.9. We have $C_{\mathcal{T}} = O(\sqrt{\log N})$ and we set

$$R_3 := \sqrt{\frac{4\beta(\mu_t^2 + C_2\sigma_t^2)\log N}{C_2}}. \tag{K.7}$$

Then, it holds

$$\int_{t_0}^{T} \int_{\mathbb{R}^{d_x}} \|u_\theta(x) - u_t(x)\|_2^2 p_t(x)\mathrm{d}x\mathrm{d}t$$

$$= \int_{t_0}^{T} \int_{\|x\|_\infty > R_3} \|u_\theta(x) - u_t(x)\|_2^2 p_t(x)\mathrm{d}x\mathrm{d}t + \int_{t_0}^{T} \int_{\|x\|_\infty \leq R_3} \|u_\theta(x) - u_t(x)\|_2^2 p_t(x)\mathrm{d}x\mathrm{d}t$$

$$\leq 2\int_{t_0}^{T} \int_{\|x\|_\infty > R_3} \left(\|u_\theta(x)\|_2^2 + \|u_t(x)\|_2^2\right) p_t(x)\mathrm{d}x\mathrm{d}t + \int_{t_0}^{T} \int_{\|x\|_\infty \leq R_3} \|u_\theta(x) - u_t(x)\|_2^2 p_t(x)\mathrm{d}x\mathrm{d}t$$
$$\left(\text{By expanding } \|\cdot\|_2^2\right)$$

$$\lesssim \int_{t_0}^{T} \int_{\|x\|_\infty > R_3} (\log N + \|u_t(x)\|_2^2) \cdot p_t(x)\mathrm{d}x\mathrm{d}t + \int_{t_0}^{T} \int_{\|x\|_\infty \leq R_3} \|u_\theta(x) - u_t(x)\|_2^2 p_t(x)\mathrm{d}x\mathrm{d}t$$
$$\left(\text{By } C_{\mathcal{T}} = O(\sqrt{\log N})\right)$$

$$\lesssim (\log N \cdot R_3^{d_x-2} + R_3^{d_x}) \exp\left(-\frac{C_2 R_3^2}{2(\mu_t^2 + C_2\sigma_t^2)}\right) + \int_{t_0}^{T} \int_{\|x\|_\infty \leq R_3} \|u_\theta(x) - u_t(x)\|_2^2 p_t(x)\mathrm{d}x\mathrm{d}t$$
$$\left(\text{By Lemma K.9}\right)$$

$$\lesssim (\log N)^{\frac{d_x}{2}} N^{-2\beta} + \int_{t_0}^{T} \int_{\|x\|_\infty \leq R_3} \|u_\theta(x) - u_t(x)\|_2^2 p_t(x)\mathrm{d}x\mathrm{d}t \qquad\qquad \left(\text{By (K.7)}\right)$$

$$\leq (\log N)^{\frac{d_x}{2}} N^{-2\beta} + B^2 N^{-2\beta}(\log N)^{k_1+d_x} \qquad\qquad\qquad\qquad \left(\text{By Lemma K.8}\right)$$

$$= O\left(B^2 N^{-2\beta}(\log N)^{k_1+d_x}\right). \qquad\qquad\qquad\qquad\qquad\qquad\qquad \left(\text{By } k_1 \leq \beta\right)$$

Furthermore, the parameter bounds in transformer network follow Lemma K.8.

This completes the proof. $\qquad\qquad\qquad\qquad\qquad\qquad\qquad\qquad\qquad\qquad\qquad\qquad\qquad\qquad$ □

# L   Proof of Theorem I.3

In this section, we prove Theorem I.3 following the three steps presented in the proof sketch.

**Organizations.**   Appendix L.1 provides fundamental definitions of flow matching and discusses key properties of the flow matching loss. Appendix L.2 introduces several auxiliary lemmas that support our proof. Finally, Appendix L.3 presents the main proof of Theorem I.3.

## L.1   Preliminaries

In this section, we consider affine conditional flows $\psi_t(x|x_1) = \mu_t x_1 + \sigma_t x$ follows Section 2. Given a velocity approximator $u_\theta$, we aim to bound the following flow matching risk $\mathcal{R}(u_\theta)$:

$$\mathcal{R}(u_\theta) := \int_{t_0}^{T} \frac{1}{T - t_0} \mathop{\mathbb{E}}_{X_t \sim p_t} [\|u_\theta(X_t, t) - u_t(X_t)\|_2^2] \, \mathrm{d}t, \tag{L.1}$$

where marginal probability path $p_t$ and marginal velocity field $u_t$ are induced by affine conditional flow $\psi_t(x|X_1) = \mu_t X_1 + \sigma_t x$ follows (2.2), (2.3), (2.5) and (2.6).

In practice, we use conditional flow matching loss to train velocity estimator $u_\theta$:

**Definition L.1** (Conditional Flow Matching).   Let $q$ be the ground truth distribution and the normal distribution $N(0, I)$ be the source distribution $p$. Considering affine conditional flows $\psi_t(x|x_1) = \mu_t X_1 + \sigma_t x$, we define the loss function and the conditional flow matching loss:

$$\ell(x; u_\theta) := \frac{1}{T - t_0} \int_{t_0}^{T} \mathop{\mathbb{E}}_{X_0 \sim N(0,I)} [\|u_\theta(\mu_t x + \sigma_t X_0, t) - (\dot\mu_t x + \dot\sigma_t X_0)\|_2^2] \mathrm{d}t,$$

$$\mathcal{L}_{\mathrm{CFM}}(u_\theta) := \frac{1}{T - t_0} \int_{t_0}^{T} \mathop{\mathbb{E}}_{X_1 \sim q, X_0 \sim N(0,I)} [\|u_\theta(\mu_t x + \sigma_t X_0, t) - (\dot\mu_t X_1 + \dot\sigma_t X_0)\|_2^2] \mathrm{d}t.$$

**Remark L.1.**   Holderrieth et al. [2025] prove that the gradients of the flow matching loss (risk) and the conditional flow matching loss coincide. Therefore, minimizing the flow matching loss (risk) $\mathcal{R}(u_\theta)$ is equivalent to minimizing the conditional flow matching loss $\mathcal{L}_{\mathrm{CFM}}(u_\theta)$.

To better evaluate the estimator $u_\theta$, now we introduce the empirical flow matching risk $\widehat{\mathcal{R}}(u_\theta)$.

**Definition L.2** (Empirical Risk).   Consider a velocity estimator $u_\theta \in \mathcal{T}_R^{h,s,r}$ and i.i.d training samples $\{x_i\}_{i=1}^n$, the empirical conditional flow matching loss $\widehat{\mathcal{L}}_{\mathrm{CFM}}(u_\theta) := \frac{1}{n} \sum_{i=1}^{n} \ell(x_i; u_\theta)$. Let $u^\star := u_t$ be the ground truth velocity field, we define empirical flow matching risk:

$$\widehat{\mathcal{R}}(u_\theta) := \widehat{\mathcal{L}}_{\mathrm{CFM}}(u_\theta) - \widehat{\mathcal{L}}_{\mathrm{CFM}}(u^\star) = \frac{1}{n} \sum_{i=1}^{n} \ell(x_i; u_\theta) - \frac{1}{n} \sum_{i=1}^{n} \ell(x_i; u^\star).$$

**Remark L.2.**   Notice that $R(u^\star) = 0$ since $u^\star$ is the ground truth velocity field. Furthermore, the fact that the gradients of the flow matching loss (risk) and the conditional flow matching loss coincide implies that $R(u_\theta) = R(u_\theta) - R(u^\star) = \mathcal{L}_{\mathrm{CFM}}(u_\theta) - \mathcal{L}_{\mathrm{CFM}}(u^\star)$.

**Remark L.3** (Unbiased Property).   We use $\widehat{\mathcal{L}}'_{\mathrm{CFM}}$ and $\widehat{\mathcal{R}}'$ to denote the conditional flow matching loss and empirical risk with training samples $\{x'_i\}_{i=1}^n$. Then for any velocity estimator $u_\theta$, the i.i.d. assumption implies that $\mathbb{E}_{\{x'_i\}_{i=1}^n}[\widehat{\mathcal{L}}'_{\mathrm{CFM}}(u_\theta)] = \mathcal{L}_{\mathrm{CFM}}(u_\theta)$, leading to $\mathbb{E}_{\{x'_i\}_{i=1}^n}[\widehat{\mathcal{R}}'(u_\theta)] = \mathcal{R}(u_\theta)$.

Next, we introduce the truncated version of (i) loss function $\ell(x; u_\theta)$, (ii) conditional flow matching loss $\mathcal{L}_{\mathrm{CFM}}(u_\theta)$, (iii) the conditional flow matching risk, $\mathcal{R}(u_\theta)$ (iv) the empirical risk $\widehat{\mathcal{R}}(u_\theta)$.

**Definition L.3** (Domain Truncation of Loss and Risk). Given $\ell(x; u_\theta)$, $\mathcal{L}_{\mathrm{CFM}}(u_\theta)$, $\mathcal{R}(u_\theta)$ and $\widehat{\mathcal{R}}(u_\theta)$, we define their truncated counterparts on a bounded domain $\mathcal{D} := [-D, D]^{d_x}$ by

$$\ell^{\mathrm{trunc}}(x; u_\theta) := \ell(x; u_\theta)\mathbb{1}\{\|x\|_\infty \leq D\}, \quad \mathcal{L}_{\mathrm{CFM}}^{\mathrm{trunc}}(u_\theta) := \mathcal{L}(u_\theta)\mathbb{1}\{\|x\|_\infty \leq D\},$$

$$\mathcal{R}^{\mathrm{trunc}}(u_\theta) := \mathcal{R}(x; u_\theta)\mathbb{1}\{\|x\|_\infty \leq D\}, \quad \widehat{\mathcal{R}}^{\mathrm{trunc}}(u_\theta) := \widehat{\mathcal{R}}(u_\theta)\mathbb{1}\{\|x\|_\infty \leq D\},$$

where $D > 0$ is a constant.

With Definition L.3, we refer to $\ell^{\mathrm{trunc}}(x; u_\theta)$, $\mathcal{L}_{\mathrm{CFM}}^{\mathrm{trunc}}(u_\theta)$, $\mathcal{R}^{\mathrm{trunc}}(u_\theta)$ and $\widehat{\mathcal{R}}^{\mathrm{trunc}}(u_\theta)$ as truncated loss, truncated CFM loss, truncated risk and truncated empirical risk respectively.

## L.2 Auxiliary lemmas

Since the target distribution $q(x_1)$ is unknown, direct computation of the risk is infeasible. Therefore, we first decompose the estimation error into four components and present supporting lemmas to bound each of them. Then, we incorporate these results in the main proof in Appendix L.3.

**Estimation Error Decomposition.** Let $\widehat{u}_\theta$ be the optimizer of the empirical conditional flow matching loss $\widehat{\mathcal{L}}_{\mathrm{CFM}}(u_\theta)$ using i.i.d samples $\{x_i\}_{i=1}^n$. Next, we introduce a different set of i.i.d samples $\{x_i'\}_{i=1}^n$ independent of the training sample $\{x_i\}_{i=1}^n$. Then, we decompose $\mathbb{E}_{\{x_i\}_{i=1}^n}[\mathcal{R}(\widehat{u}_\theta)]$:

$$\mathbb{E}_{\{x_i\}_{i=1}^n}[\mathcal{R}(\widehat{u}_\theta)] = \underbrace{\mathbb{E}_{\{x_i\}_{i=1}^n}\left[\mathbb{E}_{\{x_i'\}_{i=1}^n}\left[\widehat{\mathcal{R}}'(\widehat{u}_\theta) - \widehat{\mathcal{R}}'^{\,\mathrm{trunc}}(\widehat{u}_\theta)\right]\right]}_{(\mathrm{I})}$$

$$+ \underbrace{\mathbb{E}_{\{x_i\}_{i=1}^n}\left[\mathbb{E}_{\{x_i'\}_{i=1}^n}\left[\widehat{\mathcal{R}}'^{\,\mathrm{trunc}}(\widehat{u}_\theta) - \widehat{\mathcal{R}}^{\mathrm{trunc}}(\widehat{u}_\theta)\right]\right]}_{(\mathrm{II})}$$

$$+ \underbrace{\mathbb{E}_{\{x_i\}_{i=1}^n}\left[\widehat{\mathcal{R}}^{\mathrm{trunc}}(\widehat{u}_\theta) - \widehat{\mathcal{R}}(\widehat{u}_\theta)\right]}_{(\mathrm{III})} + \underbrace{\mathbb{E}_{\{x_i\}_{i=1}^n}\left[\widehat{\mathcal{R}}(\widehat{u}_\theta)\right]}_{(\mathrm{IV})}. \tag{L.2}$$

We refer to terms (I) and (III) as *truncation error*, and we control these errors by leveraging the sub-Gaussian assumption in Lemma L.1. Then, we derive the generalization bound to control term (II) using covering number in Lemma L.5 and Lemma L.6. Finally, we apply the approximation error using transformers to bound term (IV) in Lemma L.8.

**Truncation Error.** We apply the sub-Gaussian assumption to bound the truncation error.

**Lemma L.1** (Upper Bound on the Truncation Error). Assume Assumption I.1. Let $t_0, T \in (0, 1)$ and $u_\theta(x, t)$ be the velocity approximators in Theorem I.1 and Theorem I.2. Then, it holds

$$\mathbb{E}_x\big[|\ell(x; u_\theta) - \ell^{\mathrm{trunc}}(x; u_\theta)|\big] \lesssim D^{d_x} \exp\left(-\frac{1}{2}C_2 D^2\right) \log N \quad \text{for any} \quad t \in [t_0, T].$$

*Proof.* Our proof builds on Section D.2 of [Fu et al., 2024]. By Theorem I.1 and Theorem I.2, the transformers output bound $C_{\mathcal{T}} = O(\sqrt{\log N})$. Then, for all approximator $u_\theta \in \mathcal{T}_R^{h,s,r}$, it holds

$$\mathbb{E}_x\big[|\ell(x; u_\theta) - \ell^{\mathrm{trunc}}(x; u_\theta)|\big]$$

$$= \mathbb{E}_x[|\ell(x; u_\theta)\mathbb{1}[\|x\| \geq D]|] \hspace{4cm} (\text{By Definition L.3})$$

$$= \frac{1}{T - t_0} \int_{t_0}^T \int_{\|x\| \geq D} \mathbb{E}_{x_0 \sim N(0, I)}[\|u_\theta - (\dot{\mu}_t x + \dot{\sigma}_t x_0)\|_2^2]q(x)\mathrm{d}x\mathrm{d}t \hspace{0.5cm} (\text{By Definition L.1})$$

$$\leq \frac{2}{T - t_0} \int_{t_0}^T \int_{\|x\| \geq D} \mathbb{E}_{x_0 \sim N(0, I)}[\|u_\theta\|_2^2 + \|\dot{\mu}_t x + \dot{\sigma}_t x_0\|_2^2]q(x)\mathrm{d}x\mathrm{d}t \hspace{0.3cm} (\text{By expanding the } \ell_2\text{-norm})$$

$$\lesssim \frac{2}{T-t_0} \int_{t_0}^{T} \int_{\|x\| \geq D} \mathop{\mathbb{E}}_{x_0 \sim N(0,I)} [\|u_\theta\|_2^2 + \|\dot{\mu}_t x + \dot{\sigma}_t x_0\|_2^2] \exp\left(-\frac{1}{2}C_2\|x\|_2^2\right) \mathrm{d}x \mathrm{d}t$$
(By Assumption I.1)

$$\lesssim \frac{2}{T-t_0} \int_{t_0}^{T} \int_{\|x\| \geq D} \mathop{\mathbb{E}}_{x_0 \sim N(0,I)} [\log N + \|\dot{\mu}_t x + \dot{\sigma}_t x_0\|_2^2] \exp\left(-\frac{1}{2}C_2\|x\|_2^2\right) \mathrm{d}x \mathrm{d}t$$
(By $C_\mathcal{T} = O(\sqrt{\log N})$)

$$\lesssim \frac{1}{T-t_0} \int_{t_0}^{T} \int_{\|x\| \geq D} (\log N + \dot{\sigma}_t^2 d + \dot{\mu}_t^2 \|x\|_2^2) \exp\left(-\frac{1}{2}C_2\|x\|_2^2\right) \mathrm{d}x \mathrm{d}t \qquad \text{(By } x_0 \sim N(0,I))$$

$$\lesssim \frac{D^{d_x-2} \exp\left(-\frac{1}{2}C_2 D^2\right)}{T-t_0} \int_{t_0}^{T} (\log N + \dot{\sigma}_t^2 d) \mathrm{d}t + \frac{D^{d_x} \exp\left(-\frac{1}{2}C_2 D^2\right)}{T-t_0} \int_{t_0}^{T} \dot{\mu}_t^2 \mathrm{d}t$$
(By Lemma D.2)

$$\lesssim D^{d_x} \exp\left(-\frac{1}{2}C_2 D^2\right) \log N. \qquad \text{(By Assumption I.2)}$$

This completes the proof. $\qquad\square$

**Covering Number of Loss Function Class with Transformer Estimator.** Recall (II) in (L.2):

$$(\text{II}) = \mathop{\mathbb{E}}_{\{x_i\}_{i=1}^n} \left[ \mathop{\mathbb{E}}_{\{x_i'\}_{i=1}^n} \left[ \widehat{\mathcal{R}}'^{\,\text{trunc}}(\widehat{u}_\theta) - \widehat{\mathcal{R}}^{\text{trunc}}(\widehat{u}_\theta) \right] \right].$$

To derive an upper bound on (II), we introduce (i) the covering number technique in Lemma L.5 and Lemma L.6 (ii) the generalization error bound to bound in Lemma L.8.

We begin with the definition of covering number.

**Definition L.4** (Covering Number). Let $\Omega$ be a compact domain and $\{x_i\}_{i=1}^n$ be data drawn from distribution $P$. Denote the joint distribution $\{x_i\}_{i=1}^n \sim P^n := P \otimes \cdots \otimes P$. Given a function class $\mathcal{F}$, a $t \in \Omega$, a norm $\|\cdot\|$ and a $\epsilon_c > 0$, the $\epsilon_c$-covering number $\mathcal{N}(\epsilon, \mathcal{F}, \{x_i\}_{i=1}^n \times \Omega, \|\cdot\|)$ is the smallest size of a collection $\{f_j\}_{i=1}^N \subset \mathcal{F}$ such that for any $f \in \mathcal{F}$, there exists a $j \in [N]$ satisfying

$$\sup_t \max_i \|f(x_i, t) - f_j(x_i, t)\| \leq \epsilon. \tag{L.3}$$

Also, we define the covering number with respect to the data distribution $P$ and size $n$ as

$$\mathcal{N}(\epsilon, \mathcal{F}, P^n \times \Omega, \|\cdot\|) := \sup_{\{x_i\}_{i=1}^n \sim P^n} \mathcal{N}(\epsilon, \mathcal{F}, \{x_i\}_{i=1}^n \times \Omega, \|\cdot\|).$$

Further, for $\Omega = \emptyset$, we denote the covering number by $\mathcal{N}(\epsilon, \mathcal{F}, \{x_i\}_{i=1}^n, \|\cdot\|)$ and $\mathcal{N}(\epsilon, \mathcal{F}, P^n, \|\cdot\|)$.

**Remark L.4** (Covering Number of Transformer Network Class). We define the covering number over domain $\{x_i\}_{i=1}^n \times \Omega$ to align with the flow matching loss formulation in Definition L.1, where temporal dependence in transformers introduces no additional statistical error. Specifically, the loss averages over the time component, unlike the $n$ i.i.d. data points sampled from target distribution $q^n$.

Next, we derive an upper bound on the covering number of transformer networks. Our proof builds on [Edelman et al., 2022] and studies the class in Definition B.4, with a self-attention layer that applies softmax under inverse-temperature scaling and a feed-forward layer with ReLU activation.

**Covering Number of Linear Function Class.** Our norm-based upper bound on the covering number of the transformer network class extends the classical norm-based bound for the linear function class:

**Lemma L.2** (Covering Number of Linear Function Class, Lemma 4.6 of [Edelman et al., 2022] and [Zhang, 2002]). Let $z_1, \ldots, z_n \in \mathbb{R}^d$ be sample points satisfying $\|z_i\| \leq B_X$ for all $i \in [n]$. Then,

for linear function class $\mathcal{F} := \{f : z \to Wz \mid W \in \mathbb{R}^{d' \times d}, \|W\|_{2,1} \leq B_W, B_W > 0\}$, it holds:

$$\log \mathcal{N}(\epsilon, \mathcal{F}, \{z_i\}_{i=1}^n; \|\cdot\|_\infty) \lesssim \frac{B_X^2 B_W^2}{\epsilon^2} \log(d'n) \quad \text{for any} \quad \epsilon > 0.$$

**Remark L.5** (Covering Number Equivalence). We remark that Lemma L.2 applies to our feed-forward layer for two reasons. First, the bias terms $b_1$ and $b_2$ admit an augmented form. By appending a bottom row of ones to $Z$ and set $\widetilde{W}_1 := (W_1, b_1)$, it holds that $W_1 Z + b_1 = \widetilde{W}_1 \widetilde{Z}$, where $W_1 \in \mathbb{R}^{r \times d}, b_1 \in \mathbb{R}^r$, and $\widetilde{W}_1 \in \mathbb{R}^{r \times (d+1)}, \widetilde{Z} \in \mathbb{R}^{(d+1) \times L}$. This two forms define the same function class. Second, the norm bound in Lemma H.5 keeps the biases on the same order as the matrices operator. Since our $\lesssim$ and $O(\cdot)$ notation hides polynomial and logarithmic factors in $d$ and $L$, Lemma L.2 gives the covering number for the linear class in our feed-forward layer with biases.

Following [Edelman et al., 2022], we extend Lemma L.2 to a transformer block. We view the block as a composition of linear function classes. We construct covers for each linear function class, balance the errors across components, and minimize the size of the concatenated cover using the next lemmas:

**Lemma L.3** (Lemma A.8 of [Edelman et al., 2022]). Consider the following optimization problem:

$$\min_{x_1, \ldots, x_n} \sum_{i=1}^n \frac{\alpha_i}{x_i^2} \quad \text{subject to} \quad \sum_{i=1}^n \omega_i x_i = C \tag{L.4}$$

for some $\alpha_i, \omega_i$ and a constant $C$. Then, (L.4) has solution

$$\frac{\left(\sum_{i=1}^n \alpha_i^{\frac{1}{3}} \omega_i^{\frac{2}{3}}\right)^3}{C^2} \quad \text{when} \quad x_i = \frac{C}{\sum_{i=1}^n \alpha_i^{\frac{1}{3}} \omega^{\frac{2}{3}}} \cdot \left(\frac{\alpha_i}{\omega_i}\right)^{\frac{2}{3}}.$$

Equipped with Lemma L.3, we have the upper bound for the covering number of transformer block:

**Lemma L.4** (Covering Number of Transformer Block, Modified from Corollary 4.5 of [Edelman et al., 2022]). Let $\{x_i\}_{i=1}^n$ be sample points satisfying $\max_i \|x_i\|_\infty \leq D$ for some constant $D > 0$ and $R(\cdot) : \mathbb{R}^{d_x} \to \mathbb{R}^{d \times L}$ be the reshape layer (Definition B.3). Let $\{Z_i\}_{i=1}^n := \{R(x_i)\}_{i=1}^n, E \in \mathbb{R}^{d \times L}$ be the positional encoding and $\mathcal{T}_R^{1,s,r}(Z)$ denote a two-layer transformer class (Definition B.1) with single-head self-attention and $s$-hidden dimension and $r$-MLP dimension. Then, it holds:

$$\log \mathcal{N}\left(\epsilon, \mathcal{T}_R^{1,s,r}, \{Z_i\}_{i=1}^n; \|\cdot\|_\infty\right)$$
$$\lesssim \frac{\log(nL)}{\epsilon_c^2} \alpha^2 \left((C_F^{2,\infty})^{\frac{4}{3}} + \left(\lambda (C_F)^2 C_{OV} C_{KQ}^{2,\infty}\right)^{\frac{2}{3}} + \left((C_F)^2 C_{OV}^{2,\infty}\right)^{\frac{2}{3}}\right)^3,$$

where $\alpha := O\left(C_F^2 C_{OV} C_{KQ}(D + C_E)\right)$.

*Proof.* Our proof builds on [Edelman et al., 2022] by incorporating the scaling $\lambda > 0$ for the column-wise softmax function. For $Z \in [-D, D]^{d \times L}$ and $W_K, W_Q \in \mathbb{R}^{s \times d}$, we define

$$\mathcal{F}_{KQ} := \left\{f : Z \to (W_K Z)^\top (W_Q Z) \mid W_{KQ} = W_K^\top W_Q, \|W_{KQ}\|_2 \leq C_{KQ}\right\},$$

for some constant $D > 0$. Similarly, for $W_V \in \mathbb{R}^{s \times d}$ and $W_O \in \mathbb{R}^{d \times s}$, we define

$$\mathcal{F}_{OV} := \left\{f : Z \to W_O \cdot W_V Z \mid W_{OV} = W_O W_V, \|W_{OV}\|_2 \leq C_{OV}\right\}.$$

Recall Remark L.5. For feed-forward layer, we define:

$$\mathcal{F}^{(\text{FF})} := \left\{f : Z \to W_2 \text{ReLU}[W_1 Z] \mid \|W_1\|_2, \|W_2\|_2 \leq C_F\right\}.$$

For simplicity, we denote the $k$-th column of $Z_i$ by $z_i^{(k)}$ for all $i \in [n]$. First observe that

$$\max_{i \in [n]} \|f(Z_i) - \widehat{f}(Z_i)\|_{2,\infty} = \max_{i \in [n], k \in [L]} \|f(z_i^{(k)}) - \widehat{f}(z_i^{(k)})\|_2 \tag{L.5}$$

for any distinct $f, \widehat{f}$ in $\mathcal{F}_{KQ}, \mathcal{F}_{OV}$ or $\mathcal{F}^{(\mathrm{FF})}$. With this, we consider $\{Z_i\}_{i=1}^n$ as $nL$ samples in $[-D, D]^d$ and apply Lemma L.2 to construct covers $\mathcal{C}_{KQ}, \mathcal{C}_{OV}$ and $\mathcal{C}_{FF}$ for $\mathcal{F}_{KQ}, \mathcal{F}_{OV}$ and $\mathcal{F}^{(\mathrm{FF})}$.

**Covering Number for $\mathcal{F}_{KQ}$ and $\mathcal{F}_{OV}$.** Since $Z_i \in [-D, D]^{d \times L}$, for any $i, j \in [n]$ and $k \in [L]$, we have that $\|f(z_i^{(k)}) - f(z_j^{(k)})\|_2^2 \le dD^2 \cdot \|(W_K^\top W_Q)Z\|_2^2$. Thus, by Lemma L.2 and (L.5), it holds

$$\log \mathcal{N}\big(\epsilon_{KQ}, \mathcal{F}_{KQ}, \{Z_i\}_{i=1}^n; \|\cdot\|_\infty\big) \lesssim \frac{D^3 C_{KQ}^2}{\epsilon_{KQ}^2} \log(nL), \tag{L.6}$$

where $\lesssim$ hides polynomial factors dependent on $d$ and $L$. Similarly, for $\mathcal{F}_{OV}$, it holds

$$\log \mathcal{N}\big(\epsilon_{OV}, \mathcal{F}_{OV}, \{Z_i\}_{i=1}^n; \|\cdot\|_\infty\big) \lesssim \frac{D^2 C_{OV}^2}{\epsilon_{OV}^2} \log(nL), \tag{L.7}$$

That is, we have cover $\mathcal{C}_{KQ}, \mathcal{C}_{OV}$ for $\mathcal{F}_{KQ}, \mathcal{F}_{OV}$ whose sizes are upper-bounded by (L.6) and (L.7). By triangle inequality and by Lemma B.2, for any $W_K, W_Q \in \mathcal{F}_{KQ}$ and any $W_O, W_V \in \mathcal{F}_{OV}$, there exist some $\widehat{W}_K, \widehat{W}_Q \in \mathcal{C}_{KQ}$ and $\widehat{W}_O, \widehat{W}_V \in \mathcal{C}_{OV}$ such that for all $i \in [n]$ and all $k \in [L]$, it holds

$$\|\widehat{W}_O(\widehat{W}_V z_i^{(k)})\, \mathrm{Softmax}[(\widehat{W}_K z_i^{(k)})^\top (\widehat{W}_Q z_i^{(k)})] - W_O(W_V z_i^{(k)})\, \mathrm{Softmax}[(W_K z_i^{(k)})^\top (W_Q z_i^{(k)})]\|$$

$$\tag{L.8}$$

$$\le \underbrace{\|\widehat{W}_O(\widehat{W}_V z_i^{(k)})\, \mathrm{Softmax}[(\widehat{W}_K z_i^{(k)})^\top (\widehat{W}_Q z_i^{(k)})] - W_O(W_V z_i^{(k)})\, \mathrm{Softmax}[(\widehat{W}_K z_i^{(k)})^\top (\widehat{W}_Q z_i^{(k)})]\|}_{(\mathrm{I})}$$

$$+ \underbrace{\|W_O(W_V z_i^{(k)})\, \mathrm{Softmax}[(\widehat{W}_K z_i^{(k)})^\top (\widehat{W}_Q z_i^{(k)})] - W_O(W_V z_i^{(k)})\, \mathrm{Softmax}[(W_K z_i^{(k)})^\top (W_Q z_i^{(k)})]\|}_{(\mathrm{II})}$$

$$\left(\text{By triangle inequality}\right)$$

$$\lesssim \underbrace{\|(\widehat{W}_O \widehat{W}_V - W_O W_V) z_i^{(k)}\|}_{(\mathrm{I})} + \underbrace{\|W_O W_V z_i^{(k)}\|\|\mathrm{Softmax}[(\widehat{W}_K z_i^{(k)})^\top (\widehat{W}_Q z_i^{(k)})] - \mathrm{Softmax}[(W_K z_i^{(k)})^\top (W_Q z_i^{(k)})]\|}_{(\mathrm{II})}$$

$$\left(\text{By } \|\mathrm{Softmax}[\cdot]\|_F \le dL\right)$$

$$\lesssim \underbrace{\|(\widehat{W}_O \widehat{W}_V - W_O W_V) z_i^{(k)}\|}_{(\mathrm{I})} + \underbrace{C_{OV} D\|\mathrm{Softmax}[(\widehat{W}_K z_i^{(k)})^\top (\widehat{W}_Q z_i^{(k)})] - \mathrm{Softmax}[(W_K z_i^{(k)})^\top (W_Q z_i^{(k)})]\|}_{(\mathrm{II})}$$

$$\left(\text{By } \|W_{OV}\| \le C_{OV} \text{ and } \|z_i^{(k)}\|_\infty \le D\right)$$

$$\lesssim \epsilon_{OV} + \underbrace{C_{OV} D\|\mathrm{Softmax}[(\widehat{W}_K z_i^{(k)})^\top (\widehat{W}_Q z_i^{(k)})] - \mathrm{Softmax}[(W_K z_i^{(k)})^\top (W_Q z_i^{(k)})]\|}_{(\mathrm{II})}$$

$$\left(\text{By (L.6)}\right)$$

$$\lesssim \epsilon_{OV} + \underbrace{2\lambda C_{OV} D\|(\widehat{W}_K z_i^{(k)})^\top (\widehat{W}_Q z_i^{(k)}) - (W_K z_i^{(k)})^\top (W_Q z_i^{(k)})\|}_{(\mathrm{II})} \qquad \left(\text{By Lemma B.2}\right)$$

$$\le \epsilon_{OV} + 2\lambda C_{OV} D \epsilon_{KQ}. \qquad \left(\text{By (L.7)}\right)$$

Then, we have a cover for the self-attention layer

$$\mathcal{C}_{\mathrm{SA}} := \big\{ f: Z \to \widehat{W}_O(\widehat{W}_V Z)\, \mathrm{Softmax}[(\widehat{W}_K Z)^\top (\widehat{W}_Q Z)] \mid \widehat{W}_O \widehat{W}_V \in \mathcal{C}_{OV}, \widehat{W}_K^\top \widehat{W}_Q \in \mathcal{C}_{KQ} \big\}.$$

**Covering Number for $\mathcal{F}^{(\text{FF})}$.** Similarly, we apply Lemma L.2 twice for the inner and outer linear function function class operated by $W_1$ and $W_2$. Specifically, we construct cover $\mathcal{C}_1$ and $\mathcal{C}_2$ for

$$\mathcal{F}_1^{(\text{FF})} := \left\{ f : Z \to W_1 Z \mid \|W_1\|_2 \leq C_F \right\} \quad \text{and} \quad \mathcal{F}_2^{(\text{FF})} := \left\{ f : Z \to W_2 Z \mid \|W_2\|_2 \leq C_F \right\}$$

respectively. Then, by Lemma L.2, we have cover $\mathcal{C}_1$ and $\mathcal{C}_2$ for

$$\log \mathcal{N}\left(\epsilon_{F,1}, \mathcal{F}_1^{(\text{FF})}, \{Z_i\}_{i=1}^n; \|\cdot\|_\infty\right) \lesssim \frac{D^2 C_F^2}{\epsilon_{F,1}^2} \log(nL). \tag{L.9}$$

and

$$\log \mathcal{N}\left(\epsilon_{F,2}, \mathcal{F}_2^{(\text{FF})}, \{Z_i\}_{i=1}^n; \|\cdot\|_\infty\right) \lesssim \frac{D^2 C_F^2}{\epsilon_{F,2}^2} \log(nL). \tag{L.10}$$

That is, we have cover $\mathcal{C}_1, \mathcal{C}_2$ for $\mathcal{F}_1^{(\text{FF})}, \mathcal{F}_2^{(\text{FF})}$ whose sizes are upper-bounded by (L.9) and (L.10). Let $\widehat{\mathcal{F}}^{(\text{SA})}(Z)$ denote the self-attention layer with weights chosen from cover $\mathcal{C}_{\text{SA}}$. For any $W_1$, $W_K, W_Q, W_O, W_V$, there exist some $\widehat{W}_1 \in \mathcal{C}_1$ and $\widehat{\mathcal{F}}^{(\text{SA})}$ such that for all $i \in [n], k \in [L]$, it holds

$$\|\widehat{W}_1 \circ \widehat{\mathcal{F}}^{(\text{SA})}(z_i^{(k)}) - W_1 \circ \mathcal{F}^{(\text{SA})}(z_i^{(k)})\| \tag{L.11}$$
$$\leq \|\widehat{W}_1 \circ \widehat{\mathcal{F}}^{(\text{SA})}(z_i^{(k)}) - \widehat{W}_1 \circ \mathcal{F}^{(\text{SA})}(z_i^{(k)})\| + \|\widehat{W}_1 \circ \mathcal{F}^{(\text{SA})}(z_i^{(k)}) - W_1 \circ \mathcal{F}^{(\text{SA})}(z_i^{(k)})\|$$
$$\hspace{9cm} \left(\text{By traingle inequality}\right)$$
$$\lesssim C_F \|\widehat{\mathcal{F}}^{(\text{SA})}(z_i^{(k)}) - \mathcal{F}^{(\text{SA})}(z_i^{(k)})\| + \|\widehat{W}_1 \circ \mathcal{F}^{(\text{SA})}(z_i^{(k)}) - W_1 \circ \mathcal{F}^{(\text{SA})}(z_i^{(k)})\|$$
$$\hspace{10cm} \left(\text{By norm bound on } W_1\right)$$
$$\lesssim C_F \|\widehat{\mathcal{F}}^{(\text{SA})}(z_i^{(k)}) - \mathcal{F}^{(\text{SA})}(z_i^{(k)})\| + C_{OV} \|(\widehat{W}_1 - W_1)(z_i^{(k)})\|$$
$$\hspace{10cm} \left(\text{By the norm bound on } W_O \cdot W_V\right)$$
$$\leq C_F(\epsilon_{OV} + 2\lambda C_{OV} D \epsilon_{KQ}) + C_{OV}\epsilon_{F,1}, \hspace{3cm} \left(\text{By (L.8) and (L.9)}\right)$$

Building on (L.11), we have the cover for $\mathcal{F}^{(FF)} \circ \mathcal{F}^{(\text{SA})}$:

$$\|\widehat{W}_2 \text{ReLU}[(\widehat{W}_1 \widehat{\mathcal{F}}^{(\text{SA})}(z_i^{(k)})] - W_2 \text{ReLU}[W_1 \mathcal{F}^{(\text{SA})}(z_i^{(k)})]\| \tag{L.12}$$
$$\lesssim \underbrace{\|\widehat{W}_2 \widehat{W}_1 \widehat{\mathcal{F}}^{(\text{SA})}(z_i^{(k)}) - W_2 \widehat{W}_1 \widehat{\mathcal{F}}^{(\text{SA})}(z_i^{(k)})\|}_{(\text{I})} + \underbrace{\|W_2 \widehat{W}_1 \widehat{\mathcal{F}}^{(\text{SA})}(z_i^{(k)}) - W_2 W_1 \mathcal{F}^{(\text{SA})}(z_i^{(k)})\|}_{(\text{II})}$$
$$\hspace{8cm} \left(\text{By triangle inequality and Lipschitzness of ReLU}\right)$$
$$\lesssim \underbrace{C_F C_{OV} \|(\widehat{W}_2 - W_2) z_i^{(k)}\|}_{(\text{I})} + \underbrace{\|W_2 \widehat{W}_1 \widehat{\mathcal{F}}^{(\text{SA})}(z_i^{(k)}) - W_2 W_1 \mathcal{F}^{(\text{SA})}(z_i^{(k)})\|}_{(\text{II})}$$
$$\hspace{9cm} \left(\text{By norm bound on } W_1 \text{ and } W_{OV}\right)$$
$$\lesssim \underbrace{C_F C_{OV} \|(\widehat{W}_2 - W_2) z_i^{(k)}\|}_{(\text{I})} + C_F \underbrace{\|\widehat{W}_1 \widehat{\mathcal{F}}^{(\text{SA})}(z_i^{(k)}) - W_1 \mathcal{F}^{(\text{SA})}(z_i^{(k)})\|}_{(\text{II})} \quad \left(\text{By norm bund on } W_2\right)$$
$$\lesssim C_F C_{OV} \epsilon_{F,2} + C_F^2(\epsilon_{OV} + 2\lambda C_{OV} D \epsilon_{KQ}) + C_F C_{OV} \epsilon_{F,1}. \hspace{2cm} \left(\text{By (L.10) and (L.11)}\right)$$

The overall size of the $\epsilon$-cover of a transformer block $\mathcal{F}^{1,s,r} = \mathcal{F}^{(\text{FF})} \circ \mathcal{F}^{(\text{SA})}$ is

$$\mathcal{C}_{\text{trans}} := \{\mathcal{F}^{(\text{FF})} \circ \mathcal{F}^{(\text{SA})} \mid \widehat{W}_O \widehat{W}_V \in \mathcal{C}_{OV}, \widehat{W}_K^\top \widehat{W}_Q \in \mathcal{C}_{KQ}, \widehat{W}_1, \widehat{W}_2 \in \mathcal{C}_{\text{FF}}\}.$$

This gives

$$\log|\mathcal{C}_{\text{trans}}| = \log|\mathcal{C}_{KQ}| + \log|\mathcal{C}_{OV}| + \log|\mathcal{C}_{\text{FF}}|,$$

satisfying $C_F C_{OV}(\epsilon_{F,1} + \epsilon_{F,2}) + C_F^2(\epsilon_{OV} + 2\lambda C_{OV} D\epsilon_{KQ}) \leq \epsilon$. Let

$$\alpha_1 := D^3 C_{KQ}^2 \log nL; \quad \alpha_2 := D^2 C_{OV}^2 \log(nL); \quad \alpha_3 := D^2 C_F^2 \log(nL);$$
$$\omega_1 := 2\lambda C_F^2 C_{OV} D; \quad \omega_2 := C_F^2 C_{OV} D; \quad \omega_3 := \lambda C_F C_{OV}.$$

Finally, we apply Lemma L.3 and obtain the optimal size of the the cover $\mathcal{C}_{\text{trans}}$:

$$\frac{D^2 \log(nL)\big(C_{KQ}^{2/3} \lambda^{2/3} C_F^{4/3} C_{OV}^{2/3} D^{1/3} + C_{OV}^{4/3} C_F^{4/3} D^{2/3} + \lambda^{2/3} C_{OV}^{2/3} C_F^{4/3}\big)^3}{\epsilon^2}.$$

We extend the argument to two blocks by invoking the composition step in [Edelman et al., 2022]. For inputs with positional encoding $E$, we replace the bound $D$ with $D + C_E$, where $\|E\| \leq C_E$.

This completes the proof. $\qquad\square$

**Remark L.6** (Looseness of the Covering Number Bound). The bound in Lemma L.4 is loose yet sufficient for our two layer transformer class with inverse-temperature scaling and one attention head (recall Appendix H). The extension from one block to two follows from the composition step in Theorem A.17 of [Edelman et al., 2022], where it provides the complete induction argument.

**Lemma L.5** (Covering Number Bounds for Transformer Network Class, Modified from Theorem A.17 of [Edelman et al., 2022]). Let $\mathcal{T}_R^{h,s,r}(C_{\mathcal{T}}, C_{KQ}^{2,\infty}, C_{KQ}, C_{OV}^{2,\infty}, C_{OV}, C_E, C_F^{2,\infty}, C_F, L_{\mathcal{T}})$ be the class of functions of one transformer block satisfying the norm bound for matrix and the Lipschitz property for feed-forward layers. Then, for all data points satisfying $\|x_i\|_{2,\infty} \leq D$, it holds

$$\log \mathcal{N}(\epsilon_c, \mathcal{T}_R^{h,s,r}, P^n \times [0,1], \|\cdot\|_2) \tag{L.13}$$
$$\lesssim \frac{\log(nL/\epsilon_c)}{\epsilon_c^2} \alpha^2 \Big((C_F^{2,\infty})^{\frac{4}{3}} + \big(\lambda(C_F)^2 C_{OV} C_{KQ}^{2,\infty}\big)^{\frac{2}{3}} + \big((C_F)^2 C_{OV}^{2,\infty}\big)^{\frac{2}{3}}\Big)^3,$$

where $\alpha := O\big(C_F^2 C_{OV} C_{KQ}(D + C_E)\big)$.

*Proof.* Lemma L.4 shows Lemma L.5 with the absence of domain $[0,1]$. That is,

$$\log \mathcal{N}(\epsilon_c, \mathcal{T}_R^{h,s,r}, P^n, \|\cdot\|_2) \tag{L.14}$$
$$\lesssim \frac{\log(nL)}{\epsilon_c^2} \alpha^2 \Big((C_F^{2,\infty})^{\frac{4}{3}} + \big(\lambda(C_F)^2 C_{OV} C_{KQ}^{2,\infty}\big)^{\frac{2}{3}} + \big((C_F)^2 C_{OV}^{2,\infty}\big)^{\frac{2}{3}}\Big)^3,$$

holds with data points drawn from $P^n$. To extend it to $P^n \times [0,1]$, we discretize $[0,1]$ into a $\delta$-grid $\mathcal{G} := \big\{t_k = k \cdot \delta \mid k = 0, 1, \ldots, \lfloor 1/\delta \rfloor, \delta \in (0,1)\big\}$. For simplicity, we denote $|\mathcal{G}| := m = O(1/\delta)$.

Let $\mathcal{T}_{R,\mathcal{G}}^{h,s,r}$ be the transformer network class on domain $[-D, D]^{d_x} \times \mathcal{G}$. We first suppose that

$$\mathcal{N}\big(\epsilon_c, \mathcal{T}_R^{h,s,r}, P^n \times [0,1], \|\cdot\|_2\big) \leq \mathcal{N}\big(\epsilon_c/2, \mathcal{T}_{R,\mathcal{G}}^{h,s,r}, P^n \times \mathcal{G}, \|\cdot\|_2\big) \tag{L.15}$$

holds when $\delta := \epsilon_c/(4L_{\mathcal{T}})$, where $L_{\mathcal{T}}$ is the Lipschitz constant of the transformer block.

Then, since domain $P^n \times \mathcal{G}$ is a set of sample points with size at most $nm$, (L.14) gives

$$\mathcal{N}\big(\epsilon_c/2, \mathcal{T}_{R,\mathcal{G}}^{h,s,r}, P^n \times \mathcal{G}, \|\cdot\|_2\big) \tag{L.16}$$
$$\lesssim \frac{\log(nmL)}{(\epsilon_c/2)^2} \alpha^2 \Big((C_F^{2,\infty})^{\frac{4}{3}} + \big(\lambda(C_F)^2 C_{OV} C_{KQ}^{2,\infty}\big)^{\frac{2}{3}} + \big((C_F)^2 C_{OV}^{2,\infty}\big)^{\frac{2}{3}}\Big)^3$$

Therefore, (L.15) and (L.16) implies (L.13):

$$\mathcal{N}\big(\epsilon_c, \mathcal{T}_R^{h,s,r}, P^n \times [0,1], \|\cdot\|_2\big)$$

$$\leq \mathcal{N}\big(\epsilon_c/2, \mathcal{T}_{R,\mathcal{G}}^{h,s,r}, P^n \times \mathcal{G}, \|\cdot\|_2\big) \tag{By (L.15)}$$

$$\leq \frac{\log(nmL)}{(\epsilon_c/2)^2}\alpha^2\Big((C_F^{2,\infty})^{\frac{4}{3}} + \big(\lambda(C_F)^2 C_{OV} C_{KQ}^{2,\infty}\big)^{\frac{2}{3}} + \big((C_F)^2 C_{OV}^{2,\infty}\big)^{\frac{2}{3}}\Big)^3 \tag{By (L.16)}$$

$$\lesssim \frac{\log(nL/\epsilon_c)}{\epsilon_c^2}\alpha^2\Big((C_F^{2,\infty})^{\frac{4}{3}} + \big(\lambda(C_F)^2 C_{OV} C_{KQ}^{2,\infty}\big)^{\frac{2}{3}} + \big((C_F)^2 C_{OV}^{2,\infty}\big)^{\frac{2}{3}}\Big)^3.$$
$$\big(\text{By } \delta = \epsilon_c/L_{\mathcal{T}} \text{ and dropping lower order terms}\big)$$

Lastly, it suffices to prove that (L.15) holds when $\delta := \epsilon_c/(4L_{\mathcal{T}})$. We show this by utilizing the Lipchitz property of the transformer networks (Definition B.4.) Specifically, let $\{f_j\}_{j=1}^{\mathcal{N}}$ be a $\epsilon_c/2$ cover of $\mathcal{T}_{R,\mathcal{G}}^{h,s,r}$. Then, for any $f \in \mathcal{T}_R^{h,s,r}$ and $t \in [0,1]$, it holds:

$$\|f(x_i,t) - f_j(x_i,t)\|_2$$
$$\leq \underbrace{\|f(x_i,t) - f(x_i,t_k)\|_2}_{(A)} + \underbrace{\|f(x_i,t_k) - f_j(x_i,t_k)\|_2}_{(B)} + \underbrace{\|f_j(x_i,t_k) - f_j(x_i,t)\|_2}_{(C)}.$$
$$\big(\text{By triangle inequality}\big)$$

We then show that the RHS is bounded by $\epsilon_c$ and this implies (L.15). For (A) and (C), it holds:

$$(A), (C) \leq L_{\mathcal{T}} \cdot |t - t_k| \leq L_{\mathcal{T}} \cdot \delta \leq \frac{\epsilon_c}{4},$$

where the first inequality is by the Lipchitzness of the transformer network, the second inequality is by the definition of the $\delta$-grid $\mathcal{G}$, and the last inequality is by taking $\delta := \epsilon_c/(4L_{\mathcal{T}})$.

Further, (B) is bounded by $\epsilon_c/2$ by the definition of $\{f_j\}_{j=1}^{\mathcal{N}}$. Altogether, we have that

$$\|f(x_i,t) - f_j(x_i,t)\|_2 \leq \frac{\epsilon_c}{4} + \frac{\epsilon_c}{2} + \frac{\epsilon_c}{4} = \epsilon_c. \tag{L.17}$$

Since (L.17) holds for all $t$, (L.3) in Definition L.4 holds after taking the supremum over $t$.

This completes the proof. □

Equipped with Lemma L.5, we now derive the the covering number bounds of loss function class under transformer weights configuration in Theorem I.1 and Theorem I.2.

**Lemma L.6** (Covering Number Bounds for $\mathcal{S}(D)$). Let $\epsilon_c > 0$. We define the loss function class by $\mathcal{S}(D) := \{\ell(x; u_\theta) : \mathcal{D} \to \mathbb{R} | u_\theta \in \mathcal{T}_R^{h,s,r}\}$, where $\mathcal{D} := [-D,D]^{d_x}$ for some $D > 0$. Given a fixed set of i.i.d. sample $\{x_i\}_{i=1}^n$ drawn from the target distribution $q$, we define the norm of loss functions by $\|\ell(x; u_\theta)\|_{q^n} := \sup_{x_i} |\ell^{\mathrm{trunc}}(x_i; u_\theta(x_i))|$. Then, under parameter configuration in Theorem I.1 and Theorem I.2, the $\epsilon_c$-covering number of $\mathcal{S}(D)$ with respect to $\|\cdot\|_{\infty \mathcal{D}}$ satisfies:

$$\log \mathcal{N}(\epsilon_c, \mathcal{S}(D), \{x_i\}_{i=1}^n, \|\cdot\|_{q^n}) \leq O\Big(\frac{\log(nL/\epsilon_c)}{\epsilon_c^2} D^4 N^{16\beta d + 12\beta}(\log N)^{20d_x + 4\beta + 17}\Big).$$

For all $f(x,t) \in \mathcal{T}_R^{h,s,r}$, $t \in [0,1]$ and $\{x_i\}_{i=1}^n \sim q^n$, we equip the transformers with the norm:

$$\|f(x,t)\|_{q^n, \mathcal{D}} := \big\|f(x_i,t)\mathbb{1}\{\|x_i\|_{2,\infty} \leq D\}\big\|_2.$$

Then, the $\epsilon_c$-covering number of the transformer network class satisfies:

$$\log \mathcal{N}(\epsilon_c, \mathcal{T}_R^{h,s,r}, q^n \times [0,1], \|\cdot\|_{q^n, \mathcal{D}}) \leq O\Big(\frac{\log(nL/\epsilon_c)}{\epsilon_c^2} D^2 N^{16\beta d + 12\beta}(\log N)^{20d_x + 4\beta + 16}\Big).$$

*Proof.* First, we apply transformers parameter bounds in Theorem I.1 and Theorem I.2. Then, we extend the covering number bound to loss function calss $\mathcal{S}(D)$.

- **Log-Covering Number of Transformers Network Class.** From Theorem I.1, we have

$$C_{KQ}, C_{KQ}^{2,\infty} = O\big(\lambda^{-1} N^{4\beta d + 2\beta} (\log N)^{4d_x + 2}\big); C_{OV}, C_{OV}^{2,\infty} = O\big(N^{-\beta}\big);$$

$$C_F, C_F^{2,\infty} = O\big(N^\beta (\log N)^{\frac{d_x + \beta}{2} + 1}\big); \ C_E = O(I); \ C_{\mathcal{T}} = O(\sqrt{\log N}).$$

By Lemma L.5, the bounds on log-covering number follow

$$\log \mathcal{N}(\epsilon_c, \mathcal{T}_R^{h,s,r}, q^n \times [0,1], \|\cdot\|_2)$$

$$\leq \frac{\alpha^2 \log nL/\epsilon_c}{\epsilon_c^2} \Big( (C_F^{2,\infty})^{\frac{4}{3}} + \big(\lambda (C_F)^2 C_{OV} C_{KQ}^{2,\infty}\big)^{\frac{2}{3}} + \big((C_F)^2 C_{OV}^{2,\infty}\big)^{\frac{2}{3}} \Big)^3$$

$$\lesssim \frac{\alpha^2 \log(nL/\epsilon_c)}{\epsilon_c^2} \big(\lambda (C_F)^2 C_{OV} C_{KQ}^{2,\infty}\big)^2, \qquad \text{(By dropping lower order terms)}$$

where

$$(C_F)^2 C_{OV} C_{KQ}^{2,\infty}$$

$$= O\big(\underbrace{N^{4\beta} (\log N)^{2d_x + 2\beta + 4}}_{(C_F)^4} \underbrace{N^{-2\beta}}_{(C_{OV})^2} \underbrace{N^{8\beta d + 4\beta} (\log N)^{8d_x + 4}}_{(\lambda C_{KQ}^{2,\infty})^2}\big)$$

$$= O(N^{8\beta d + 6\beta} (\log N)^{10d_x + 2\beta + 8}).$$

Therefore,

$$\log \mathcal{N}(\epsilon_c, \mathcal{T}_R^{h,s,r}, q^n \times [0,1], \|\cdot\|_2) \lesssim \frac{\alpha^2 \log(nL_{\mathcal{T}})}{\epsilon_c^2} (N^{8\beta d + 6\beta} (\log N)^{10d_x + 2\beta + 8}).$$

By Lemma L.5, we have

$$\alpha \lesssim (C_F)^2 C_{OV} C_{KQ} (D + C_E)$$

$$\lesssim \underbrace{N^{2\beta} (\log N)^{d_x + \beta + 2}}_{(C_F)^2} \underbrace{N^{-\beta}}_{(C_{OV})} \underbrace{N^{4\beta d + 2\beta} (\log N)^{4d_x + 2}}_{(\lambda C_{KQ})} (D + C_E) \qquad \text{(By the definition of } \alpha)$$

$$= O(DN^{4\beta d + 3\beta} (\log N)^{5d_x + \beta + 4}).$$

Altogether, we have

$$\log \mathcal{N}(\epsilon_c, \mathcal{T}_R^{h,s,r}, q^n \times [0,1], \|\cdot\|_2) \lesssim \frac{\log(nL_{\mathcal{T}})}{\epsilon_c^2} D^2 N^{16\beta d + 12\beta} (\log N)^{20d_x + 4\beta + 16}.$$

Further, by $\|\cdot\|_\infty \leq \|\cdot\|_2$, we have that

$$\log \mathcal{N}(\epsilon_c, \mathcal{T}_R^{h,s,r}, q^n \times [0,1], \|\cdot\|_\infty) \lesssim \frac{\log(nL_{\mathcal{T}})}{\epsilon_c^2} D^2 N^{16\beta d + 12\beta} (\log N)^{20d_x + 4\beta + 16}. \quad \text{(L.18)}$$

- **Log-Covering Number of Loss Function Class.** Recall the of loss function Definition L.1 and its truncated counterpart Definition L.3. Let $\delta > 0$ and $u_1(x,t), u_2(x,t) \in \mathcal{T}_R^{h,r,s}$ be any two transformers satisfying $\max_i \|u_1(x_i,t) - u_2(x_i,t)\|_\infty \leq \delta$ for all $i \in [n]$.

First, we derive the upper bound on the expectation of $\|u_t(x|x_1)\|$:

$$\mathbb{E}_{X_0 \sim N(0,I)}[\|u_t(x|x_1)\|_2] \qquad\qquad \text{(L.19)}$$

$$= \mathbb{E}_{X_0 \sim N(0,I)}[\|\dot{\mu}_t x + \dot{\sigma}_t X_0\|] \qquad\qquad \text{(By Definition L.1)}$$

$$\leq \sqrt{\underset{X_0 \sim N(0,I)}{\mathbb{E}}[\|\dot{\mu}_t x + \dot{\sigma}_t X_0\|_2^2]} \qquad \left(\text{By Jensen's inequality}\right)$$

$$\leq \sqrt{\underset{X_0 \sim N(0,I)}{\mathbb{E}}[\dot{\mu}_t^2 \|x\|_2^2 + \dot{\sigma}_t^2 \|X_0\|_2^2]} \qquad \left(\text{By expanding the } \ell_2 \text{ norm}\right)$$

$$= \sqrt{\underset{X_0 \sim N(0,I)}{\mathbb{E}}[\dot{\mu}_t^2 \|x\|_2^2] + \dot{\sigma}_t^2} \qquad \left(\text{By } X_0 \sim N(0,I)\right)$$

$$\leq \sqrt{\dot{\mu}_t^2 D^2 + \dot{\sigma}_t^2}. \qquad \left(\text{By } x \in [-D, D]^{d_x}\right)$$

Then, the distance between loss function $\ell_1(x; u_1)$ and $\ell_2(x; u_2)$ follows:

$$|\ell_1(x; u_1) - \ell_2(x; u_2)| \qquad \text{(L.20)}$$

$$= \frac{1}{T - t_0} \left| \int_{t_0}^T \underset{X_0 \sim N(0,I)}{\mathbb{E}}[\|u_1(x,t) - u_t(x|x_1)\|_2^2 - \|u_2(x,t) - u_t(x|x_1)\|_2^2]\mathrm{d}t \right|$$

$$\qquad \left(\text{By Definition L.1}\right)$$

$$= \frac{1}{T - t_0} \left| \int_{t_0}^T \underset{X_0 \sim N(0,I)}{\mathbb{E}}[(u_1(x,t) + u_2(x,t) - 2u_t(x|x_1))^\top (u_1(x,t) - u_2(x,t))]\mathrm{d}t \right|$$

$$\leq \frac{\delta}{T - t_0} \int_{t_0}^T \underset{X_0 \sim N(0,I)}{\mathbb{E}}[\|u_1(x,t) + u_2(x,t) - 2u_t(x|x_1)\|]\mathrm{d}t \qquad \left(\text{By } \|u_1 - u_2\| \leq \delta\right)$$

$$\leq \frac{\delta}{T - t_0} \int_{t_0}^T \sqrt{\underset{X_0 \sim N(0,I)}{\mathbb{E}}[\|u_1(x,t) + u_2(x,t) - 2u_t(x|x_1)\|_2^2]}\mathrm{d}t \qquad \left(\text{By Jensen's inequality}\right)$$

$$\leq \frac{\delta}{T - t_0} \int_{t_0}^T \sqrt{2 \underset{X_0 \sim N(0,I)}{\mathbb{E}}[\|u_1(x,t) + u_2(x,t)\|_2^2 + 2\|u_t(x|x_1)\|_2^2]}\mathrm{d}t$$

$$\qquad \left(\text{By expanding the } \ell_2 \text{ norm}\right)$$

$$\lesssim \frac{\delta}{T - t_0} \int_{t_0}^T \sqrt{\underset{X_0 \sim N(0,I)}{\mathbb{E}}[\log N + 2\|u_t(x|x_1)\|_2^2]}\mathrm{d}t \qquad \left(\text{By } C_{\mathcal{T}} = O(\sqrt{\log N})\right)$$

$$\lesssim \frac{\delta}{T - t_0} \int_{t_0}^T \sqrt{\log N + \dot{\mu}_t^2 D^2 + 4\dot{\sigma}_t^2}\mathrm{d}t \qquad \left(\text{By (L.19)}\right)$$

$$\lesssim \delta\sqrt{\log N + D^2}. \qquad \left(\text{By Assumption I.2}\right)$$

Finally, we extend the log covering number to the loss function class $\mathcal{S}(D)$ by setting

$$\epsilon_c' := \Omega\big(\epsilon_c \sqrt{\log N + D^2}\big).$$

This gives

$$\log \mathcal{N}(\epsilon_c', \mathcal{S}(D), \|\cdot\|_{\infty \mathcal{D}}) \leq \log \mathcal{N}(\epsilon_c, \mathcal{T}_R^{h,s,r}, \|\cdot\|_\infty). \qquad \text{(By (L.20))}$$

Therefore,

$$\log \mathcal{N}(\epsilon_c', \mathcal{S}(D), \|\cdot\|_{\infty \mathcal{D}})$$

$$\leq \log \mathcal{N}(\epsilon_c, \mathcal{T}_R^{h,s,r}, \|\cdot\|_\infty)$$

$$\lesssim \frac{\log(nL)}{\epsilon_c^2} \cdot D^2 N^{16\beta d + 12\beta} (\log N)^{20d_x + 4\beta + 16} \qquad \left(\text{By (L.18)}\right)$$

$$= O\Big(\frac{\log(nL)}{(\epsilon_c')^2} D^4 N^{16\beta d + 12\beta} (\log N)^{20d_x + 4\beta + 17}\Big). \qquad \left(\text{By the definition of } \epsilon_c'\right)$$

This completes the proof. □

**Generalization Bound.** Based on covering number bounds results in Lemma L.6, we analyze the upper bound of generalization error $\left| \mathbb{E}_{\{x_i\}_{i=1}^n} [\mathcal{R}^{\mathrm{trunc}}(\widehat{u}_\theta) - \widehat{\mathcal{R}}^{\mathrm{trunc}}(\widehat{u}_\theta)] \right|$. Note that the following distinction separates generalization bound for the flow-matching loss from classical learning theory.

The empirical risk (Definition L.2) takes the form $\ell(x; u_\theta) - \ell(x; u^\star)$, where $u^\star$ denotes the ground truth velocity. While most standard losses stay non-negative almost everywhere, the flow matching loss may take negative values. We use the next lemma in (L.25), which bounds the second moment of the flow matching loss in terms of its first moment. Without it, the sign issue breaks the derivation.

---

**Lemma L.7** (Bounds on Second Moment of Flow Matching Loss, Modified from Lemma C.1 of [Yakovlev and Puchkin, 2025] ). Assume Assumption I.1 and Assumption I.3. Then, it holds

$$\mathbb{E}_{x \sim q} \left[ \left| \ell^{\mathrm{trunc}}(x; u_\theta) - \ell^{\mathrm{trunc}}(x; u^\star) \right|^2 \right] \lesssim \kappa \cdot \mathbb{E}_{x \sim q} \left[ \ell^{\mathrm{trunc}}(x; u_\theta) - \ell^{\mathrm{trunc}}(x; u^\star) \right],$$

where $\kappa := D^2 + \sqrt{\log N}$.

---

*Proof.* Recall Definition L.1 and Definition L.2. We have

$$\ell^{\mathrm{trunc}}(x; u_\theta) := \ell(x; u_\theta) \mathbb{1}\{\|x\|_\infty \le D\} \quad \text{and} \quad \widehat{\mathcal{R}}(u_\theta) = \frac{1}{n} \sum_{i=1}^n \ell(x_i; u_\theta) - \frac{1}{n} \sum_{i=1}^n \ell(x_i; u^\star),$$

where $u^\star(x, t) = \frac{1}{p_t(x)} \cdot \int_{\mathbb{R}^{d_x}} u_t(x|x_1) p_t(x|x_1) q(x_1) \, \mathrm{d}x_1$ is the ground truth velocity and

$$\ell(x; u_\theta) := \frac{1}{T - t_0} \int_{t_0}^T \mathbb{E}_{X_0 \sim N(0, I)} [\|u_\theta(\mu_t x + \sigma_t X_0, t) - (\dot{\mu}_t x + \dot{\sigma}_t X_0)\|_2^2] \mathrm{d}t.$$

For any $x_i$, the flow matching loss takes the form $\ell(x_i; u_\theta) - \ell(x_i; u^\star)$. To simplify notation, we omit the indicator $\mathbb{1}\{\|x\|_\infty \le D\}$ when expanding $\ell^{\mathrm{trunc}}$, with the understanding that we focus only on the bounded domain where the flow matching loss is defined. Then, we compute

$$\left| \ell^{\mathrm{trunc}}(x; u_\theta) - \ell^{\mathrm{trunc}}(x; u^\star) \right|$$

$$= \left| \int_{t_0}^T \frac{1}{T - t_0} \mathbb{E}_{X_0 \sim N(0, I)} \left[ \|u_\theta - (\dot{\mu}_t x + \dot{\sigma}_t X_0)\|_2^2 - \|u^\star - (\dot{\mu}_t x + \dot{\sigma}_t X_0)\|_2^2 \right] \mathrm{d}t \right|$$

$$= \left| \int_{t_0}^T \frac{1}{T - t_0} \mathbb{E}_{X_0 \sim N(0, I)} \left[ (u_\theta - u^\star)^\top (u_\theta + u^\star - 2 \cdot (\dot{\mu}_t x + \dot{\sigma}_t X_0)) \right] \mathrm{d}t \right|$$

$$\le \left( \int_{t_0}^T \frac{1}{T - t_0} \mathbb{E} \left[ \|u_\theta - u^\star\|_2^2 \right] \mathrm{d}t \right)^{\frac{1}{2}} \cdot \underbrace{\left( \int_{t_0}^T \frac{1}{T - t_0} \mathbb{E} \left[ \|u_\theta + u^\star - 2 \cdot (\dot{\mu}_t x + \dot{\sigma}_t X_0)\|_2^2 \right] \right)^{\frac{1}{2}}}_{(A)},$$

(L.21)

where we apply the Cauchy-Schwarz inequality for the last inequality. Next, we bound (A) using previous results for the bounds on the true velocity, conditional velocity and transformer network.

Recall Lemma J.4. It holds

$$\|u^\star\|_\infty \le \frac{|\dot{\mu}_t|}{\mu_t} \cdot \|x\|_\infty + C_5 \left| \frac{\dot{\mu}_t}{\mu_t} - \frac{\dot{\sigma}_t}{\sigma_t} \right| \cdot (\|x\|_2 + 1),$$

and by Assumption I.3 we have $\|u^\star\|_\infty^2 \lesssim \|x\|_2^2$ and here we consider bounded domain $\|x\|_\infty \le D$.

Further, under the transformer network configuration in either Theorem I.1 or Theorem I.2, we have the transformer output bounds $C_\mathcal{T} = O(\sqrt{\log N})$. Lastly, for $\dot{\mu}_t x + \dot{\sigma}_t X_0$, it holds:

$$\mathop{\mathbb{E}}_{X_0 \sim N(0,I)} \left[\|\dot{\mu}_t x + \dot{\sigma}_t X_0\|_2^2\right] \leq \mathop{\mathbb{E}}_{X_0 \sim N(0,I)} \left[\|\dot{\mu}_t x\|_2^2 + \|\dot{\sigma}_t X_0\|_2^2\right] \lesssim D^2,$$

where we invoke Assumption I.3 and $\|x\|_2^2 \leq d_x D^2$ in the last inequality.

Altogether, we have

$$\text{(A)} \leq \int_{t_0}^{T} \frac{1}{T - t_0} \mathbb{E}\left[\|u_\theta\|_2^2 + \|u^\star\|_2^2 + \|2 \cdot (\dot{\mu}_t x + \dot{\sigma}_t X_0)\|_2^2 \lesssim D^2 + \sqrt{\log N}.$$

Therefore, (L.21) becomes:

$$\left|\ell^{\text{trunc}}(x; u_\theta) - \ell^{\text{trunc}}(x; u^\star)\right|^2 \lesssim \left(\int_{t_0}^{T} \frac{1}{T - t_0} \mathbb{E}\left[\|u_\theta - u^\star\|_2^2\right] dt\right) \cdot \left(D^2 + \sqrt{\log N}\right).$$

Then, we conclude that

$$\mathop{\mathbb{E}}_{x \sim q} \left[\left|\ell^{\text{trunc}}(x; u_\theta) - \ell^{\text{trunc}}(x; u^\star)\right|^2\right]$$

$$\lesssim \left(D^2 + \sqrt{\log N}\right) \cdot \int_{t_0}^{T} \frac{1}{T - t_0} \mathop{\mathbb{E}}_{x \sim q}\left[\mathop{\mathbb{E}}_{X_0 \sim N(0,I)}\left[\|u_\theta - u^\star\|_2^2\right] dt\right]$$

$$= \left(D^2 + \sqrt{\log N}\right) \cdot \underbrace{\int_{t_0}^{T} \frac{1}{T - t_0} \mathop{\mathbb{E}}_{x_t \sim p_t}\left[\|u_\theta - u^\star\|_2^2 dt\right]}_{\text{(B)}} \qquad \text{(By tower property)}$$

$$= \left(D^2 + \sqrt{\log N}\right) \cdot \mathop{\mathbb{E}}_{x \sim q}\left[\ell^{\text{trunc}}(x; u_\theta) - \ell^{\text{trunc}}(x; u^\star)\right]. \qquad \text{( By Remark L.2 )}$$

We remark that (B) is the conditional flow matching risk $\mathcal{R}(u_\theta)$ defined in (L.1).

This completes the proof. $\qquad\square$

---

**Lemma L.8** (Generalization Bound, Modified from the Theorem C.4 of [Oko et al., 2023]). Let $\widehat{u}_\theta$ be the velocity estimator trained by optimizing $\mathcal{L}_{\text{CFM}}(u_\theta)$ following Definition L.1 with i.i.d training samples $\{x_i\}_{i=1}^n$. For $\epsilon_c > 0$, let $\mathcal{N} := \mathcal{N}(\epsilon_c, \mathcal{S}(D), q^n, \|\cdot\|_\infty)$ be the covering number of function class of loss $\mathcal{S}(D)$ following Lemma L.6. Then we bound the generalization error:

$$\mathop{\mathbb{E}}_{\{x_i\}_{i=1}^n} \left[\mathcal{R}^{\text{trunc}}(\widehat{u}_\theta) - \widehat{\mathcal{R}}^{\text{trunc}}(\widehat{u}_\theta)\right] \lesssim \mathop{\mathbb{E}}_{\{x_i\}_{i=1}^n} [\widehat{\mathcal{R}}^{\text{trunc}}(\widehat{u}_\theta)] + O\left(\frac{1}{n} \log \mathcal{N} + \epsilon_c\right).$$

---

*Proof.* We use $\widehat{\mathcal{L}}'_{\text{CFM}}$ and $\widehat{\mathcal{R}}'$ to denote the conditional flow matching loss and empirical risk with ghost training samples $\{x_i'\}_{i=1}^n$. Further, let $u^\star$ denote the ground truth velocity field.

Then, following Remark L.3, we rewrite the generalization error:

$$\left|\mathop{\mathbb{E}}_{\{x_i\}_{i=1}^n} \left[\mathcal{R}^{\text{trunc}}(\widehat{u}_\theta) - \widehat{\mathcal{R}}^{\text{trunc}}(\widehat{u}_\theta)\right]\right| \qquad\qquad (\text{L.22})$$

$$= \left|\mathop{\mathbb{E}}_{\{x_i\}_{i=1}^n} \left[\mathop{\mathbb{E}}_{\{x_i'\}_{i=1}^n} \left[\widehat{\mathcal{R}}'^{\text{trunc}}(\widehat{u}_\theta)\right] - \widehat{\mathcal{R}}^{\text{trunc}}(\widehat{u}_\theta)\right]\right| \qquad\qquad (\text{By Remark L.3 })$$

$$= \left|\mathop{\mathbb{E}}_{\{x_i, x_i'\}_{i=1}^n} \left[\widehat{\mathcal{R}}'^{\text{trunc}}(\widehat{u}_\theta) - \widehat{\mathcal{R}}^{\text{trunc}}(\widehat{u}_\theta)\right]\right| \qquad\qquad (\text{By the independence between } x_i' \text{ and } \widehat{\mathcal{R}}(\widehat{u}_\theta))$$

$$= \left| \frac{1}{n} \mathop{\mathbb{E}}_{\{x_i,x_i'\}_{i=1}^n} \left[ \Big( \sum_{i=1}^n \ell^{\text{trunc}}(x_i'; \widehat{u}_\theta) - \sum_{i=1}^n \ell^{\text{trunc}}(x_i'; u^\star) \Big) - \Big( \sum_{i=1}^n \ell^{\text{trunc}}(x_i; \widehat{u}_\theta) - \sum_{i=1}^n \ell^{\text{trunc}}(x_i; u^\star) \Big) \right] \right|.$$

$$\text{(By Definition L.2)}$$

For $\epsilon_c > 0$ to be chosen later, let $\mathcal{J} := \{\ell_1, \ell_2, \dots, \ell_{\mathcal{N}}\}$ be a $\epsilon_c$-covering of the loss function class $\mathcal{S}(\mathcal{D})$ with the minimum cardinality in the $L_\infty$ metric. Note that $\ell_1, \dots, \ell_{\mathcal{N}}$ have domain $\mathcal{D} = [-D, D]^{d_x}$ by Definition L.3 and Definition L.4. Further, let $J$ be the random variable such that $\|\ell(\cdot, \widehat{u}_\theta) - \ell_J(\cdot, u_J)\|_\infty \le \epsilon_c$. Moreover, we introduce following definitions for simplicity:

$$\omega(x) := \ell^{\text{trunc}}(x; \widehat{u}_\theta) - \ell^{\text{trunc}}(x; u^\star),$$
$$\omega_j(x) := \ell_j(x; u_j) - \ell^{\text{trunc}}(x; u^\star),$$
$$h_j := \max \left\{ A, \sqrt{\mathop{\mathbb{E}}_z [\ell_j(z; u_j) - \ell^{\text{trunc}}(z; u^\star)]} \right\},$$
$$\Omega := \max_{1 \le j \le \mathcal{N}} \left| \sum_{i=1}^n \frac{\omega_j(x_i') - \omega_j(x_i)}{h_j} \right|,$$

where $z \sim q$ is independent of $\{x_i, x_i'\}_{i=1}^n$. Then we can further bound (L.22) as follows:

$$\left| \frac{1}{n} \mathop{\mathbb{E}}_{\{x_i,x_i'\}_{i=1}^n} \left[ \Big( \sum_{i=1}^n \ell^{\text{trunc}}(x_i'; \widehat{u}_\theta) - \sum_{i=1}^n \ell^{\text{trunc}}(x_i'; u^\star) \Big) - \Big( \sum_{i=1}^n \ell^{\text{trunc}}(x_i; \widehat{u}_\theta) - \sum_{i=1}^n \ell^{\text{trunc}}(x_i; u^\star) \Big) \right] \right|$$

$$\text{(L.23)}$$

$$\le \left| \frac{1}{n} \mathop{\mathbb{E}}_{\{x_i,x_i'\}_{i=1}^n} \Big[ \Big( \sum_{i=1}^n (\omega_J(x_i') - \omega_J(x_i)) \Big) \Big] \right| + 2\epsilon_c \qquad \text{(By the definitions of } \omega_J \text{ and covering number)}$$

$$\le \frac{1}{n} \mathop{\mathbb{E}}_{\{x_i,x_i'\}_{i=1}^n} \Big[ \Big| \Big( \sum_{i=1}^n (\omega_J(x_i') - \omega_J(x_i)) \Big) \Big| \Big] + 2\epsilon_c \qquad \text{(By the property of expectation)}$$

$$\le \frac{1}{n} \mathop{\mathbb{E}}_{\{x_i,x_i'\}_{i=1}^n} [h_J \Omega] + 2\epsilon_c \qquad \text{(By the definitions of } h_j \text{ and } \Omega)$$

$$\le \frac{1}{n} \sqrt{\mathop{\mathbb{E}}_{\{x_i,x_i'\}_{i=1}^n} [h_J^2] \mathop{\mathbb{E}}_{\{x_i,x_i'\}_{i=1}^n} [\Omega^2]} + 2\epsilon_c \qquad \text{(By Cauchy-Schwarz inequality)}$$

$$\le \frac{1}{n} \Big( \frac{n}{2} \mathop{\mathbb{E}}_{\{x_i,x_i'\}_{i=1}^n} [h_J^2] + \frac{1}{2n} \mathop{\mathbb{E}}_{\{x_i,x_i'\}_{i=1}^n} [\Omega^2] \Big) + 2\epsilon_c \qquad \text{(By AM-GM Inequality)}$$

$$= \frac{1}{2} \mathop{\mathbb{E}}_{\{x_i,x_i'\}_{i=1}^n} [h_J^2] + \frac{1}{2n^2} \mathop{\mathbb{E}}_{\{x_i,x_i'\}_{i=1}^n} [\Omega^2] + 2\epsilon_c.$$

Now we bound $\mathbb{E}_{\{x_i,x_i'\}_{i=1}^n}[h_J^2]$ and $\mathbb{E}_{\{x_i,x_i'\}_{i=1}^n}[\Omega^2]$ separately. For $\mathbb{E}_{\{x_i,x_i'\}_{i=1}^n}[h_J^2]$, we have

$$\mathop{\mathbb{E}}_{\{x_i,x_i'\}_{i=1}^n} [h_J^2] \le A^2 + \mathop{\mathbb{E}}_{\{x_i,x_i'\}_{i=1}^n} [\mathop{\mathbb{E}}_z [\ell_j(z; u_J) - \ell^{\text{trunc}}(z; u^\star)]] \qquad \text{(By the definition of } h_j)$$

$$\le A^2 + \mathop{\mathbb{E}}_{\{x_i,x_i'\}_{i=1}^n} [\mathop{\mathbb{E}}_z [\ell^{\text{trunc}}(z; \widehat{u}_\theta) - \ell^{\text{trunc}}(z; u^\star)]] + 2\epsilon_c \quad \text{(By the definition of } \epsilon_c)$$

$$\le A^2 + \mathop{\mathbb{E}}_{\{x_i\}_{i=1}^n} [\mathcal{R}^{\text{trunc}}(\widehat{u}_\theta)] + 2\epsilon_c. \qquad \text{(By Remark L.3)}$$

Then we start to bound $\mathbb{E}_{\{x_i,x_i'\}_{i=1}^n}[\Omega^2]$. By the definition of $\omega_j(x)$ and the independence between $\{x_i\}_{i=1}^n$ and $\{x_i'\}_{i=1}^n$, we have

$$\mathop{\mathbb{E}}_{x_i,x_i'} \left[ \frac{\omega_j(x_i)\omega_j(x_i')}{h_j^2} \right]$$

$$= \frac{1}{h_j^2} \mathop{\mathbb{E}}_{x_i} [\omega_j(x_i)] \cdot \mathop{\mathbb{E}}_{x_i'} [\omega_j(x_i')] \qquad \text{(By the independence between } h_j \text{ and } \{x_i, x_i'\}_{i=1}^n)$$

$$= \frac{1}{h_j^2}\Big(\mathop{\mathbb{E}}_{x_i}\big[\omega_j(x_i)\big]\Big)^2 \qquad\qquad \left(\text{By the independence between } w_j \text{ and } \{x_i, x_i'\}_{i=1}^n\right)$$

$$\geq 0. \tag{L.24}$$

To use Bernstein's Inequality, for any $j$, we bound the following expectation as

$$\mathop{\mathbb{E}}_{\{x_i,x_i'\}_{i=1}^n}\Big[\sum_{i=1}^n\Big(\frac{\omega_j(x_i) - \omega_j(x_i')}{h_j}\Big)^2\Big]$$

$$= \sum_{i=1}^n\Big(\mathop{\mathbb{E}}_{x_i,x_i'}\Big[\Big(\frac{\omega_j(x_i)}{h_j}\Big)^2 + \Big(\frac{\omega_j(x_i')}{h_j}\Big)^2\Big] - 2\mathop{\mathbb{E}}_{x_i,x_i'}\Big[\frac{\omega_j(x_i)\omega_j(x_i')}{h_j^2}\Big]\Big)$$

$$\leq \sum_{i=1}^n\mathop{\mathbb{E}}_{x_i,x_i'}\Big[\Big(\frac{\omega_j(x)}{h_j}\Big)^2 + \Big(\frac{\omega_j(x')}{h_j}\Big)^2\Big]. \qquad\qquad \big(\text{By (L.24)}\big)$$

Recall that for any $j \in [\mathcal{N}]$, $\omega_j(x) := \ell_j(x; u_j) - \ell^{\text{trunc}}(x; u^\star)$. For any $\ell \in \mathcal{S}(D)$, assume $|\ell^{\text{trunc}}(\cdot; u_\theta)| \leq \kappa$, then for any $i \in [n], j \in [\mathcal{N}]$, we have $\mathbb{E}_{x_i,x_i'}[\omega_j(x_i)] = \mathbb{E}_{x_i,x_i'}[\omega_j(x_i')]$, which leads to

$$\mathop{\mathbb{E}}_{x_i,x_i'}[\omega_j(x_i)] = \mathop{\mathbb{E}}_{x_i,x_i'}[\omega_j(x_i')]$$

$$= \mathop{\mathbb{E}}_{x_i,x_i'}[\ell_j(x_i'; u_j) - \ell^{\text{trunc}}(x_i'; u^\star)] \qquad\qquad \big(\text{By the definition of } \omega_j(x)\big)$$

$$= \mathop{\mathbb{E}}_{z}[\ell_j(z; u_j) - \ell^{\text{trunc}}(z; u^\star)]$$

$$\leq \mathop{\mathbb{E}}_{x_i,x_i'}[h_j^2]. \qquad\qquad \big(\text{By the definition of } h_j\big)$$

Then, it holds

$$\mathop{\mathbb{E}}_{\{x_i,x_i'\}_{i=1}^n}\Big[\sum_{i=1}^n\Big(\frac{\omega_j(x_i) - \omega_j(x_i')}{h_j}\Big)^2\Big]$$

$$\leq \sum_{i=1}^n\mathop{\mathbb{E}}_{x_i,x_i'}\Big[\Big(\frac{\omega_j(x_i)}{h_j}\Big)^2 + \Big(\frac{\omega_j(x_i')}{h_j}\Big)^2\Big]$$

$$\leq 2\kappa\sum_{i=1}^n\mathop{\mathbb{E}}_{x_i,x_i'}\Big[\Big(\frac{\omega_j(x_i)}{h_j^2}\Big) + \Big(\frac{\omega_j(x_i')}{h_j^2}\Big)\Big] \qquad\qquad \big(\text{By Lemma L.7}\big)$$

$$\leq 4n\kappa. \tag{L.25}$$

Since $\left|\frac{\omega_j(x_i) - \omega_j(x_i')}{h_j}\right| \leq \frac{\kappa}{A}$ and $\mathbb{E}_{\{x_i,x_i'\}_{i=1}^n}[\frac{\omega_j(x_i)-\omega_j(x_i')}{h_j}] = 0$, by Bernstein's Inequality, we have for any $j \in [\mathcal{N}], h > 0$,

$$\Pr\Big[(\sum_{i=1}^n\frac{\omega_j(x_i) - \omega_j(x_i')}{h_j})^2 \geq h\Big] = 2\Pr\Big[\sum_{i=1}^n\frac{\omega_j(x_i) - \omega_j(x_i')}{h_j} \geq \sqrt{h}\Big]$$

$$\leq 2\exp\Big(-\frac{h/2}{\kappa(4n + \frac{\sqrt{h}}{3A})}\Big).$$

Thus, we have

$$\Pr\big[\Omega^2 \geq h\big] \leq \sum_{j=1}^{\mathcal{N}}\Pr\Big[(\sum_{i=1}^n\frac{\omega_j(x_i) - \omega_j(x_i')}{h_j})^2 \geq h\Big] \qquad\qquad \big(\text{By union bound.}\big)$$

$$\le 2\mathcal{N} \exp\left(-\frac{h/2}{\kappa(4n + \frac{\sqrt{h}}{3A})}\right).$$

Thus, for any $h_0 > 0$, we bound $\mathbb{E}_{\{x_i,x_i'\}_{i=1}^n}[\Omega^2]$ as

$$\mathbb{E}_{\{x_i,x_i'\}_{i=1}^n}[\Omega^2]$$

$$= \int_0^{h_0} \Pr[\Omega^2 \ge h]\mathrm{d}h + \int_{h_0}^\infty \Pr[\Omega^2 \ge h]\mathrm{d}h$$

$$\le h_0 + \int_{h_0}^\infty 2\mathcal{N} \exp\left(-\frac{h/2}{\kappa(4n + \frac{\sqrt{h}}{3A})}\right)\mathrm{d}h \qquad \text{(By tail-sum formula)}$$

$$\le h_0 + 2\mathcal{N} \int_{h_0}^\infty [\exp\left(-\frac{h}{16\kappa n}\right) + \exp\left(-\frac{3A\sqrt{h}}{4\kappa}\right)]\mathrm{d}h$$

$$\le h_0 + 2\mathcal{N}[16\kappa n \exp\left(-\frac{h_0}{16\kappa n}\right) + (\frac{8\kappa\sqrt{h_0}}{3A} + \frac{32\kappa}{9A^2})\exp\left(-\frac{3A\sqrt{h_0}}{4\kappa}\right)].$$

Taking $A = \frac{\sqrt{h_0}}{12n}$ and $h_0 = 16\kappa n \log\mathcal{N}$, we have

$$\mathbb{E}_{\{x_i,x_i'\}_{i=1}^n}[\Omega^2] \lesssim n\kappa \log\mathcal{N}.$$

Combining above, we bound the generalization error as

$$\left|\mathbb{E}_{\{x_i\}_{i=1}^n}[\mathcal{R}^{\mathrm{trunc}}(\widehat{u}_\theta) - \widehat{\mathcal{R}}^{\mathrm{trunc}}(\widehat{u}_\theta)]\right|$$

$$\le \frac{1}{2}\mathbb{E}_{\{x_i,x_i'\}_{i=1}^n}[h_j^2] + \frac{1}{2n^2}\mathbb{E}_{\{x_i,x_i'\}_{i=1}^n}[\Omega^2] + 2\epsilon_c \qquad \text{(By (L.22))}$$

$$\le \frac{1}{2}\left(A^2 + \mathbb{E}_{\{x_i\}_{i=1}^n}[\mathcal{R}^{\mathrm{trunc}}(\widehat{u}_\theta)] + 2\epsilon_c\right) + \frac{1}{2n^2}O(n\kappa\log\mathcal{N})$$

$$\lesssim \frac{1}{2}\mathbb{E}_{\{x_i\}_{i=1}^n}[\mathcal{R}^{\mathrm{trunc}}(\widehat{u}_\theta)] + O\left(\frac{\kappa}{n}\log\mathcal{N} + \epsilon_c\right).$$

This implies

$$\mathbb{E}_{\{x_i\}_{i=1}^n}\left[\mathcal{R}^{\mathrm{trunc}}(\widehat{u}_\theta)\right] \lesssim 2 \cdot \mathbb{E}_{\{x_i\}_{i=1}^n}\left[\widehat{\mathcal{R}}^{\mathrm{trunc}}(\widehat{u}_\theta)\right] + O\left(\frac{\kappa}{n}\log\mathcal{N} + \epsilon_c\right).$$

Therefore,

$$\mathbb{E}_{\{x_i\}_{i=1}^n}\left[\mathcal{R}^{\mathrm{trunc}}(\widehat{u}_\theta) - \widehat{\mathcal{R}}^{\mathrm{trunc}}(\widehat{u}_\theta)\right] \lesssim \mathbb{E}_{\{x_i\}_{i=1}^n}\left[\widehat{\mathcal{R}}^{\mathrm{trunc}}(\widehat{u}_\theta)\right] + O\left(\frac{\kappa}{n}\log\mathcal{N} + \epsilon_c\right).$$

This completes the proof. $\qquad\qquad \square$

## L.3 Main Proof of Theorem I.3

We now give the formal proof of Theorem I.3.

**Theorem L.1** (Theorem I.3 Restated: Velocity Estimation with Transformer). Let $d$ be the feature dimension. Suppose we choose the transformers as in Theorem I.1 and Theorem I.2 correspondingly, then we have

- Assume Assumption I.1 and Assumption I.2. Then,

$$\mathop{\mathbb{E}}_{\{x_i\}_{i=1}^n}[\mathcal{R}(\widehat{u}_\theta)] = O(n^{-\frac{1}{16d+15}}(\log n)^{20d_x+4\beta+20}).$$

- Assume Assumption I.2 and Assumption I.3. Then,

$$\mathop{\mathbb{E}}_{\{x_i\}_{i=1}^n}[\mathcal{R}(\widehat{u}_\theta)] = O(n^{-\frac{1}{8d+9}}(\log n)^{20d_x+4\beta+20}).$$

*Proof of Theorem I.3.* Let $\{x_i'\}_{i=1}^n$ be a different set of i.i.d samples independent of the training sample $\{x_i\}_{i=1}^n$. Further, we use $\widehat{\mathcal{R}}'$ to denote the empirical risk with samples $\{x_i'\}_{i=1}^n$. Then, following (L.2), we decompose $\mathbb{E}_{\{x_i\}_{i=1}^n}[\mathcal{R}(\widehat{u}_\theta)]$ as:

$$\mathop{\mathbb{E}}_{\{x_i\}_{i=1}^n}[\mathcal{R}(\widehat{u}_\theta)] = \underbrace{\mathop{\mathbb{E}}_{\{x_i\}_{i=1}^n}\left[\mathop{\mathbb{E}}_{\{x_i'\}_{i=1}^n}\left[\widehat{\mathcal{R}}'(\widehat{u}_\theta) - \widehat{\mathcal{R}}'^{\text{trunc}}(\widehat{u}_\theta)\right]\right]}_{\text{(I)}}$$

$$+ \underbrace{\mathop{\mathbb{E}}_{\{x_i\}_{i=1}^n}\left[\mathop{\mathbb{E}}_{\{x_i'\}_{i=1}^n}\left[\widehat{\mathcal{R}}'^{\text{trunc}}(\widehat{u}_\theta) - \widehat{\mathcal{R}}^{\text{trunc}}(\widehat{u}_\theta)\right]\right]}_{\text{(II)}}$$

$$+ \underbrace{\mathop{\mathbb{E}}_{\{x_i\}_{i=1}^n}\left[\widehat{\mathcal{R}}^{\text{trunc}}(\widehat{u}_\theta) - \widehat{\mathcal{R}}(\widehat{u}_\theta)\right]}_{\text{(III)}} + \underbrace{\mathop{\mathbb{E}}_{\{x_i\}_{i=1}^n}\left[\widehat{\mathcal{R}}(\widehat{u}_\theta)\right]}_{\text{(IV)}}.$$

Then, we bound each term and incorporate them to obtain the upper bound on the estimation error.

- **Bound** (I) **and** (III). By Lemma L.1, term (I) and term (III) are upper bounded by

$$\text{(I)}, \text{(III)} \lesssim D^{d_x}\exp\left(-\frac{1}{2}C_2 D^2\right)\log N.$$

- **Bound** (II). By the generalization error bound (Lemma L.8), we have

$$\begin{aligned}
\text{(II)} &= \mathop{\mathbb{E}}_{\{x_i\}_{i=1}^n}\left[\mathop{\mathbb{E}}_{\{x_i'\}_{i=1}^n}\left[\widehat{\mathcal{R}}'^{\text{trunc}}(\widehat{u}_\theta)\right] - \widehat{\mathcal{R}}^{\text{trunc}}(\widehat{u}_\theta)\right] &\text{(L.26)}\\
&= \mathop{\mathbb{E}}_{\{x_i\}_{i=1}^n}[\mathcal{R}^{\text{trunc}}(\widehat{u}_\theta) - \widehat{\mathcal{R}}^{\text{trunc}}(\widehat{u}_\theta)] &\left(\text{By } \mathbb{E}_{\{x_i'\}_{i=1}^n}[\widehat{\mathcal{R}}'^{\text{trunc}}] = \mathcal{R}^{\text{trunc}}\right)\\
&\lesssim \mathop{\mathbb{E}}_{\{x_i\}_{i=1}^n}[\mathcal{R}^{\text{trunc}}(\widehat{u}_\theta)] + O(\frac{1}{n}\log\mathcal{N} + \epsilon_c) &\left(\text{By Lemma L.8}\right)\\
&\lesssim \text{(IV)} + D^{d_x}\exp\left(-\frac{1}{2}C_2 D^2\right)\log N + O(\frac{1}{n}\log\mathcal{N} + \epsilon_c). &\left(\text{By Lemma L.1}\right)
\end{aligned}$$

where $\mathcal{N}(\epsilon_c, \mathcal{S}(D), \|\cdot\|_{\infty\mathcal{D}})$ is the covering number (Definition L.4) of loss function class.

- **Bound** (IV). Recall that $\widehat{\mathcal{R}}(\widehat{u}_\theta) := \widehat{\mathcal{L}}_{\text{CFM}}(\widehat{u}_\theta) - \widehat{\mathcal{L}}_{\text{CFM}}(u^\star)$ and $\widehat{u}_\theta$ is trained by optimizing $\widehat{\mathcal{L}}_{\text{CFM}}(u_\theta)$ following Definition L.2. Therefore, for any velocity estimator $u_\theta$, it holds

$$\widehat{\mathcal{R}}(\widehat{u}_\theta) \le \widehat{\mathcal{L}}_{\text{CFM}}(u_\theta) - \widehat{\mathcal{L}}_{\text{CFM}}(u^\star) = \widehat{\mathcal{R}}(u_\theta).$$

Then, for any velocity estimator $u_\theta$, it holds

$$\mathbb{E}_{\{x_i\}_{i=1}^n}[\widehat{\mathcal{R}}(\widehat{u}_\theta)] \le \mathbb{E}_{\{x_i\}_{i=1}^n}[\widehat{\mathcal{R}}(u_\theta)] = \mathcal{R}(u_\theta). \tag{L.27}$$

Altogether, the estimation error is upper bounded by

$$\mathbb{E}_{\{x_i\}_{i=1}^n}[\mathcal{R}(\widehat{u}_\theta)] \tag{L.28}$$
$$= \text{(I)} + \text{(II)} + \text{(III)} + \text{(IV)}$$
$$\lesssim D^{d_x}\exp(-C_2 D^2)\log N + O(\frac{1}{n}\log\mathcal{N} + \epsilon_c) + 2\text{(IV)}$$
$$\le O(N^{-2\beta}(\log N)^{d_x/2+1}) + O(\frac{1}{n}\log\mathcal{N} + \epsilon_c) + 2\text{(IV)}. \qquad \left(\text{By setting } D \coloneqq \sqrt{2\beta\log N/C_2}\right)$$

Furthermore, the log covering number is upper bounded by

$$\log\mathcal{N}(\epsilon_c, \mathcal{S}(D), \|\cdot\|_{\infty\mathcal{D}}) \tag{L.29}$$
$$\le O\left(\frac{\log(nL/\epsilon_c)}{\epsilon_c^2}D^4 N^{16\beta d+12\beta}(\log N)^{20d_x+4\beta+17}\right) \qquad \left(\text{By Lemma L.6}\right)$$
$$\le O\left(\frac{\log(nL/\epsilon_c)}{\epsilon_c^2}N^{16\beta d+12\beta}(\log N)^{20d_x+4\beta+19}\right). \qquad \left(\text{By } D \coloneqq \sqrt{2\beta\log N/C_2}\right)$$

Next, we bound the velocity field estimation error.

- **Estimation Rates under Generic Hölder Smoothness.** By Theorem I.1, it holds

$$\text{(IV)} \le \mathcal{R}(u_\theta(x,t)) \qquad \left(\text{By (L.27)}\right)$$
$$= \int\frac{1}{T-t_0}\int_{\mathbb{R}^{d_x}}\|u_t(x) - u_\theta(x,t)\|_2^2 p_t(x)\mathrm{d}x\mathrm{d}t$$
$$= O(B^2 N^{-\beta}\cdot(\log N)^{d_x+\frac{\beta}{2}+1}). \qquad \left(\text{By Theorem I.1}\right)$$

Then, (L.28) becomes

$$\mathbb{E}_{\{x_i\}_{i=1}^n}[\mathcal{R}(\widehat{u}_\theta)]$$
$$\le O(N^{-2\beta}(\log N)^{d_x/2+1}) + O(\frac{1}{n}\log\mathcal{N} + \epsilon_c) + O(B^2 N^{-\beta}(\log N)^{d_x+\frac{\beta}{2}+1})$$
$$\le O(N^{-2\beta}(\log N)^{d_x/2+1}) + O(\frac{\log(nL/\epsilon_c)}{n\epsilon_c^2}N^\nu(\log N)^{20d_x+4\beta+19} + \epsilon_c) + O(B^2 N^{-\beta}(\log N)^{d_x+\frac{\beta}{2}+1}),$$
$$\left(\text{By (L.29)}\right)$$

where $\nu \coloneqq 16\beta d + 12\beta$.

Let $\gamma_1, \gamma_2 \in (0,1)$ be two arbitrary numbers. We take $N = n^{\gamma_1/\nu}$ and $\epsilon_c = n^{-\gamma_2}$. Then,

$$\mathbb{E}_{\{x_i\}_{i=1}^n}[\mathcal{R}(\widehat{u}_\theta)]$$
$$\le O(n^{-\frac{2\beta\gamma_1}{\nu}}(\log n)^{\frac{d_x}{2}+1}) + O(n^{-1+\gamma_1+2\gamma_2}(\log N)^{20d_x+4\beta+20}\log n + n^{-\gamma_2}) + O(B^2 n^{-\frac{\beta\gamma_1}{\nu}}(\log n)^{d_x+\frac{\beta}{2}+1})$$
$$\le O(n^{-\min\{\frac{\beta\gamma_1}{\nu}, 1-\gamma_1-2\gamma_2, \gamma_2\}}(\log n)^{20d_x+4\beta+21}).$$

For any $\gamma_1, \gamma_2 \in (0,1)$ satisfying

$$\gamma_1 + 2\gamma_2 < 1,$$

we consider

$$\min\{\frac{\beta\gamma_1}{\nu}, 1 - \gamma_1 - 2\gamma_2, \gamma_2\}.$$

To simplify, we set

$$\frac{\beta\gamma_1}{\nu} = 1 - \gamma_1 - 2\gamma_2 = \gamma_2,$$

giving

$$\gamma_1 = \frac{\nu}{\nu + 3\beta}, \quad \gamma_2 = \frac{\beta}{\nu + 3\beta}.$$

Therefore,

$$\mathbb{E}_{\{x_i\}_{i=1}^n} [\mathcal{R}(\widehat{u}_\theta)] = O(n^{-\frac{1}{16d+15}} (\log n)^{20d_x + 4\beta + 21}).$$

- **Estimation Rates under Stronger Hölder Smoothness.** By Theorem I.2, it holds

$$\text{(IV)} \le \mathcal{R}(u_\theta(x, t)) \hspace{4cm} \text{(By (L.27))}$$
$$= \int \int \|u_t(x) - u_\theta(x, t)\|_2^2 \cdot p_t(x) \mathrm{d}x \mathrm{d}t$$
$$= O(B^2 N^{-2\beta} (\log N)^{d_x + \beta}).$$

Then, (L.28) becomes

$$\mathbb{E}_{\{x_i\}_{i=1}^n} [\mathcal{R}(\widehat{u}_\theta)]$$
$$\le O(N^{-2\beta} (\log N)^{\frac{d_x}{2}+1}) + O(\frac{1}{n} \log \mathcal{N} + \epsilon_c) + O(B^2 N^{-2\beta} (\log N)^{d_x + \beta})$$
$$\le O(N^{-2\beta} (\log N)^{\frac{d_x}{2}+1}) + O(\frac{\log n}{n\epsilon_c^2} N^\nu (\log N)^{20d_x + 4\beta + 19} + \epsilon_c) + O(B^2 N^{-2\beta} (\log N)^{d_x + \beta}),$$
$$\hspace{9cm} \text{(By (L.29))}$$

where $\nu := 16\beta d + 12\beta$.

Let $\gamma_3, \gamma_4 \in (0, 1)$ be two arbitrary numbers. We take $N = n^{\gamma_3/\nu}$ and $\epsilon_c = n^{-\gamma_4}$. Then,

$$\mathbb{E}_{\{x_i\}_{i=1}^n} [\mathcal{R}(\widehat{u}_\theta)]$$
$$\le O(n^{-\frac{2\beta\gamma_3}{\nu}} (\log n)^{\frac{d_x}{2}+1}) + O(n^{-1} n^{2\gamma_4} n^{\gamma_3} (\log n)^{20d_x + 4\beta + 20} + n^{-\gamma_4}) + O(B^2 n^{-\frac{2\beta\gamma_3}{\nu}} (\log n)^{d_x + \beta})$$
$$\le O(n^{-\min\{\frac{2\beta\gamma_3}{\nu}, 1 - \gamma_3 - 2\gamma_4, \gamma_4\}} (\log n)^{20d_x + 4\beta + 20}).$$

For any $\gamma_3, \gamma_4 \in (0, 1)$ satisfying

$$\gamma_3 + 2\gamma_4 < 1,$$

we consider

$$\min\{\frac{2\beta\gamma_3}{\nu}, 1 - \gamma_3 - 2\gamma_4, \gamma_4\}.$$

To simplify, we set

$$\frac{2\beta\gamma_3}{\nu} = 1 - \gamma_3 - 2\gamma_4 = \gamma_4,$$

giving

$$\gamma_3 = \frac{\nu}{\nu + 6\beta}, \quad \gamma_4 = \frac{2\beta}{\nu + 6\beta}.$$

Therefore,

$$\mathbb{E}_{\{x_i\}_{i=1}^n} [\mathcal{R}(\widehat{u}_\theta)] = O(n^{-\frac{1}{8d+9}} (\log n)^{20d_x + 4\beta + 20}).$$

This completes the proof. $\square$

# M  Proof of Theorem I.4

In this section, we apply the Grönwall's inequality and the Alekseev–Gröbner lemma to extend the velocity estimation to distribution estimation under 2-Wasserstein distance.

**Organizations.**  Appendix M.1 introduces auxiliary lemmas. Appendix M.2 presents the main proof.

## M.1  Auxiliary Lemmas

In this section, we introduce auxiliary lemmas for extending the velocity estimation to distribution estimation in 2-Wasserstein distance. Specifically, we state the Grönwall's inequality in Lemma M.1. Furthermore, we introduce the Alekseev–Gröbner lemma that quantifies the deviation between solutions of two distinct ODEs in terms of the discrepancy between their velocity in Lemma M.2.

We begin with the Grönwall's inequality.

**Lemma M.1** (Grönwall's Inequality, [Gronwall, 1919]).  Let $a, b \in \mathbb{R}$ with $a < b$. Let $g(t)$ and $y(t)$ be two real-valued continuous functions defined on $[a, b]$. Then, if $y(t)$ is differentiable on $[a, b]$ and satisfies:

$$\frac{\mathrm{d}}{\mathrm{d}t}y(t) \leq y(t)g(t), \quad t \in [a, b],$$

it holds

$$y(t) \leq y(a)\exp\left(\int_a^b g(s)\mathrm{d}s\right).$$

Next, we introduce the Alekseev-Gröbner lemma.

**Lemma M.2** (Alekseev-Gröbner Lemma, Lemma 16 of [Fukumizu et al., 2024], Proposition 2 of [Benton et al., 2023], Theorem 14.5 of [Hairer et al., 1993]).  Let $u(x, t)$ and $u_\theta(x, t)$ be smooth vector fields and $\psi(x, s, t)$ and $\psi_\theta(x, s, t)$ be the respective flows defined for $t \geq s$ that satisfy

$$\frac{\mathrm{d}}{\mathrm{d}t}\psi(x, s, t) = u(\psi(x, s, t), t), \quad \psi(x, s, s) = x$$
$$\frac{\mathrm{d}}{\mathrm{d}t}\psi_\theta(x, s, t) = u_\theta(\psi_\theta(x, s, t), t), \quad \psi_\theta(x, s, s) = x.$$

Then,

$$\psi_\theta(x, t_0, T) - \psi(x, t_0, T) = \int_{t_0}^T D\psi_\theta(\psi(x, t_0, s), s, T)(u_\theta(\psi(x, t_0, s), s) - u(\psi(x, t_0, s), s))\mathrm{d}s,$$

where the partial derivatives in the Jacobian matrix $D\psi_\theta(\psi(x, t_0, s), s, T)$ is with respect to its first argument.

## M.2  Main Proof of Theorem I.4

We now present the main proof of Theorem I.4.

**Theorem M.1** (Theorem I.4 Restated: Distribution Estimation under Wasserstein Distance).  Let $\widehat{P}_T$ denote the estimated distribution at time $T$. Further, we define a constant $\nu := 16(L + 1) + 12/d$.

- Assume Assumption I.1 and Assumption I.2. It holds

$$\mathbb{E}_{\{x_i\}_{i=1}^n}[W_2(\widehat{P}_T, P_T)] = O(n^{-\frac{1}{32d+30}}(\log n)^{10d_x + 2\beta + 10}).$$

- Assume Assumption I.2 and Assumption I.3. It holds

$$\mathop{\mathbb{E}}_{\{x_i\}_{i=1}^n}[W_2(\widehat{P}_T, P_T)] = O(n^{-\frac{1}{16d+18}}(\log n)^{10d_x+2\beta+10}).$$

*Proof of Theorem I.4.* We bound the 2-Wasserstein distance between the estimated and true distributions with the $\ell_2$ difference of the velocity field network and the true velocity field. Our proof structure follows [Fukumizu et al., 2024, Theorem 3] and [Benton et al., 2023, Theorem 1].

The distributions $\widehat{P}_T$ and $P_T$ are the pushforwards of $P_{t_0}$ by $\psi_\theta(\cdot, t_0, T)$ and $\psi(\cdot, t_0, T)$. Thus, using the definition of the 2-Wasserstein metric, it follows that

$$W_2(\widehat{P}_T, P_T) \leq \sqrt{\mathop{\mathbb{E}}_{x \sim p_{t_0}}[\|\psi_\theta(x, t_0, T) - \psi(x, t_0, T)\|_2^2]}.$$

We use Lemma M.2 to bound the $\ell_2$ difference of the flows. To that end, let us first bound the Jacobian matrix $D\psi_\theta(\psi(x, t_0, s), s, t)$. We have

$$\frac{\partial}{\partial t}\|D\psi_\theta(\psi(x, t_0, s), s, t)\|_2$$
$$\leq \|\frac{\partial}{\partial t}D\psi_\theta(\psi(x, t_0, s), s, t)\|_2$$
$$= \|Du_\theta(\psi_\theta(\psi(x, t_0, s), t), s, t)D\psi_\theta(\psi(x, t_0, s), s, t)\|_2$$
$$\leq L_\mathcal{T}\|D\psi_\theta(\psi(x, t_0, s), s, t)\|_2,$$

where the first inequality follows from triangle inequality of the $\|\cdot\|_2$-norm, and the second equality follows from the flow ODE in the assumption of Lemma M.2, and the third inequality follows from the Lipschitzness of transformer network (Definition B.2). Therefore,

$$\|D\psi_\theta(\psi(x, t_0, s), s, t)\|_2 \lesssim \exp\left\{\int_s^t L_\mathcal{T} du\right\} \leq \exp\left\{\int_0^1 L_\mathcal{T} du\right\} =: M. \qquad \text{(By Lemma M.1)}$$

Now we have

$$\|\psi_\theta(x, t_0, T) - \psi(x, t_0, T)\|_2^2$$
$$\leq M^2 \cdot (\int_{t_0}^T \|u_\theta(\psi(x, t_0, s), s) - u(\psi(x, t_0, s), s)\|_2 ds)^2$$
$$\leq M^2 \int_{t_0}^T \|u_\theta(\psi(x, t_0, s), s) - u(\psi(x, t_0, s), s)\|_2^2 ds,$$

where in the first line we apply Lemma M.2 and in the second line we apply the Hölder's inequality. Then, we take expectation with respect to $x \sim p_{t_0}$ on both sides of the above inequality

$$\mathop{\mathbb{E}}_{x \sim p_{t_0}}[\|\psi_\theta(x, t_0, T) - \psi(x, t_0, T)\|_2^2] \leq M^2 \mathop{\mathbb{E}}_{x \sim p_{t_0}}[\int_{t_0}^T \|u_\theta(\psi(x, t_0, s), s) - u(\psi(x, t_0, s), s)\|_2^2 ds]$$
$$= M^2 \int_{t_0}^T \mathop{\mathbb{E}}_{x \sim p_s}[\|u_\theta(x, s) - u(x, s)\|_2^2] ds,$$

where the last equality follows since the samples $\psi(x, t_0, s)$ with $x \sim p_{t_0}$ are the same as the samples $x \sim p_s$ by construction of the flow.

Therefore, we have

$$W_2(\widehat{P}_T, P_T) \leq M \cdot (\int_{t_0}^T \mathop{\mathbb{E}}_{x \sim p_s}[\|u_\theta(x, s) - u(x, s)\|_2^2] ds)^{\frac{1}{2}},$$

where

$$\int_{t_0}^{T} \mathop{\mathbb{E}}_{x \sim p_s} [\|u_\theta(x,s) - u(x,s)\|_2^2] \mathrm{d}s = (T - t_0)\mathcal{R}(u_\theta). \qquad \text{\color{gray}(By Definition I.2)}$$

Then, by Assumption I.2, we have

$$\mathop{\mathbb{E}}_{\{x_i\}_{i=1}^n} [W_2(\widehat{P}_T, P_T)] \leq M \cdot (T - t_0) \mathop{\mathbb{E}}_{\{x_i\}_{i=1}^n} [\sqrt{\mathcal{R}(\widehat{u}_\theta)}] \lesssim M \mathop{\mathbb{E}}_{\{x_i\}_{i=1}^n} [\sqrt{\mathcal{R}(\widehat{u}_\theta)}].$$

Finally, we apply the flow estimation results in Theorem I.3 and get

$$\mathop{\mathbb{E}}_{\{x_i\}_{i=1}^n} [\mathcal{R}(\widehat{u}_\theta)] = O(n^{-\frac{1}{16d+15}} (\log n)^{20d_x + 4\beta + 20}),$$

$$\mathop{\mathbb{E}}_{\{x_i\}_{i=1}^n} [\mathcal{R}(\widehat{u}_\theta)] = O(n^{-\frac{1}{8d+9}} (\log n)^{20d_x + 4\beta + 20}),$$

under Assumption I.1 and Assumption I.3 respectively. These imply

$$\mathop{\mathbb{E}}_{\{x_i\}_{i=1}^n} [W_2(\widehat{P}_T, P_T)] \lesssim M \mathop{\mathbb{E}}_{\{x_i\}_{i=1}^n} [\sqrt{\mathcal{R}(\widehat{u}_\theta)}] = O(n^{-\frac{1}{32d+30}} (\log n)^{10d_x + 2\beta + 10}),$$

$$\mathop{\mathbb{E}}_{\{x_i\}_{i=1}^n} [W_2(\widehat{P}_T, P_T)] \lesssim M \mathop{\mathbb{E}}_{\{x_i\}_{i=1}^n} [\sqrt{\mathcal{R}(\widehat{u}_\theta)}] = O(n^{-\frac{1}{16d+18}} (\log n)^{10d_x + 2\beta + 10}).$$

This completes the proof. $\qquad\qquad\qquad\qquad\qquad\qquad\qquad\qquad\qquad\qquad\qquad\qquad$ $\square$

# N   Proof of Theorem I.5

In this section, we prove the nearly minimax optimality results of flow matching transformers under specified settings (Theorem I.5).

We begin with the definition of modulus of smoothness following [Oko et al., 2023].

**Definition N.1** (Modulus of Smoothness).   Let $\Omega$ be a domain in $\mathbb{R}^{d_x}$ and $f \in L^{p'}(\Omega)$ be a function for some $p' \in (0, \infty]$. We define the $r$-th modulus of smoothness of $f$ by:

$$\omega_{r,p'}(f,t) := \sup_{\|h\|_2 \le t} \|\Delta_h^r(f)\|_{p'},$$

where $\Delta_h^r(\Omega)$ is the difference operator defined by

$$\Delta_h^r(f)(x) := \begin{cases} \sum_{j=0}^r \binom{r}{j} (-1)^{r-j} f(x + jh), & \text{if } x + jh \in \Omega \text{ for all } j, \\ 0, & \text{otherwise.} \end{cases}$$

Next, we define the Besov space.

**Definition N.2** (Besov Space $B_{p',q'}^s$).   Let $0 < p', q' \le \infty$, $s > 0$ and $r := \lfloor s \rfloor + 1$. The Besov norm of a function $f \in L^{p'}(\Omega)$ is defined by $\|f\|_{B_{p',q'}^s} := \|f\|_{p'} + |f|_{B_{p',q'}^s}$, where

$$|f|_{B_{p',q'}^s} := \begin{cases} \left( \int_0^\infty \left( (t^{-s} \omega_{r,p'}(f,t))^{q'} \frac{\mathrm{d}t}{t} \right)^{\frac{1}{q'}}, & q' < \infty, \\ 0, & q' = \infty. \end{cases}$$

Given $m, L > 0$ we have the Besov space $B_{p'q'}^s(L, m) := \{f \in L^{p'}(\Omega) \mid \|f\|_{B_{p'q'}^s} < L, f \ge m\}$.

The next lemma provides the minimax optimal rate for density in the Besov space $B_{p',q'}^s$.

**Lemma N.1** (Theorem 3 of [Niles-Weed and Berthet, 2022]).   Let $\Omega := [-1, 1]^{d_x}$ be the domain of density $q(x_1)$ in Besov space $B_{p',q'}^s(L, m)$. Then, for any $r, p', q' \ge 1$ and $s > 0$,

$$\inf_{\widehat{P}} \sup_{q \in B_{p',q'}^s(L,m)} \mathbb{E}_{\{x_i\}_{i=1}^n} [W_r(\widehat{P}, P)] \gtrsim n^{-\frac{s+1}{d_x + 2s}},$$

where $\{x_i\}_{i=1}^n$ is a set of i.i.d samples drawn from distribution $P$, and $\widehat{P}$ runs over all possible estimators constructed from the data.

Then, we revisit the definition of Wasserstein distance:

**Definition N.3** (2-Wasserstein Distance).   Let $X$ and $Y$ be two random variables with marginal densities $\mu_x$ and $\mu_y$ respectively. We define the 2-Wasserstein distance by:

$$W_2(\mu_x, \mu_y) := \left( \inf_{\pi \in \mathcal{M}(\mu_x, \mu_y)} \int \|x - y\|^p \mathrm{d}\pi(x,y) \right)^{\frac{1}{p}},$$

where $\mathcal{M}(\mu_x, \mu_y)$ denotes the set of joint measures $\pi$ with marginals $\mu_x$ and $\mu_y$.

We then give the minimax optimal rate in the Hölder density function spaces.

**Lemma N.2** (Modified from Theorem 3 of [Niles-Weed and Berthet, 2022]).   Consider the task of estimating a probability distribution $P(x_1)$ with density function belonging to the space

$$\mathcal{P} := \{q(x_1) | q(x_1) \in \mathcal{H}^\beta([-1, 1]^{d_x}, B), q(x_1) \ge C\},$$

Then, for any $r \geq 1$, $\beta > 0$ and $d_x > 2$, we have

$$\inf_{\widehat{P}} \sup_{q(x_1) \in \mathcal{P}} \mathbb{E}_{\{x_i\}_{i=1}^n} [W_r(\widehat{P}, P)] \gtrsim n^{-\frac{\beta+1}{d_x+2\beta}},$$

where $\{x_i\}_{i=1}^n$ is a set of i.i.d samples drawn from distribution $P$, and $\widehat{P}$ runs over all possible estimators constructed from the data.

*Proof.* Let $\Omega$ be some domains. Since $B_{\infty,\infty}^s(\Omega) = \mathcal{H}^s(\Omega)$ for any $s \in \mathbb{R}_+ \setminus \mathbb{Z}_+$, Lemma N.1 directly implies Lemma N.2. This completes the proof. $\square$

Next, we present the proof of Theorem I.5.

**Theorem N.1** (Theorem I.5 Restated: Minimax Optimality of Flow Matching Transformers). Under the setting of $(16d + 18)(\beta + 1) = d_x + 2\beta$, the distribution estimation rate of flow matching transformers (Theorem I.4) matches the minimax lower bound of Hölder distribution class in 2-Wasserstein distance up to a $\log n$ and Lipschitz constants factors.

*Proof of Theorem I.5.* By Theorem I.4, we have the distribution estimation rate in 2-Wasserstein distance under Assumption I.2 and Assumption I.3:

$$\mathbb{E}_{\{x_i\}_{i=1}^n} [W_2(\widehat{P}_T, P_T)] = O(n^{-\frac{1}{16d+18}}(\log n)^{10d_x+2\beta+10}).$$

Then, by Lemma N.2, the distribution rates matches the minimax lower bound up to a $\log n$ and Lipschitz constant factors under the setting

$$(16d + 18)(\beta + 1) = d_x + 2\beta.$$

This completes the proof. $\square$

# O Experimental Validation

To provide empirical support for the proposed High-Order Flow Matching (HOFM) framework, we conduct a series of synthetic experiments designed to evaluate the practical benefits of incorporating higher-order dynamics. We compare the performance of standard first-order flow matching (equivalent to our framework with $K = 1$) against second-order flow matching ($K = 2$).

## O.1 Experimental Setup

**Task and Datasets.** We evaluate the models on 2D density matching tasks, transitioning a standard multivariate Gaussian distribution, $\pi_0$, to three complex target distributions, $\pi_1$. Following the experimental setting in [Chen et al., 2025], we use target distributions shaped as: (1) a square, (2) two intertwined spirals, and (3) three intertwined spirals. These datasets are chosen to test the models' ability to learn distributions with sharp corners and high-curvature manifolds.

**Evaluation Metric.** To quantify the quality of the generated samples, we measure the 2-Wasserstein distance between the generated distribution and the target distribution. A lower Wasserstein distance indicates a better match and, therefore, superior performance.

## O.2 Results and Discussion

The results of our comparison are summarized in Appendix O.2. The findings demonstrate the advantages of using second-order dynamics.

Across all three target distributions and for every sampling step count (10, 50, and 100), the **second-order model achieves a lower Wasserstein distance** than the first-order model. This suggests that incorporating higher-order information allows the model to learn more accurate and stable generation paths, which aligns with the motivations discussed in Section 1.

| Distribution | Sampling Steps | First Order ($K = 1$) | Second Order ($K = 2$) |
|---|---|---|---|
| **Square** | 10 | 8.51 | **7.09** |
| | 50 | 6.45 | **6.08** |
| | 100 | 5.48 | **2.82** |
| **Two Spirals** | 10 | 114.39 | **74.57** |
| | 50 | 73.37 | **68.47** |
| | 100 | 66.15 | **46.71** |
| **Three Spirals** | 10 | 192.19 | **109.93** |
| | 50 | 123.53 | **87.70** |
| | 100 | 93.26 | **68.81** |

Table 1: Comparison of first-order and second-order flow matching on synthetic 2D datasets.

Furthermore, these results highlight a notable improvement in **sampling efficiency**. For instance, in the Three Spirals task, the second-order model with only $50$ sampling steps (Wasserstein distance of $87.70$) outperforms the first-order model with $100$ steps ($93.26$). This empirical evidence supports the theoretical premise that HOFM lead to more efficient sampling strategies (Section 5).

