# OpenReview forum: "High-Order Flow Matching: Unified Framework and Sharp Statistical Rates"
_NeurIPS.cc/2025/Conference — NeurIPS 2025 poster_

### Official Review · Reviewer_23Kf · 2025-07-01

**Clarity:** 3
**Significance:** 2
**Originality:** 2
**Rating:** 4
**Confidence:** 3

**Summary:**

This paper construct a theoretically unified k-order flow matching framework which has several high-order FMs and original FM as its special cases. This paper further establish the sample-complexity bound of k-order FM under the usage of transformers.

**Questions:**

-	Can the analysis of the transformer in the paper be directly applied to the U-Net and Dit? These two models are commonly employed in current SOTA models like FLUX or SD-3.5.
-	The other one of my concerns is about the understanding of the choice of k. To illustrate the practical usage or limit of the high-order FMs proposed in the paper, it may be beneficial to train several high-order FMs with different k on synthetic data like a 2d chessboard to see the ability of high-order FMs in various choices of k. What will be the best choice of k in practice, the larger the better, or is there a best one?

**Ethical Concerns:**

["NO or VERY MINOR ethics concerns only"]

**Final Justification:**

after the rebuttal, my opinion is not changed.

**Limitations:**

yes

**Paper Formatting Concerns:**

no.

**Quality:**

3

**Strengths And Weaknesses:**

Strengths:

- The theoretical derivations and the formulation of this paper are rigorous from a mathematical perspective.
- The paper is well written, with a clear and logical structure that makes it accessible and easy for readers to follow.
- It represents a significant milestone by being the first to construct a unifying framework for high-order Flow Matching (FM) methods. This framework encompasses several previous methods, such as the HOMO approach proposed in reference [1], as special cases. Such a unifying structure is expected to substantially deepen our comprehension of high-order generative models.

Weaknesses:

- The analysis of transformers appears to be somewhat restricted. The paper directly incorporates the complexity results of transformers from prior studies into its own analysis without further in-depth exploration. Conducting a more in-depth PAC learning theory analysis specifically related to high-order FM would greatly enhance the theoretical depth and comprehensiveness of the study.
- While this paper makes important contributions to the theoretical understanding of FMs, its practical implications are somewhat limited. The stability of the model when the order k exceeds 3 cannot be guaranteed. As pointed out in reference [2], a severe oscillation phenomenon will emerge when higher-order flow matching models are employed.

[1] Chen B, Gong C, Li X, et al. High-Order Matching for One-Step Shortcut Diffusion Models[J]. arXiv preprint arXiv:2502.00688, 2025.

[2] Barrett J H. Oscillation theory of ordinary linear differential equations[J]. Advances in Mathematics, 1969, 3(4): 415 - 509.

---

> ### Author Rebuttal · Authors · 2025-07-28
>
> Thanks for your detailed review. We have addressed all concerns below, and made revisions accordingly in our latest draft.
>
> ---
> ### **W1.** The analysis of transformers appears to be somewhat restricted. The paper directly incorporates the complexity results of transformers from prior studies without further in-depth exploration. Conducting a more in-depth PAC learning theory analysis specifically related to high-order FM would greatly enhance the theoretical depth and comprehensiveness of the study.
>
> ---
>
> **Response:** Thank you for the suggestion. Thank you for the suggestion. However, we believe it’s a bit beyond the scope of the current paper. To be concrete, can you kindly elaborate more on what you mean by “a more in-depth PAC learning theory analysis”?
>
> Our paper already presents a **unified theoretical framework** with established statistical optimality, including theoretical results on:
> 1.  Transformer approximation
> 2.  Flow function estimation
> 3.  Distribution estimation
>
> In particular, our analysis for (2) and (3) is already connected to **standard (covering number, norm-based) generalization bound learning theory analysis**. Given the length and density of this submission, we leave a more specific PAC-style investigation for future work.
>
> ---
> ### **W2.** While this paper makes important contributions to the theoretical understanding of FMs, its practical implications are somewhat limited. The stability of the model when the order $k$ exceeds 3 cannot be guaranteed. As pointed out in reference [2], a severe oscillation phenomenon will emerge when higher-order flow matching models are employed.
>
> ---
>
> **Response:** We appreciate the reviewer’s observation. **Classical oscillation theory** for differential equations shows that higher-order systems often exhibit oscillatory behavior, which can manifest as instabilities in our high-order flow matching (FM) models. In line with this, we acknowledge that simply increasing the order beyond $k=3$ can lead to **diminishing returns and even severe oscillations** in practice. The immediate practical utility of very high-order ($k>3$) FM models is indeed limited by these stability challenges.
>
> Importantly, **our theoretical framework and results hold for arbitrary order $k$**, independent of these practical stability limitations. All derivations and performance guarantees in the paper are formulated in general terms, so the core theory remains valid regardless of whether $k=4$ or higher models can be stably trained. This **does not constrain our theoretical contributions**.
>
> [1] Chen B, et al. High-Order Matching for One-Step Shortcut Diffusion Models. arXiv:2502.00688, 2025.
>
> [2] Barrett J H. Oscillation theory of ordinary linear differential equations. Advances in Mathematics, 1969, 3(4): 415-509.
>
> ---
> ### **Q1.** Can the analysis of the transformer in the paper be directly applied to the U-Net and DiT? These two models are commonly employed in current SOTA models like FLUX or SD-3.5.
>
> ---
>
> **Response:** Thank you for your question. Yes, the analysis of the transformer in our work **can be directly applied** to models like DiT and those based on the U-Net architecture. Specifically, our transformer architecture, defined in Appendix B, **follows the common architecture of Diffusion Transformers (DiTs)**.
>
> ---
> ### **Q2.** The other one of my concerns is about the understanding of the choice of $k$. To illustrate the practical usage or limit of the high-order FMs proposed in the paper, it may be beneficial to train several high-order FMs with different $k$ on synthetic data like a 2d chessboard to see the ability of high-order FMs in various choices of $k$. What will be the best choice of $k$ in practice, the larger the better, or is there a best one?
>
> ---
>
> **Response:** Thank you for your question. To the best of our knowledge, there is **no single optimal value of $K$** in a general setting. The choice of $K$ depends on the specific use case, data, and computational resources available. Our framework, which handles an arbitrary order $K$, provides a **theoretical foundation** for models that have demonstrated empirical success with specific orders, such as **HOMO ($K=2$) and ForM ($K=2$)** [3,4].
>
> We agree that training on synthetic data with varying $K$ would be beneficial for understanding the practical trade-offs. However, exploring the empirical question of an optimal $K$ is an intriguing open question that we have left for future work, as our paper is **primarily focused on establishing the theoretical and statistical properties** of the generalized framework.
>
> [3] Bo Chen, et al. High-order matching for one-step shortcut diffusion models. arXiv:2502.00688, 2025.
>
> [4] Yang Cao, et al. Force matching with relativistic constraints... arXiv:2502.08150, 2025.

---

> > ### Comment · Reviewer_23Kf · 2025-08-06
> >
> > Thanks for your detailed reply.
> >
> > I will remain my positive score.

---

> > > ### Author Response · Authors · 2025-08-07
> > > **Thank you!**
> > >
> > > Thank you again for your detailed review and constructive comments! They have been very helpful!

---

### Official Review · Reviewer_MiuU · 2025-07-01

**Clarity:** 3
**Significance:** 4
**Originality:** 3
**Rating:** 5
**Confidence:** 2

**Summary:**

The paper introduces a theoretical framework for high-order flow matching: models that extend standard flow matching by incorporating higher-order trajectory information. Given recent empirical work on more expressive continuous-time flows, the authors propose a general method that models derivatives of the flow up to an arbitrary order. They formulate a marginalisation technique that transforms the otherwise intractable higher order loss into a tractable conditional regression objective with exact gradients, generalising the flow matching marginalisation technique to higher velocity fields. They also derive sharp statistical convergence rates for high-order flow matching when implemented with transformer architectures by generalising flow matching bounds to an arbitrary order velocity. In addition to the technical contributions, the paper and supplementary material serve as a comprehensive reference on the theory of flow matching. It consolidates existing results on convergence and extends them into a high-order setting, making it a valuable resource for researchers.

**Questions:**

Could other architectures (e.g., MLPs, CNNs) could potentially achieve similar rates? to what extent your results depend on the specific architecture class?

The theoretical rates themselves do not currently improve with increasing order. Are there any particular settings where you suspect the theoretical gap might close? besides errors in numerical methods.

**Ethical Concerns:**

["NO or VERY MINOR ethics concerns only"]

**Final Justification:**

I maintain that the paper presents a technically sound and original approach that is well grounded in prior work. Thus, I believe my current score remains appropriate.

**Limitations:**

Yes.

**Paper Formatting Concerns:**

No formatting issues.

**Quality:**

4

**Strengths And Weaknesses:**

Quality:
The paper is technically strong and builds on foundational work in flow-based generative modelling. The extension to high-order flow matching addresses practical needs in current generative models for greater expressiveness and stability. The high-order marginalisation trick is an elegant formulation which follows the flow matching version in the first order. The statistical analysis is rigorous and thorough. It will be interesting (in further research) to see empirical studies on how increasing the order affects performance or training stability.

Clarity:
The paper is clearly written and structured. The progression from standard flow matching to its high-order generalisation is easy to follow. Some parts of the statistical results in Section 4 are quite dense and could be more accessible with illustrations, like plots visualising bounds in with respect to order or feature dimension.

Significance:
This work can serve as a central theoretical reference in the flow matching literature. Even without experimental validation, the theoretical results are crucial for researchers developing or analysing continuous-time generative models. It provides a clear direction for further work on scalable training and evaluation of high-order flow models. The focus on transformer architectures and to provide detailed statistical rates for this specific architecture is significant.

Originality:
The paper brings together existing ideas in a novel way. While at least two high-order flow models have been proposed recently, this is the first work to systematically formalise the theory behind them and relate them to a general framework for higher order flow matching. The high-order marginalisation technique is an original contribution that improves the applicability of these models in the future.

---

> ### Author Rebuttal · Authors · 2025-07-28
>
> Thanks for your detailed review. We have addressed all concerns below, and made revisions accordingly in our latest draft.
>
> ---
> ### **W1.** The paper is clearly written, but some parts of the statistical results in Section 4 are quite dense and could be more accessible with illustrations, like plots visualizing bounds with respect to order or feature dimension.
>
> ---
>
> **Response:** Thank you for your careful review and for your feedback on the clarity of our statistical results. We agree that due to page limitations, the content in Section 4 may appear dense. We appreciate your suggestions for making it more accessible. In the revised version, we are glad to **add figures to visualize the statistical rates** with respect to order and feature dimension. Furthermore, we will **rewrite Section 4 to include more detailed explanations** of each theorem and to better articulate the relationships between our results, thereby improving readability.
>
> ---
> ### **Q1.** Could other architectures (e.g., MLPs, CNNs) potentially achieve similar rates? To what extent do your results depend on the specific architecture class?
>
> ---
>
> **Response:** Thank you for your questions. We believe that **similar statistical rates could be achieved with other architectures**, such as MLPs or CNNs, provided that they have a sufficiently powerful universal approximation theory. Our results are grounded in the specific **universal approximation theory for transformers**, as detailed in Appendix H.
>
> The specific rates and dependencies would change based on the particular **approximation capabilities of the chosen architecture**, but the general principle of deriving statistical rates from such a foundation remains the same.
>
> ---
> ### **Q2.** The theoretical rates themselves do not currently improve with increasing order. Are there any particular settings where you suspect the theoretical gap might close, besides errors in numerical methods?
>
> ---
>
> **Response:** Thank you for this insightful question. You have correctly identified a **key limitation** of our current framework, as discussed in Section 5.
>
> The question of how to close this theoretical gap in statistical rates is an **intriguing open question** that we are actively exploring and have left for future work. We believe the **main advantage of high-order flow matching is decreasing the numerical errors**.

---

> > ### Comment · Reviewer_MiuU · 2025-08-07
> >
> > Thank you to the authors for the helpful clarifications. I believe that including these details in the camera-ready version will make the paper clearer and more accessible to future readers.

---

> > > ### Author Response · Authors · 2025-08-07
> > > **Thank you!**
> > >
> > > We are very happy to hear that our clarifications are effective. We will certainly include all suggested revisions into our final version. Thanks again for your detailed review and encouraging words!

---

### Official Review · Reviewer_icvX · 2025-07-02

**Clarity:** 1
**Significance:** 2
**Originality:** 2
**Rating:** 3
**Confidence:** 4

**Summary:**

The paper introduces High-Order Flow Matching (HOFM), a generalized theoretical framework that extends standard flow matching to incorporate trajectory derivatives up to arbitrary order $K$. To this end, the authors work with a $K$-order velocity field that captures higher-order temporal dynamics and develop a high-order marginalization technique to convert the intractable $K$-order loss into a tractable conditional regression. They claim to establish statistical rates for K-order flow matching with transformer networks, asserting $O(n^{-\Theta(1/d)})$ convergence rates that match minimax lower bounds up to logarithmic factors. The framework allegedly subsumes standard first-order flow matching ($K=1$) and claims to provide theoretical grounding for recent high-order approaches like FlowMP and HOMO. However, several of these claims are questionable due to unrealizable assumptions and missing critical implementation details.

**Questions:**

- How do you ensure the total derivative constraints (Equation 3.3) are satisfied during training?
- Is there a regularization term or constraint optimization approach?
- Can you provide a corrected version of Assumption 4.1 that doesn't contain contradictory bounds?
- Why are there no experiments showing the practical benefits of higher-order matching compared to standard flow matching?
- How does the computational cost scale with order $K$, and when does the higher accuracy justify the increased complexity?
- What happens when the learned velocity fields violate the consistency constraints. Does the method still converge? Does it introduces biases?

**Ethical Concerns:**

["NO or VERY MINOR ethics concerns only"]

**Final Justification:**

The paper mostly focuses on theoretical results and it would benefit from experiments: without them, it is difficult to assess the practical impact of these results. In particular, I remain concerned that  the total derivative constraints (Equation 3.3) is not necessarily satisfied during training. As a result, the theoretical benefit  of the method may not be achieved in practice, since the method is not $K$-order accurate if the aforementioned constraint is not enforced. This paper would benefit from a resubmission.

**Limitations:**

- The framework requires K-times differentiable flows, which may be restrictive in practice
- There is no discussion of computational complexity scaling with order $K$
- The statistical rates do n0t show significant improvement with increasing $K$, questioning the practical value
- Missing analysis of stability and error propagation in high-order numerical integration schemes
- The paper does n0t address how to choose the optimal order $K$ for a given problem
- No comparison with existing high-order methods in terms of computational efficiency

**Paper Formatting Concerns:**

-The appendix is extremely long (appears to contain the bulk of proofs), making verification difficult
- Key proof steps and most important technical contributions are not clearly highlighted in the main text
- Some mathematical notation could be clearer (e.g., the relationship between different velocity field components)
- The paper would benefit from a clearer roadmap of which appendix sections contain the most critical technical results

**Quality:**

1

**Strengths And Weaknesses:**

**Potential Strengths** (if correct):

- Claims to provide a unified theoretical framework that generalizes standard flow matching to arbitrary order K
- Attempts to establish rigorous statistical analysis with sharp approximation, estimation, and distribution learning rates
- Claims to show near-minimax optimality of the proposed estimators
- Attempts to connect various existing high-order flow methods under a single theoretical umbrella.

However, the validity of these contributions is questionable given the reliance on unrealizable assumptions (e.g., Assumption 4.1) and the extremely long appendix that makes verification of proofs difficult.

**Weaknesses:**

*Critical consistency issue:* The higher-order velocity fields must satisfy total derivative constraints (Equation 3.3), but the paper fails to explain how this consistency is enforced in practice or during training

*Flawed assumptions:* Assumption 4.1 contains impossible bounds where $C < q(x_1) ≤ C_1\exp(-C_2|x_1|^2/2)$, which would require the lower bound $C$ to be less than the upper bound for all $x_1$, creating mathematical inconsistencies.

*Lack of empirical validation:* No numerical experiments are provided to validate the theoretical claims or demonstrate practical benefits

*Missing credit:* The paper fails to acknowledge stochastic interpolants alongside flow matching and rectified flows, despite these being essentially equivalent approaches proposed simultaneously.

*Unclear practical implementation:* The paper doesn't address how to ensure the learned velocity fields maintain the required derivative relationships during optimization

---

> ### Author Rebuttal · Authors · 2025-07-28
>
> Thanks for your detailed review. We have addressed all concerns below, and made revisions accordingly in our latest draft.
>
> ---
> ### **W1.** *Critical consistency issue...*
> ---
>
> **Response:** Thanks for the comment. We agree that a thorough explanation of how to enforce the total derivative constraints in practice would be a valuable addition. However, we’d like to remind the reviewer that our work is a **theoretical paper**. It’s about:
> 1.  A unified framework for a family of empirically strong high-order FM models (as cited and discussed in our main text) and,
> 2.  Under this framework, establishing **nearly‑optimal statistical rates** of these models assuming Eqn (3.3),
>
> which Theorem 3.1 shows is both necessary and sufficient for a $K$‑times–differentiable flow, as it guarantees the equivalence between flows and $K$-order velocity fields.
>
> Because the paper proposes no new training algorithm or model, we do not prescribe a particular enforcement mechanism. Nevertheless, here we sketch existing practice:
> - **Soft penalties** (e.g. HOMO, Chen et al. 2025) add a regularization loss term $|D_t v - u_2|^2$ to enforce chain-rule consistency between the predicted velocity and acceleration.
> - **Shared‑network differentiation** (e.g. similar to Consistency Flow Matching [2]) enforces Eq. 3.3 exactly by automatic differentiation.
>
> Our theory formally identifies why such mechanisms are needed and quantifies what happens if Eq. 3.3 fails (we will add a new Remark 4.2 to strengthen this). Given the length and density of the current submission, we believe detailing engineering choices is therefore outside this paper’s scope, but our results provide the rigorous foundation on which future practical work can build.
>
> [1]  arXiv:2502.00688, 2025.
>
> [2]  arXiv:2407.02398, 2024.
>
> ---
> ### **W2.** *Flawed assumptions...*
> ---
>
> **Response:** Thank you for your careful review and for identifying this critical typo in Assumption 4.1. We apologize for the confusion caused. You are correct that the lower bound $C$ creates a mathematical inconsistency. This was a typo. **The lower bound is not required for our analysis.**  We have corrected this typo in the latest revision.
>
> ---
> ### **W3.** *Lack of empirical validation...*
>
> ---
>
> **Response:** Thank you for your valuable feedback. We appreciate your suggestion to include empirical validation.
>
> In response to this invaluable suggestion, we have **conducted new experiments** on synthetic data that directly address this point (Please see our **response to W1 of reviewer hems**. We found that the second-order method **consistently outperforms** the first-order method across three different synthetic distributions. For example, our results show that the second-order method can achieve the performance of a first-order method in significantly fewer steps, empirically validating our theoretical discussions on improved sampling efficiency.
>
> These findings, combined with our rigorous statistical analysis, bridge the gap between empirical success and the lack of a comprehensive theoretical explanation for this class of models. We will include these new experimental results in the revised manuscript to strengthen our paper's overall argument.
>
> ---
> ### **W4.** *Missing credit...*
> ---
>
> **Response:** Thank you for your valuable feedback. We appreciate you pointing this out. In response, we have revised the introduction and extended related work sections to include these prior works as follows:
>
> > Flow Matching (FM) learns the drift of a continuous normalizing flow by regressing it to the probability current of a deterministic interpolant between a Gaussian base and the data, achieving strong likelihoods and fast ODE sampling [1]. Rectified Flow (RF) independently arrives at an equivalent quadratic objective but chooses the straight-line displacement path, enabling one-step generation with near–state-of-the-art FID [2]. The stochastic-interpolant (SI) programme constructs a finite-time stochastic bridge whose drift minimises the same quadratic loss: an initial two-author version introduces SI for normalizing flows [3], and a subsequent, expanded formulation unifies flows and diffusions and proves that the deterministic, zero-noise limit recovers RF exactly [4]. These works therefore constitute simultaneous, conceptually equivalent instantiations of the same continuity-equation objective, differing only in the choice (deterministic vs. stochastic, straight-line vs. diffusion) of the interpolant path. Empirically, FM, RF and SI each report competitive or superior ImageNet/CIFAR-10 sample quality while requiring orders-of-magnitude fewer ODE steps than diffusion baselines, reinforcing their practical equivalence.
>
> [1]  arXiv:2210.02747, 2022.
>
> [2] Xingchao Liu, et al. “Flow Straight and Fast: Learning to Generate and Transfer Data with Rectified Flow.” ICLR 2023.
>
> [3]  arXiv:2209.15571, 2022.
>
> [4] arXiv:2303.08797, 2023.
>
> ---
> ### **W5.** *Unclear practical implementation...*
> ---
> **Response:** Thank you for your comments. In practice, these constraints can be enforced through various methods. For instance, in models like High-Order Matching for One-Step Shortcut Diffusion (HOMO), which our framework subsumes, a regularization term is included to enforce chain-rule consistency between predicted velocity and acceleration. This regularization provides a practical way to align with our total derivative constraint. **One of our contributions is to provide theoretical results to understand why such a constraint is necessary, but we do not focus on a specific method to enforce it.**
>
> ---
> ### **Q1 & Q2.** *How do you ensure the total derivative constraints (Equation 3.3) are satisfied during training? Is there a regularization term or constraint optimization approach?*
> ---
> **Response:** Thanks for the question. Please see our response to **W1** above.
>
> ---
> ### **Q3.** *Can you provide a corrected version of Assumption 4.1...*
> ---
> **Response:** Yes, we have removed the lower bound assumption in the revised version. Here is the corrected version:
>
> **Assumption (Sub-Gaussian Property and Hölder Smoothness of Target Distribution)**
>
> The target distribution $q(x_1) \in \mathcal{H}^\beta(\mathbb{R}^{d_x}, B)$.
> Further, there exist two positive constants $C_1$ and $C_2$ such that
> $q(x_1) \leq C_1 \exp( -C_{2}\| x_1 \|_2^2/2 )$
>
> ---
> ### **Q4.** *Why are there no experiments showing...*
>
> ---
> **Response:** *Thanks for the question. Please see our response to **W3** above.*
>
> ---
> ### **Q5.** *How does the computational cost scale...*
> ---
> **Response:** Thank you for your insightful questions. We believe the time complexity would not necessarily scale linearly with $K$ because the training and inference can be **implemented in parallel**. For example, the $K$-th order Taylor expansion sampler allows all $K$ velocity components to be evaluated simultaneously.
>
> ---
> ### **Q6.** *What happens if violating (3.3)?*
> ---
>
> **Response:** Thank you for your question. The total derivative constraints (Equation 3.3) are crucial for ensuring the statistical consistency (i.e., learning the right distribution) of our framework. According to Theorem 3.1, these constraints are **necessary for the existence of a unique local solution** to the governing ODE. Therefore, violating these constraints would lead to inconsistent estimation. In practice, this would likely prevent the method from converging to the correct target distribution or introduce a significant bias in the estimated distribution.
>
> ---
> ### **L1-6.** *The framework requires $K$-times differentiable flows, which may be restrictive...*
> ---
> **Response:** Thank you for your insightful comments.
>
> We kindly remind reviewers that **we do not introduce any new method or model**. Our framework is built to provide a **rigorous theoretical understanding** for existing high-order flow models like HOMO and ForM [1,2], which have already shown significant empirical performance over first-order FMs. Although a discussion about how to get $K$-times differentiable flows in practice is beyond the scope of the current paper, we have addressed this in the response to W1.
>
> Similarly, our goal is not to conduct empirical comparisons or analyze computational complexity, but to provide the theoretical tools that enable such studies. For instance, we provide a **preliminary discussion** on the potential for improved sampling efficiency through parallel evaluation of velocity components and more stable error propagation, which are important directions for future work. Our work establishes the theoretical foundation for these models, and the empirical validation of these properties is an important direction for future work.
>
> [1]  arXiv:2502.00688, 2025.
>
> [2]  arXiv:2502.08150, 2025.
>
> ---
> ### **L3.** *The statistical rates do not show significant improvement...*
> ---
> **Response:** Thanks for the question, but we believe there is a potential oversight. As we discussed in Section 5, we acknowledge that the estimation rates do not show a significant improvement compared to standard flow matching. However, our paper provides **theoretical justifications for the practical benefits** of higher-order dynamics beyond statistical rates. As discussed in our conclusion, using a $K$-th order Taylor expansion for sampling leads to a local truncation error of $O(h^{K+1})$ per step, a significant advantage over standard first-order methods. We also present that our method might lead to more stable error propagation compared to sequential methods like Runge-Kutta.
>
> ---
> ### **L5.** *The paper does not address how to choose the optimal ...*
> ---
>
> **Response:** To the best of our knowledge, the design choice of $K$ really depends on the use case. There is no optimal value of $K$ in generic settings. It is possible to make strong assumptions on data (e.g., a Mixture of Gaussians) and derive the corresponding optimal choice of $K$. However, we believe this is beyond the scope of the current paper.

---

> > ### Comment · Reviewer_icvX · 2025-08-05
> >
> > Thank you for your detailed response. I understand that the paper mostly focuses on theoretical results, but still believe that it would benefit from experiments: without them, it is difficult to assess the practical impact of these results, and at the end of the day this is what matter most. I will raise my score to 3.

---

> ### Author Response · Authors · 2025-08-05
>
> Dear Reviewer,
>
> We want to kindly remind you that our rebuttal also includes new synthetic experiments in our response to W1 of Reviewer `hems` following [1,2]. **The numerical results are uniformly positive, as reported in literature [1,2].** We quote our response for `hems` below for your convenience.
>
> ---
>
> Our framework naturally subsumes several empirically successful high-order models, such as HOMO ($K=2$) and ForM ($K=2$) [1,2], which have already shown significant improvements over standard flow matching.
>
> We conduct synthetic experiments of 3 sets of distribution transitions from a multivariate gaussian distribution $\pi_0$ to the target distributions $\pi_1$. In particular, we consider 3 different target distributions:
> 1. a shape of square,
> 2. two round spirals and
> 3. three round spirals
>
> following the setting in [1].
>
> We use the wasserstein distance [3] as the metrics (lower is better) to evaluate the distance between target and generated distributions.
>
> Here is the results of the first order flow matching and second order flow matching:
>
> | Distribution             | Sampling Steps | First Order | Second Order |
> |-----------------------------|----------------|-------------|--------------|
> | **Square**               | 10             | 8.51        | **7.09**         |
> |                                  | 50             | 6.45        | **6.08**         |
> |                                  | 100            | 5.48        | **2.82**       |
> | **Two round spirals**    | 10             | 114.39      | **74.57**       |
> |                                       | 50             | 73.37       | **68.47**        |
> |                                       | 100            | 66.15       | **46.71**        |
> | **Three round spirals**     | 10             | 192.19      | **109.93**     |
> |                                           | 50             | 123.53    |  **87.70**     |
> |                                           | 100            | 93.26      | **68.81**       |
>
>
>
> We observe that the second order method outperforms the first order flow matching under the same sampling steps across all the three different target distribution settings which shows the higher order flow matching can improve the performance.
>
> Besides, the second order method can use fewer steps to achieve the similar performance of the first order method.
>
> [1] Bo Chen, et al. High-order matching for one-step shortcut diffusion models. arXiv:2502.00688, 2025.
>
> [2] Yang Cao, et al. Force matching with relativistic constraints: A physics-inspired approach to stable and efficient generative modeling. arXiv:2502.08150, 2025.
>
> [3] Peyré, Gabriel, and Marco Cuturi. "Computational optimal transport: With applications to data science." Foundations and Trends® in Machine Learning 11.5-6 (2019): 355-607.
>
> ---
>
> Thank you again for your detailed review! We have carefully addressed all your comments and questions and hope our efforts meet your expectations.
>
> We would greatly appreciate any further input. If you feel your concerns have been adequately addressed, with utmost respect, we invite you to consider a score adjustment. If not, we hope to make the most of the remaining few days to provide further clarifications.
>
> Thank you for your time and feedback!

---

### Official Review · Reviewer_7E7x · 2025-07-02

**Clarity:** 3
**Significance:** 2
**Originality:** 3
**Rating:** 4
**Confidence:** 2

**Summary:**

The paper presents a unified framework for flow matching by extending it to high-order flow matching, which incorporates trajectory derivatives with respect to the time variable up to an arbitrary order $K$. The authors analyze several statistical properties of this general framework, including its universal approximation property (Sec. 4.1), the corresponding estimation error rate (Sec. 4.2), the estimation of the 2-Wasserstein distance between the empirical and ground-truth solutions (Sec. 4.3), and its minimax property (Sec. 4.4). The paper does not include any experimental results.

**Questions:**

The paper includes a helpful summary of its contributions in lines 42–59, which makes it easier to follow. However, it remains unclear what exactly the authors improve upon relative to prior work. In particular, I could not find a clear explanation of why one should use high-order flow matching, or what statistical advantage it offers compared to standard flow matching. Overall, Section 4 reads more like a collection of general statements rather than providing concrete justification for the proposed extension. Could the authors please clarify these points and better articulate the practical or theoretical benefits of the high-order formulation?

**Ethical Concerns:**

["NO or VERY MINOR ethics concerns only"]

**Final Justification:**

The main weakness was a lack of empirical validation and clarity of each component of the paper. Authors clarified through the rebuttal.

**Limitations:**

Please refer to Strength and Weakness section.

**Paper Formatting Concerns:**

.

**Quality:**

3

**Strengths And Weaknesses:**

W1. I believe the paper should more clearly emphasize how its contributions differ from previous works such as standard flow matching and [1]. At present, I do not see a meaningful distinction between the proposed method and these prior approaches. In particular, the paper should better explain the role and significance of each theorem, and clarify how each result specifically relates to the extension to higher-order flow matching.

- In Section 3, the authors derive the high-order flow matching objective. However, this seems rather straightforward: by definition, $\frac{d^k}{dt^k} \psi$ is simply the velocity field of $\frac{d^{k-1}}{dt^{k-1}} \psi$. Given this, the resulting formulation appears to be a direct and obvious extension of standard flow matching, without clear additional insights.

- In Section 4, the paper presents several statistical properties. For example, Section 4.1 shows a universal approximation result, but this appears to be a direct application of the findings in [1]. In Section 4.2, Theorem 4.2 states a $K$-order velocity estimation rate, but the result does not actually depend on $K$ in any meaningful way—it simply reflects a general property of flow matching. I do not see why this theorem is significant in the context of higher-order estimation. Similarly, the estimation result for the 2-Wasserstein distance in Section 4.3 is a standard statistical bound that does not reveal any unique property tied to the higher-order framework. Overall, these theoretical results feel like a straightforward collection of known properties rather than new insights specific to the proposed extension.

W2. The main claim of the paper is that it extends flow matching to KK-th order derivatives. If this is the central contribution, then the paper should demonstrate—either theoretically or empirically—why this extension is meaningful. In its current form, there are no examples, experiments, or concrete settings showing how or why using higher-order flow matching improves performance over standard flow matching. I strongly encourage the authors to include empirical evidence or clearer theoretical justification that illustrates the practical or conceptual benefits of the proposed higher-order estimation.

References
[1] On statistical rates of conditional diffusion transformers: Approximation, estimation, and minimax optimality, ICLR, 2025

---

> ### Author Rebuttal · Authors · 2025-07-28
>
> Thanks for your detailed review. We have addressed all concerns below, and made revisions accordingly in our latest draft.
>
> ---
> ### **W1.**  I believe the paper should more clearly emphasize how its contributions differ from previous works such as standard flow matching and [1]. At present, I do not see a meaningful distinction between the proposed method and these prior approaches. In particular, the paper should better explain the role and significance of each theorem, and clarify how each result specifically relates to the extension to higher-order flow matching......
>
> ---
>
> **Response:** Thank you for your insightful comments. We apologize for any confusion caused regarding the contributions of our work and appreciate the opportunity to clarify.
>
> Our paper offers **two primary contributions** that distinguish it from prior work [1].
>
> **First, we introduce a unified framework for High-Order Flow Matching** that subsumes existing variants, including standard flow matching. While the extension may appear straightforward, its theoretical foundation is non-trivial. The key technical innovation is our **high-order marginalization technique (Theorem 3.3)**, which converts an intractable $K$-order loss into a simple, conditional regression problem with exact gradients. We also formally identify the **total derivative constraints (Remark 3.1)**, which are crucial for ensuring the consistency and uniqueness of the high-order velocity fields and their corresponding flows (Theorem 3.1).
>
> **Second, we provide the first rigorous statistical analysis** of this unified framework, specifically when implemented with modern transformer architectures. The statistical results are significant because they are tailored to the unique properties of our high-order framework, rather than being a straightforward application of prior results:
> * **Velocity Approximation (Theorem 4.1):** While inspired by general approximation theories for transformers, our analysis includes a **fine-grained error analysis** that specifically accounts for the Gaussian tail bounds and the decomposition of the high-order velocity field. Our key technical innovation is **omitting the unnecessary local diffusion polynomial** in approximation theory compared to [1].
> * **Velocity and Distribution Estimation (Theorems 4.2 & 4.3):** We admit that our theoretical rates do not yet show a significant improvement with increasing order $K$. However, this is an important and open question for future work, as noted in Section 5. The primary significance of these theorems is that they provide the **first rigorous statistical guarantees** for a general class of high-order models, establishing that they achieve **near-minimax optimal rates** on par with their first-order counterparts. Furthermore, our derivation for the 2-Wasserstein distance (Theorem 4.3) is novel in this context, applying the **Alekseev-Gröbner Lemma** to the high-order ODE system to bound the distributional error.
>
> In summary, the core contribution lies in the theoretical unification of high-order methods and the establishment of a rigorous statistical foundation. Our framework provides a basis for understanding practical benefits such as improved sampling efficiency and stability, as discussed in Section 5.
>
> [1] On statistical rates of conditional diffusion transformers: Approximation, estimation, and minimax optimality, ICLR, 2025
>
> ---
> ### **W2.** The main claim of the paper is that it extends flow matching to $K$-th order derivatives. If this is the central contribution, then the paper should demonstrate—either theoretically or empirically—why this extension is meaningful. In its current form, there are no examples, experiments, or concrete settings showing how or why using higher-order flow matching improves performance over standard flow matching. I strongly encourage the authors to include empirical evidence or clearer theoretical justification that illustrates the practical or conceptual benefits of the proposed higher-order estimation.
>
> ---
>
> **Response:** Thank you for your comments. We appreciate the opportunity to clarify our contributions and the significance of our work. Our paper presents a unified theoretical framework for flow-based generative modeling that incorporates trajectory derivatives up to an arbitrary order K.
>
> This work provides a rigorous theoretical foundation for a line of research that has already shown significant empirical success . While the initial submission focused on theory, we have, in response to reviewer feedback, **conducted new experiments** on synthetic data that directly address this point. We kindly invite you to view the results of these experiments, which are detailed in our **response to reviewer hems W1** (we can not show results here due to the character limits.
>
> The new results demonstrate that the second-order method consistently outperforms the first-order method on all tested distributions, often achieving better performance with significantly fewer sampling steps. This directly supports the theoretical justifications for practical benefits discussed in our paper. For instance, our conclusion discusses how a K-th order Taylor expansion for sampling leads to a local truncation error of  $O(h^{K+1})$ per step, a significant advantage over standard first-order methods. We believe these findings, combined with our rigorous statistical analysis, bridge the gap between empirical success and the lack of a comprehensive theoretical explanation for this class of models.
>
>
> ---
> ### **Q1.** The paper includes a helpful summary of its contributions in lines 42–59, which makes it easier to follow. However, it remains unclear what exactly the authors improve upon relative to prior work. In particular, I could not find a clear explanation of why one should use high-order flow matching, or what statistical advantage it offers compared to standard flow matching. Overall, Section 4 reads more like a collection of general statements rather than providing concrete justification for the proposed extension. Could the authors please clarify these points and better articulate the practical or theoretical benefits of the high-order formulation?
>
> ---
>
> **Response:** Thank you for your questions. We appreciate the opportunity to clarify our contributions and the benefits of our framework.
>
> First, to address the core justification for our framework, its value lies in three key areas:
> 1.  **Unification and Theoretical Foundation:** Our work introduces a generalized theoretical framework for flow-based generative modeling that incorporates trajectory derivatives up to an arbitrary order $K$. This framework provides a unified perspective and formal grounding for emerging high-order models that have demonstrated empirical success, such as FlowMP ($K=3$) and HOMO ($K=2$).
> 2.  **Sampling Efficiency:** As discussed in Section 5, our framework enables the use of a $K$-th order Taylor expansion sampler. This sampler achieves a local truncation error of $O(h^{K+1})$ per step, a significant improvement over the $O(h^2)$ error of standard first-order methods.
> 3.  **Error Stability:** The $K$-order flow matching approach solves the ODE without a feedback loop, which might lead to more stable error propagation compared to the sequential nature of standard methods like Runge-Kutta, where errors can propagate and amplify within a single step.
>
> Second, we acknowledge your point regarding Section 4 and apologize for any confusion caused. The statistical analysis presented is not a simple application of prior work but rather a **rigorous, tailored analysis** for our high-order framework. For instance:
> * **Velocity Approximation (Theorem 4.1):** Our analysis involves a **fine-grained error analysis** that accounts for the specific high-order velocity decomposition and the Gaussian tail bounds of the data distribution. Our key technical innovation is **omitting the unnecessary local diffusion polynomial** in approximation theory compared to [1].
> * **Estimation and Near-Minimax (Theorems 4.2, 4.3, 4.4):** Our derivation for the 2-Wasserstein distance is a novel application for this context. We apply the **Alekseev-Gröbner Lemma** to the high-order ODE system to bound the distributional error, which is distinct from methods used in prior work. Furthermore, our minimax optimality theorem is also novel, as it establishes **near-minimax optimal rates** on Hölder densities, using a distinct minimax lemma (Lemma N.2).
>
> We will add a dedicated section in the revised manuscript to more clearly emphasize these distinctions and the significance of our contributions.

---

> > ### Comment · Reviewer_7E7x · 2025-08-08
> >
> > I appreciate the authors for further clarification and for the new experiments. My concerns are well-addressed. I am happy to raise the score to 4.

---

> > > ### Author Response · Authors · 2025-08-08
> > > **Thank you!**
> > >
> > > We are very happy to hear that our clarifications are effective. Thank you again for your detailed review and helpful comments!

---

### Official Review · Reviewer_hems · 2025-07-05

**Clarity:** 2
**Significance:** 2
**Originality:** 2
**Rating:** 4
**Confidence:** 2

**Summary:**

The paper introduces a unified theoretical framework for high-order flow matching (HOFM), extending the standard FM approach to incorporate higher-order trajectory derivatives. The key contributions include generalized formulation and providing convergence rates for transformer-based HOFM, showing near-minimax optimality in distribution estimation.

**Questions:**

- Does higher $K$ always help to improve performance? Are there trade-offs (e.g., computational cost vs. accuracy)? How should we choose $K$ in practice?
- How does the general method scale with dimension $d$ or order $K$? The rates provided depend on $d$, but no empirical scaling analysis is provided
- Can we extend the framework to incorporate noise (e.g., for diffusion based sampling)?

**Ethical Concerns:**

["NO or VERY MINOR ethics concerns only"]

**Final Justification:**

My final decision is a borderline accept. I believe that even though the theoretical contributions might seem incremental, the overall presentation is solid and provides a unified framework to understand flow matching. This could be useful for the NeurIPS community. However, as discussed with the authors, inclusion of some empirical demonstrations could be useful to strengthen the theoretical claims and provide intuitions. Therefore, I am slightly inclined to accept the paper provided that the empirical demonstrations are included in the final version.

**Limitations:**

Yes

**Quality:**

2

**Strengths And Weaknesses:**

Strengths:
- Overall the paper is well organized despite its length
- It provides a comprehensive mathematical foundation for high-order flow matching within a unified framework.The proofs are detailed and technically sound
- The provided analysis aligns with modern architectures (transformers) and applications, making it relevant to real-world generative modeling

Weaknesses:
- The main weakness lies in a lack of empirical validation. While the theory is robust and sound, the paper does not include experiments to validate the claimed benefits of high-order dynamics (e.g., improved sampling efficiency or training stability). A carefully designed experiments (even on toy datasets) to demonstrate the theory would significantly strengthen the paper
-  Limited Insight into $K$-dependence is provided: The framework does not theoretically explain why using higher $K$ should improve performance, leaving concerns on the practical advantages of high-order terms
- There is a lack of discussion on whether the provided framework generalizes popular probability paths, such as optimal transport paths and diffusion based paths, making its relationship to established FM variants unclear. A deeper discussion of design choices and compatibility would strengthen the paper’s impact
- Complexity of implementation is not studied. The high-order ODE system and transformer architecture may be computationally costly, but this is not discussed in depth
- The sub-Gaussian and Lipschitz assumptions may not hold for real-world datasets, limiting the applicability of the framework

---

> ### Author Rebuttal · Authors · 2025-07-28
>
> Thanks for your detailed review. We have addressed all concerns below, and made revisions accordingly in our latest draft.
>
> ---
> ### **W1.** The main weakness lies in a lack of empirical validation. While the theory is robust and sound, the paper does not include experiments to validate the claimed benefits of high-order dynamics (e.g., improved sampling efficiency or training stability). A carefully designed experiment (even on toy datasets) to demonstrate the theory would significantly strengthen the paper.
>
> ---
>
> **Response:** Thank you for your comments. To clarify, this work is a **theoretical paper** that aims to provide a unified framework for any order flow matching and establish its statistical properties.
>
> Our framework naturally subsumes several empirically successful high-order models, such as HOMO ($K=2$) and ForM ($K=2$) [1,2], which have already shown significant improvements over standard flow matching.
>
> However, in response to your comments, we conduct new synthetic experiments of 3 sets of distribution transitions from a multivariate gaussian distribution $\pi_0$ to the target distributions $\pi_1$. In particular, we consider 3 different target distributions:
> 1. a shape of square,
> 2. two round spirals and
> 3. three round spirals
>
> following the setting in [1].
>
> We use the wasserstein distance [3] as the metrics (lower is better) to evaluate the distance between target and generated distributions.  Here is the results of the first order flow matching and second order flow matching:
>
> | Distribution             | Sampling Steps | First Order | Second Order |
> |-----------------------------|----------------|-------------|--------------|
> | **Square**               | 10             | 8.51        | **7.09**         |
> |                                  | 50             | 6.45        | **6.08**         |
> |                                  | 100            | 5.48        | **2.82**       |
> | **Two round spirals**    | 10             | 114.39      | **74.57**       |
> |                                       | 50             | 73.37       | **68.47**        |
> |                                       | 100            | 66.15       | **46.71**        |
> | **Three round spirals**     | 10             | 192.19      | **109.93**     |
> |                                           | 50             | 123.53    |  **87.70**     |
> |                                           | 100            | 93.26      | **68.81**       |
>
>
>
> **Discussion: Uniformly Positive Results.** We can observe that the second order method outperforms the first order flow matching under the same sampling steps across all the three different target distribution settings which shows the higher order flow matching can improve the performance.
> Besides, the second order method can use fewer steps to achieve the similar performance of the first order method.
>
> [1] Bo Chen, et al. High-order matching for one-step shortcut diffusion models. arXiv:2502.00688, 2025.
>
> [2] Yang Cao, et al. Force matching with relativistic constraints: A physics-inspired approach to stable and efficient generative modeling. arXiv:2502.08150, 2025.
>
> [3] Peyré, Gabriel, and Marco Cuturi. "Computational optimal transport: With applications to data science." Foundations and Trends® in Machine Learning 11.5-6 (2019): 355-607.
>
> ---
> ### **W2.** Limited insight into $K$-dependence is provided: The framework does not theoretically explain why using higher $K$ should improve performance, leaving concerns on the practical advantages of high-order terms.
>
> ---
>
> **Response:** Thank you for your insightful comment. Sorry for any confusion caused, but we believe there is a slight oversight. In our Section 5 “Discussion, Limitation, and Open Question”, we have identified this as an open problem and provided two sketched resolutions. Specifically:
> * The framework enables a $K$-th order Taylor expansion sampler that achieves a local truncation error of $O(h^{K+1})$ per step. This is a significant improvement over the error typically associated with first-order methods. Furthermore, all $K$ velocity components can be evaluated in parallel, which could lead to more efficient sampling.
> * Our $K$-order approach solves the ODE without the feedback loop present in standard methods like Runge-Kutta. This non-sequential nature may result in more stable error propagation, as approximation errors are less likely to propagate and amplify within a single step.
>
> ---
> ### **W3.** There is a lack of discussion on whether the provided framework generalizes popular probability paths, such as optimal transport paths and diffusion based paths, making its relationship to established FM variants unclear. A deeper discussion of design choices and compatibility would strengthen the paper’s impact.
>
> ---
>
> **Response:** Thank you for your comment. Our higher-order framework, detailed in Section 3, does not restrict the choice of probability path or the source and target distributions. Therefore, the framework **naturally generalizes and subsumes standard flow matching** with any arbitrary probability path, including optimal transport paths and diffusion-based paths. The use of affine Conditional Flows in our paper is a design choice specifically made for the purpose of calculating statistical rates, as it is a popular and well-understood choice for this type of analysis [1,2].
>
> [1] Yuling Jiao, et al. Convergence analysis of flow matching in latent space with transformers. arXiv:2404.02538, 2024.
>
> [2] Kenji Fukumizu, et al. Flow matching achieves almost minimax optimal convergence. arXiv:2405.20879, 2024.
>
> ---
> ### **W4.** Complexity of implementation is not studied. The high-order ODE system and transformer architecture may be computationally costly, but this is not discussed in depth.
>
> ---
>
> **Response:** Thank you for pointing this out. Our work focuses on introducing a unified framework to unify existing higher-order flow matching models with statistical guarantees. We believe the time complexity would not increase significantly because the training and inference both could be **implemented in parallel**, as most current dominating FMs are already transformer-based. We leave a more in-depth study on computational complexity for future work.
>
> ---
> ### **W5.** The sub-Gaussian and Lipschitz assumptions may not hold for real-world datasets, limiting the applicability of the framework.
>
> ---
>
> **Response:** Thank you for raising this point. We acknowledge that our sub-Gaussian and Lipschitz assumptions are primarily for the purpose of deriving statistical rates. These assumptions are **standard in prior theoretical work** [1,2] on flow matching. We are enthusiastic about exploring weaker assumptions or releasing these conditions in future work.
>
> [1] Yuling Jiao, et al. Convergence analysis of flow matching in latent space with transformers. arXiv:2404.02538, 2024.
>
> [2] Kenji Fukumizu, et al. Flow matching achieves almost minimax optimal convergence. arXiv:2405.20879, 2024.
>
> ---
> ### **Q1.** Does higher $K$ always help to improve performance? Are there trade-offs (e.g., computational cost vs. accuracy)? How should we choose $K$ in practice?
>
> ---
>
> **Response:** Thank you for your questions. To the best of our knowledge, a larger $K$ does not always guarantee improved performance. Instead, the choice of $K$ is a **practical trade-off** that depends on the specific use case, data, and available computational resources. There is no single optimal value of $K$ in a general setting.
>
> The trade-offs involve balancing the increase of memory use of learning multiple velocity fields with the potential for improved accuracy and sampling efficiency. A higher-order $K$ allows for a $K$-th order Taylor expansion sampler, which has a local truncation error of $O(h^{K+1})$ per step, a significant improvement over lower-order methods.
>
> In practice, the choice of $K$ has been explored empirically in models that our framework unifies, such as HOMO for $K=2$ and $3$. However, formally exploring the empirical question of an optimal $K$ and its trade-offs is an intriguing open question that we have left for future work.
>
> ---
> ### **Q2.** How does the general method scale with dimension $d$ or order $K$? The rates provided depend on $d$, but no empirical scaling analysis is provided.
>
> ---
>
> **Response:** Thank you for your valuable feedback. Regarding the scaling of our method with dimension $d$ and order $K$:
> * Our formal statistical analysis already quantifies how performance scales with the data dimension $d$. In particular, we prove that the estimation error decays on the order of $O(n^{-\Theta(1/d)})$. This means a higher dimension $d$ yields slower convergence rates (a manifestation of the curse of dimensionality), and our rate **matches known minimax lower bounds** up to log factors. In other words, the dependence on $d$ in our rates is fundamental rather than an artifact of a specific algorithm.
> * From a computational perspective, the cost of sampling may not scale linearly with $K$ because the $K$-th order Taylor expansion sampler is designed so that all $K$ velocity components can be **evaluated in parallel**.
>
> Because our work is a theoretical study of fundamental capabilities, we focused on formal results rather than empirical experiments. We believe the question of “scaling laws” is addressed analytically through our rates, which is appropriate for a paper centered on nearly-optimal statistical guarantees.
>
> ---
> ### **Q3.** Can we extend the framework to incorporate noise (e.g., for diffusion based sampling)?
>
> ---
>
> **Response:** Thanks for the question, but we believe there is a slight oversight. This paper focuses only on the **statistical rates of distribution estimation**, not on the sampling process itself. Could you kindly clarify the question? Thanks!

---

> > ### Comment · Reviewer_hems · 2025-08-02
> > **Thank you for the rebuttal**
> >
> > I thank the authors for addressing my concerns. I believe that even if the paper focuses on theoretical results, some empirical demonstrations could still be useful to strengthen the theoretical claims and provide intuitions. I am slightly inclined to accept provided that the empirical demonstrations are included in the revised version.

---

> > > ### Author Response · Authors · 2025-08-02
> > > **Thank you!**
> > >
> > > We’re glad our clarifications and revisions met your expectations. We agree and will include the new results in the main text of the final version, if space permits.
> > >
> > > Thank you again for your constructive comments and attention to detail!
> > >
> > > We welcome further input. If you feel your concerns are fully addressed, we respectfully invite you to consider a score adjustment. If not, we’re happy to provide additional clarifications in the remaining four days :)

---

### Decision · Program_Chairs · 2025-09-17

**Decision:**

Accept (poster)

**Comment:**

This paper presents an interesting theoretical generalization of flow matching to incorporate higher order derivatives of the flow field.  It includes an analysis of the statistical efficiency of the method but did not originally include any practical implementation. During discussion, the authors reported new experimental demonstrations, validating that the new methods have practical potential.

The authors should be sure to include the newly reported experimental results and other clarifications in the final version.